Extensive microbial diversity within the chicken gut microbiome revealed by metagenomics and culture

Gilroy Rachel 1
Ravi Anuradha 1
http://orcid.org/0000-0002-2937-3420 Getino Maria 2
Pursley Isabella 2
Horton Daniel L. 2
http://orcid.org/0000-0002-1243-0767 Alikhan Nabil-Fareed 1
Baker Dave 1
Gharbi Karim 3
http://orcid.org/0000-0003-2808-0009 Hall Neil 3 4
http://orcid.org/0000-0003-4211-0358 Watson Mick 5
http://orcid.org/0000-0003-4826-5406 Adriaenssens Evelien M. 1
http://orcid.org/0000-0001-6620-9403 Foster-Nyarko Ebenezer 1
Jarju Sheikh 6
Secka Arss 7
Antonio Martin 6
Oren Aharon 8
http://orcid.org/0000-0001-5037-2695 Chaudhuri Roy R. 9
La Ragione Roberto 2
Hildebrand Falk 1 3 Falk.Hildebrand@quadram.ac.uk
http://orcid.org/0000-0003-1807-3657 Pallen Mark J. 1 2 4 mark.pallen@quadram.ac.uk
1 Quadram Institute Bioscience , Norwich , UK
2 School of Veterinary Medicine, University of Surrey , Guildford , UK
3 Earlham Institute, Norwich Research Park , Norwich , UK
4 University of East Anglia , Norwich , UK
5 Roslin Institute, University of Edinburgh , Edinburgh , UK
6 Medical Research Council Unit The Gambia at the London School of Hygiene and Tropical Medicine, Atlantic Boulevard , Banjul , The Gambia
7 West Africa Livestock Innovation Centre , Banjul , The Gambia
8 Department of Plant and Environmental Sciences, The Alexander Silberman Institute of Life Sciences, Edmond J. Safra Campus, Hebrew University of Jerusalem , Jerusalem , Israel
9 Department of Molecular Biology and Biotechnology, University of Sheffield , Sheffield , UK
Driscoll Timothy
Electronic publication date: 2021 Apr 6
Publication date: 2021
Volume: 9
Electronic Location ID: e10941
Received 2020 Dec 4; Accepted 2021 Jan 22
Copyright: © 2021 Gilroy et al.
Copyright year: 2021
Copyright holder: Gilroy et al.
License: This is an open access article distributed under the terms of the Creative Commons Attribution License, which permits unrestricted use, distribution, reproduction and adaptation in any medium and for any purpose provided that it is properly attributed. For attribution, the original author(s), title, publication source (PeerJ) and either DOI or URL of the article must be cited.
License URL: https://creativecommons.org/licenses/by/4.0/

Keywords: Chickens, Gut microbiome, Biodiversity, Metagenomics, Metagenome-assembled genome, Bacterial nomenclature, Uncultured bacteria, Candidatus

Funding: Quadram Institute Bioscience BBSRC BB/R012504/1 and BBS/E/F/000PR10351 Medical Research Council CLIMB MR/L015080/1 British Egg Marketing Board Research and Education Trust BSRC Institute BB/r012490/1, BBS/E/F/000Pr10353 and BBS/E/F/000PR10356 This research is supported by the Quadram Institute Bioscience BBSRC-funded Strategic Programme: Microbes in the Food Chain (project no. BB/R012504/1) and its constituent project BBS/E/F/000PR10351 (Theme 3, Microbial Communities in the Food Chain) and by the Medical Research Council CLIMB grant (MR/L015080/1) and the British Egg Marketing Board Research and Education Trust. Evelien M. Adriaenssens and Falk Hildebrand were funded by the BBSRC Institute Strategic Programme Gut Microbes and Health BB/r012490/1, its constituent project BBS/E/F/000Pr10353 and BBS/E/F/000PR10356. The funders had no role in study design, data collection and analysis, decision to publish, or preparation of the manuscript.

==============================
Background

The chicken is the most abundant food animal in the world. However, despite its importance, the chicken gut microbiome remains largely undefined. Here, we exploit culture-independent and culture-dependent approaches to reveal extensive taxonomic diversity within this complex microbial community.

Results

We performed metagenomic sequencing of fifty chicken faecal samples from two breeds and analysed these, alongside all (n = 582) relevant publicly available chicken metagenomes, to cluster over 20 million non-redundant genes and to construct over 5,500 metagenome-assembled bacterial genomes. In addition, we recovered nearly 600 bacteriophage genomes. This represents the most comprehensive view of taxonomic diversity within the chicken gut microbiome to date, encompassing hundreds of novel candidate bacterial genera and species. To provide a stable, clear and memorable nomenclature for novel species, we devised a scalable combinatorial system for the creation of hundreds of well-formed Latin binomials. We cultured and genome-sequenced bacterial isolates from chicken faeces, documenting over forty novel species, together with three species from the genus Escherichia, including the newly named species Escherichia whittamii.

Conclusions

Our metagenomic and culture-based analyses provide new insights into the bacterial, archaeal and bacteriophage components of the chicken gut microbiome. The resulting datasets expand the known diversity of the chicken gut microbiome and provide a key resource for future high-resolution taxonomic and functional studies on the chicken gut microbiome.

Introduction

The domestic chicken is the most abundant bird and most abundant food animal on Earth, accounting for a larger fraction of the planet’s biomass than all species of wild birds combined (Bennett et al., 2018). Consumption of chicken meat is growing faster than any other type of meat and is seen as a cheaper, healthier, low-carbon alternative to meat from mammalian livestock (Eshel et al., 2014; Willett et al., 2019). Chicken eggs remain a nutritious, affordable food across the globe (Réhault-Godbert, Guyot & Nys, 2019).

The chicken gastrointestinal tract is home to a complex community of microbes and their genes—the chicken gut microbiome—that underpins links between diet, health and productivity in poultry, as evidenced by the ability of antibiotics to promote growth in chicks (Bedford, 2000). This microbial community also acts as a source of pathogens associated with disease in birds or in humans—including Campylobacter, Salmonella, and Escherichia coli—as well as providing a reservoir of antimicrobial resistance genes (Florez-Cuadrado et al., 2018; Jørgensen et al., 2019; Hermans et al., 2012).

Previous studies of this community have documented a rich variety of microorganisms (dominated by bacteria, but including viruses, archaea and microbial eukaryotes) and have shown that the taxonomic composition of this community varies with age, breed and disease status (Shang et al., 2018; Rychlik, 2020). However, these earlier efforts have largely relied on analyses of molecular barcodes (in particular short 16S rRNA gene sequences), which fail to provide species-level resolution, are unable to detect viruses and reveal nothing about the genome sequences, population structures or functional repertoires of microbial species (Hillmann et al., 2018).

Two strategies have proven productive for exploring taxonomic and functional diversity in complex microbial communities (Almeida et al., 2019; Forster et al., 2019). Culture-independent approaches rely on shotgun metagenomic sequencing of DNA extracted from relevant samples, followed by bioinformatics-based community profiling and analysis (Glendinning et al., 2020; Sergeant et al., 2014). Culture-dependent approaches combine large-scale isolation of microorganisms in pure culture with whole-genome sequencing and phylogenomic analysis (Medvecky et al., 2018). To explore taxonomic novelty in the chicken gut microbiome, we generated phylogenetic profiles to document known and unknown diversity and then exploited culture-dependent and culture-independent approaches to create an unprecedented high-quality reference collection of microbial genes and genomes from the chicken gut, revealing and naming hundreds of new candidate species from this commonplace but important ecological setting.

Materials and Methods

Sample collection and storage

Faecal samples were collected in South-East England from adult Lohmann Brown laying hens and adult Silkie hens in 2018. Birds were housed in a large outdoor run with a substrate of stone chippings and small turf enrichment beds during the day and kept in a coop overnight. They were fed a commercial layer feed, Farmgate Layer pellets and mash (ForFarmers UK Limited, Rougham, Bury St Edmunds), according to the manufacturer’s instructions and no antibiotics were used. Faecal sampling was approved by the University of Surrey’s NASPA ethics committee.

Sixty faecal samples were collected from the Lohmann Brown laying hens and thirty samples from the Silkie hens (six and three samples per day, respectively, for 10 days). Freshly evacuated faeces from individual birds were collected in sterile containers and immediately stored at −20 °C. Samples were then transferred to the laboratory for culture and/or DNA extraction. DNA was extracted using DNeasy PowerSoil kit (Qiagen, Hilden, Germany), following manufacturer’s instructions and then stored at −20 °C.

Sequencing and subsequent workflow

Workflow from this point forward is summarised in Fig. 1.The fifty samples yielding >20 ng DNA were processed according to the Low Input, Transpose Enabled (LITE) library construction pipeline (Perez-Sepulveda et al., 2020) before being subjected to paired-end (2 × 150 bp) metagenomic sequencing on the Illumina Novaseq 6000 platform. Bioinformatics analyses were performed on the Earlham Institute’s High Performance Computing cluster and on the Cloud Infrastructure for Microbial Bioinformatics (Connor et al., 2016). Sequences were assessed for quality using FastQC Version 0.11.8 and trimmed using Trimmomatic Version 0.36, configured to a minimum read length of 40, ‘leading’ and ‘trailing’ settings of 3 (SLIDINGWINDOW:4:20) (http://www.bioinformatics.babraham.ac.uk/projects/fastqc; Bolger, Lohse & Usadel, 2014). Metagenomic sequences for all samples have been uploaded to the Sequence Read Archive under Bioproject ID PRJNA543206.

Figure 1 Analytical Workflow.

An asterisk (*) indicates read numbers are detailed post-filtering of diet and host associated reads.

Reference-based metagenomic analysis

An initial analysis of our chicken faecal sequences using the Kraken 2 taxonomic classifier (Wood, Lu & Langmead, 2019) was performed on custom databases representing the domestic chicken genome (GenBank assembly accession GCF_000002315.6) and the food plants Triticum aestivum (wheat), Aegilops tauschii (diploid progenitor of the D genome of hexaploid wheat) and Glycine max (soy bean): GenBank assembly accessions GCF_001957025.1, GCA_900519105.1, GCF_000004515.5. Kraken 2 revealed that 8% of reads originated from the chicken and at least 16% originated from the diet. These sequences were filtered from our dataset and excluded from subsequent analyses by keeping only reads ‘Unclassified’ by Kraken 2 after comparison with each database in turn.

The remaining dataset underwent taxonomic profiling using Kraken 2 against a microbial database built from all complete/representative archaeal, bacterial, fungal, protozoan, viral and UniVec_Core sequences in RefSeq (O’Leary et al., 2016) in January 2020. Bracken (Lu et al., 2017) was used to estimate taxon abundance from the Kraken 2 profiles, accepting only those taxa with ≥1,000 assigned reads. Bracken-database files were generated using ‘bracken-build’ on our microbial database and visualised using Pavian (Breitwieser & Salzberg, 2016).

Metagenomic assembly

We searched the NCBI BioProjects database (https://www.ncbi.nlm.nih.gov/bioproject/) in November 2019 with the term ‘chicken gut microbiome’ and then selected nine publicly available projects that contained at least one metagenomic sequence dataset >1 GByte in size: PRJEB33338, PRJNA193217, PRJNA291299, PRJNA375762, PRJNA415593, PRJNA417359, PRJEB22062, PRJNA543206, PRJNA417359, PRJNA385038, PRJNA616250. Only four of these studies were linked to research publications at the time of publication (Glendinning et al., 2020; Sergeant et al., 2014; Foster-Nyarko et al., 2020; Luiken et al., 2020)

All shotgun metagenomic reads were quality-filtered by removing reads shorter than 70% of the maximum expected read length (100 bp, 250 bp for MiSeq data), an estimated accumulated error >2.5 with a probability of ≥0.01 (Puente-Sánchez, Aguirre & Parro, 2016) or with an observed accumulated error >2, or >1 ambiguous position to assist assembly. If base quality dropped below 20 in a window of 15 bases at the 3′ end, or if the accumulated error exceeded 2, reads were trimmed. All these filter steps are integrated in sdm (Hildebrand et al., 2014). Reads mapping to the chicken genome and diet were removed from the metagenomic data as described previously, classifying reads with Kraken 2 against custom databases built on the aforementioned genomes.

Sequence datasets from our fifty samples—together with 582 samples from the selected BioProjects—were assembled using MegaHIT (Li et al., 2016) under the option ‘--k-list 25,43,67,87,101,127’. To avoid artefacts that sometimes result from co-assembly of sequences from different samples and different sources, we performed individual assemblies on each sample, with the exception of BioProject PRJNA417359. For that BioProject, as multiple metagenomic samples had been sourced from different tissues of the same individual bird, we co-assembled reads from the 120 BioSamples from that project.

Bacteriophage identification and characterisation

Contig sequences from the MegaHIT assemblies of our 50 samples that were ≥10 kb were analysed with VirSorter v1.0.5 with the ‘-db 2’ option to identify viral genomes (Roux et al., 2015). VirSorter Category 1 and 2 contig sequences were collapsed at 95% nucleotide identity over 70% of the sequence length using CD-Hit Est v4.6.1 (Fu et al., 2012). Classification of bacteriophage sequences relied on nucleotide searches using BLASTN against the NCBI NT database (Completed April 2020) and protein searches using Kaiju Version 1.7.3 against the RefSeq database (Completed April 2020) (Menzel, Ng & Krogh, 2016). Only bacteriophage genomes with BLASTN hit E-Value < 0.05, percentage identity >70% and query covering >50% were selected as reliable hits.

A taxonomic assignment was drawn from the highest scoring BLASTN (or in rare cases BLASTP) hit ranked by query cover and percentage ID. Synteny between predicted coliphages and their respective reference genomes were visualised using EasyFig (Sullivan, Petty & Beatson, 2011). Escherichia bacteriophage coverage per sample was determined using Anvi’o v6.1 (Eren et al., 2015) using default parameters and visualised in R using the Pheatmap package (https://www.rdocumentation.org/packages/pheatmap). Remaining viral genomes were filtered for completeness, retaining those that were circular and encoded a complete terminase gene (as predicted by VirSorter). Taxonomic assignments to the level of family were performed on viral genomes using Demovir (https://github.com/feargalr/Demovir).

Gene catalogue

Complete genes identified by Prodigal v2.6.1 (Hyatt et al., 2010) were clustered at 95% nucleotide identity using CD-HIT-Est v4.6.1 (Fu et al., 2012). Incomplete genes were then mapped to this complete gene list using Bowtie2 v 2.3.4.1 (Langmead & Salzberg, 2012) and any mapping at 95% nucleotide identity were incorporated into the relevant gene clusters. Finally, genes representing the 40 conserved marker genes defined by Mende et al. (2013) were clustered separately and then merged with the existing set of gene clusters. We thus obtained a gene catalogue of >20 million genes, defined as non-redundant at 95% average nucleotide identity (ANI). The final gene catalogue was uploaded to FigShare (https://doi.org/10.6084/m9.figshare.13116809.v4)

Abundance estimates of contigs and genes

Prodigal (Hyatt et al., 2010) was applied in metagenome-mode to all contigs from the MegaHIT assemblies. Unfiltered reads from each sample were mapped against their respective assembly to provide an estimate of contig and gene abundance using Bowtie2 (Langmead & Salzberg, 2012) with the options ‘--no-unal--end-to-end -score-min L, −0.6,−0.6’. Samtools 1.3.1 was used to sort and index all resulting Bam files (Li et al., 2009). Only reads with mapping quality >20, >95% nucleotide identity and >75% overall alignment length were retained. BEDTools v2.21.0 (Quinlan, 2014) was used to create depth profiles from the Bam files. These depth profiles were then translated with rdCover (https://github.com/hildebra/rdCover) into average coverage (in a 50 bp window) per contig or per gene predicted from each contig. Bam files were translated to abundances using the ‘jgi_summarize_bam_contig_depths’ script from the MetaBAT 2 package (Kang et al., 2019).

Gene abundances were linked to their respective gene clusters and originating samples. Redundant genes representing the same orthologue were removed.

Binning

We identified metagenomic species (MGSs) using the combinatorial approach described by Hildebrand et al. (2019), incorporating single-assembly binning in the creation of metagenome-assembled genomes (MAGs), gene catalogue binning in the creation of canopy clusters (Nielsen et al., 2014) and hierarchical clustering of candidate genes using the R function hclust, method = complete. To start with, we used MetaBAT 2 v2.15 (Kang et al., 2019) to bin contigs ≥400 bp. These were quality filtered using CheckM v1.0.11 (Parks et al., 2015) to obtain 5,595 bins at >80% completeness and <5% contamination.

Species-level clusters were formed using a combination of two distinct approaches. One approach removed redundancy between samples by pre-clustering bins if ≥30% of their genes overlapped with a higher-quality bin to create a set of pre-MGS bins. Lower-quality bins (>60% completeness and <10% contamination) were also included in the analysis but were not used to form new species clusters. To recover prokaryotic species usually obscured using single-sample assemblies and conventional binning techniques, we refined all species bins into ‘hcl-clusters’ using gene correlations and hierarchical clustering, as described by Hildebrand et al. (2019). We chose genes occurring in ≥10% of all associated MAGs as representatives for each pre-MGS bin and used these to fish for additional co-occurring genes from the gene catalogue, using a threshold of >0.75 Pearson correlation and >0.85 spearman rho to identify gene co-occurrences within this core gene set. We then merged MetaBAT 2 bins, canopy bins and co-occurring genes into our species bins. We used the presence of 40 known single-copy marker genes, without duplicates, as a quality criterion in selection of sub-clusters, before extracting the final set of MGS gene representatives using MATAFILER (https://github.com/hildebra/MATAFILER). The final collection of MGS bins (canopy clusters + hcl-clusters) was re-assessed for contamination and completeness using CheckM (Parks et al., 2015), so that we could be confident that each bin represents a single species.

A second approach de-replicated all MAGs at 95% ANI (species-level) and 99% ANI (strain-level) using dRep Version 2.0 (Olm et al., 2017) and only species not identified in approach one were added to the resulting non-redundant species catalogue. The minimum aligned fraction used during ANI genome alignment was 60%. A single representative MAG for each novel species cluster was uploaded to NCBI SRA under BioProject PRJNA543206 and all MAGs generated were uploaded to FigShare (https://doi.org/10.6084/m9.figshare.13116809.v4.). CompareM Version0.1.1 (https://github.com/dparks1134/CompareM) was used to calculate average amino acid identity (AAI) when identifying novel genera, using a cut-off of 60% for the percentage identity and 70% for the minimum alignment length used to delineate genus boundaries.

Taxonomy of metagenomic species

We used the Genome Taxonomy Database Toolkit (GTDB-Tk Release 95) to perform taxonomic assignments on strain-level dereplicated MAGs (Chaumeil et al., 2019). In addition, genes from each MGS were analysed through GTDB-Tk (Release 95), proGenomes resource (Mende et al., 2017) and underwent k-mer-based taxonomic profiling using Kraken 2. In assigning taxonomy, we allowed GTDB assignments to take precedence—only when no GTDB taxonomy was available would we adopt taxonomies assigned by ProGenomes and Kraken 2 and, then, only where genus and family assignments from these sources matched. When exploiting the taxonomy assigned according genes from metagenomic species, we applied a least-common-ancestor approach to unplaced taxa at higher taxonomic levels. Species distribution analyses were conducted using the Vegan package in R (R-Core-Team, 2018), before visualisation using ggplot2 (Wickham, 2016) and Pheatmap R packages (https://www.rdocumentation.org/packages/pheatmap). Pan-genome analysis was conducted using Roary v3.11.2 and visualised using the roary2svg.pl script (Page et al., 2015). Comparison of our derived metagenomes with those of Glendinning et al. (2020) was performed at 95% ANI using dRep and visualised using web-tool BioVenn (Hulsen, De Vlieg & Alkema, 2008).

Bacterial culture

To estimate species richness and diversity, the Phyloseq package of R (R-Core-Team, 2018) was applied to the output from Bracken (Lu et al., 2017) on all of our chicken faecal metagenomic datasets. The six faecal samples that showed highest species richness and taxonomic diversity were selected for culture-based studies. Frozen faecal samples were thawed, vortexed and two 0.5 g aliquots (once processed aerobically, the other anaerobically) from each sample were suspended in 5 ml PBS. Each aliquot was vortexed until homogenised, before performing serial dilutions in duplicate down to 1 × 10−5. Processing of samples for aerobic and anaerobic culture was identical, except that, for anaerobic culture, all culture media, diluent and consumables were pre-reduced to anaerobic conditions for at least 24 h before faecal samples were processed in a Whitley A95TG workstation.

For dilutions 10−3–105, 200 µl was plated directly on to a set of three agar plates for each culture medium (Brain Heart Infusion, Colombia Blood Agar, Yeast extract, casitone and fatty acid) with or without vancomycin supplementation at a concentration of 6 µg/ml (Table S1). Cultures were incubated at 37 °C for 72 h in their respective conditions before assessment of colony growth. Well-isolated colonies were picked according to colonial morphotype distinctive in colour, shape and size, before being re-streaked on to the growth medium from which they were sourced to confirm purity. Individual colonies were subsequently used to inoculate 2 ml of broth based on the source culture medium, incubated at 37 °C for a further 24 h before bacterial DNA extraction. All isolates were archived at −80 °C in glycerol at 20% concentration.

Genome sequencing and analysis

Genomic DNA was extracted using a DNeasy UltraClean DNA isolation kit according to the manufacturer’s instructions (Qiagen, Hilden, Germany). DNA was quantified using a Qubit® fluorometer (Invitrogen, Carlsbad, CA, USA) high-sensitivity assay, before dilution to the required concentration in RNase-free water and purification on AMPure XP beads (Beckman Coulter, Brea, CA, USA). Sequencing libraries were prepared from 0.5 ng/µl of RNA free genomic DNA. A total of 282 isolates were included for genomic sequencing using the Nextera-XT DNA sample preparation kit (Illumina, San Diego, CA, USA) and whole-genome sequencing performed using the Illumina NextSeq sequencing platform, generating paired-end reads (2 × 150 bp).

Paired-end reads were quality-assessed and trimmed using FastQC and Trimmomatic as described above. Trimmed reads were assembled into contigs using SPAdes version 3.13.1 (Bankevich et al., 2012). Contigs shorter than 500 bp were discarded from analysis. Genome contamination and completeness was assessed using CheckM version 1.0.13. To confirm assembly quality, only genomes conforming to all the following criteria were included in further analysis: (i) contig N50 of >20 kbp (ii) 90% of assembled bases at >5× read coverage (iii) completeness of >95% (iv) contamination of <5% (v) complete 16S rRNA gene sequence.

Genome sequence taxonomic assignment

Barrnap Version 0.9 (https://github.com/tseemann/barrnap) was applied to all genomes that passed the quality filters to extract full-length 16S rRNA gene sequences. These were then compared to NCBI 16S rRNA gene sequences from RefSeq genomes using the NCBI’s web-based BLASTN facility (Altschul et al., 1990). 16S rRNA gene sequences that showed an identity of <98.7% to known sequences were assigned to novel species, using the conservative approach in proposed minimal standards (Chun et al., 2018). We used ReferenceSeeker Version 1.6.2 (Schwengers et al., 2019) to determine ANI and conserved DNA values compared to RefSeq bacterial genomes (Completed March 2020) (O’Leary et al., 2016). Genomes that showed ANI ≤95% and conserved DNA ≤69% to the closest relative were designated novel species. The Genome Taxonomy Database Toolkit (GTDB-Tk Release 95) was used to perform taxonomic assignments on isolate genomes (Chaumeil et al., 2019). Genomes were clustered at 95% and 99% ANI before selection of a single representative isolate per species using dRep (Olm et al., 2017). Where a genome previously designated as novel clustered with a genome of assigned taxonomy, this taxonomy was then applied to the previously designated ‘novel’ genome. Final taxonomic assignments were based on genome-based ANI values derived from RefSeq and GTDB—with GTDB assignments taking precedence. A single representative genome for each novel or renamed species cluster was uploaded to NCBI SRA under BioProject PRJNA543206 and all genomes alongside respective 16S rRNA gene sequences generated were uploaded to FigShare (https://doi.org/10.6084/m9.figshare.13234556).

Phylogenetic analysis

For phylogenetic analysis of all MGS and genome sequenced isolates we used PhyloPhlAn v3.0.58 (Asnicar et al., 2020) with the ‘diversity high’ and a proteome input predicted from all genome sequences using Prodigal v2.6.1 (Hyatt et al., 2010). Diamond v0.9.34 (Buchfink, Xie & Huson, 2015) was used to perform a search against 400 universal PhyloPhlAn markers. MAFFT v.7.271 (Katoh et al., 2002) was used to perform multiple sequence alignment before refinement with trimAl v.1.4 (Capella-Gutiérrez, Silla-Martínez & Gabaldón, 2009) and reconstruction into trees using FastTree v2.1 and RAxML v. 8.2.12 (Price, Dehal & Arkin, 2010; Stamatakis, 2014). All trees were visualised and annotated manually using the online iTOLv5.7 platform (Letunic & Bork, 2016). Trees were scrutinised to confirm that species and genera were monophyletic. Phylogeny for all cultured genomes unassigned at species level was confirmed as previously described against all available reference proteomes of that respective genus downloaded from NCBI.

To investigate the phylogenetic placement of cultured isolates designated as Escherichia marmotae and Escherichia sp001660175 by GTDB, we constructed a core genome phylogenetic tree. The genomes from cultured isolates were compared to genomes representing the full diversity of the genus Escherichia. Three Salmonella genomes were included as an outgroup. The genome sequences were aligned using Mugsy (Angiuoli & Salzberg, 2011), and alignment blocks conserved across all genomes were concatenated to produce a core genome alignment. A phylogenetic tree was constructed by maximum likelihood with 100 rapid bootstrap replicates, using the general time reversible model of nucleotide substitution with gamma correction for rate heterogeneity, as implemented in RAxML version 8.2.12 (Stamatakis, 2014).

Results

Reference-based profiling documents novel diversity

We collected faecal samples from 90 chickens reared in the UK belonging to two breeds: Lohman Browns (n = 60) and Silkies (n = 30). Short-read sequencing of 50 of these faecal samples generated a metagenomic dataset in excess of a billion paired-end reads or three hundred billion base pairs (Table S2).

We initially analysed the faecal samples using the k-mer-based programme Kraken 2, followed by refined phylogenetic analysis using the allied programme Bracken (Lu et al., 2017) (Table S3). Unsurprisingly, these programmes assigned sequence reads from the faecal samples to all three domains of life, as well as to viruses (Table S4), although relative abundance assignments show that bacteria predominate in this environment. Sequences were assigned to a wide range of bacterial phyla, including the three expected as predominant in the vertebrate gut (Bacteroidetes, Firmicutes, Proteobacteria), but also including over twenty additional phyla. Searches of the PubMed database with each phylum name and the term ‘chicken’ reveal that round half of these have been previously documented in the chicken gut. However, at least a dozen appear to be novel in this setting, including the Aquificae, Balneolaeota, Calditrichaeota, Chlorobi, Dictyoglomi, Fibrobacteres, Gemmatimonadetes, Ignavibacteriae, Kiritimatiellaeota, Lentisphaerae, Nitrospirae, and the Thermodesulfobacteria.

When we rank-ordered the species identified by Bracken according to maximum abundance in any one sample, we found, as expected, that species from the family Lactobacillaceae dominated among the top 20 most abundant organisms. However, we found that two species of Escherichia— Escherichia coli and Escherichia marmotae—accounted for ≥5% of reads in nearly half of the samples (22/50) and in two samples, accounted for more than 50%. Such monodominance of the gut microbiome by bacterial species has been described in diseased humans (Hildebrand et al., 2019; Ravi et al., 2019), but is surprising in the context of poultry reported as apparently healthy by their handlers. We also noted a high relative abundance of the recently described chicken pathogen Gallibacterium anatis (Narasinakuppe Krishnegowda et al., 2020) in most birds (with four birds showing >5% reads assigned to this organism), despite their healthy status. Similarly, Fusobacterium mortiferum—an opportunistic pathogen of humans (Almohaya et al., 2020)—accounted for >10% of sequences in 11 birds, corroborating a recent report of high abundance of 16S rRNA gene sequences from this organism obtained from the chicken caecum (Kollarcikova et al., 2019).

Bracken assigned sequences to over a hundred bacteriophage genomes, predominately phages infecting members of the Enterobacteriaceae assigned to the families Myoviridae and Podoviridae. Particularly noteworthy was the high abundance of reads in some samples from two distinct bacteriophages that prey on E. coli: phiEcoM-GJ1—a lytic bacteriophage isolated in Canada from pig sewage (Jamalludeen et al., 2008)—which accounted for 6.5% reads in a single sample and phAPEC8—a lytic bacteriophage with a large 147 kb genome, isolated from a Belgian poultry farm—which accounted for 10% of reads in a single sample and for >1% of reads in three others (Tsonos et al., 2012).

Although these k-mer-based analyses can provide interesting insights into taxonomic diversity within the chicken gut, we quickly realised that they provide an incomplete and misleading picture of this important microbiome for several reasons: (1) they often report the presence of highly implausible organisms—for example, Kraken 2 reported the presence of human pathogens such as Shigella flexneri and Plasmodium falciparum that are simply not credible in this context on clinical grounds; (2) as with studies on 16S rRNA gene sequences, they fail to provide genomic data or insights into the functional diversity or population structure of the microbial species that they identify and; (3) they rely on a reference database and so can only report previously known organisms and can never uncover ‘unknown unknowns’.

The scale of the problem of unknown diversity is clear from the observation that nearly three quarters (73%) of sequence reads from our chicken samples cannot be confidently classified by Kraken 2 to species level and more than half of the reads (52%) cannot be classified at all and are simply designated as ‘Unassigned’. We therefore sought to extend our understanding of this community through two powerful reference-free approaches: assembly-based metagenome analyses and high-throughput culture.

Metagenomic assembly uncovers a wealth of viral diversity

Assembly of metagenomic sequences is a reference-free approach that involves aligning and merging short sequence reads into long contiguous sequences (contigs) contigs.

To confirm the presence of bacteriophages inferred through the reference-based analysis and to identify novel viral genomes, we assembled sequence reads from our fifty chicken faecal samples into contigs. Contigs ≥10 kb were analysed with VirSorter—a programme designed to detect viral signals in microbial sequence data to find novel viruses (Roux et al., 2015).

VirSorter identified 184 of our chicken faecal contigs as Category 1 (‘most confident’) bacteriophage sequences and identified an additional 1,840 contigs as Category 2 (‘likely’) bacteriophage sequences. This was de-replicated to 1,455 genomes using similarity thresholds of 95% ANI over 70% of the genome (Table S5). BLASTN analysis revealed only 10 of these bacteriophage genomes showed high similarity (percentage identity > 70%; query covering > 50%) to known phages at the nucleotide level (Table S6). These included close relatives of the two phages (phiEcoM-GJ1 and phAPEC8) found highly abundant in the Bracken analyses (Fig. 2). Interestingly, more than one genus of coliphage (e.g. Jilinvirus, Phapecoctavirus, or Gamaleyavirus) was often detected in the same sample, along with an abundance of reads from their predicted prey (Escherichia) suggesting interesting dynamics in phage-host and phage-phage interactions (Fig. 3; Table S7).

Figure 2 Genome synteny of recovered phage genomes.

Synteny plots comparing four novel coliphage genomes recovered from chicken faecal metagenomes (in red) to closest reference genomes. The figure was generated using EasyFig.

Figure 3 Coliphage abundance within chicken faecal samples.

Coverage of four coliphages and of putative host bacterial species. Only samples in which at least one genome had ≥1× coverage are shown (n = 29). All coverage values have been Log10 transformed with blue depicting low abundance and red high abundance.

Of the remaining 1,445 unclassified bacteriophage genomes, nearly 600 encoded either an obvious terminase region or were circular and as such were suggested as being near-complete. Classification of these genomes revealed all genomes were predicted to belong to the order Caudovirales of tailed phages, with the majority belonging to the family Siphoviridae (n = 429), but we also found representatives from the Myoviridae (n = 87) and Podoviridae (n = 27), plus some bacteriophages unclassified at family level (n = 28) (Table S8).

Remarkable microbial genome diversity in the chicken gut

Next, we subjected our samples to computational binning—a process of grouping contigs on the basis of sequence composition and depth of coverage into discrete population bins representing metagenome-assembled genomes (MAGs). However, to carry out a definitive survey of bacterial and archaeal diversity in the chicken gut microbiome—in addition to analysing the fifty faecal samples mentioned and before we started the binning—we retrieved all publicly available chicken gut metagenomic datasets, to create an expansive dataset representing >630 samples, drawn from ten studies and twelve countries (Belgium, China, France, Germany, Italy, Malaysia, Netherlands, Poland, Spain, The Gambia, UK, USA) (Figs. S1A and S1B; Table S9).

Sequence assembly and binning on all these samples generated 5,595 MAGs that passed our quality threshold of ≥80% completion and ≤5% contamination (Fig. S1C). Of these 3,131 could be considered high-quality draft genomes, with >90% completion and <5% contamination, as judged by recently published criteria (Table S10) (Bowers et al., 2017). Genome sizes of the MAGs ranged from ~0.4 to 6.8 Mbp, while GC content ranged from 24% to 73%.

Then, we grouped the MAGs into metagenomic species (MGSs). Initially, this involved de-replicating MAGs at the widely accepted 95% average nucleotide identity (ANI) for defining bacterial and archaeal species and 99% ANI for defining bacterial and archaeal strains (Jain et al., 2018; Luo, Rodriguez-R & Konstantinidis, 2014). De-replication of MAGs at 95% ANI resulted in 846 clusters representing bacterial and archaeal species, while de-replication at 99% ANI resulted in 2182 clusters, representing strains. However, to improve recovery of MAGs, MGSs and associated gene sets, we used gene correlations to identify species-representative genes and then applied hierarchical clustering to co-occurring genes across the samples. This allowed us to identify additional genes from the core genome of a species, even when they show divergent nucleotide compositions (such as genes from genomic islands and plasmids) (Hildebrand et al., 2019). Similarly, using canopy clustering (Nielsen et al., 2014), we could identify commonly occurring species of low abundance. Using these approaches, we were able to identify an additional seven MGSs (Table S11). These MGS were prevalent at >1× coverage in 53% of all analysed samples spanning at least 4 different BioProjects.

Analysis of bacterial metagenomic species, primarily using the Genome Taxonomy Database (GTDB) taxonomy (Parks et al., 2020), confirmed and extended the taxonomic novelty uncovered by reference-based community profiling (Fig. 4), recovering species spanning nineteen of the bacterial phyla defined by GTDB (Table S12). These include Cyanobacteria (12 species; 32 strains); Deferribacterota (1 species; 1 strain) Synergistota (2 species; 5 strains) and the Verrucomicrobiota (7 species; 8 strains).

Figure 4 Phylogenetic tree of draft MGS genomes derived from 820 metagenomic samples of the chicken gut and draft genomes of 93 species cultured from chicken faecal samples.

Phylum, generally as assigned by GTDB, is indicated by colour range. Data symbols in the outer layers have been used to describe further characteristics for each draft genomes. Triangles indicate sequence novelty and status of binomial designation within publicly available databases or published research with filled symbols indicating novel species assigned a binomial as part of this research, hollow symbol indicated a known species assigned a binomial as part of this research and no symbol indicated a known species with a well-formed binomial already assigned. Stars are used to indicate isolation source, with filled symbols indicating isolation of species in both culture and metagenomic assembly and hollow symbols indicating isolation in culture alone. Tree branches have been collapsed where duplicate species have been identified by different methodologies. The tree was reconstructed using PhyloPhlAn 3.0.58 against 400 marker genes before reconstruction using FastTree and RAxML of a MAFFT sequence alignment and visualised using the online iTOLv5.7 tool including provision of a heat map according to individual genome length.

Of the 853 de-replicated bacterial metagenomic species, 321 represented previously delineated species catalogued in publicly available databases (Table S13). Following direct comparison, a further 165 metagenomic species had been previously identified by Glendinning et al. (2020), with these sequences not currently available in public archives. However, only 158 of our metagenomic species possess validly published names based on Latin binomials.

We performed a search of PubMed with the species name and ‘chicken’, leaving aside the 33 species named by Glendinning et al. (2020). This suggested that our study provides the first-evidence-in-chickens for the majority (81/125) of these species (Table S14). Examples include: Jeotgalicoccus halophilus, first isolated from the traditional fermented seafood, Jeotgal (Yoon et al., 2003) and present in 197 chicken samples; Aliicoccus persicus, first isolated from a hypersaline lake (Amoozegar et al., 2014) and present in 241 chicken samples; and Bacteroides reticulotermitis, first isolated from the gut of a termite (Sakamoto & Ohkuma, 2013) and present in 39 chicken samples.

We found that 309 of our metagenomic species could be assigned a taxonomy only at the level of genus and so represent novel candidate species. A further 56 species could be assigned a taxonomy only at the level of family and, after AAI clustering at 60%, were assigned to 36 novel candidate genera. One candidate bacterial species could be assigned a taxonomy only at the level of order (Oscillospirales) and so represent a new family.

Three MAGs were assigned to the domain Archaea. One represents the species Methanobrevibacter woesei—which is already known to inhabit the chicken gut (Saengkerdsub et al., 2007)—while the other two represent novel species within the genera Methanocorpusculum and UBA71, which we have renamed Candidatus Methanospyradousia.

Linnaean binomials for hundreds of new candidate species

Linnaeus first proposed the assignment of Latin binomials to provide a universal nomenclature for biological species (Linnaeus, 1759). The International Code of Nomenclature of Prokaryotes (ICNP) sets the rules for naming prokaryotic species (Parker, Tindall & Garrity, 2019), but currently precludes the valid publication of names of uncultivated organisms, represented by MAGs or other sequences. Furthermore, high-throughput generation of MAGs and of sequence-based taxonomies for bacteria, such as the GTDB (Parks et al., 2020) is often assumed to preclude the detailed attention usually given to one-by-one construction of Linnaean binomials. As a result, most uncultured taxa, as well as many taxa defined on sequence-based criteria, have been assigned unstable, confusing and hard to-remember alphanumerical identifiers.

To provide a stable, clear and memorable nomenclature for novel and/or previously unnamed bacterial and archaeal species from the chicken gut, we exploited the provision within the ICNP for naming uncultivated taxa via Candidatus assignments, which, although provisional, provide the scientific community with well-formed Latin binomials (Oren, 2017; Oren et al., 2020). However, this prompted us into an unprecedented effort to create hundreds of new names for the purpose of this single research study—an effort that required us to devise a scalable combinatorial system for the creation of binomials. Here, we made extensive combinatorial use of several dozen Latin and Greek roots pertaining to poultry (avi-, galli-, pulli-, alektryo, ptero, kotto-, ornitho-), intestines (intestini- entero-), faeces (faec-, kakke, merd-, kopro-, excrement-) or microbial life (-monas, -bacterium, -microbium, -coccus, -bacillus, -bium, -cola)—twinned with addition of these roots (singly or in tandem) and/or prefixes (allo, hetero, meta-, para-, crypto-) to existing genus names—to create over 150 Candidatus genus names. For genera with alphanumeric designations in GTDB Release 05-RS95 (Parks et al., 2020) known to occur also in gut microbiomes of other animals, we adopted a similar combinatorial approach, but avoided roots pertaining to poultry and stuck instead with combinations that simply meant ‘gut or faecal microbe’, for example Fimicola, Caccocola. An additional source of diversity stemmed from repetitive use of around forty Candidatus species epithets built from similar roots, which when combined with genus names gave us a total of over 650 distinctive binomials for new Candidatus species (Table 1; Table S15).

Table 1 Protologues for new Candidatus taxa identified from metagenomic analysis of chicken gut samples.

Description of Candidatus Acetatifactor stercoripullorum sp. nov.	
Candidatus Acetatifactor stercoripullorum (ster.co.ri.pul.lo’rum. L. neut. n. stercus dung; L. masc. n. pullus a young chicken; N.L. gen. n. stercoripullorum of the faceces of young chickens)	
A bacterial species identified by metagenomic analyses. This species includes all bacteria with genomes that show ≥95% average nucleotide identity (ANI) to the type genome, which has been assigned the MAG ID CHK195-6426 and which is available via NCBI BioSample SAMN15816622. The GC content of the type genome is 48.46% and the genome length is 3.1 Mbp.	
Description of Candidatus Acinetobacter avistercoris sp. nov.	
Candidatus Acinetobacter avistercoris (a.vi.ster’co.ris. L. fem. n. avis bird; L. neut. n. stercus dung; N.L. gen. n. avistercoris of bird faeces)	
A bacterial species identified by metagenomic analyses. This species includes all bacteria with genomes that show ≥95% average nucleotide identity (ANI) to the type genome, which has been assigned the MAG ID 5402 and which is available via NCBI BioSample SAMN15816735. The GC content of the type genome is 38.29% and the genome length is 3.9 Mbp.	
Description of Candidatus Acutalibacter ornithocaccae sp. nov.	
Candidatus Acutalibacter ornithocaccae (or.ni.tho.cac’cae. Gr. masc. or fem. n. ornis, ornithos bird; Gr. fem. n. kakke faeces; N.L. gen. n. ornithocaccae of bird faeces)	
A bacterial species identified by metagenomic analyses. This species includes all bacteria with genomes that show ≥95% average nucleotide identity (ANI) to the type genome, which has been assigned the MAG ID ChiBcolR8-3208 and which is available via NCBI BioSample SAMN15816822. This is a new name for the alphanumeric GTDB species sp000435395. The GC content of the type genome is 62.02% and the genome length is 2.1 Mbp.	
Description of Candidatus Acutalibacter pullicola sp. nov.	
Candidatus Acutalibacter pullicola (pul.li’co.la. L. masc. n. pullus a young chicken; L. suff. -cola inhabitant of; N.L. n. pullicola an inhabitant of young chickens)	
A bacterial species identified by metagenomic analyses. This species includes all bacteria with genomes that show ≥95% average nucleotide identity (ANI) to the type genome, which has been assigned the MAG ID CHK185-1770 and which is available via NCBI BioSample SAMN15816590. The GC content of the type genome is 58.43% and the genome length is 2.1 Mbp.	
Description of Candidatus Acutalibacter pullistercoris sp. nov.	
Candidatus Acutalibacter pullistercoris (pul.li.ster’co.ris. L. masc. n. pullus a young chicken; L. neut. n. stercus dung; N.L. gen. n. pullistercoris of young chicken faeces)	
A bacterial species identified by metagenomic analyses. This species includes all bacteria with genomes that show ≥95% average nucleotide identity (ANI) to the type genome, which has been assigned the MAG ID 1282 and which is available via NCBI BioSample SAMN15816718. The GC content of the type genome is 63.65% and the genome length is 2.0 Mbp.	
Description of Candidatus Acutalibacter stercoravium sp. nov.	
Candidatus Acutalibacter stercoravium (ster.cor.a’vi.um. L. neut. n. stercus dung; L. fem. n. avis bird; N.L. gen. n. stercoravium of bird faeces)	
A bacterial species identified by metagenomic analyses. This species includes all bacteria with genomes that show ≥95% average nucleotide identity (ANI) to the type genome, which has been assigned the MAG ID ChiBcolR1-495 and which is available via NCBI BioSample SAMN15816868. This is a new name for the alphanumeric GTDB species sp900543555. The GC content of the type genome is 60.31% and the genome length is 2.0 Mbp.	
Description of Candidatus Acutalibacter stercorigallinarum sp. nov.	
Candidatus Acutalibacter stercorigallinarum (ster.co.ri.gal.li.na’rum. L. neut. n. stercus dung; L. fem. n. gallina hen; N.L. gen. n. stercorigallinarum of hen faeces)	
A bacterial species identified by metagenomic analyses. This species includes all bacteria with genomes that show ≥95% average nucleotide identity (ANI) to the type genome, which has been assigned the MAG ID ChiGjej2B2-2649 and which is available via NCBI BioSample SAMN15816629. The GC content of the type genome is 63.77% and the genome length is 2.1 Mbp.	
Description of Candidatus Agathobaculum intestinigallinarum sp. nov.	
Candidatus Agathobaculum intestinigallinarum (in.tes.ti.ni.gal.li.na’rum. L. neut. n. intestinum gut; L. fem. n. gallina hen; N.L. gen. n. intestinigallinarum of the gut of the hens)	
A bacterial species identified by metagenomic analyses. This species includes all bacteria with genomes that show ≥95% average nucleotide identity (ANI) to the type genome, which has been assigned the MAG ID ChiGjej6B6-20540 and which is available via NCBI BioSample SAMN15816816. This is a new name for the alphanumeric GTDB species sp900555465. The GC content of the type genome is 60.79% and the genome length is 2.0 Mbp.	

Description of Candidatus Agathobaculum intestinipullorum sp. nov.	
Candidatus Agathobaculum intestinipullorum (in.tes.ti.ni.pul.lo’rum. L. neut. n. intestinum gut; L. masc. n. pullus a young chicken; N.L. gen. n. intestinipullorum of the gut of young chickens)	
A bacterial species identified by metagenomic analyses. This species includes all bacteria with genomes that show ≥95% average nucleotide identity (ANI) to the type genome, which has been assigned the MAG ID ChiBcec16-9926 and which is available via NCBI BioSample SAMN15816670. The GC content of the type genome is 57.76% and the genome length is 1.9 Mbp.	
Description of Candidatus Agathobaculum merdavium sp. nov.	
Candidatus Agathobaculum merdavium (merd.a’vi.um. L. fem. n. merda faeces; L. fem. n. avis bird; N.L. gen. n. merdavium of bird faeces)	
A bacterial species identified by metagenomic analyses. This species includes all bacteria with genomes that show ≥95% average nucleotide identity (ANI) to the type genome, which has been assigned the MAG ID ChiBcec15-6302 and which is available via NCBI BioSample SAMN15816712. The GC content of the type genome is 57.98% and the genome length is 2.0 Mbp.	
Description of Candidatus Agathobaculum merdigallinarum sp. nov.	
Candidatus Agathobaculum merdigallinarum (mer.di.gal.li.na’rum. L. fem. n. merda faeces; L. fem. n. gallina hen; N.L. gen. n. merdigallinarum of hen faeces)	
A bacterial species identified by metagenomic analyses. This species includes all bacteria with genomes that show ≥95% average nucleotide identity (ANI) to the type genome, which has been assigned the MAG ID ChiSjej1B19-3834 and which is available via NCBI BioSample SAMN15816715. The GC content of the type genome is 57.98% and the genome length is 2.0 Mbp.	
Description of Candidatus Agathobaculum merdipullorum sp. nov.	
Candidatus Agathobaculum merdipullorum (mer.di.pul.lo’rum. L. fem. n. merda faeces; L. masc. n. pullus a young chicken; N.L. gen. n. merdipullorum of the faeces of young chickens)	
A bacterial species identified by metagenomic analyses. This species includes all bacteria with genomes that show ≥95% average nucleotide identity (ANI) to the type genome, which has been assigned the MAG ID CHK149-1869 and which is available via NCBI BioSample SAMN15816722. The GC content of the type genome is 56.28% and the genome length is 1.7 Mbp.	
Description of Candidatus Agathobaculum pullicola sp. nov.	
Candidatus Agathobaculum pullicola (pul.li’co.la. L. masc. n. pullus a young chicken; L. suff. -cola inhabitant of; N.L. n. pullicola an inhabitant of young chickens)	
A bacterial species identified by metagenomic analyses. This species includes all bacteria with genomes that show ≥95% average nucleotide identity (ANI) to the type genome, which has been assigned the MAG ID 2940 and which is available via NCBI BioSample SAMN15816725. The GC content of the type genome is 54.80% and the genome length is 2.0 Mbp.	
Description of Candidatus Agathobaculum pullistercoris sp. nov.	
Candidatus Agathobaculum pullistercoris (pul.li.ster’co.ris. L. masc. n. pullus a young chicken; L. neut. n. stercus dung; N.L. gen. n. pullistercoris of young chicken faeces)	
A bacterial species identified by metagenomic analyses. This species includes all bacteria with genomes that show ≥95% average nucleotide identity (ANI) to the type genome, which has been assigned the MAG ID CHK180-9785 and which is available via NCBI BioSample SAMN15816619. The GC content of the type genome is 58.01% and the genome length is 2.3 Mbp.	
Description of Candidatus Agathobaculum stercoravium sp. nov.	
Candidatus Agathobaculum stercoravium (ster.cor.a’vi.um. L. neut. n. stercus dung; L. fem. n. avis bird; N.L. gen. n. stercoravium of bird faeces)	
A bacterial species identified by metagenomic analyses. This species includes all bacteria with genomes that show ≥95% average nucleotide identity (ANI) to the type genome, which has been assigned the MAG ID ChiW21-6059 and which is available via NCBI BioSample SAMN15816625. The GC content of the type genome is 59.83% and the genome length is 2.3 Mbp.	
Description of Candidatus Agrococcus pullicola sp. nov.	
Candidatus Agrococcus pullicola (pul.li’co.la. L. masc. n. pullus a young chicken; L. suff. -cola inhabitant of; N.L. n. pullicola an inhabitant of young chickens)	
A bacterial species identified by metagenomic analyses. This species includes all bacteria with genomes that show ≥95% average nucleotide identity (ANI) to the type genome, which has been assigned the MAG ID ChiGjej1B1-98 and which is available via NCBI BioSample SAMN15816710. The GC content of the type genome is 63.86% and the genome length is 3.0 Mbp.	

Description of Candidatus Akkermansia intestinavium sp. nov.	
Candidatus Akkermansia intestinavium (in.tes.tin.a’vi.um. L. neut. n. intestinum gut; L. fem. n. avis bird; N.L. gen. n. intestinavium of the gut of birds)	
A bacterial species identified by metagenomic analyses. This species includes all bacteria with genomes that show ≥95% average nucleotide identity (ANI) to the type genome, which has been assigned the MAG ID ChiGjej6B6-8097 and which is available via NCBI BioSample SAMN15816856. This is a new name for the alphanumeric GTDB species sp900548895. The GC content of the type genome is 65.09% and the genome length is 2.2 Mbp.	
Description of Candidatus Akkermansia intestinigallinarum sp. nov.	
Candidatus Akkermansia intestinigallinarum (in.tes.ti.ni.gal.li.na’rum. L. neut. n. intestinum gut; L. fem. n. gallina hen; N.L. gen. n. intestinigallinarum of the gut of the hens)	
A bacterial species identified by metagenomic analyses. This species includes all bacteria with genomes that show ≥95% average nucleotide identity (ANI) to the type genome, which has been assigned the MAG ID 14975 and which is available via NCBI BioSample SAMN15816742. The GC content of the type genome is 63.40% and the genome length is 2.1 Mbp.	
Description of Candidatus Alectryobacillus gen. nov.	
Candidatus Alectryobacillus (A.lec.try.o.ba.cil’lus. Gr. neut. n. alektryon chicken; L. masc. n. bacillus a rod; N.L. masc. n. Alectryobacillus a bacillus found in poultry)	
A bacterial genus identified by metagenomic analyses. The genus includes all bacteria with genomes that show ≥60% average amino acid identity (AAI) to the type genome from the type species Candidatus Alectryobacillus merdavium. This genus has been assigned by GTDB-Tk v1.3.0 working on GTDB Release 05-RS95 (Chaumeil et al., 2019; Parks et al., 2020) to the order RFN20 and to the family CAG-826.	
Description of Candidatus Alectryobacillus merdavium sp. nov.	
Candidatus Alectryobacillus merdavium (merd.a’vi.um. L. fem. n. merda faeces; L. fem. n. avis bird; N.L. gen. n. merdavium of bird faeces)	
A bacterial species identified by metagenomic analyses. This species includes all bacteria with genomes that show ≥95% average nucleotide identity (ANI) to the type genome, which has been assigned the MAG ID 13038 and which is available via NCBI BioSample SAMN15816966. The GC content of the type genome is 27.10% and the genome length is 1.2 Mbp.	
Description of Candidatus Alectryocaccobium gen. nov.	
Candidatus Alectryocaccobium (A.lec.try.o.cac.co’bi.um. Gr. neut. n. alektryon chicken; Gr. fem. n. kakke faeces; Gr. masc. n. bios life; N.L. neut. n. Alectryocaccobium A life form found in chicken faceces)	
A bacterial genus identified by metagenomic analyses. The genus includes all bacteria with genomes that show ≥60% average amino acid identity (AAI) to the type genome from the type species Candidatus Alectryocaccobium stercorigallinarum. This genus has been assigned by GTDB-Tk v1.3.0 working on GTDB Release 05-RS95 (Chaumeil et al., 2019; Parks et al., 2020) to the order Lachnospirales and to the family Lachnospiraceae.	
Description of Candidatus Alectryocaccobium stercorigallinarum sp. nov.	
Candidatus Alectryocaccobium stercorigallinarum (ster.co.ri.gal.li.na’rum. L. neut. n. stercus dung; L. fem. n. gallina hen; N.L. gen. n. stercorigallinarum of hen faeces)	
A bacterial species identified by metagenomic analyses. This species includes all bacteria with genomes that show ≥95% average nucleotide identity (ANI) to the type genome, which has been assigned the MAG ID ChiGjej2B2-785 and which is available via NCBI BioSample SAMN15816998. The GC content of the type genome is 46.32% and the genome length is 1.5 Mbp.	
Description of Candidatus Alectryocaccomicrobium gen. nov.	
Candidatus Alectryocaccomicrobium (A.lec.try.o.cac.co.mi.cro’bi.um. Gr. neut. n. alektryon chicken; Gr. fem. n. kakke faeces; N.L. neut. n. microbium a microbe; N.L. neut. n. Alectryocaccomicrobium A microbe found in chicken faceces)	
A bacterial genus identified by metagenomic analyses. The genus includes all bacteria with genomes that show ≥60% average amino acid identity (AAI) to the type genome from the type species Candidatus Alectryocaccomicrobium excrementavium. This genus has been assigned by GTDB-Tk v1.3.0 working on GTDB Release 05-RS95 (Chaumeil et al., 2019; Parks et al., 2020) to the order Christensenellales and to the family CAG-74.	
Description of Candidatus Alectryocaccomicrobium excrementavium sp. nov.	
Candidatus Alectryocaccomicrobium excrementavium (ex.cre.ment.a’vi.um. L. neut. n. excrementum excrement; L. fem. n. avis bird; N.L. gen. n. excrementavium of bird excrement)	
A bacterial species identified by metagenomic analyses. This species includes all bacteria with genomes that show ≥95% average nucleotide identity (ANI) to the type genome, which has been assigned the MAG ID 13766 and which is available via NCBI BioSample SAMN15816965. The GC content of the type genome is 59.90% and the genome length is 3.0 Mbp.	

Description of Candidatus Alistipes avicola sp. nov.	
Candidatus Alistipes avicola (a.vi’co.la. L. fem. n. avis bird; L. suff. -cola inhabitant of; N.L. n. avicola inhabitant of birds)	
A bacterial species identified by metagenomic analyses. This species includes all bacteria with genomes that show ≥95% average nucleotide identity (ANI) to the type genome, which has been assigned the MAG ID CHK169-11906 and which is available via NCBI BioSample SAMN15816659. The GC content of the type genome is 53.79% and the genome length is 1.6 Mbp.	
Description of Candidatus Alistipes avistercoris sp. nov.	
Candidatus Alistipes avistercoris (a.vi.ster’co.ris. L. fem. n. avis bird; L. neut. n. stercus dung; N.L. gen. n. avistercoris of bird faeces)	
A bacterial species identified by metagenomic analyses. This species includes all bacteria with genomes that show ≥95% average nucleotide identity (ANI) to the type genome, which has been assigned the MAG ID 653 and which is available via NCBI BioSample SAMN15816855. This is a new name for the alphanumeric GTDB species sp000434235. The GC content of the type genome is 62.33% and the genome length is 2.4 Mbp.	
Description of Candidatus Alistipes cottocaccae sp. nov.	
Candidatus Alistipes cottocaccae (cot.to.cac’cae. Gr. masc. n. kottos chicken Gr. fem. n. kakke faeces; N.L. gen. n. cottocaccae of chicken faeces)	
A bacterial species identified by metagenomic analyses. This species includes all bacteria with genomes that show ≥95% average nucleotide identity (ANI) to the type genome, which has been assigned the MAG ID ChiBcec16-1783 and which is available via NCBI BioSample SAMN15816853. This is a new name for the alphanumeric GTDB species sp002161445. The GC content of the type genome is 60.94% and the genome length is 2.4 Mbp.	
Description of Candidatus Alistipes excrementavium sp. nov.	
Candidatus Alistipes excrementavium (ex.cre.ment.a’vi.um. L. neut. n. excrementum excrement; L. fem. n. avis bird; N.L. gen. n. excrementavium of bird excrement)	
A bacterial species identified by metagenomic analyses. This species includes all bacteria with genomes that show ≥95% average nucleotide identity (ANI) to the type genome, which has been assigned the MAG ID CHK15-232 and which is available via NCBI BioSample SAMN15816809. This is a new name for the alphanumeric GTDB species sp900021155. The GC content of the type genome is 61.18% and the genome length is 2.2 Mbp.	
Description of Candidatus Alistipes excrementigallinarum sp. nov.	
Candidatus Alistipes excrementigallinarum (ex.cre.men.ti.gal.li.na’rum. L. neut. n. excrementum excrement; L. fem. n. gallina hen; N.L. gen. n. excrementigallinarum of hen excrement)	
A bacterial species identified by metagenomic analyses. This species includes all bacteria with genomes that show ≥95% average nucleotide identity (ANI) to the type genome, which has been assigned the MAG ID CHK106-249 and which is available via NCBI BioSample SAMN15816875. The GC content of the type genome is 63.33% and the genome length is 2.3 Mbp.	
Description of Candidatus Alistipes excrementipullorum sp. nov.	
Candidatus Alistipes excrementipullorum (ex.cre.men.ti.pul.lo’rum. L. neut. n. excrementum excrement; L. masc. n. pullus a young chicken; N.L. gen. n. excrementipullorum of young chicken excrement)	
A bacterial species identified by metagenomic analyses. This species includes all bacteria with genomes that show ≥95% average nucleotide identity (ANI) to the type genome, which has been assigned the MAG ID ChiHjej8B7-9065 and which is available via NCBI BioSample SAMN15816799. This is a new name for the alphanumeric GTDB species. The GC content of the type genome is 56.25% and the genome length is 1.7 Mbp.	
Description of Candidatus Alistipes faecavium sp. nov.	
Candidatus Alistipes faecavium (faec.a’vi.um. L. fem. n. faex, faecis excrement; L. fem. n. avis bird; N.L. gen. n. faecavium of bird faeces)	
A bacterial species identified by metagenomic analyses. This species includes all bacteria with genomes that show ≥95% average nucleotide identity (ANI) to the type genome, which has been assigned the MAG ID ChiGjej2B2-19477 and which is available via NCBI BioSample SAMN15816800. The GC content of the type genome is 62.24% and the genome length is 2.3 Mbp.	
Description of Candidatus Alistipes faecigallinarum sp. nov.	
Candidatus Alistipes faecigallinarum (fae.ci.gal.li.na’rum. L. fem. n. faex, faecis excrement; L. fem. n. gallina hen; N.L. gen. n. faecigallinarum of chicken faeces)	
A bacterial species identified by metagenomic analyses. This species includes all bacteria with genomes that show ≥95% average nucleotide identity (ANI) to the type genome, which has been assigned the MAG ID 6451 and which is available via NCBI BioSample SAMN15816915. The GC content of the type genome is 61.37% and the genome length is 2.2 Mbp.	
Description of Candidatus Alistipes intestinigallinarum sp. nov.	
Candidatus Alistipes intestinigallinarum (in.tes.ti.ni.gal.li.na’rum. L. neut. n. intestinum gut; L. fem. n. gallina hen; N.L. gen. n. intestinigallinarum of the gut of the hens)	
A bacterial species identified by metagenomic analyses. This species includes all bacteria with genomes that show ≥95% average nucleotide identity (ANI) to the type genome, which has been assigned the MAG ID 5134 and which is available via NCBI BioSample SAMN15816708. The GC content of the type genome is 59.58% and the genome length is 2.7 Mbp.	

Description of Candidatus Alistipes intestinipullorum sp. nov.	
Candidatus Alistipes intestinipullorum (in.tes.ti.ni.pul.lo’rum. L. neut. n. intestinum gut; L. masc. n. pullus a young chicken; N.L. gen. n. intestinipullorum of the gut of young chickens)	
A bacterial species identified by metagenomic analyses. This species includes all bacteria with genomes that show ≥95% average nucleotide identity (ANI) to the type genome, which has been assigned the MAG ID ChiGjej2B2-5998 and which is available via NCBI BioSample SAMN15816759. The GC content of the type genome is 59.58% and the genome length is 2.3 Mbp.	
Description of Candidatus Alistipes merdavium sp. nov.	
Candidatus Alistipes merdavium (merd.a’vi.um. L. fem. n. merda faeces; L. fem. n. avis bird; N.L. gen. n. merdavium of bird faeces)	
A bacterial species identified by metagenomic analyses. This species includes all bacteria with genomes that show ≥95% average nucleotide identity (ANI) to the type genome, which has been assigned the MAG ID ChiBcolR5-1230 and which is available via NCBI BioSample SAMN15816813. This is a new name for the alphanumeric GTDB species sp900544265. The GC content of the type genome is 63.44% and the genome length is 2.2 Mbp.	
Description of Candidatus Alistipes merdigallinarum sp. nov.	
Candidatus Alistipes merdigallinarum (mer.di.gal.li.na’rum. L. fem. n. merda faeces; L. fem. n. gallina hen; N.L. gen. n. merdigallinarum of hen faeces)	
A bacterial species identified by metagenomic analyses. This species includes all bacteria with genomes that show ≥95% average nucleotide identity (ANI) to the type genome, which has been assigned the MAG ID 2432 and which is available via NCBI BioSample SAMN15816893. Although GTDB has assigned this species to the genus it calls Alistipes_A, this genus designation cannot be incorporated into a well-formed binomial, so in naming this species, we have used the current validly published name for the genus. The GC content of the type genome is 49.96% and the genome length is 2.2 Mbp.	
Description of Candidatus Alistipes merdipullorum sp. nov.	
Candidatus Alistipes merdipullorum (mer.di.pul.lo’rum. L. fem. n. merda faeces; L. masc. n. pullus a young chicken; N.L. gen. n. merdipullorum of the faeces of young chickens)	
A bacterial species identified by metagenomic analyses. This species includes all bacteria with genomes that show ≥95% average nucleotide identity (ANI) to the type genome, which has been assigned the MAG ID ChiHjej9B8-3741 and which is available via NCBI BioSample SAMN15816807. This is a new name for the alphanumeric GTDB species sp900546065. The GC content of the type genome is 57.66% and the genome length is 2.3 Mbp.	
Description of Candidatus Alistipes pullicola sp. nov.	
Candidatus Alistipes pullicola (pul.li’co.la. L. masc. n. pullus a young chicken; L. suff. -cola inhabitant of; N.L. n. pullicola an inhabitant of young chickens)	
A bacterial species identified by metagenomic analyses. This species includes all bacteria with genomes that show ≥95% average nucleotide identity (ANI) to the type genome, which has been assigned the MAG ID ChiHjej10B9-11434 and which is available via NCBI BioSample SAMN15816929. This is a new name for the alphanumeric GTDB species sp900546005. Although GTDB has assigned this species to the genus it calls Alistipes_A, this genus designation cannot be incorporated into a well-formed binomial, so in naming this species, we have used the current validly published name for the genus. The GC content of the type genome is 52.02% and the genome length is 1.9 Mbp.	
Description of Candidatus Alistipes pullistercoris sp. nov.	
Candidatus Alistipes pullistercoris (pul.li.ster’co.ris. L. masc. n. pullus a young chicken; L. neut. n. stercus dung; N.L. gen. n. pullistercoris of young chicken faeces)	
A bacterial species identified by metagenomic analyses. This species includes all bacteria with genomes that show ≥95% average nucleotide identity (ANI) to the type genome, which has been assigned the MAG ID 3244 and which is available via NCBI BioSample SAMN15816930. This is a new name for the alphanumeric GTDB species sp900240235. Although GTDB has assigned this species to the genus it calls Alistipes_A, this genus designation cannot be incorporated into a well-formed binomial, so in naming this species, we have used the current validly published name for the genus. The GC content of the type genome is 56.88% and the genome length is 2.0 Mbp.	
Description of Candidatus Alistipes stercoravium sp. nov.	
Candidatus Alistipes stercoravium (ster.cor.a’vi.um. L. neut. n. stercus dung; L. fem. n. avis bird; N.L. gen. n. stercoravium of bird faeces)	
A bacterial species identified by metagenomic analyses. This species includes all bacteria with genomes that show ≥95% average nucleotide identity (ANI) to the type genome, which has been assigned the MAG ID ChiHjej8B7-9257 and which is available via NCBI BioSample SAMN15816640. The GC content of the type genome is 61.39% and the genome length is 2.0 Mbp.	
Description of Candidatus Alistipes stercorigallinarum sp. nov.	
Candidatus Alistipes stercorigallinarum (ster.co.ri.gal.li.na’rum. L. neut. n. stercus dung; L. fem. n. gallina hen; N.L. gen. n. stercorigallinarum of hen faeces)	
A bacterial species identified by metagenomic analyses. This species includes all bacteria with genomes that show ≥95% average nucleotide identity (ANI) to the type genome, which has been assigned the MAG ID ChiHcolR4-13572 and which is available via NCBI BioSample SAMN15816817. This is a new name for the alphanumeric GTDB species sp900542505. The GC content of the type genome is 62.42% and the genome length is 2.2 Mbp.	

Description of Candidatus Alistipes stercoripullorum sp. nov.	
Candidatus Alistipes stercoripullorum (ster.co.ri.pul.lo’rum. L. neut. n. stercus dung; L. masc. n. pullus a young chicken; N.L. gen. n. stercoripullorum of the faceces of young chickens)	
A bacterial species identified by metagenomic analyses. This species includes all bacteria with genomes that show ≥95% average nucleotide identity (ANI) to the type genome, which has been assigned the MAG ID ChiBcec8-6454 and which is available via NCBI BioSample SAMN15816818. This is a new name for the alphanumeric GTDB species sp006542685. The GC content of the type genome is 62.87% and the genome length is 2.4 Mbp.	
Description of Candidatus Anaerobiospirillum merdipullorum sp. nov.	
Candidatus Anaerobiospirillum merdipullorum (mer.di.pul.lo’rum. L. fem. n. merda faeces; L. masc. n. pullus a young chicken; N.L. gen. n. merdipullorum of the faeces of young chickens)	
A bacterial species identified by metagenomic analyses. This species includes all bacteria with genomes that show ≥95% average nucleotide identity (ANI) to the type genome, which has been assigned the MAG ID 687 and which is available via NCBI BioSample SAMN15816911. Although GTDB has assigned this species to the genus it calls Anaerobiospirillum_A, this genus designation cannot be incorporated into a well-formed binomial, so in naming this species, we have used the current validly published name for the genus. The GC content of the type genome is 49.84% and the genome length is 2.0 Mbp.	
Description of Candidatus Anaerobiospirillum pullicola sp. nov.	
Candidatus Anaerobiospirillum pullicola (pul.li’co.la. L. masc. n. pullus a young chicken; L. suff. -cola inhabitant of; N.L. n. pullicola an inhabitant of young chickens)	
A bacterial species identified by metagenomic analyses. This species includes all bacteria with genomes that show ≥95% average nucleotide identity (ANI) to the type genome, which has been assigned the MAG ID 378 and which is available via NCBI BioSample SAMN15816727. The GC content of the type genome is 52.37% and the genome length is 3.9 Mbp.	
Description of Candidatus Anaerobiospirillum pullistercoris sp. nov.	
Candidatus Anaerobiospirillum pullistercoris (pul.li.ster’co.ris. L. masc. n. pullus a young chicken; L. neut. n. stercus dung; N.L. gen. n. pullistercoris of young chicken faeces)	
A bacterial species identified by metagenomic analyses. This species includes all bacteria with genomes that show ≥95% average nucleotide identity (ANI) to the type genome, which has been assigned the MAG ID USASDec5-558 and which is available via NCBI BioSample SAMN15816730. The GC content of the type genome is 49.01% and the genome length is 3.3 Mbp.	
Description of Candidatus Anaerobiospirillum stercoravium sp. nov.	
Candidatus Anaerobiospirillum stercoravium (ster.cor.a’vi.um. L. neut. n. stercus dung; L. fem. n. avis bird; N.L. gen. n. stercoravium of bird faeces)	
A bacterial species identified by metagenomic analyses. This species includes all bacteria with genomes that show ≥95% average nucleotide identity (ANI) to the type genome, which has been assigned the MAG ID USASDcec2-551 and which is available via NCBI BioSample SAMN15816778. The GC content of the type genome is 56.27% and the genome length is 2.9 Mbp.	
Description of Candidatus Anaerobutyricum avicola sp. nov.	
Candidatus Anaerobutyricum avicola (a.vi’co.la. L. fem. n. avis bird; L. suff. -cola inhabitant of; N.L. n. avicola inhabitant of birds)	
A bacterial species identified by metagenomic analyses. This species includes all bacteria with genomes that show ≥95% average nucleotide identity (ANI) to the type genome, which has been assigned the MAG ID ChiSxjej6B18-9268 and which is available via NCBI BioSample SAMN15816760. The GC content of the type genome is 50.20% and the genome length is 2.5 Mbp.	
Description of Candidatus Anaerobutyricum faecale sp. nov.	
Candidatus Anaerobutyricum faecale (fae.ca’le. L. neut. adj. faecale of faeces)	
A bacterial species identified by metagenomic analyses. This species includes all bacteria with genomes that show ≥95% average nucleotide identity (ANI) to the type genome, which has been assigned the MAG ID CHK182-24705 and which is available via NCBI BioSample SAMN15816814. This is a new name for the alphanumeric GTDB species sp002161065. The GC content of the type genome is 48.07% and the genome length is 2.8 Mbp.	
Description of Candidatus Anaerobutyricum stercoripullorum sp. nov.	
Candidatus Anaerobutyricum stercoripullorum (ster.co.ri.pul.lo’rum. L. neut. n. stercus dung; L. masc. n. pullus a young chicken; N.L. gen. n. stercoripullorum of the faceces of young chickens)	
A bacterial species identified by metagenomic analyses. This species includes all bacteria with genomes that show ≥95% average nucleotide identity (ANI) to the type genome, which has been assigned the MAG ID ChiSxjej3B15-1167 and which is available via NCBI BioSample SAMN15816729. The GC content of the type genome is 52.36% and the genome length is 2.3 Mbp.	
Description of Candidatus Anaerobutyricum stercoris sp. nov.	
Candidatus Anaerobutyricum stercoris (ster’co.ris. L. gen. n. stercoris of dung, excrement)	
A bacterial species identified by metagenomic analyses. This species includes all bacteria with genomes that show ≥95% average nucleotide identity (ANI) to the type genome, which has been assigned the MAG ID CHK179-28034 and which is available via NCBI BioSample SAMN15816848. This is a new name for the alphanumeric GTDB species sp900016875. The GC content of the type genome is 47.36% and the genome length is 3.0 Mbp.	

Description of Candidatus Anaerofilum excrementigallinarum sp. nov.	
Candidatus Anaerofilum excrementigallinarum (ex.cre.men.ti.gal.li.na’rum. L. neut. n. excrementum excrement; L. fem. n. gallina hen; N.L. gen. n. excrementigallinarum of hen excrement)	
A bacterial species identified by metagenomic analyses. This species includes all bacteria with genomes that show ≥95% average nucleotide identity (ANI) to the type genome, which has been assigned the MAG ID 3951 and which is available via NCBI BioSample SAMN15816720. The GC content of the type genome is 61.37% and the genome length is 2.5 Mbp.	
Description of Candidatus Anaerofilum faecale sp. nov.	
Candidatus Anaerofilum faecale (fae.ca’le. L. neut. adj. faecale of faeces)	
A bacterial species identified by metagenomic analyses. This species includes all bacteria with genomes that show ≥95% average nucleotide identity (ANI) to the type genome, which has been assigned the MAG ID ChiGjej6B6-374 and which is available via NCBI BioSample SAMN15816865. This is a new name for the alphanumeric GTDB species sp002160015. The GC content of the type genome is 63.11% and the genome length is 2.3 Mbp.	
Description of Candidatus Anaeromassilibacillus stercoravium sp. nov.	
Candidatus Anaeromassilibacillus stercoravium (ster.cor.a’vi.um. L. neut. n. stercus dung; L. fem. n. avis bird; N.L. gen. n. stercoravium of bird faeces)	
A bacterial species identified by metagenomic analyses. This species includes all bacteria with genomes that show ≥95% average nucleotide identity (ANI) to the type genome, which has been assigned the MAG ID ChiSjej5B23-4625 and which is available via NCBI BioSample SAMN15816824. This is a new name for the alphanumeric GTDB species sp002159845. The GC content of the type genome is 54.17% and the genome length is 2.2 Mbp.	
Description of Candidatus Anaerostipes avicola sp. nov.	
Candidatus Anaerostipes avicola (a.vi’co.la. L. fem. n. avis bird; L. suff. -cola inhabitant of; N.L. n. avicola inhabitant of birds)	
A bacterial species identified by metagenomic analyses. This species includes all bacteria with genomes that show ≥95% average nucleotide identity (ANI) to the type genome, which has been assigned the MAG ID CHK189-27985 and which is available via NCBI BioSample SAMN15816576. The GC content of the type genome is 43.22% and the genome length is 2.5 Mbp.	
Description of Candidatus Anaerostipes avistercoris sp. nov.	
Candidatus Anaerostipes avistercoris (a.vi.ster’co.ris. L. fem. n. avis bird; L. neut. n. stercus dung; N.L. gen. n. avistercoris of bird faeces)	
A bacterial species identified by metagenomic analyses. This species includes all bacteria with genomes that show ≥95% average nucleotide identity (ANI) to the type genome, which has been assigned the MAG ID ChiSjej3B21-8574 and which is available via NCBI BioSample SAMN15816634. The GC content of the type genome is 44.43% and the genome length is 2.6 Mbp.	
Description of Candidatus Anaerostipes excrementavium sp. nov.	
Candidatus Anaerostipes excrementavium (ex.cre.ment.a’vi.um. L. neut. n. excrementum excrement; L. fem. n. avis bird; N.L. gen. n. excrementavium of bird excrement)	
A bacterial species identified by metagenomic analyses. This species includes all bacteria with genomes that show ≥95% average nucleotide identity (ANI) to the type genome, which has been assigned the MAG ID CHK191-13928 and which is available via NCBI BioSample SAMN15816615. The GC content of the type genome is 41.56% and the genome length is 2.7 Mbp.	
Description of Candidatus Anaerotignum merdipullorum sp. nov.	
Candidatus Anaerotignum merdipullorum (mer.di.pul.lo’rum. L. fem. n. merda faeces; L. masc. n. pullus a young chicken; N.L. gen. n. merdipullorum of the faeces of young chickens)	
A bacterial species identified by metagenomic analyses. This species includes all bacteria with genomes that show ≥95% average nucleotide identity (ANI) to the type genome, which has been assigned the MAG ID CHK190-6203 and which is available via NCBI BioSample SAMN15816613. The GC content of the type genome is 44.75% and the genome length is 2.2 Mbp.	
Description of Candidatus Anaerotruncus excrementipullorum sp. nov.	
Candidatus Anaerotruncus excrementipullorum (ex.cre.men.ti.pul.lo’rum. L. neut. n. excrementum excrement; L. masc. n. pullus a young chicken; N.L. gen. n. excrementipullorum of young chicken excrement)	
A bacterial species identified by metagenomic analyses. This species includes all bacteria with genomes that show ≥95% average nucleotide identity (ANI) to the type genome, which has been assigned the MAG ID CHK188-5543 and which is available via NCBI BioSample SAMN15816616. The GC content of the type genome is 64.05% and the genome length is 1.9 Mbp.	
Description of Candidatus Aphodenecus gen. nov.	
Candidatus Aphodenecus (Aph.od.en.e’cus. Gr. fem. n. aphodos dung; Gr. masc. enoikos inhabitant; N.L. masc. n. Aphodenecus a microbe associated with faeces)	
A bacterial genus identified by metagenomic analyses. The genus includes all bacteria with genomes that show ≥60% average amino acid identity (AAI) to the type genome from the type species Candidatus Aphodenecus pullistercoris. This is a name for the alphanumeric GTDB genus Spiro-01. This genus has been assigned by GTDB-Tk v1.3.0 working on GTDB Release 05-RS95 (Chaumeil et al., 2019; Parks et al., 2020) to the order Sphaerochaetales and to the family Sphaerochaetaceae.	

Description of Candidatus Aphodenecus pullistercoris sp. nov.	
Candidatus Aphodenecus pullistercoris (pul.li.ster’co.ris. L. masc. n. pullus a young chicken; L. neut. n. stercus dung; N.L. gen. n. pullistercoris of young chicken faeces)	
A bacterial species identified by metagenomic analyses. This species includes all bacteria with genomes that show ≥95% average nucleotide identity (ANI) to the type genome, which has been assigned the MAG ID 11167 and which is available via NCBI BioSample SAMN15817123. The GC content of the type genome is 59.34% and the genome length is 2.0 Mbp.	
Description of Candidatus Aphodocola gen. nov.	
Candidatus Aphodocola (Aph.o.do’co.la. Gr. fem. n. aphodos dung; L. suff. -cola inhabitant of; N.L. fem. n. Aphodocola a microbe associated with faeces)	
A bacterial genus identified by metagenomic analyses. The genus includes all bacteria with genomes that show ≥60% average amino acid identity (AAI) to the type genome from the type species Candidatus Aphodocola excrementigallinarum. This is a name for the alphanumeric GTDB genus CAG-594. This genus has been assigned by GTDB-Tk v1.3.0 working on GTDB Release 05-RS95 (Chaumeil et al., 2019; Parks et al., 2020) to the order RF39 and to the family CAG-433.	
Description of Candidatus Aphodocola excrementigallinarum sp. nov.	
Candidatus Aphodocola excrementigallinarum (ex.cre.men.ti.gal.li.na’rum. L. neut. n. excrementum excrement; L. fem. n. gallina hen; N.L. gen. n. excrementigallinarum of hen excrement)	
A bacterial species identified by metagenomic analyses. This species includes all bacteria with genomes that show ≥95% average nucleotide identity (ANI) to the type genome, which has been assigned the MAG ID CHK193-30670 and which is available via NCBI BioSample SAMN15817049. The GC content of the type genome is 27.74% and the genome length is 1.2 Mbp.	
Description of Candidatus Aphodomonas gen. nov.	
Candidatus Aphodomonas (Aph.o.do.mo’nas. Gr. fem. n. aphodos dung; L. fem. n. monas a monad; N.L. fem. n. Aphodomonas a microbe associated with faeces)	
A bacterial genus identified by metagenomic analyses. The genus includes all bacteria with genomes that show ≥60% average amino acid identity (AAI) to the type genome from the type species Candidatus Aphodomonas merdavium. This is a name for the alphanumeric GTDB genus SFFS01. This genus has been assigned by GTDB-Tk v1.3.0 working on GTDB Release 05-RS95 (Chaumeil et al., 2019; Parks et al., 2020) to the order Christensenellales and to the family CAG-74.	
Description of Candidatus Aphodomonas merdavium sp. nov.	
Candidatus Aphodomonas merdavium (merd.a’vi.um. L. fem. n. merda faeces; L. fem. n. avis bird; N.L. gen. n. merdavium of bird faeces)	
A bacterial species identified by metagenomic analyses. This species includes all bacteria with genomes that show ≥95% average nucleotide identity (ANI) to the type genome, which has been assigned the MAG ID ChiGjej2B2-35035 and which is available via NCBI BioSample SAMN15817117. The GC content of the type genome is 59.45% and the genome length is 2.1 Mbp.	
Description of Candidatus Aphodomorpha gen. nov.	
Candidatus Aphodomorpha (Aph.o.do.mor’pha. Gr. fem. n. aphodos dung; Gr. fem. n. morphe a form, shape; N.L. fem. n. Aphodomorpha a microbe associated with faeces)	
A bacterial genus identified by metagenomic analyses. The genus includes all bacteria with genomes that show ≥60% average amino acid identity (AAI) to the type genome from the type species Candidatus Aphodomorpha intestinavium. This is a name for the alphanumeric GTDB genus UMGS1241. This genus has been assigned by GTDB-Tk v1.3.0 working on GTDB Release 05-RS95 (Chaumeil et al., 2019; Parks et al., 2020) to the order Christensenellales and to the family CAG-138.	
Description of Candidatus Aphodomorpha intestinavium sp. nov.	
Candidatus Aphodomorpha intestinavium (in.tes.tin.a’vi.um. L. neut. n. intestinum gut; L. fem. n. avis bird; N.L. gen. n. intestinavium of the gut of birds)	
A bacterial species identified by metagenomic analyses. This species includes all bacteria with genomes that show ≥95% average nucleotide identity (ANI) to the type genome, which has been assigned the MAG ID ChiGjej2B2-16831 and which is available via NCBI BioSample SAMN15817204. This is a new name for the alphanumeric GTDB species sp900550525. The GC content of the type genome is 68.13% and the genome length is 1.6 Mbp.	
Description of Candidatus Aphodoplasma gen. nov.	
Candidatus Aphodoplasma (Aph.o.do.plas’ma. Gr. fem. n. aphodos dung; Gr. neut. n. plasma a form; N.L. neut. n. Aphodoplasma a microbe associated with faeces)	
A bacterial genus identified by metagenomic analyses. The genus includes all bacteria with genomes that show ≥60% average amino acid identity (AAI) to the type genome from the type species Candidatus Aphodoplasma excrementigallinarum. This is a name for the alphanumeric GTDB genus UMGS1253. This genus has been assigned by GTDB-Tk v1.3.0 working on GTDB Release 05-RS95 (Chaumeil et al., 2019; Parks et al., 2020) to the order Monoglobales_A and to the family UMGS1253.	

Description of Candidatus Aphodoplasma excrementigallinarum sp. nov.	
Candidatus Aphodoplasma excrementigallinarum (ex.cre.men.ti.gal.li.na’rum. L. neut. n. excrementum excrement; L. fem. n. gallina hen; N.L. gen. n. excrementigallinarum of hen excrement)	
A bacterial species identified by metagenomic analyses. This species includes all bacteria with genomes that show ≥95% average nucleotide identity (ANI) to the type genome, which has been assigned the MAG ID 4920 and which is available via NCBI BioSample SAMN15817155. The GC content of the type genome is 54.59% and the genome length is 1.8 Mbp.	
Description of Candidatus Aphodosoma gen. nov.	
Candidatus Aphodosoma (Aph.o.do.so’ma. Gr. fem. n. aphodos dung; Gr. neut. n. soma a body; N.L. neut. n. Aphodosoma a microbe associated with faeces)	
A bacterial genus identified by metagenomic analyses. The genus includes all bacteria with genomes that show ≥60% average amino acid identity (AAI) to the type genome from the type species Candidatus Aphodosoma intestinipullorum. This is a name for the alphanumeric GTDB genus SFVR01. This genus has been assigned by GTDB-Tk v1.3.0 working on GTDB Release 05-RS95 (Chaumeil et al., 2019; Parks et al., 2020) to the order Bacteroidales and to the family Paludibacteraceae.	
Description of Candidatus Aphodosoma intestinipullorum sp. nov.	
Candidatus Aphodosoma intestinipullorum (in.tes.ti.ni.pul.lo’rum. L. neut. n. intestinum gut; L. masc. n. pullus a young chicken; N.L. gen. n. intestinipullorum of the gut of young chickens)	
A bacterial species identified by metagenomic analyses. This species includes all bacteria with genomes that show ≥95% average nucleotide identity (ANI) to the type genome, which has been assigned the MAG ID 3924 and which is available via NCBI BioSample SAMN15817132. The GC content of the type genome is 52.56% and the genome length is 2.4 Mbp.	
Description of Candidatus Aphodousia gen. nov.	
Candidatus Aphodousia (Aph.od.ou’si.a. Gr. fem. n. aphodos dung; Gr. fem. n. ousia an essence; N.L. fem. n. Aphodousia a microbe associated with faeces)	
A bacterial genus identified by metagenomic analyses. The genus includes all bacteria with genomes that show ≥60% average amino acid identity (AAI) to the type genome from the type species Candidatus Aphodousia faecavium. This is a name for the alphanumeric GTDB genus CAG-521. This genus has been assigned by GTDB-Tk v1.3.0 working on GTDB Release 05-RS95 (Chaumeil et al., 2019; Parks et al., 2020) to the order Burkholderiales and to the family Burkholderiaceae.	
Description of Candidatus Aphodousia faecalis sp. nov.	
Candidatus Aphodousia faecalis (fae.ca’lis. L. fem. adj. faecalis of faeces)	
A bacterial species identified by metagenomic analyses. This species includes all bacteria with genomes that show ≥95% average nucleotide identity (ANI) to the type genome, which has been assigned the MAG ID ChiW13-1064 and which is available via NCBI BioSample SAMN15817170. This is a new name for the alphanumeric GTDB species sp000437635. The GC content of the type genome is 47.35% and the genome length is 1.7 Mbp.	
Description of Candidatus Aphodousia faecavium sp. nov.	
Candidatus Aphodousia faecavium (faec.a’vi.um. L. fem. n. faex, faecis excrement; L. fem. n. avis bird; N.L. gen. n. faecavium of bird faeces)	
A bacterial species identified by metagenomic analyses. This species includes all bacteria with genomes that show ≥95% average nucleotide identity (ANI) to the type genome, which has been assigned the MAG ID 10345 and which is available via NCBI BioSample SAMN15817126. The GC content of the type genome is 48.23% and the genome length is 1.7 Mbp.	
Description of Candidatus Aphodousia faecigallinarum sp. nov.	
Candidatus Aphodousia faecigallinarum (fae.ci.gal.li.na’rum. L. fem. n. faex, faecis excrement; L. fem. n. gallina hen; N.L. gen. n. faecigallinarum of hen faeces)	
A bacterial species identified by metagenomic analyses. This species includes all bacteria with genomes that show ≥95% average nucleotide identity (ANI) to the type genome, which has been assigned the MAG ID 7463 and which is available via NCBI BioSample SAMN15817137. The GC content of the type genome is 48.37% and the genome length is 1.5 Mbp.	
Description of Candidatus Aphodousia faecipullorum sp. nov.	
Candidatus Aphodousia faecipullorum (fae.ci.pul.lo’rum. L. fem. n. faex, faecis excrement; L. masc. n. pullus a young chicken; N.L. gen. n. faecipullorum of young chicken faeces)	
A bacterial species identified by metagenomic analyses. This species includes all bacteria with genomes that show ≥95% average nucleotide identity (ANI) to the type genome, which has been assigned the MAG ID CHK135-12538 and which is available via NCBI BioSample SAMN15817146. The GC content of the type genome is 48.08% and the genome length is 1.8 Mbp.	

Description of Candidatus Aphodousia gallistercoris sp. nov.	
Candidatus Aphodousia gallistercoris (gal.li.ster’co.ris. L. masc. n gallus chicken; L. neut. n. stercus dung; N.L. gen. n. gallistercoris of chicken faeces)	
A bacterial species identified by metagenomic analyses. This species includes all bacteria with genomes that show ≥95% average nucleotide identity (ANI) to the type genome, which has been assigned the MAG ID CHK121-301 and which is available via NCBI BioSample SAMN15817147. The GC content of the type genome is 52.58% and the genome length is 1.8 Mbp.	
Description of Candidatus Aphodovivens gen. nov.	
Candidatus Aphodovivens (Aph.o.do.vi’vens. Gr. fem. n. aphodos dung; N.L. pres. part. vivens living; N.L. fem. n. Aphodovivens a microbe associated with faeces)	
A bacterial genus identified by metagenomic analyses. The genus includes all bacteria with genomes that show ≥60% average amino acid identity (AAI) to the type genome from the type species Candidatus Aphodovivens avicola. This is a name for the alphanumeric GTDB genus UMGS1293. This genus has been assigned by GTDB-Tk v1.3.0 working on GTDB Release 05-RS95 (Chaumeil et al., 2019; Parks et al., 2020) to the order Coriobacteriales and to the family Eggerthellaceae.	
Description of Candidatus Aphodovivens avicola sp. nov.	
Candidatus Aphodovivens avicola (a.vi’co.la. L. fem. n. avis bird; L. suff. -cola inhabitant of; N.L. n. avicola inhabitant of birds)	
A bacterial species identified by metagenomic analyses. This species includes all bacteria with genomes that show ≥95% average nucleotide identity (ANI) to the type genome, which has been assigned the MAG ID ChiGjej6B6-21069 and which is available via NCBI BioSample SAMN15817067. The GC content of the type genome is 65.54% and the genome length is 2.2 Mbp.	
Description of Candidatus Aphodovivens avistercoris sp. nov.	
Candidatus Aphodovivens avistercoris (a.vi.ster’co.ris. L. fem. n. avis bird; L. neut. n. stercus dung; N.L. gen. n. avistercoris of bird faeces)	
A bacterial species identified by metagenomic analyses. This species includes all bacteria with genomes that show ≥95% average nucleotide identity (ANI) to the type genome, which has been assigned the MAG ID ChiGjej5B5-3278 and which is available via NCBI BioSample SAMN15817093. The GC content of the type genome is 66.86% and the genome length is 2.4 Mbp.	
Description of Candidatus Aphodovivens excrementavium sp. nov.	
Candidatus Aphodovivens excrementavium (ex.cre.ment.a’vi.um. L. neut. n. excrementum excrement; L. fem. n. avis bird; N.L. gen. n. excrementavium of bird excrement)	
A bacterial species identified by metagenomic analyses. This species includes all bacteria with genomes that show ≥95% average nucleotide identity (ANI) to the type genome, which has been assigned the MAG ID ChiGjej2B2-30709 and which is available via NCBI BioSample SAMN15817109. The GC content of the type genome is 58.74% and the genome length is 2.1 Mbp.	
Description of Candidatus Aquabacterium excrementipullorum sp. nov.	
Candidatus Aquabacterium excrementipullorum (ex.cre.men.ti.pul.lo’rum. L. neut. n. excrementum excrement; L. masc. n. pullus a young chicken; N.L. gen. n. excrementipullorum of young chicken excrement)	
A bacterial species identified by metagenomic analyses. This species includes all bacteria with genomes that show ≥95% average nucleotide identity (ANI) to the type genome, which has been assigned the MAG ID ChiHile3-4534 and which is available via NCBI BioSample SAMN15816783. The GC content of the type genome is 67.11% and the genome length is 4.7 Mbp.	
Description of Candidatus Atopostipes pullistercoris sp. nov.	
Candidatus Atopostipes pullistercoris (pul.li.ster’co.ris. L. masc. n. pullus a young chicken; L. neut. n. stercus dung; N.L. gen. n. pullistercoris of young chicken faeces)	
A bacterial species identified by metagenomic analyses. This species includes all bacteria with genomes that show ≥95% average nucleotide identity (ANI) to the type genome, which has been assigned the MAG ID CHK169-4300 and which is available via NCBI BioSample SAMN15816688. The GC content of the type genome is 34.84% and the genome length is 1.9 Mbp.	
Description of Candidatus Avacholeplasma gen. nov.	
Candidatus Avacholeplasma (Av.a.cho.le.plas’ma. L. fem. n. avis bird; N.L. neut. n. Acholeplasma a genus name; N.L. neut n. Avacholeplasma a genus related to the genus Acholeplasma but distinct from it and found in poultry)	
A bacterial genus identified by metagenomic analyses. The genus includes all bacteria with genomes that show ≥60% average amino acid identity (AAI) to the type genome from the type species Candidatus Avacholeplasma faecigallinarum. This genus has been assigned by GTDB-Tk v1.3.0 working on GTDB Release 05-RS95 (Chaumeil et al., 2019; Parks et al., 2020) to the order Acholeplasmatales and to the family Anaeroplasmataceae.	

Description of Candidatus Avacholeplasma faecigallinarum sp. nov.	
Candidatus Avacholeplasma faecigallinarum (fae.ci.gal.li.na’rum. L. fem. n. faex, faecis excrement; L. fem. n. gallina hen; N.L. gen. n. faecigallinarum of hen faeces)	
A bacterial species identified by metagenomic analyses. This species includes all bacteria with genomes that show ≥95% average nucleotide identity (ANI) to the type genome, which has been assigned the MAG ID 3263 and which is available via NCBI BioSample SAMN15816972. The GC content of the type genome is 29.88% and the genome length is 1.3 Mbp.	
Description of Candidatus Avacidaminococcus gen. nov.	
Candidatus Avacidaminococcus (Av.a.cid.a.mi.no.coc’cus. L. fem. n. avis bird; N.L. masc. n. Acidaminococcus a genus name; N.L. masc. n. Avacidaminococcus a genus related to the genus Acidaminococcus but distinct from it and found in poultry)	
A bacterial genus identified by metagenomic analyses. The genus includes all bacteria with genomes that show ≥60% average amino acid identity (AAI) to the type genome from the type species Candidatus Avacidaminococcus intestinavium. This genus has been assigned by GTDB-Tk v1.3.0 working on GTDB Release 05-RS95 (Chaumeil et al., 2019; Parks et al., 2020) to the order Acidaminococcales and to the family Acidaminococcaceae.	
Description of Candidatus Avacidaminococcus intestinavium sp. nov.	
Candidatus Avacidaminococcus intestinavium (in.tes.tin.a’vi.um. L. neut. n. intestinum gut; L. fem. n. avis bird; N.L. gen. n. intestinavium of the gut of birds)	
A bacterial species identified by metagenomic analyses. This species includes all bacteria with genomes that show ≥95% average nucleotide identity (ANI) to the type genome, which has been assigned the MAG ID CHK160-1198 and which is available via NCBI BioSample SAMN15816987. The GC content of the type genome is 37.45% and the genome length is 1.6 Mbp.	
Description of Candidatus Avamphibacillus gen. nov.	
Candidatus Avamphibacillus (Av.am.phi.ba.cil’lus. L. fem. n. avis bird; N.L. masc. n. Amphibacillus a genus name; N.L. masc. n. Avamphibacillus a genus related to the genus Amphibacillus but distinct from it and found in poultry)	
A bacterial genus identified by metagenomic analyses. The genus includes all bacteria with genomes that show ≥60% average amino acid identity (AAI) to the type genome from the type species Candidatus Avamphibacillus intestinigallinarum. This genus has been assigned by GTDB-Tk v1.3.0 working on GTDB Release 05-RS95 (Chaumeil et al., 2019; Parks et al., 2020) to the order Bacillales and to the family Amphibacillaceae.	
Description of Candidatus Avamphibacillus intestinigallinarum sp. nov.	
Candidatus Avamphibacillus intestinigallinarum (in.tes.ti.ni.gal.li.na’rum. L. neut. n. intestinum gut; L. fem. n. gallina hen; N.L. gen. n. intestinigallinarum of the gut of the hens)	
A bacterial species identified by metagenomic analyses. This species includes all bacteria with genomes that show ≥95% average nucleotide identity (ANI) to the type genome, which has been assigned the MAG ID CHK125-3527 and which is available via NCBI BioSample SAMN15816959. The GC content of the type genome is 36.77% and the genome length is 2.0 Mbp.	
Description of Candidatus Avanaerovorax gen. nov.	
Candidatus Avanaerovorax (Av.an.a.e.ro.vo’rax. L. fem. n. avis bird; N.L. masc. n. Anaerovorax a genus name; N.L. masc. n. Avanaerovorax a genus related to the genus Anaerovorax but distinct from it and found in poultry)	
A bacterial genus identified by metagenomic analyses. The genus includes all bacteria with genomes that show ≥60% average amino acid identity (AAI) to the type genome from the type species Candidatus Avanaerovorax faecigallinarum. This genus has been assigned by GTDB-Tk v1.3.0 working on GTDB Release 05-RS95 (Chaumeil et al., 2019; Parks et al., 2020) to the order Peptostreptococcales and to the family Anaerovoracaceae.	
Description of Candidatus Avanaerovorax faecigallinarum sp. nov.	
Candidatus Avanaerovorax faecigallinarum (fae.ci.gal.li.na’rum. L. fem. n. faex, faecis excrement; L. fem. n. gallina hen; N.L. gen. n. faecigallinarum of hen faeces)	
A bacterial species identified by metagenomic analyses. This species includes all bacteria with genomes that show ≥95% average nucleotide identity (ANI) to the type genome, which has been assigned the MAG ID Gambia13-1450 and which is available via NCBI BioSample SAMN15816994. The GC content of the type genome is 48.68% and the genome length is 1.8 Mbp.	
Description of Candidatus Aveggerthella gen. nov.	
Candidatus Aveggerthella (Av.eg.ger.thel’la. L. fem. n. avis bird; N.L. fem. n. Eggerthella a genus name; N.L. fem. n. Aveggerthella a genus related to the genus Eggerthella but distinct from it and found in poultry)	
A bacterial genus identified by metagenomic analyses. The genus includes all bacteria with genomes that show ≥60% average amino acid identity (AAI) to the type genome from the type species Candidatus Avieggerthella excrementigallinarum. This genus has been assigned by GTDB-Tk v1.3.0 working on GTDB Release 05-RS95 (Chaumeil et al., 2019; Parks et al., 2020) to the order Coriobacteriales and to the family Eggerthellaceae.	

Description of Candidatus Aveggerthella excrementigallinarum sp. nov.	
Candidatus Aveggerthella excrementigallinarum (ex.cre.men.ti.gal.li.na’rum. L. neut. n. excrementum excrement; L. fem. n. gallina hen; N.L. gen. n. excrementigallinarum of hen excrement)	
A bacterial species identified by metagenomic analyses. This species includes all bacteria with genomes that show ≥95% average nucleotide identity (ANI) to the type genome, which has been assigned the MAG ID ChiGjej4B4-3573 and which is available via NCBI BioSample SAMN15816976. The GC content of the type genome is 65.93% and the genome length is 2.0 Mbp.	
Description of Candidatus Aveggerthella stercoripullorum sp. nov.	
Candidatus Aveggerthella stercoripullorum (ster.co.ri.pul.lo’rum. L. neut. n. stercus dung; L. masc. n. pullus a young chicken; N.L. gen. n. stercoripullorum of the faceces of young chickens)	
A bacterial species identified by metagenomic analyses. This species includes all bacteria with genomes that show ≥95% average nucleotide identity (ANI) to the type genome, which has been assigned the MAG ID ChiGjej1B1-2707 and which is available via NCBI BioSample SAMN15816950. The GC content of the type genome is 61.50% and the genome length is 2.1 Mbp.	
Description of Candidatus Avelusimicrobium gen. nov.	
Candidatus Avelusimicrobium (Av.e.lu.si.mi.cro’bi.um. L. fem. n. avis bird; N.L. neut. n. Elusimicrobium a genus name; N.L. neut. n. Avelusimicrobium a genus related to the genus Elusimicrobium but distinct from it and found in poultry)	
A bacterial genus identified by metagenomic analyses. The genus includes all bacteria with genomes that show ≥60% average amino acid identity (AAI) to the type genome from the type species Candidatus Avielusimicrobium excrementipullorum. This genus has been assigned by GTDB-Tk v1.3.0 working on GTDB Release 05-RS95 (Chaumeil et al., 2019; Parks et al., 2020) to the order Elusimicrobiales and to the family Elusimicrobiaceae.	
Description of Candidatus Avelusimicrobium excrementipullorum sp. nov.	
Candidatus Avelusimicrobium excrementipullorum (ex.cre.men.ti.pul.lo’rum. L. neut. n. excrementum excrement; L. masc. n. pullus a young chicken; N.L. gen. n. excrementipullorum of young chicken excrement)	
A bacterial species identified by metagenomic analyses. This species includes all bacteria with genomes that show ≥95% average nucleotide identity (ANI) to the type genome, which has been assigned the MAG ID CHK136-6324 and which is available via NCBI BioSample SAMN15817002. The GC content of the type genome is 53.46% and the genome length is 1.3 Mbp.	
Description of Candidatus Avibacteroides gen. nov.	
Candidatus Avibacteroides (A.vi.bac.te.ro’i.des. L. fem. n. avis bird; N.L. masc. n. Bacteroides a genus name; N.L. masc. n. Avibacteroides a genus related to the genus Bacteroides but distinct from it and found in poultry)	
A bacterial genus identified by metagenomic analyses. The genus includes all bacteria with genomes that show ≥60% average amino acid identity (AAI) to the type genome from the type species Candidatus Avibacteroides excrementipullorum. This genus has been assigned by GTDB-Tk v1.3.0 working on GTDB Release 05-RS95 (Chaumeil et al., 2019; Parks et al., 2020) to the order Bacteroidales and to the family Bacteroidaceae.	
Description of Candidatus Avibacteroides avistercoris sp. nov.	
Candidatus Avibacteroides avistercoris (a.vi.ster’co.ris. L. fem. n. avis bird; L. neut. n. stercus dung; N.L. gen. n. avistercoris of bird faeces)	
A bacterial species identified by metagenomic analyses. This species includes all bacteria with genomes that show ≥95% average nucleotide identity (ANI) to the type genome, which has been assigned the MAG ID MalCec1-1739 and which is available via NCBI BioSample SAMN15816974. The GC content of the type genome is 53.14% and the genome length is 2.2 Mbp.	
Description of Candidatus Avibacteroides excrementipullorum sp. nov.	
Candidatus Avibacteroides excrementipullorum (ex.cre.men.ti.pul.lo’rum. L. neut. n. excrementum excrement; L. masc. n. pullus a young chicken; N.L. gen. n. excrementipullorum of young chicken excrement)	
A bacterial species identified by metagenomic analyses. This species includes all bacteria with genomes that show ≥95% average nucleotide identity (ANI) to the type genome, which has been assigned the MAG ID ChiHjej12B11-16860 and which is available via NCBI BioSample SAMN15816958. The GC content of the type genome is 47.79% and the genome length is 2.2 Mbp.	
Description of Candidatus Avibacteroides faecavium sp. nov.	
Candidatus Avibacteroides faecavium (faec.a’vi.um. L. fem. n. faex, faecis excrement; L. fem. n. avis bird; N.L. gen. n. faecavium of bird faeces)	
A bacterial species identified by metagenomic analyses. This species includes all bacteria with genomes that show ≥95% average nucleotide identity (ANI) to the type genome, which has been assigned the MAG ID 3702 and which is available via NCBI BioSample SAMN15816980. The GC content of the type genome is 55.42% and the genome length is 2.1 Mbp.	

Description of Candidatus Avichristensenella gen. nov.	
Candidatus Avichristensenella (A.vi.chris.ten.sen.el’la. L. fem. n. avis bird; N.L. fem. n. Christensenella a genus name; N.L. fem. n. Avichristensenella a genus related to the genus Christensenella but distinct from it and found in poultry)	
A bacterial genus identified by metagenomic analyses. The genus includes all bacteria with genomes that show ≥60% average amino acid identity (AAI) to the type genome from the type species Candidatus Avichristensenella intestinipullorum. This genus has been assigned by GTDB-Tk v1.3.0 working on GTDB Release 05-RS95 (Chaumeil et al., 2019; Parks et al., 2020) to the order Christensenellales and to the family CAG-74.	
Description of Candidatus Avichristensenella intestinipullorum sp. nov.	
Candidatus Avichristensenella intestinipullorum (in.tes.ti.ni.pul.lo’rum. L. neut. n. intestinum gut; L. masc. n. pullus a young chicken; N.L. gen. n. intestinipullorum of the gut of young chickens)	
A bacterial species identified by metagenomic analyses. This species includes all bacteria with genomes that show ≥95% average nucleotide identity (ANI) to the type genome, which has been assigned the MAG ID ChiHile30-977 and which is available via NCBI BioSample SAMN15816947. The GC content of the type genome is 63.80% and the genome length is 2.3 Mbp.	
Description of Candidatus Avidehalobacter gen. nov.	
Candidatus Avidehalobacter (A.vi.de.ha.lo.bac’ter. L. fem. n. avis bird; N.L. masc. n. Dehalobacter a genus name; N.L. masc. n. Avidehalobacter a genus related to the genus Dehalobacter but distinct from it and found in poultry)	
A bacterial genus identified by metagenomic analyses. The genus includes all bacteria with genomes that show ≥60% average amino acid identity (AAI) to the type genome from the type species Candidatus Avidehalobacter gallistercoris. This genus has been assigned by GTDB-Tk v1.3.0 working on GTDB Release 05-RS95 (Chaumeil et al., 2019; Parks et al., 2020) to the order UBA4068 and to the family UBA5755.	
Description of Candidatus Avidehalobacter gallistercoris sp. nov.	
Candidatus Avidehalobacter gallistercoris (gal.li.ster’co.ris. L. masc. n gallus chicken; L. neut. n. stercus dung; N.L. gen. n. gallistercoris of chicken faeces)	
A bacterial species identified by metagenomic analyses. This species includes all bacteria with genomes that show ≥95% average nucleotide identity (ANI) to the type genome, which has been assigned the MAG ID 2830 and which is available via NCBI BioSample SAMN15816981. The GC content of the type genome is 52.20% and the genome length is 1.4 Mbp.	
Description of Candidatus Avidesulfovibrio gen. nov.	
Candidatus Avidesulfovibrio (A.vi.de.sul.fo.vi’bri.o. L. fem. n. avis bird; N.L. masc. n. Desulfovibrio a genus name; N.L. masc. n. Avidesulfovibrio a genus related to the genus Desulfovibrio but distinct from it and found in poultry)	
A bacterial genus identified by metagenomic analyses. The genus includes all bacteria with genomes that show ≥60% average amino acid identity (AAI) to the type genome from the type species Candidatus Avidesulfovibrio excrementigallinarum. This genus has been assigned by GTDB-Tk v1.3.0 working on GTDB Release 05-RS95 (Chaumeil et al., 2019; Parks et al., 2020) to the order Desulfovibrionales and to the family Desulfovibrionaceae.	
Description of Candidatus Avidesulfovibrio excrementigallinarum sp. nov.	
Candidatus Avidesulfovibrio excrementigallinarum (ex.cre.men.ti.gal.li.na’rum. L. neut. n. excrementum excrement; L. fem. n. gallina hen; N.L. gen. n. excrementigallinarum of hen excrement)	
A bacterial species identified by metagenomic analyses. This species includes all bacteria with genomes that show ≥95% average nucleotide identity (ANI) to the type genome, which has been assigned the MAG ID ChiHcec4-2777 and which is available via NCBI BioSample SAMN15816982. The GC content of the type genome is 60.70% and the genome length is 2.2 Mbp.	
Description of Candidatus Avigastranaerophilus gen. nov.	
Candidatus Avigastranaerophilus (A.vi.gastr.an.a.e.ro’phi.lus. L. fem. n. avis bird; N.L. masc. n. Gastranaerophilus a genus name; N.L. masc. n. Avigastranaerophilus a genus related to the genus Gastranaerophilus but distinct from it and found in poultry)	
A bacterial genus identified by metagenomic analyses. The genus includes all bacteria with genomes that show ≥60% average amino acid identity (AAI) to the type genome from the type species Candidatus Avigastranaerophilus faecigallinarum. This genus has been assigned by GTDB-Tk v1.3.0 working on GTDB Release 05-RS95 (Chaumeil et al., 2019; Parks et al., 2020) to the order Gastranaerophilales and to the family Gastranaerophilaceae.	
Description of Candidatus Avigastranaerophilus faecigallinarum sp. nov.	
Candidatus Avigastranaerophilus faecigallinarum (fae.ci.gal.li.na’rum. L. fem. n. faex, faecis excrement; L. fem. n. gallina hen; N.L. gen. n. faecigallinarum of hen faeces)	
A bacterial species identified by metagenomic analyses. This species includes all bacteria with genomes that show ≥95% average nucleotide identity (ANI) to the type genome, which has been assigned the MAG ID 5572 and which is available via NCBI BioSample SAMN15816968. The GC content of the type genome is 29.33% and the genome length is 2.2 Mbp.	
Description of Candidatus Avilachnospira gen. nov.	
Candidatus Avilachnospira (A.vi.lach.no.spi’ra. L. fem. n. avis bird; N.L. fem. n. Lachnospira a genus name; N.L. fem. n. Avilachnospira a genus related to the genus Lachnospira but distinct from it and found in poultry)	
A bacterial genus identified by metagenomic analyses. The genus includes all bacteria with genomes that show ≥60% average amino acid identity (AAI) to the type genome from the type species Candidatus Avilachnospira avistercoris. This genus has been assigned by GTDB-Tk v1.3.0 working on GTDB Release 05-RS95 (Chaumeil et al., 2019; Parks et al., 2020) to the order Lachnospirales and to the family Lachnospiraceae.	

Description of Candidatus Avilachnospira avicola sp. nov.	
Candidatus Avilachnospira avicola (a.vi’co.la. L. fem. n. avis bird; L. suff. -cola inhabitant of; N.L. n. avicola inhabitant of birds)	
A bacterial species identified by metagenomic analyses. This species includes all bacteria with genomes that show ≥95% average nucleotide identity (ANI) to the type genome, which has been assigned the MAG ID ChiHecec3B27-5021 and which is available via NCBI BioSample SAMN15816990. The GC content of the type genome is 49.15% and the genome length is 1.6 Mbp.	
Description of Candidatus Avilachnospira avistercoris sp. nov.	
Candidatus Avilachnospira avistercoris (a.vi.ster’co.ris. L. fem. n. avis bird; L. neut. n. stercus dung; N.L. gen. n. avistercoris of bird faeces)	
A bacterial species identified by metagenomic analyses. This species includes all bacteria with genomes that show ≥95% average nucleotide identity (ANI) to the type genome, which has been assigned the MAG ID ChiGjej5B5-15814 and which is available via NCBI BioSample SAMN15816991. The GC content of the type genome is 50.02% and the genome length is 1.6 Mbp.	
Description of Candidatus Avimonoglobus gen. nov.	
Candidatus Avimonoglobus (A.vi.mo.no.glo’bus. L. fem. n. avis bird; N.L. masc. n. Monoglobus a genus name; N.L. masc. n. Avimonoglobus a genus related to the genus Monoglobus but distinct from it and found in poultry)	
A bacterial genus identified by metagenomic analyses. The genus includes all bacteria with genomes that show ≥60% average amino acid identity (AAI) to the type genome from the type species Candidatus Avimonoglobus intestinipullorum. This genus has been assigned by GTDB-Tk v1.3.0 working on GTDB Release 05-RS95 (Chaumeil et al., 2019; Parks et al., 2020) to the order Monoglobales_A and to the family UBA1381.	
Description of Candidatus Avimonoglobus intestinipullorum sp. nov.	
Candidatus Avimonoglobus intestinipullorum (in.tes.ti.ni.pul.lo’rum. L. neut. n. intestinum gut; L. masc. n. pullus a young chicken; N.L. gen. n. intestinipullorum of the gut of young chickens)	
A bacterial species identified by metagenomic analyses. This species includes all bacteria with genomes that show ≥95% average nucleotide identity (ANI) to the type genome, which has been assigned the MAG ID ChiSjej4B22-9803 and which is available via NCBI BioSample SAMN15816985. The GC content of the type genome is 51.95% and the genome length is 1.8 Mbp.	
Description of Candidatus Avimuribaculum gen. nov.	
Candidatus Avimuribaculum (A.vi.mu.ri.ba’cu.lum. L. fem. n. avis bird; N.L. neut. n. Muribaculum a genus name; N.L. neut. n. Avimuribaculum a genus related to the genus Muribaculum but distinct from it and found in poultry)	
A bacterial genus identified by metagenomic analyses. The genus includes all bacteria with genomes that show ≥60% average amino acid identity (AAI) to the type genome from the type species Candidatus Avimuribaculum pullicola. This genus has been assigned by GTDB-Tk v1.3.0 working on GTDB Release 05-RS95 (Chaumeil et al., 2019; Parks et al., 2020) to the order Bacteroidales and to the family Muribaculaceae.	
Description of Candidatus Avimuribaculum pullicola sp. nov.	
Candidatus Avimuribaculum pullicola (pul.li’co.la. L. masc. n. pullus a young chicken; L. suff. -cola inhabitant of; N.L. n. pullicola an inhabitant of young chickens)	
A bacterial species identified by metagenomic analyses. This species includes all bacteria with genomes that show ≥95% average nucleotide identity (ANI) to the type genome, which has been assigned the MAG ID ChiHecec3B27-9160 and which is available via NCBI BioSample SAMN15816969. The GC content of the type genome is 47.58% and the genome length is 2.2 Mbp.	
Description of Candidatus Avipropionibacterium gen. nov.	
Candidatus Avipropionibacterium (A.vi.pro.pi.o.ni.bac.te’ri.um. L. fem. n. avis bird; N.L. neut. n. Propionibacterium a genus name; N.L. neut. n. Avipropionibacterium a genus related to the genus Propionibacterium but distinct from it and found in poultry)	
A bacterial genus identified by metagenomic analyses. The genus includes all bacteria with genomes that show ≥60% average amino acid identity (AAI) to the type genome from the type species Candidatus Avipropionibacterium avicola. This genus has been assigned by GTDB-Tk v1.3.0 working on GTDB Release 05-RS95 (Chaumeil et al., 2019; Parks et al., 2020) to the order Propionibacteriales and to the family Propionibacteriaceae.	
Description of Candidatus Avipropionibacterium avicola sp. nov.	
Candidatus Avipropionibacterium avicola (a.vi’co.la. L. fem. n. avis bird; L. suff. -cola inhabitant of; N.L. n. avicola inhabitant of birds)	
A bacterial species identified by metagenomic analyses. This species includes all bacteria with genomes that show ≥95% average nucleotide identity (ANI) to the type genome, which has been assigned the MAG ID ChiGjej1B1-24693 and which is available via NCBI BioSample SAMN15816979. The GC content of the type genome is 69.14% and the genome length is 3.2 Mbp.	

Description of Candidatus Avirikenella gen. nov.	
Candidatus Avirikenella (A.vi.ri.ke.nel’la. L. fem. n. avis bird; N.L. fem. n. Rikenella a genus name; N.L. fem. n. Avirikenella a genus related to the genus Rikenella but distinct from it and found in poultry)	
A bacterial genus identified by metagenomic analyses. The genus includes all bacteria with genomes that show ≥60% average amino acid identity (AAI) to the type genome from the type species Candidatus Avirikenella pullistercoris. This genus has been assigned by GTDB-Tk v1.3.0 working on GTDB Release 05-RS95 (Chaumeil et al., 2019; Parks et al., 2020) to the order Bacteroidales and to the family Rikenellaceae.	
Description of Candidatus Avirikenella pullistercoris sp. nov.	
Candidatus Avirikenella pullistercoris (pul.li.ster’co.ris. L. masc. n. pullus a young chicken; L. neut. n. stercus dung; N.L. gen. n. pullistercoris of young chicken faeces)	
A bacterial species identified by metagenomic analyses. This species includes all bacteria with genomes that show ≥95% average nucleotide identity (ANI) to the type genome, which has been assigned the MAG ID 9321 and which is available via NCBI BioSample SAMN15816960. The GC content of the type genome is 41.91% and the genome length is 1.9 Mbp.	
Description of Candidatus Avisuccinivibrio gen. nov.	
Candidatus Avisuccinivibrio (A.vi.suc.ci.ni.vi’bri.o. L. fem. n. avis bird; N.L. masc. n. Succinivibrio a genus name; N.L. masc. n. Avisuccinivibrio a genus related to the genus Succinivibrio but distinct from it and found in poultry)	
A bacterial genus identified by metagenomic analyses. The genus includes all bacteria with genomes that show ≥60% average amino acid identity (AAI) to the type genome from the type species Candidatus Avisuccinivibrio stercorigallinarum. This genus has been assigned by GTDB-Tk v1.3.0 working on GTDB Release 05-RS95 (Chaumeil et al., 2019; Parks et al., 2020) to the order Enterobacterales and to the family Succinivibrionaceae.	
Description of Candidatus Avisuccinivibrio pullicola sp. nov.	
Candidatus Avisuccinivibrio pullicola (pul.li’co.la. L. masc. n. pullus a young chicken; L. suff. -cola inhabitant of; N.L. n. pullicola an inhabitant of young chickens)	
A bacterial species identified by metagenomic analyses. This species includes all bacteria with genomes that show ≥95% average nucleotide identity (ANI) to the type genome, which has been assigned the MAG ID 3820 and which is available via NCBI BioSample SAMN15816999. The GC content of the type genome is 55.94% and the genome length is 2.4 Mbp.	
Description of Candidatus Avisuccinivibrio stercorigallinarum sp. nov.	
Candidatus Avisuccinivibrio stercorigallinarum (ster.co.ri.gal.li.na’rum. L. neut. n. stercus dung; L. fem. n. gallina hen; N.L. gen. n. stercorigallinarum of hen faeces)	
A bacterial species identified by metagenomic analyses. This species includes all bacteria with genomes that show ≥95% average nucleotide identity (ANI) to the type genome, which has been assigned the MAG ID 17213 and which is available via NCBI BioSample SAMN15817000. The GC content of the type genome is 54.49% and the genome length is 2.4 Mbp.	
Description of Candidatus Avitreponema gen. nov.	
Candidatus Avitreponema (A.vi.tre.po.ne’ma. L. fem. n. avis bird; N.L. neut. n. Treponema a genus name; N.L. neut. n. Avitreponema a genus related to the genus Treponema but distinct from it and found in poultry)	
A bacterial genus identified by metagenomic analyses. The genus includes all bacteria with genomes that show ≥60% average amino acid identity (AAI) to the type genome from the type species Candidatus Avitreponema avistercoris. This genus has been assigned by GTDB-Tk v1.3.0 working on GTDB Release 05-RS95 (Chaumeil et al., 2019; Parks et al., 2020) to the order Treponematales and to the family Treponemataceae.	
Description of Candidatus Avitreponema avistercoris sp. nov.	
Candidatus Avitreponema avistercoris (a.vi.ster’co.ris. L. fem. n. avis bird; L. neut. n. stercus dung; N.L. gen. n. avistercoris of bird faeces)	
A bacterial species identified by metagenomic analyses. This species includes all bacteria with genomes that show ≥95% average nucleotide identity (ANI) to the type genome, which has been assigned the MAG ID B3-4054 and which is available via NCBI BioSample SAMN15816977. The GC content of the type genome is 55.36% and the genome length is 1.9 Mbp.	
Description of Candidatus Avoscillospira gen. nov.	
Candidatus Avoscillospira (Av.os.cil.lo.spi’ra. L. fem. n. avis bird; N.L. fem. n. Oscillospira a genus name; N.L. fem. n. Avoscillospira a genus related to the genus Oscillospira but distinct from it and found in poultry)	
A bacterial genus identified by metagenomic analyses. The genus includes all bacteria with genomes that show ≥60% average amino acid identity (AAI) to the type genome from the type species Candidatus Avioscillospira stercorigallinarum. This genus has been assigned by GTDB-Tk v1.3.0 working on GTDB Release 05-RS95 (Chaumeil et al., 2019; Parks et al., 2020) to the order Oscillospirales and to the family Oscillospiraceae.	

Description of Candidatus Avoscillospira avicola sp. nov.	
Candidatus Avoscillospira avicola (a.vi’co.la. L. fem. n. avis bird; L. suff. -cola inhabitant of; N.L. n. avicola inhabitant of birds)	
A bacterial species identified by metagenomic analyses. This species includes all bacteria with genomes that show ≥95% average nucleotide identity (ANI) to the type genome, which has been assigned the MAG ID ChiBcec15-4380 and which is available via NCBI BioSample SAMN15816934. The GC content of the type genome is 61.76% and the genome length is 2.5 Mbp.	
Description of Candidatus Avoscillospira avistercoris sp. nov.	
Candidatus Avoscillospira avistercoris (a.vi.ster’co.ris. L. fem. n. avis bird; L. neut. n. stercus dung; N.L. gen. n. avistercoris of bird faeces)	
A bacterial species identified by metagenomic analyses. This species includes all bacteria with genomes that show ≥95% average nucleotide identity (ANI) to the type genome, which has been assigned the MAG ID ChiBcec16-1751 and which is available via NCBI BioSample SAMN15816964. The GC content of the type genome is 58.00% and the genome length is 2.4 Mbp.	
Description of Candidatus Avoscillospira stercorigallinarum sp. nov.	
Candidatus Avoscillospira stercorigallinarum (ster.co.ri.gal.li.na’rum. L. neut. n. stercus dung; L. fem. n. gallina hen; N.L. gen. n. stercorigallinarum of hen faeces)	
A bacterial species identified by metagenomic analyses. This species includes all bacteria with genomes that show ≥95% average nucleotide identity (ANI) to the type genome, which has been assigned the MAG ID ChiSjej2B20-13462 and which is available via NCBI BioSample SAMN15816948. The GC content of the type genome is 63.01% and the genome length is 2.2 Mbp.	
Description of Candidatus Avoscillospira stercoripullorum sp. nov.	
Candidatus Avoscillospira stercoripullorum (ster.co.ri.pul.lo’rum. L. neut. n. stercus dung; L. masc. n. pullus a young chicken; N.L. gen. n. stercoripullorum of the faceces of young chickens)	
A bacterial species identified by metagenomic analyses. This species includes all bacteria with genomes that show ≥95% average nucleotide identity (ANI) to the type genome, which has been assigned the MAG ID ChiHjej9B8-7071 and which is available via NCBI BioSample SAMN15816951. The GC content of the type genome is 60.69% and the genome length is 1.9 Mbp.	
Description of Candidatus Bacteroides avicola sp. nov.	
Candidatus Bacteroides avicola (a.vi’co.la. L. fem. n. avis bird; L. suff. -cola inhabitant of; N.L. n. avicola inhabitant of birds)	
A bacterial species identified by metagenomic analyses. This species includes all bacteria with genomes that show ≥95% average nucleotide identity (ANI) to the type genome, which has been assigned the MAG ID ChiHjej12B11-9795 and which is available via NCBI BioSample SAMN15816830. This is a new name for the alphanumeric GTDB species sp002160055. The GC content of the type genome is 50.12% and the genome length is 3.0 Mbp.	
Description of Candidatus Bacteroides intestinavium sp. nov.	
Candidatus Bacteroides intestinavium (in.tes.tin.a’vi.um. L. neut. n. intestinum gut; L. fem. n. avis bird; N.L. gen. n. intestinavium of the gut of birds)	
A bacterial species identified by metagenomic analyses. This species includes all bacteria with genomes that show ≥95% average nucleotide identity (ANI) to the type genome, which has been assigned the MAG ID ChiHecec1B25-7008 and which is available via NCBI BioSample SAMN15816665. The GC content of the type genome is 54.17% and the genome length is 2.6 Mbp.	
Description of Candidatus Bacteroides intestinigallinarum sp. nov.	
Candidatus Bacteroides intestinigallinarum (in.tes.ti.ni.gal.li.na’rum. L. neut. n. intestinum gut; L. fem. n. gallina hen; N.L. gen. n. intestinigallinarum of the gut of the hens)	
A bacterial species identified by metagenomic analyses. This species includes all bacteria with genomes that show ≥95% average nucleotide identity (ANI) to the type genome, which has been assigned the MAG ID 2926 and which is available via NCBI BioSample SAMN15816831. This is a new name for the alphanumeric GTDB species sp003463205. The GC content of the type genome is 41.82% and the genome length is 5.8 Mbp.	
Description of Candidatus Bacteroides intestinipullorum sp. nov.	
Candidatus Bacteroides intestinipullorum (in.tes.ti.ni.pul.lo’rum. L. neut. n. intestinum gut; L. masc. n. pullus a young chicken; N.L. gen. n. intestinipullorum of the gut of young chickens)	
A bacterial species identified by metagenomic analyses. This species includes all bacteria with genomes that show ≥95% average nucleotide identity (ANI) to the type genome, which has been assigned the MAG ID B3-3758 and which is available via NCBI BioSample SAMN15816671. The GC content of the type genome is 53.99% and the genome length is 2.7 Mbp.	

Description of Candidatus Bacteroides merdavium sp. nov.	
Candidatus Bacteroides merdavium (merd.a’vi.um. L. fem. n. merda faeces; L. fem. n. avis bird; N.L. gen. n. merdavium of bird faeces)	
A bacterial species identified by metagenomic analyses. This species includes all bacteria with genomes that show ≥95% average nucleotide identity (ANI) to the type genome, which has been assigned the MAG ID CHK118-2852 and which is available via NCBI BioSample SAMN15816687. The GC content of the type genome is 49.76% and the genome length is 2.9 Mbp.	
Description of Candidatus Bacteroides merdigallinarum sp. nov.	
Candidatus Bacteroides merdigallinarum (mer.di.gal.li.na’rum. L. fem. n. merda faeces; L. fem. n. gallina hen; N.L. gen. n. merdigallinarum of hen faeces)	
A bacterial species identified by metagenomic analyses. This species includes all bacteria with genomes that show ≥95% average nucleotide identity (ANI) to the type genome, which has been assigned the MAG ID ChiHjej9B8-1298 and which is available via NCBI BioSample SAMN15816694. The GC content of the type genome is 54.52% and the genome length is 2.7 Mbp.	
Description of Candidatus Bacteroides merdipullorum sp. nov.	
Candidatus Bacteroides merdipullorum (mer.di.pul.lo’rum. L. fem. n. merda faeces; L. masc. n. pullus a young chicken; N.L. gen. n. merdipullorum of the faeces of young chickens)	
A bacterial species identified by metagenomic analyses. This species includes all bacteria with genomes that show ≥95% average nucleotide identity (ANI) to the type genome, which has been assigned the MAG ID ChiHjej12B11-24981 and which is available via NCBI BioSample SAMN15816699. The GC content of the type genome is 53.73% and the genome length is 2.5 Mbp.	
Description of Candidatus Bacteroides pullicola sp. nov.	
Candidatus Bacteroides pullicola (pul.li’co.la. L. masc. n. pullus a young chicken; L. suff. -cola inhabitant of; N.L. n. pullicola an inhabitant of young chickens)	
A bacterial species identified by metagenomic analyses. This species includes all bacteria with genomes that show ≥95% average nucleotide identity (ANI) to the type genome, which has been assigned the MAG ID Gambia2-208 and which is available via NCBI BioSample SAMN15816704. The GC content of the type genome is 55.18% and the genome length is 2.7 Mbp.	
Description of Candidatus Bariatricus faecipullorum sp. nov.	
Candidatus Bariatricus faecipullorum (fae.ci.pul.lo’rum. L. fem. n. faex, faecis excrement; L. masc. n. pullus a young chicken; N.L. gen. n. faecipullorum of young chicken faeces)	
A bacterial species identified by metagenomic analyses. This species includes all bacteria with genomes that show ≥95% average nucleotide identity (ANI) to the type genome, which has been assigned the MAG ID 9095 and which is available via NCBI BioSample SAMN15816662. The GC content of the type genome is 51.54% and the genome length is 2.5 Mbp.	
Description of Candidatus Barnesiella excrementavium sp. nov.	
Candidatus Barnesiella excrementavium (ex.cre.ment.a’vi.um. L. neut. n. excrementum excrement; L. fem. n. avis bird; N.L. gen. n. excrementavium of bird excrement)	
A bacterial species identified by metagenomic analyses. This species includes all bacteria with genomes that show ≥95% average nucleotide identity (ANI) to the type genome, which has been assigned the MAG ID 4398 and which is available via NCBI BioSample SAMN15816714. The GC content of the type genome is 52.73% and the genome length is 2.7 Mbp.	
Description of Candidatus Barnesiella excrementigallinarum sp. nov.	
Candidatus Barnesiella excrementigallinarum (ex.cre.men.ti.gal.li.na’rum. L. neut. n. excrementum excrement; L. fem. n. gallina hen; N.L. gen. n. excrementigallinarum of hen excrement)	
A bacterial species identified by metagenomic analyses. This species includes all bacteria with genomes that show ≥95% average nucleotide identity (ANI) to the type genome, which has been assigned the MAG ID CHK169-14362 and which is available via NCBI BioSample SAMN15816739. The GC content of the type genome is 47.04% and the genome length is 2.6 Mbp.	
Description of Candidatus Barnesiella excrementipullorum sp. nov.	
Candidatus Barnesiella excrementipullorum (ex.cre.men.ti.pul.lo’rum. L. neut. n. excrementum excrement; L. masc. n. pullus a young chicken; N.L. gen. n. excrementipullorum of young chicken excrement)	
A bacterial species identified by metagenomic analyses. This species includes all bacteria with genomes that show ≥95% average nucleotide identity (ANI) to the type genome, which has been assigned the MAG ID ChiHjej12B11-16260 and which is available via NCBI BioSample SAMN15816862. This is a new name for the alphanumeric GTDB species sp900542255. The GC content of the type genome is 50.69% and the genome length is 2.1 Mbp.	

Description of Candidatus Barnesiella merdigallinarum sp. nov.	
Candidatus Barnesiella merdigallinarum (mer.di.gal.li.na’rum. L. fem. n. merda faeces; L. fem. n. gallina hen; N.L. gen. n. merdigallinarum of hen faeces)	
A bacterial species identified by metagenomic analyses. This species includes all bacteria with genomes that show ≥95% average nucleotide identity (ANI) to the type genome, which has been assigned the MAG ID CHK136-6590 and which is available via NCBI BioSample SAMN15816819. This is a new name for the alphanumeric GTDB species sp002159975. The GC content of the type genome is 54.15% and the genome length is 2.7 Mbp.	
Description of Candidatus Barnesiella merdipullorum sp. nov.	
Candidatus Barnesiella merdipullorum (mer.di.pul.lo’rum. L. fem. n. merda faeces; L. masc. n. pullus a young chicken; N.L. gen. n. merdipullorum of the faeces of young chickens)	
A bacterial species identified by metagenomic analyses. This species includes all bacteria with genomes that show ≥95% average nucleotide identity (ANI) to the type genome, which has been assigned the MAG ID 5648 and which is available via NCBI BioSample SAMN15816849. This is a new name for the alphanumeric GTDB species sp002161555. The GC content of the type genome is 52.62% and the genome length is 2.6 Mbp.	
Description of Candidatus Bilophila faecipullorum sp. nov.	
Candidatus Bilophila faecipullorum (fae.ci.pul.lo’rum. L. fem. n. faex, faecis excrement; L. masc. n. pullus a young chicken; N.L. gen. n. faecipullorum of young chicken faeces)	
A bacterial species identified by metagenomic analyses. This species includes all bacteria with genomes that show ≥95% average nucleotide identity (ANI) to the type genome, which has been assigned the MAG ID ChiSxjej5B17-1746 and which is available via NCBI BioSample SAMN15816754. The GC content of the type genome is 63.22% and the genome length is 2.8 Mbp.	
Description of Candidatus Blautia avicola sp. nov.	
Candidatus Blautia avicola (a.vi’co.la. L. fem. n. avis bird; L. suff. -cola inhabitant of; N.L. n. avicola inhabitant of birds)	
A bacterial species identified by metagenomic analyses. This species includes all bacteria with genomes that show ≥95% average nucleotide identity (ANI) to the type genome, which has been assigned the MAG ID ChiBcec6-4105 and which is available via NCBI BioSample SAMN15816794. The GC content of the type genome is 46.40% and the genome length is 3.1 Mbp.	
Description of Candidatus Blautia avistercoris sp. nov.	
Candidatus Blautia avistercoris (a.vi.ster’co.ris. L. fem. n. avis bird; L. neut. n. stercus dung; N.L. gen. n. avistercoris of bird faeces)	
A bacterial species identified by metagenomic analyses. This species includes all bacteria with genomes that show ≥95% average nucleotide identity (ANI) to the type genome, which has been assigned the MAG ID 5548 and which is available via NCBI BioSample SAMN15816924. This is a new name for the alphanumeric GTDB species sp002159835. Although GTDB has assigned this species to the genus it calls Blautia_A, this genus designation cannot be incorporated into a well-formed binomial, so in naming this species, we have used the current validly published name for the genus. The GC content of the type genome is 45.38% and the genome length is 2.5 Mbp.	
Description of Candidatus Blautia excrementigallinarum sp. nov.	
Candidatus Blautia excrementigallinarum (ex.cre.men.ti.gal.li.na’rum. L. neut. n. excrementum excrement; L. fem. n. gallina hen; N.L. gen. n. excrementigallinarum of hen excrement)	
A bacterial species identified by metagenomic analyses. This species includes all bacteria with genomes that show ≥95% average nucleotide identity (ANI) to the type genome, which has been assigned the MAG ID ChiSjej6B24-370 and which is available via NCBI BioSample SAMN15816894. Although GTDB has assigned this species to the genus it calls Blautia_A, this genus designation cannot be incorporated into a well-formed binomial, so in naming this species, we have used the current validly published name for the genus. The GC content of the type genome is 49.44% and the genome length is 2.3 Mbp.	
Description of Candidatus Blautia excrementipullorum sp. nov.	
Candidatus Blautia excrementipullorum (ex.cre.men.ti.pul.lo’rum. L. neut. n. excrementum excrement; L. masc. n. pullus a young chicken; N.L. gen. n. excrementipullorum of young chicken excrement)	
A bacterial species identified by metagenomic analyses. This species includes all bacteria with genomes that show ≥95% average nucleotide identity (ANI) to the type genome, which has been assigned the MAG ID CHK197-7439 and which is available via NCBI BioSample SAMN15816885. Although GTDB has assigned this species to the genus it calls Blautia_A, this genus designation cannot be incorporated into a well-formed binomial, so in naming this species, we have used the current validly published name for the genus. The GC content of the type genome is 46.79% and the genome length is 3.3 Mbp.	

Description of Candidatus Blautia faecavium sp. nov.	
Candidatus Blautia faecavium (faec.a’vi.um. L. fem. n. faex, faecis excrement; L. fem. n. avis bird; N.L. gen. n. faecavium of bird faeces)	
A bacterial species identified by metagenomic analyses. This species includes all bacteria with genomes that show ≥95% average nucleotide identity (ANI) to the type genome, which has been assigned the MAG ID ChiSjej1B19-5720 and which is available via NCBI BioSample SAMN15816886. Although GTDB has assigned this species to the genus it calls Blautia_A, this genus designation cannot be incorporated into a well-formed binomial, so in naming this species, we have used the current validly published name for the genus. The GC content of the type genome is 45.35% and the genome length is 3.5 Mbp.	
Description of Candidatus Blautia faecigallinarum sp. nov.	
Candidatus Blautia faecigallinarum (fae.ci.gal.li.na’rum. L. fem. n. faex, faecis excrement; L. fem. n. gallina hen; N.L. gen. n. faecigallinarum of hen faeces)	
A bacterial species identified by metagenomic analyses. This species includes all bacteria with genomes that show ≥95% average nucleotide identity (ANI) to the type genome, which has been assigned the MAG ID 14324 and which is available via NCBI BioSample SAMN15816901. Although GTDB has assigned this species to the genus it calls Blautia_A, this genus designation cannot be incorporated into a well-formed binomial, so in naming this species, we have used the current validly published name for the genus. The GC content of the type genome is 48.52% and the genome length is 2.9 Mbp.	
Description of Candidatus Blautia faecipullorum sp. nov.	
Candidatus Blautia faecipullorum (fae.ci.pul.lo’rum. L. fem. n. faex, faecis excrement; L. masc. n. pullus a young chicken; N.L. gen. n. faecipullorum of young chicken faeces)	
A bacterial species identified by metagenomic analyses. This species includes all bacteria with genomes that show ≥95% average nucleotide identity (ANI) to the type genome, which has been assigned the MAG ID ChiSxjej6B18-2004 and which is available via NCBI BioSample SAMN15816906. Although GTDB has assigned this species to the genus it calls Blautia_A, this genus designation cannot be incorporated into a well-formed binomial, so in naming this species, we have used the current validly published name for the genus. The GC content of the type genome is 48.18% and the genome length is 3.5 Mbp.	
Description of Candidatus Blautia gallistercoris sp. nov.	
Candidatus Blautia gallistercoris (gal.li.ster’co.ris. L. masc. n gallus chicken; L. neut. n. stercus dung; N.L. gen. n. gallistercoris of chicken faeces)	
A bacterial species identified by metagenomic analyses. This species includes all bacteria with genomes that show ≥95% average nucleotide identity (ANI) to the type genome, which has been assigned the MAG ID ChiSjej1B19-8411 and which is available via NCBI BioSample SAMN15816925. This is a new name for the alphanumeric GTDB species sp900542045. Although GTDB has assigned this species to the genus it calls Blautia_A, this genus designation cannot be incorporated into a well-formed binomial, so in naming this species, we have used the current validly published name for the genus. The GC content of the type genome is 48.96% and the genome length is 2.8 Mbp.	
Description of Candidatus Blautia intestinavium sp. nov.	
Candidatus Blautia intestinavium (in.tes.tin.a’vi.um. L. neut. n. intestinum gut; L. fem. n. avis bird; N.L. gen. n. intestinavium of the gut of birds)	
A bacterial species identified by metagenomic analyses. This species includes all bacteria with genomes that show ≥95% average nucleotide identity (ANI) to the type genome, which has been assigned the MAG ID CHK186-553 and which is available via NCBI BioSample SAMN15816890. Although GTDB has assigned this species to the genus it calls Blautia_A, this genus designation cannot be incorporated into a well-formed binomial, so in naming this species, we have used the current validly published name for the genus. The GC content of the type genome is 47.27% and the genome length is 2.8 Mbp.	
Description of Candidatus Blautia intestinigallinarum sp. nov.	
Candidatus Blautia intestinigallinarum (in.tes.ti.ni.gal.li.na’rum. L. neut. n. intestinum gut; L. fem. n. gallina hen; N.L. gen. n. intestinigallinarum of the gut of the hens)	
A bacterial species identified by metagenomic analyses. This species includes all bacteria with genomes that show ≥95% average nucleotide identity (ANI) to the type genome, which has been assigned the MAG ID CHK186-9876 and which is available via NCBI BioSample SAMN15816891. Although GTDB has assigned this species to the genus it calls Blautia_A, this genus designation cannot be incorporated into a well-formed binomial, so in naming this species, we have used the current validly published name for the genus. The GC content of the type genome is 46.77% and the genome length is 2.7 Mbp.	
Description of Candidatus Blautia intestinipullorum sp. nov.	
Candidatus Blautia intestinipullorum (in.tes.ti.ni.pul.lo’rum. L. neut. n. intestinum gut; L. masc. n. pullus a young chicken; N.L. gen. n. intestinipullorum of the gut of young chickens)	
A bacterial species identified by metagenomic analyses. This species includes all bacteria with genomes that show ≥95% average nucleotide identity (ANI) to the type genome, which has been assigned the MAG ID ChiW16-4312 and which is available via NCBI BioSample SAMN15816892. Although GTDB has assigned this species to the genus it calls Blautia_A, this genus designation cannot be incorporated into a well-formed binomial, so in naming this species, we have used the current validly published name for the genus. The GC content of the type genome is 47.05% and the genome length is 2.4 Mbp.	

Description of Candidatus Blautia merdavium sp. nov.	
Candidatus Blautia merdavium (merd.a’vi.um. L. fem. n. merda faeces; L. fem. n. avis bird; N.L. gen. n. merdavium of bird faeces)	
A bacterial species identified by metagenomic analyses. This species includes all bacteria with genomes that show ≥95% average nucleotide identity (ANI) to the type genome, which has been assigned the MAG ID ChiBcec2-3848 and which is available via NCBI BioSample SAMN15816633. The GC content of the type genome is 48.60% and the genome length is 3.3 Mbp.	
Description of Candidatus Blautia merdigallinarum sp. nov.	
Candidatus Blautia merdigallinarum (mer.di.gal.li.na’rum. L. fem. n. merda faeces; L. fem. n. gallina hen; N.L. gen. n. merdigallinarum of hen faeces)	
A bacterial species identified by metagenomic analyses. This species includes all bacteria with genomes that show ≥95% average nucleotide identity (ANI) to the type genome, which has been assigned the MAG ID ChiSxjej6B18-287 and which is available via NCBI BioSample SAMN15816815. This is a new name for the alphanumeric GTDB species sp900543715. The GC content of the type genome is 45.18% and the genome length is 3.3 Mbp.	
Description of Candidatus Blautia merdipullorum sp. nov.	
Candidatus Blautia merdipullorum (mer.di.pul.lo’rum. L. fem. n. merda faeces; L. masc. n. pullus a young chicken; N.L. gen. n. merdipullorum of the faeces of young chickens)	
A bacterial species identified by metagenomic analyses. This species includes all bacteria with genomes that show ≥95% average nucleotide identity (ANI) to the type genome, which has been assigned the MAG ID 17058 and which is available via NCBI BioSample SAMN15816655. The GC content of the type genome is 45.05% and the genome length is 3.2 Mbp.	
Description of Candidatus Blautia ornithocaccae sp. nov.	
Candidatus Blautia ornithocaccae (or.ni.tho.cac’cae. Gr. masc. or fem. n. ornis, ornithos bird Gr. fem. n. kakke faeces; N.L. gen. n. ornithocaccae of bird faeces)	
A bacterial species identified by metagenomic analyses. This species includes all bacteria with genomes that show ≥95% average nucleotide identity (ANI) to the type genome, which has been assigned the MAG ID ChiBcec1-3711 and which is available via NCBI BioSample SAMN15816880. This is a new name for the alphanumeric GTDB species sp002161285. The GC content of the type genome is 44.79% and the genome length is 3.1 Mbp.	
Description of Candidatus Blautia pullicola sp. nov.	
Candidatus Blautia pullicola (pul.li’co.la. L. masc. n. pullus a young chicken; L. suff. -cola inhabitant of; N.L. n. pullicola an inhabitant of young chickens)	
A bacterial species identified by metagenomic analyses. This species includes all bacteria with genomes that show ≥95% average nucleotide identity (ANI) to the type genome, which has been assigned the MAG ID 1068 and which is available via NCBI BioSample SAMN15816689. The GC content of the type genome is 45.62% and the genome length is 3.0 Mbp.	
Description of Candidatus Blautia pullistercoris sp. nov.	
Candidatus Blautia pullistercoris (pul.li.ster’co.ris. L. masc. n. pullus a young chicken; L. neut. n. stercus dung; N.L. gen. n. pullistercoris of young chicken faeces)	
A bacterial species identified by metagenomic analyses. This species includes all bacteria with genomes that show ≥95% average nucleotide identity (ANI) to the type genome, which has been assigned the MAG ID ChiHjej12B11-1927 and which is available via NCBI BioSample SAMN15816618. The GC content of the type genome is 45.73% and the genome length is 3.2 Mbp.	
Description of Candidatus Blautia stercoravium sp. nov.	
Candidatus Blautia stercoravium (ster.cor.a’vi.um. L. neut. n. stercus dung; L. fem. n. avis bird; N.L. gen. n. stercoravium of bird faeces)	
A bacterial species identified by metagenomic analyses. This species includes all bacteria with genomes that show ≥95% average nucleotide identity (ANI) to the type genome, which has been assigned the MAG ID 3268 and which is available via NCBI BioSample SAMN15816738. The GC content of the type genome is 44.17% and the genome length is 2.7 Mbp.	
Description of Candidatus Blautia stercorigallinarum sp. nov.	
Candidatus Blautia stercorigallinarum (ster.co.ri.gal.li.na’rum. L. neut. n. stercus dung; L. fem. n. gallina hen; N.L. gen. n. stercorigallinarum of hen faeces)	
A bacterial species identified by metagenomic analyses. This species includes all bacteria with genomes that show ≥95% average nucleotide identity (ANI) to the type genome, which has been assigned the MAG ID CHK195-9823 and which is available via NCBI BioSample SAMN15816627. The GC content of the type genome is 45.83% and the genome length is 3.1 Mbp.	

Description of Candidatus Blautia stercoripullorum sp. nov.	
Candidatus Blautia stercoripullorum (ster.co.ri.pul.lo’rum. L. neut. n. stercus dung; L. masc. n. pullus a young chicken; N.L. gen. n. stercoripullorum of the faceces of young chickens)	
A bacterial species identified by metagenomic analyses. This species includes all bacteria with genomes that show ≥95% average nucleotide identity (ANI) to the type genome, which has been assigned the MAG ID ChiW19-6364 and which is available via NCBI BioSample SAMN15816793. The GC content of the type genome is 44.78% and the genome length is 3.1 Mbp.	
Description of Candidatus Borkfalkia avicola sp. nov.	
Candidatus Borkfalkia avicola (a.vi’co.la. L. fem. n. avis bird; L. suff. -cola inhabitant of; N.L. n. avicola inhabitant of birds)	
A bacterial species identified by metagenomic analyses. This species includes all bacteria with genomes that show ≥95% average nucleotide identity (ANI) to the type genome, which has been assigned the MAG ID CHK192-19661 and which is available via NCBI BioSample SAMN15816606. The GC content of the type genome is 58.92% and the genome length is 1.7 Mbp.	
Description of Candidatus Borkfalkia avistercoris sp. nov.	
Candidatus Borkfalkia avistercoris (a.vi.ster’co.ris. L. fem. n. avis bird; L. neut. n. stercus dung; N.L. gen. n. avistercoris of bird faeces)	
A bacterial species identified by metagenomic analyses. This species includes all bacteria with genomes that show ≥95% average nucleotide identity (ANI) to the type genome, which has been assigned the MAG ID CHK187-5294 and which is available via NCBI BioSample SAMN15816607. The GC content of the type genome is 53.99% and the genome length is 1.7 Mbp.	
Description of Candidatus Borkfalkia excrementavium sp. nov.	
Candidatus Borkfalkia excrementavium (ex.cre.ment.a’vi.um. L. neut. n. excrementum excrement; L. fem. n. avis bird; N.L. gen. n. excrementavium of bird excrement)	
A bacterial species identified by metagenomic analyses. This species includes all bacteria with genomes that show ≥95% average nucleotide identity (ANI) to the type genome, which has been assigned the MAG ID CHK199-9574 and which is available via NCBI BioSample SAMN15816608. The GC content of the type genome is 52.76% and the genome length is 1.6 Mbp.	
Description of Candidatus Borkfalkia excrementigallinarum sp. nov.	
Candidatus Borkfalkia excrementigallinarum (ex.cre.men.ti.gal.li.na’rum. L. neut. n. excrementum excrement; L. fem. n. gallina hen; N.L. gen. n. excrementigallinarum of hen excrement)	
A bacterial species identified by metagenomic analyses. This species includes all bacteria with genomes that show ≥95% average nucleotide identity (ANI) to the type genome, which has been assigned the MAG ID 1345 and which is available via NCBI BioSample SAMN15816609. The GC content of the type genome is 53.42% and the genome length is 1.9 Mbp.	
Description of Candidatus Borkfalkia excrementipullorum sp. nov.	
Candidatus Borkfalkia excrementipullorum (ex.cre.men.ti.pul.lo’rum. L. neut. n. excrementum excrement; L. masc. n. pullus a young chicken; N.L. gen. n. excrementipullorum of young chicken excrement)	
A bacterial species identified by metagenomic analyses. This species includes all bacteria with genomes that show ≥95% average nucleotide identity (ANI) to the type genome, which has been assigned the MAG ID CHK192-2667 and which is available via NCBI BioSample SAMN15816611. The GC content of the type genome is 55.63% and the genome length is 1.6 Mbp.	
Description of Candidatus Borkfalkia faecavium sp. nov.	
Candidatus Borkfalkia faecavium (faec.a’vi.um. L. fem. n. faex, faecis excrement; L. fem. n. avis bird; N.L. gen. n. faecavium of bird faeces)	
A bacterial species identified by metagenomic analyses. This species includes all bacteria with genomes that show ≥95% average nucleotide identity (ANI) to the type genome, which has been assigned the MAG ID 2189 and which is available via NCBI BioSample SAMN15816731. The GC content of the type genome is 58.98% and the genome length is 1.7 Mbp.	
Description of Candidatus Borkfalkia faecigallinarum sp. nov.	
Candidatus Borkfalkia faecigallinarum (fae.ci.gal.li.na’rum. L. fem. n. faex, faecis excrement; L. fem. n. gallina hen; N.L. gen. n. faecigallinarum of hen faeces)	
A bacterial species identified by metagenomic analyses. This species includes all bacteria with genomes that show ≥95% average nucleotide identity (ANI) to the type genome, which has been assigned the MAG ID 26628 and which is available via NCBI BioSample SAMN15816617. The GC content of the type genome is 62.49% and the genome length is 1.6 Mbp.	

Description of Candidatus Borkfalkia faecipullorum sp. nov.	
Candidatus Borkfalkia faecipullorum (fae.ci.pul.lo’rum. L. fem. n. faex, faecis excrement; L. masc. n. pullus a young chicken; N.L. gen. n. faecipullorum of young chicken faeces)	
A bacterial species identified by metagenomic analyses. This species includes all bacteria with genomes that show ≥95% average nucleotide identity (ANI) to the type genome, which has been assigned the MAG ID 811 and which is available via NCBI BioSample SAMN15816621. The GC content of the type genome is 54.29% and the genome length is 1.9 Mbp.	
Description of Candidatus Borkfalkia stercoripullorum sp. nov.	
Candidatus Borkfalkia stercoripullorum (ster.co.ri.pul.lo’rum. L. neut. n. stercus dung; L. masc. n. pullus a young chicken; N.L. gen. n. stercoripullorum of the faceces of young chickens)	
A bacterial species identified by metagenomic analyses. This species includes all bacteria with genomes that show ≥95% average nucleotide identity (ANI) to the type genome, which has been assigned the MAG ID CHK196-13738 and which is available via NCBI BioSample SAMN15816588. The GC content of the type genome is 55.22% and the genome length is 1.8 Mbp.	
Description of Candidatus Brachybacterium intestinipullorum sp. nov.	
Candidatus Brachybacterium intestinipullorum (in.tes.ti.ni.pul.lo’rum. L. neut. n. intestinum gut; L. masc. n. pullus a young chicken; N.L. gen. n. intestinipullorum of the gut of young chickens)	
A bacterial species identified by metagenomic analyses. This species includes all bacteria with genomes that show ≥95% average nucleotide identity (ANI) to the type genome, which has been assigned the MAG ID CHK130-7132 and which is available via NCBI BioSample SAMN15816812. This is a new name for the alphanumeric GTDB species sp003711805. The GC content of the type genome is 72.59% and the genome length is 3.5 Mbp.	
Description of Candidatus Brachybacterium merdavium sp. nov.	
Candidatus Brachybacterium merdavium (merd.a’vi.um. L. fem. n. merda faeces; L. fem. n. avis bird; N.L. gen. n. merdavium of bird faeces)	
A bacterial species identified by metagenomic analyses. This species includes all bacteria with genomes that show ≥95% average nucleotide identity (ANI) to the type genome, which has been assigned the MAG ID ChiHjej13B12-24818 and which is available via NCBI BioSample SAMN15816666. The GC content of the type genome is 70.22% and the genome length is 3.7 Mbp.	
Description of Candidatus Brachybacterium merdigallinarum sp. nov.	
Candidatus Brachybacterium merdigallinarum (mer.di.gal.li.na’rum. L. fem. n. merda faeces; L. fem. n. gallina hen; N.L. gen. n. merdigallinarum of hen faeces)	
A bacterial species identified by metagenomic analyses. This species includes all bacteria with genomes that show ≥95% average nucleotide identity (ANI) to the type genome, which has been assigned the MAG ID ChiHjej13B12-7362 and which is available via NCBI BioSample SAMN15816717. The GC content of the type genome is 71.16% and the genome length is 3.1 Mbp.	
Description of Candidatus Brevibacterium intestinavium sp. nov.	
Candidatus Brevibacterium intestinavium (in.tes.tin.a’vi.um. L. neut. n. intestinum gut; L. fem. n. avis bird; N.L. gen. n. intestinavium of the gut of birds)	
A bacterial species identified by metagenomic analyses. This species includes all bacteria with genomes that show ≥95% average nucleotide identity (ANI) to the type genome, which has been assigned the MAG ID 5295 and which is available via NCBI BioSample SAMN15816668. The GC content of the type genome is 66.76% and the genome length is 3.2 Mbp.	
Description of Candidatus Brevibacterium intestinigallinarum sp. nov.	
Candidatus Brevibacterium intestinigallinarum (in.tes.ti.ni.gal.li.na’rum. L. neut. n. intestinum gut; L. fem. n. gallina hen; N.L. gen. n. intestinigallinarum of the gut of the hens)	
A bacterial species identified by metagenomic analyses. This species includes all bacteria with genomes that show ≥95% average nucleotide identity (ANI) to the type genome, which has been assigned the MAG ID CHK132-2174 and which is available via NCBI BioSample SAMN15816673. The GC content of the type genome is 70.54% and the genome length is 2.7 Mbp.	
Description of Candidatus Butyricicoccus avicola sp. nov.	
Candidatus Butyricicoccus avicola (a.vi’co.la. L. fem. n. avis bird; L. suff. -cola inhabitant of; N.L. n. avicola inhabitant of birds)	
A bacterial species identified by metagenomic analyses. This species includes all bacteria with genomes that show ≥95% average nucleotide identity (ANI) to the type genome, which has been assigned the MAG ID ChiSjej6B24-14740 and which is available via NCBI BioSample SAMN15816587. The GC content of the type genome is 59.67% and the genome length is 2.0 Mbp.	

Description of Candidatus Borkfalkia faecipullorum sp. nov.	
Candidatus Borkfalkia faecipullorum (fae.ci.pul.lo’rum. L. fem. n. faex, faecis excrement; L. masc. n. pullus a young chicken; N.L. gen. n. faecipullorum of young chicken faeces)	
A bacterial species identified by metagenomic analyses. This species includes all bacteria with genomes that show ≥95% average nucleotide identity (ANI) to the type genome, which has been assigned the MAG ID 811 and which is available via NCBI BioSample SAMN15816621. The GC content of the type genome is 54.29% and the genome length is 1.9 Mbp.	
Description of Candidatus Borkfalkia stercoripullorum sp. nov.	
Candidatus Borkfalkia stercoripullorum (ster.co.ri.pul.lo’rum. L. neut. n. stercus dung; L. masc. n. pullus a young chicken; N.L. gen. n. stercoripullorum of the faceces of young chickens)	
A bacterial species identified by metagenomic analyses. This species includes all bacteria with genomes that show ≥95% average nucleotide identity (ANI) to the type genome, which has been assigned the MAG ID CHK196-13738 and which is available via NCBI BioSample SAMN15816588. The GC content of the type genome is 55.22% and the genome length is 1.8 Mbp.	
Description of Candidatus Brachybacterium intestinipullorum sp. nov.	
Candidatus Brachybacterium intestinipullorum (in.tes.ti.ni.pul.lo’rum. L. neut. n. intestinum gut; L. masc. n. pullus a young chicken; N.L. gen. n. intestinipullorum of the gut of young chickens)	
A bacterial species identified by metagenomic analyses. This species includes all bacteria with genomes that show ≥95% average nucleotide identity (ANI) to the type genome, which has been assigned the MAG ID CHK130-7132 and which is available via NCBI BioSample SAMN15816812. This is a new name for the alphanumeric GTDB species sp003711805. The GC content of the type genome is 72.59% and the genome length is 3.5 Mbp.	
Description of Candidatus Brachybacterium merdavium sp. nov.	
Candidatus Brachybacterium merdavium (merd.a’vi.um. L. fem. n. merda faeces; L. fem. n. avis bird; N.L. gen. n. merdavium of bird faeces)	
A bacterial species identified by metagenomic analyses. This species includes all bacteria with genomes that show ≥95% average nucleotide identity (ANI) to the type genome, which has been assigned the MAG ID ChiHjej13B12-24818 and which is available via NCBI BioSample SAMN15816666. The GC content of the type genome is 70.22% and the genome length is 3.7 Mbp.	
Description of Candidatus Brachybacterium merdigallinarum sp. nov.	
Candidatus Brachybacterium merdigallinarum (mer.di.gal.li.na’rum. L. fem. n. merda faeces; L. fem. n. gallina hen; N.L. gen. n. merdigallinarum of hen faeces)	
A bacterial species identified by metagenomic analyses. This species includes all bacteria with genomes that show ≥95% average nucleotide identity (ANI) to the type genome, which has been assigned the MAG ID ChiHjej13B12-7362 and which is available via NCBI BioSample SAMN15816717. The GC content of the type genome is 71.16% and the genome length is 3.1 Mbp.	
Description of Candidatus Brevibacterium intestinavium sp. nov.	
Candidatus Brevibacterium intestinavium (in.tes.tin.a’vi.um. L. neut. n. intestinum gut; L. fem. n. avis bird; N.L. gen. n. intestinavium of the gut of birds)	
A bacterial species identified by metagenomic analyses. This species includes all bacteria with genomes that show ≥95% average nucleotide identity (ANI) to the type genome, which has been assigned the MAG ID 5295 and which is available via NCBI BioSample SAMN15816668. The GC content of the type genome is 66.76% and the genome length is 3.2 Mbp.	
Description of Candidatus Brevibacterium intestinigallinarum sp. nov.	
Candidatus Brevibacterium intestinigallinarum (in.tes.ti.ni.gal.li.na’rum. L. neut. n. intestinum gut; L. fem. n. gallina hen; N.L. gen. n. intestinigallinarum of the gut of the hens)	
A bacterial species identified by metagenomic analyses. This species includes all bacteria with genomes that show ≥95% average nucleotide identity (ANI) to the type genome, which has been assigned the MAG ID CHK132-2174 and which is available via NCBI BioSample SAMN15816673. The GC content of the type genome is 70.54% and the genome length is 2.7 Mbp.	
Description of Candidatus Butyricicoccus avicola sp. nov.	
Candidatus Butyricicoccus avicola (a.vi’co.la. L. fem. n. avis bird; L. suff. -cola inhabitant of; N.L. n. avicola inhabitant of birds)	
A bacterial species identified by metagenomic analyses. This species includes all bacteria with genomes that show ≥95% average nucleotide identity (ANI) to the type genome, which has been assigned the MAG ID ChiSjej6B24-14740 and which is available via NCBI BioSample SAMN15816587. The GC content of the type genome is 59.67% and the genome length is 2.0 Mbp.	

Description of Candidatus Caccocola faecigallinarum sp. nov.	
Candidatus Caccocola faecigallinarum (fae.ci.gal.li.na’rum. L. fem. n. faex, faecis excrement; L. fem. n. gallina hen; N.L. gen. n. faecigallinarum of hen faeces)	
A bacterial species identified by metagenomic analyses. This species includes all bacteria with genomes that show ≥95% average nucleotide identity (ANI) to the type genome, which has been assigned the MAG ID ChiHjej11B10-3130 and which is available via NCBI BioSample SAMN15817215. This is a new name for the alphanumeric GTDB species sp900544635. The GC content of the type genome is 60.31% and the genome length is 2.3 Mbp.	
Description of Candidatus Caccocola faecipullorum sp. nov.	
Candidatus Caccocola faecipullorum (fae.ci.pul.lo’rum. L. fem. n. faex, faecis excrement; L. masc. n. pullus a young chicken; N.L. gen. n. faecipullorum of young chicken faeces)	
A bacterial species identified by metagenomic analyses. This species includes all bacteria with genomes that show ≥95% average nucleotide identity (ANI) to the type genome, which has been assigned the MAG ID ChiHcec3-7374 and which is available via NCBI BioSample SAMN15817103. The GC content of the type genome is 59.08% and the genome length is 2.2 Mbp.	
Description of Candidatus Caccomonas gen. nov.	
Candidatus Caccomonas (Cac.co.mo’nas. Gr. fem. n. kakke dung; L. fem. n. monas a monad; N.L. fem. n. Caccomonas a microbe associated with faeces)	
A bacterial genus identified by metagenomic analyses. The genus includes all bacteria with genomes that show ≥60% average amino acid identity (AAI) to the type genome from the type species Candidatus Caccomonas pullistercoris. This is a name for the alphanumeric GTDB genus CAG-617. This genus has been assigned by GTDB-Tk v1.3.0 working on GTDB Release 05-RS95 (Chaumeil et al., 2019; Parks et al., 2020) to the order Bacteroidales and to the family Bacteroidaceae.	
Description of Candidatus Caccomonas pullistercoris sp. nov.	
Candidatus Caccomonas pullistercoris (pul.li.ster’co.ris. L. masc. n. pullus a young chicken; L. neut. n. stercus dung; N.L. gen. n. pullistercoris of young chicken faeces)	
A bacterial species identified by metagenomic analyses. This species includes all bacteria with genomes that show ≥95% average nucleotide identity (ANI) to the type genome, which has been assigned the MAG ID 5345 and which is available via NCBI BioSample SAMN15817152. The GC content of the type genome is 59.45% and the genome length is 2.2 Mbp.	
Description of Candidatus Caccomorpha gen. nov.	
Candidatus Caccomorpha (Cac.co.mor’pha. Gr. fem. n. kakke dung; Gr. fem. n. morphe a form, shape; N.L. fem. n. Caccomorpha a microbe associated with faeces)	
A bacterial genus identified by metagenomic analyses. The genus includes all bacteria with genomes that show ≥60% average amino acid identity (AAI) to the type genome from the type species Candidatus Caccomorpha excrementavium. This is a name for the alphanumeric GTDB genus SZUA-448. This genus has been assigned by GTDB-Tk v1.3.0 working on GTDB Release 05-RS95 (Chaumeil et al., 2019; Parks et al., 2020) to the order Lachnospirales and to the family Lachnospiraceae.	
Description of Candidatus Caccomorpha excrementavium sp. nov.	
Candidatus Caccomorpha excrementavium (ex.cre.ment.a’vi.um. L. neut. n. excrementum excrement; L. fem. n. avis bird; N.L. gen. n. excrementavium of bird excrement)	
A bacterial species identified by metagenomic analyses. This species includes all bacteria with genomes that show ≥95% average nucleotide identity (ANI) to the type genome, which has been assigned the MAG ID 1999 and which is available via NCBI BioSample SAMN15817145. The GC content of the type genome is 50.75% and the genome length is 2.7 Mbp.	
Description of Candidatus Caccoplasma gen. nov.	
Candidatus Caccoplasma (Cac.co.plas’ma. Gr. fem. n. kakke dung; Gr. neut. n. plasma a form; N.L. neut. n. Caccoplasma a microbe associated with faeces)	
A bacterial genus identified by metagenomic analyses. The genus includes all bacteria with genomes that show ≥60% average amino acid identity (AAI) to the type genome from the type species Candidatus Caccoplasma merdavium. This is a name for the alphanumeric GTDB genus UBA11471. This genus has been assigned by GTDB-Tk v1.3.0 working on GTDB Release 05-RS95 (Chaumeil et al., 2019; Parks et al., 2020) to the order Bacteroidales and to the family UBA11471.	
Description of Candidatus Caccoplasma intestinavium sp. nov.	
Candidatus Caccoplasma intestinavium (in.tes.tin.a’vi.um. L. neut. n. intestinum gut; L. fem. n. avis bird; N.L. gen. n. intestinavium of the gut of birds)	
A bacterial species identified by metagenomic analyses. This species includes all bacteria with genomes that show ≥95% average nucleotide identity (ANI) to the type genome, which has been assigned the MAG ID 21143 and which is available via NCBI BioSample SAMN15817187. This is a new name for the alphanumeric GTDB species sp000434215. The GC content of the type genome is 45.83% and the genome length is 2.4 Mbp.	

Description of Candidatus Caccoplasma merdavium sp. nov.	
Candidatus Caccoplasma merdavium (merd.a’vi.um. L. fem. n. merda faeces; L. fem. n. avis bird; N.L. gen. n. merdavium of bird faeces)	
A bacterial species identified by metagenomic analyses. This species includes all bacteria with genomes that show ≥95% average nucleotide identity (ANI) to the type genome, which has been assigned the MAG ID 6821 and which is available via NCBI BioSample SAMN15817190. This is a new name for the alphanumeric GTDB species sp900542765. The GC content of the type genome is 52.79% and the genome length is 2.4 Mbp.	
Description of Candidatus Caccoplasma merdipullorum sp. nov.	
Candidatus Caccoplasma merdipullorum (mer.di.pul.lo’rum. L. fem. n. merda faeces; L. masc. n. pullus a young chicken; N.L. gen. n. merdipullorum of the faeces of young chickens)	
A bacterial species identified by metagenomic analyses. This species includes all bacteria with genomes that show ≥95% average nucleotide identity (ANI) to the type genome, which has been assigned the MAG ID G3-4614 and which is available via NCBI BioSample SAMN15817135. The GC content of the type genome is 46.70% and the genome length is 2.0 Mbp.	
Description of Candidatus Caccopulliclostridium gen. nov.	
Candidatus Caccopulliclostridium (Cac.co.pul.li.clos.tri’di.um. Gr. fem. n. kakke faeces; L. masc. n. pullus a young chicken; N.L. neut. n. Clostridium a genus name; N.L. neut. n. Caccopulliclostridium a genus related to the genus Clostridium but distinct from it and found in poultry faeces)	
A bacterial genus identified by metagenomic analyses. The genus includes all bacteria with genomes that show ≥60% average amino acid identity (AAI) to the type genome from the type species Candidatus Caccopulliclostridium gallistercoris. This genus was identified but not named by Glendinning et al. (2020). This genus has been assigned by GTDB-Tk v1.3.0 working on GTDB Release 05-RS95 (Chaumeil et al., 2019; Parks et al., 2020) to the order 4C28d-15 and to the family UBA1242.	
Description of Candidatus Caccopulliclostridium gallistercoris sp. nov.	
Candidatus Caccopulliclostridium gallistercoris (gal.li.ster’co.ris. L. masc. n gallus chicken; L. neut. n. stercus dung; N.L. gen. n. gallistercoris of chicken faeces)	
A bacterial species identified by metagenomic analyses. This species includes all bacteria with genomes that show ≥95% average nucleotide identity (ANI) to the type genome, which has been assigned the MAG ID CHK186-9395 and which is available via NCBI BioSample SAMN15816943. The GC content of the type genome is 35.40% and the genome length is 1.0 Mbp.	
Description of Candidatus Caccosoma gen. nov.	
Candidatus Caccosoma (Cac.co.so’ma. Gr. fem. n. kakke dung; Gr. neut. n. soma a body; N.L. neut. n. Caccosoma a microbe associated with faeces)	
A bacterial genus identified by metagenomic analyses. The genus includes all bacteria with genomes that show ≥60% average amino acid identity (AAI) to the type genome from the type species Candidatus Caccosoma faecigallinarum. This is a name for the alphanumeric GTDB genus CAG-631. This genus has been assigned by GTDB-Tk v1.3.0 working on GTDB Release 05-RS95 (Chaumeil et al., 2019; Parks et al., 2020) to the order RFN20 and to the family CAG-631.	
Description of Candidatus Caccosoma faecigallinarum sp. nov.	
Candidatus Caccosoma faecigallinarum (fae.ci.gal.li.na’rum. L. fem. n. faex, faecis excrement; L. fem. n. gallina hen; N.L. gen. n. faecigallinarum of hen faeces)	
A bacterial species identified by metagenomic analyses. This species includes all bacteria with genomes that show ≥95% average nucleotide identity (ANI) to the type genome, which has been assigned the MAG ID 14508 and which is available via NCBI BioSample SAMN15817185. This is a new name for the alphanumeric GTDB species sp000433015. The GC content of the type genome is 30.38% and the genome length is 1.3 Mbp.	
Description of Candidatus Caccousia gen. nov.	
Candidatus Caccousia (Cacc.ou’si.a. Gr. fem. n. kakke dung; Gr. fem. n. ousia an essence; N.L. fem. n. Caccousia a microbe associated with faeces)	
A bacterial genus identified by metagenomic analyses. The genus includes all bacteria with genomes that show ≥60% average amino acid identity (AAI) to the type genome from the type species Candidatus Caccousia avicola. This is a name for the alphanumeric GTDB genus An200. This genus has been assigned by GTDB-Tk v1.3.0 working on GTDB Release 05-RS95 (Chaumeil et al., 2019; Parks et al., 2020) to the order Oscillospirales and to the family Acutalibacteraceae.	
Description of Candidatus Caccousia avicola sp. nov.	
Candidatus Caccousia avicola (a.vi’co.la. L. fem. n. avis bird; L. suff. -cola inhabitant of; N.L. n. avicola inhabitant of birds)	
A bacterial species identified by metagenomic analyses. This species includes all bacteria with genomes that show ≥95% average nucleotide identity (ANI) to the type genome, which has been assigned the MAG ID ChiSxjej1B13-7958 and which is available via NCBI BioSample SAMN15817070. The GC content of the type genome is 58.28% and the genome length is 2.2 Mbp.	

Description of Candidatus Caccousia avistercoris sp. nov.	
Candidatus Caccousia avistercoris (a.vi.ster’co.ris. L. fem. n. avis bird; L. neut. n. stercus dung; N.L. gen. n. avistercoris of bird faeces)	
A bacterial species identified by metagenomic analyses. This species includes all bacteria with genomes that show ≥95% average nucleotide identity (ANI) to the type genome, which has been assigned the MAG ID 3024 and which is available via NCBI BioSample SAMN15817047. The GC content of the type genome is 60.88% and the genome length is 2.6 Mbp.	
Description of Candidatus Caccousia stercoris sp. nov.	
Candidatus Caccousia stercoris (ster’co.ris. L. gen. n. stercoris of dung, excrement)	
A bacterial species identified by metagenomic analyses. This species includes all bacteria with genomes that show ≥95% average nucleotide identity (ANI) to the type genome, which has been assigned the MAG ID 6086 and which is available via NCBI BioSample SAMN15817184. This is a new name for the alphanumeric GTDB species sp002160025. The GC content of the type genome is 56.84% and the genome length is 2.3 Mbp.	
Description of Candidatus Caccovicinus gen. nov.	
Candidatus Caccovicinus (Cac.co.vi.ci’nus. Gr. fem. n. kakke dung; L. masc. n. vicinus a neighbour; N.L. masc. n. Caccovicinus a microbe associated with faeces)	
A bacterial genus identified by metagenomic analyses. The genus includes all bacteria with genomes that show ≥60% average amino acid identity (AAI) to the type genome from the type species Candidatus Caccovicinus merdipullorum. This is a name for the alphanumeric GTDB genus UMGS1370. This genus has been assigned by GTDB-Tk v1.3.0 working on GTDB Release 05-RS95 (Chaumeil et al., 2019; Parks et al., 2020) to the order Lachnospirales and to the family Lachnospiraceae.	
Description of Candidatus Caccovicinus merdipullorum sp. nov.	
Candidatus Caccovicinus merdipullorum (mer.di.pul.lo’rum. L. fem. n. merda faeces; L. masc. n. pullus a young chicken; N.L. gen. n. merdipullorum of the faeces of young chickens)	
A bacterial species identified by metagenomic analyses. This species includes all bacteria with genomes that show ≥95% average nucleotide identity (ANI) to the type genome, which has been assigned the MAG ID CHK198_11255 and which is available via NCBI BioSample SAMN15817041. The GC content of the type genome is 50.44% and the genome length is 3.0 Mbp.	
Description of Candidatus Caccovivens gen. nov.	
Candidatus Caccovivens (Cac.co.vi’vens. Gr. fem. n. kakke dung; N.L. pres. part. vivens living; N.L. fem. n. Caccovivens a microbe associated with faeces)	
A bacterial genus identified by metagenomic analyses. The genus includes all bacteria with genomes that show ≥60% average amino acid identity (AAI) to the type genome from the type species Candidatus Caccovivens faecavium. This is a name for the alphanumeric GTDB genus UBA11517. This genus has been assigned by GTDB-Tk v1.3.0 working on GTDB Release 05-RS95 (Chaumeil et al., 2019; Parks et al., 2020) to the order Christensenellales and to the family UBA1242.	
Description of Candidatus Caccovivens faecavium sp. nov.	
Candidatus Caccovivens faecavium (faec.a’vi.um. L. fem. n. faex, faecis excrement; L. fem. n. avis bird; N.L. gen. n. faecavium of bird faeces)	
A bacterial species identified by metagenomic analyses. This species includes all bacteria with genomes that show ≥95% average nucleotide identity (ANI) to the type genome, which has been assigned the MAG ID ChiW6-1002 and which is available via NCBI BioSample SAMN15817119. The GC content of the type genome is 34.22% and the genome length is 1.1 Mbp.	
Description of Candidatus Cellulosilyticum pullistercoris sp. nov.	
Candidatus Cellulosilyticum pullistercoris (pul.li.ster’co.ris. L. masc. n. pullus a young chicken; L. neut. n. stercus dung; N.L. gen. n. pullistercoris of young chicken faeces)	
A bacterial species identified by metagenomic analyses. This species includes all bacteria with genomes that show ≥95% average nucleotide identity (ANI) to the type genome, which has been assigned the MAG ID B5-657 and which is available via NCBI BioSample SAMN15816752. The GC content of the type genome is 33.78% and the genome length is 2.3 Mbp.	
Description of Candidatus Choladocola gen. nov.	
Candidatus Choladocola (Cho.la.do’co.la. Gr. fem. n. cholas guts; L. suff. -cola inhabitant of; N.L. fem. n. Choladocola a microbe associated with the intestines)	
A bacterial genus identified by metagenomic analyses. The genus includes all bacteria with genomes that show ≥60% average amino acid identity (AAI) to the type genome from the type species Candidatus Choladocola avistercoris. This is a name for the alphanumeric GTDB genus UBA7182. This genus has been assigned by GTDB-Tk v1.3.0 working on GTDB Release 05-RS95 (Chaumeil et al., 2019; Parks et al., 2020) to the order Lachnospirales and to the family Lachnospiraceae.	

Description of Candidatus Choladocola avistercoris sp. nov.	
Candidatus Choladocola avistercoris (a.vi.ster’co.ris. L. fem. n. avis bird; L. neut. n. stercus dung; N.L. gen. n. avistercoris of bird faeces)	
A bacterial species identified by metagenomic analyses. This species includes all bacteria with genomes that show ≥95% average nucleotide identity (ANI) to the type genome, which has been assigned the MAG ID ChiBcec18-1958 and which is available via NCBI BioSample SAMN15817180. This is a new name for the alphanumeric GTDB species sp002160135. The GC content of the type genome is 50.79% and the genome length is 2.4 Mbp.	
Description of Candidatus Choladousia gen. nov.	
Candidatus Choladousia (Cho.lad.ou’si.a. Gr. fem. n. cholas guts; Gr. fem. n. ousia an essence; N.L. fem. n. Choladousia a microbe associated with the intestines)	
A bacterial genus identified by metagenomic analyses. The genus includes all bacteria with genomes that show ≥60% average amino acid identity (AAI) to the type genome from the type species Candidatus Choladousia intestinavium. This is a name for the alphanumeric GTDB genus UBA7160. This genus has been assigned by GTDB-Tk v1.3.0 working on GTDB Release 05-RS95 (Chaumeil et al., 2019; Parks et al., 2020) to the order Lachnospirales and to the family Lachnospiraceae.	
Description of Candidatus Choladousia intestinavium sp. nov.	
Candidatus Choladousia intestinavium (in.tes.tin.a’vi.um. L. neut. n. intestinum gut; L. fem. n. avis bird; N.L. gen. n. intestinavium of the gut of birds)	
A bacterial species identified by metagenomic analyses. This species includes all bacteria with genomes that show ≥95% average nucleotide identity (ANI) to the type genome, which has been assigned the MAG ID ChiSjej4B22-8148 and which is available via NCBI BioSample SAMN15817012. The GC content of the type genome is 49.57% and the genome length is 2.9 Mbp.	
Description of Candidatus Choladousia intestinigallinarum sp. nov.	
Candidatus Choladousia intestinigallinarum (in.tes.ti.ni.gal.li.na’rum. L. neut. n. intestinum gut; L. fem. n. gallina hen; N.L. gen. n. intestinigallinarum of the gut of the hens)	
A bacterial species identified by metagenomic analyses. This species includes all bacteria with genomes that show ≥95% average nucleotide identity (ANI) to the type genome, which has been assigned the MAG ID ChiBcec11-13528 and which is available via NCBI BioSample SAMN15817065. The GC content of the type genome is 48.69% and the genome length is 3.1 Mbp.	
Description of Candidatus Choladousia intestinipullorum sp. nov.	
Candidatus Choladousia intestinipullorum (in.tes.ti.ni.pul.lo’rum. L. neut. n. intestinum gut; L. masc. n. pullus a young chicken; N.L. gen. n. intestinipullorum of the gut of young chickens)	
A bacterial species identified by metagenomic analyses. This species includes all bacteria with genomes that show ≥95% average nucleotide identity (ANI) to the type genome, which has been assigned the MAG ID ChiSjej5B23-16397 and which is available via NCBI BioSample SAMN15817078. The GC content of the type genome is 50.54% and the genome length is 2.2 Mbp.	
Description of Candidatus Collinsella stercoripullorum sp. nov.	
Candidatus Collinsella stercoripullorum (ster.co.ri.pul.lo’rum. L. neut. n. stercus dung; L. masc. n. pullus a young chicken; N.L. gen. n. stercoripullorum of the faceces of young chickens)	
A bacterial species identified by metagenomic analyses. This species includes all bacteria with genomes that show ≥95% average nucleotide identity (ANI) to the type genome, which has been assigned the MAG ID ChiGjej6B6-20822 and which is available via NCBI BioSample SAMN15816681. The GC content of the type genome is 68.61% and the genome length is 2.3 Mbp.	
Description of Candidatus Companilactobacillus pullicola sp. nov.	
Candidatus Companilactobacillus pullicola (pul.li’co.la. L. masc. n. pullus a young chicken; L. suff. -cola inhabitant of; N.L. n. pullicola an inhabitant of young chickens)	
A bacterial species identified by metagenomic analyses. This species includes all bacteria with genomes that show ≥95% average nucleotide identity (ANI) to the type genome, which has been assigned the MAG ID 3204 and which is available via NCBI BioSample SAMN15816700. The GC content of the type genome is 35.87% and the genome length is 2.9 Mbp.	
Description of Candidatus Coprenecus gen. nov.	
Candidatus Coprenecus (Copr.en.e’cus. Gr. fem. n. kopros dung; Gr. masc. enoikos inhabitant; N.L. masc. n. Coprenecus a microbe associated with faeces)	
A bacterial genus identified by metagenomic analyses. The genus includes all bacteria with genomes that show ≥60% average amino acid identity (AAI) to the type genome from the type species Candidatus Coprenecus pullicola. This is a name for the alphanumeric GTDB genus CAG-831. This genus has been assigned by GTDB-Tk v1.3.0 working on GTDB Release 05-RS95 (Chaumeil et al., 2019; Parks et al., 2020) to the order Bacteroidales and to the family UBA932.	

Description of Candidatus Coprenecus avistercoris sp. nov.	
Candidatus Coprenecus avistercoris (a.vi.ster’co.ris. L. fem. n. avis bird; L. neut. n. stercus dung; N.L. gen. n. avistercoris of bird faeces)	
A bacterial species identified by metagenomic analyses. This species includes all bacteria with genomes that show ≥95% average nucleotide identity (ANI) to the type genome, which has been assigned the MAG ID ChiHjej13B12-12457 and which is available via NCBI BioSample SAMN15817175. This is a new name for the alphanumeric GTDB species sp000432775. The GC content of the type genome is 56.44% and the genome length is 2.0 Mbp.	
Description of Candidatus Coprenecus merdigallinarum sp. nov.	
Candidatus Coprenecus merdigallinarum (mer.di.gal.li.na’rum. L. fem. n. merda faeces; L. fem. n. gallina hen; N.L. gen. n. merdigallinarum of hen faeces)	
A bacterial species identified by metagenomic analyses. This species includes all bacteria with genomes that show ≥95% average nucleotide identity (ANI) to the type genome, which has been assigned the MAG ID ChiHecolR1B25-18470 and which is available via NCBI BioSample SAMN15817066. The GC content of the type genome is 55.94% and the genome length is 2.2 Mbp.	
Description of Candidatus Coprenecus merdipullorum sp. nov.	
Candidatus Coprenecus merdipullorum (mer.di.pul.lo’rum. L. fem. n. merda faeces; L. masc. n. pullus a young chicken; N.L. gen. n. merdipullorum of the faeces of young chickens)	
A bacterial species identified by metagenomic analyses. This species includes all bacteria with genomes that show ≥95% average nucleotide identity (ANI) to the type genome, which has been assigned the MAG ID Gambia11-1358 and which is available via NCBI BioSample SAMN15817068. The GC content of the type genome is 54.29% and the genome length is 2.1 Mbp.	
Description of Candidatus Coprenecus pullicola sp. nov.	
Candidatus Coprenecus pullicola (pul.li’co.la. L. masc. n. pullus a young chicken; L. suff. -cola inhabitant of; N.L. n. pullicola an inhabitant of young chickens)	
A bacterial species identified by metagenomic analyses. This species includes all bacteria with genomes that show ≥95% average nucleotide identity (ANI) to the type genome, which has been assigned the MAG ID ChiHjej9B8-15444 and which is available via NCBI BioSample SAMN15817077. The GC content of the type genome is 52.53% and the genome length is 2.1 Mbp.	
Description of Candidatus Coprenecus pullistercoris sp. nov.	
Candidatus Coprenecus pullistercoris (pul.li.ster’co.ris. L. masc. n. pullus a young chicken; L. neut. n. stercus dung; N.L. gen. n. pullistercoris of young chicken faeces)	
A bacterial species identified by metagenomic analyses. This species includes all bacteria with genomes that show ≥95% average nucleotide identity (ANI) to the type genome, which has been assigned the MAG ID Gambia18-42 and which is available via NCBI BioSample SAMN15817084. The GC content of the type genome is 52.64% and the genome length is 2.0 Mbp.	
Description of Candidatus Coprenecus stercoravium sp. nov.	
Candidatus Coprenecus stercoravium (ster.cor.a’vi.um. L. neut. n. stercus dung; L. fem. n. avis bird; N.L. gen. n. stercoravium of bird faeces)	
A bacterial species identified by metagenomic analyses. This species includes all bacteria with genomes that show ≥95% average nucleotide identity (ANI) to the type genome, which has been assigned the MAG ID Gambia16-554 and which is available via NCBI BioSample SAMN15817085. The GC content of the type genome is 51.84% and the genome length is 1.8 Mbp.	
Description of Candidatus Coprenecus stercorigallinarum sp. nov.	
Candidatus Coprenecus stercorigallinarum (ster.co.ri.gal.li.na’rum. L. neut. n. stercus dung; L. fem. n. gallina hen; N.L. gen. n. stercorigallinarum of hen faeces)	
A bacterial species identified by metagenomic analyses. This species includes all bacteria with genomes that show ≥95% average nucleotide identity (ANI) to the type genome, which has been assigned the MAG ID 3382 and which is available via NCBI BioSample SAMN15817086. The GC content of the type genome is 53.66% and the genome length is 2.1 Mbp.	
Description of Candidatus Coprenecus stercoripullorum sp. nov.	
Candidatus Coprenecus stercoripullorum (ster.co.ri.pul.lo’rum. L. neut. n. stercus dung; L. masc. n. pullus a young chicken; N.L. gen. n. stercoripullorum of the faceces of young chickens)	
A bacterial species identified by metagenomic analyses. This species includes all bacteria with genomes that show ≥95% average nucleotide identity (ANI) to the type genome, which has been assigned the MAG ID 7141 and which is available via NCBI BioSample SAMN15817115. The GC content of the type genome is 50.92% and the genome length is 1.7 Mbp.	

Description of Candidatus Coprocola gen. nov.	
Candidatus Coprocola (Co.pro’co.la. Gr. fem. n. kopros dung; L. suff. -cola inhabitant of; N.L. fem. n. Coprocola a microbe associated with faeces)	
A bacterial genus identified by metagenomic analyses. The genus includes all bacteria with genomes that show ≥60% average amino acid identity (AAI) to the type genome from the type species Candidatus Coprocola pullicola. This is a name for the alphanumeric GTDB genus ASF356. This genus has been assigned by GTDB-Tk v1.3.0 working on GTDB Release 05-RS95 (Chaumeil et al., 2019; Parks et al., 2020) to the order Lachnospirales and to the family Anaerotignaceae.	
Description of Candidatus Coprocola pullicola sp. nov.	
Candidatus Coprocola pullicola (pul.li’co.la. L. masc. n. pullus a young chicken; L. suff. -cola inhabitant of; N.L. n. pullicola an inhabitant of young chickens)	
A bacterial species identified by metagenomic analyses. This species includes all bacteria with genomes that show ≥95% average nucleotide identity (ANI) to the type genome, which has been assigned the MAG ID CHK193-15662 and which is available via NCBI BioSample SAMN15817043. The GC content of the type genome is 33.91% and the genome length is 3.0 Mbp.	
Description of Candidatus Copromonas gen. nov.	
Candidatus Copromonas (Co.pro.mo’nas. Gr. fem. n. kopros dung; L. fem. n. monas a monad; N.L. fem. n. Copromonas a microbe associated with faeces)	
A bacterial genus identified by metagenomic analyses. The genus includes all bacteria with genomes that show ≥60% average amino acid identity (AAI) to the type genome from the type species Candidatus Copromonas avistercoris. This is a name for the alphanumeric GTDB genus CAG-81. This genus has been assigned by GTDB-Tk v1.3.0 working on GTDB Release 05-RS95 (Chaumeil et al., 2019; Parks et al., 2020) to the order Lachnospirales and to the family Lachnospiraceae.	
Description of Candidatus Copromonas avistercoris sp. nov.	
Candidatus Copromonas avistercoris (a.vi.ster’co.ris. L. fem. n. avis bird; L. neut. n. stercus dung; N.L. gen. n. avistercoris of bird faeces)	
A bacterial species identified by metagenomic analyses. This species includes all bacteria with genomes that show ≥95% average nucleotide identity (ANI) to the type genome, which has been assigned the MAG ID ChiSjej3B21-5768 and which is available via NCBI BioSample SAMN15817005. The GC content of the type genome is 52.88% and the genome length is 2.2 Mbp.	
Description of Candidatus Copromonas faecavium sp. nov.	
Candidatus Copromonas faecavium (faec.a’vi.um. L. fem. n. faex, faecis excrement; L. fem. n. avis bird; N.L. gen. n. faecavium of bird faeces)	
A bacterial species identified by metagenomic analyses. This species includes all bacteria with genomes that show ≥95% average nucleotide identity (ANI) to the type genome, which has been assigned the MAG ID CHK180-2868 and which is available via NCBI BioSample SAMN15817009. The GC content of the type genome is 50.20% and the genome length is 2.7 Mbp.	
Description of Candidatus Copromorpha gen. nov.	
Candidatus Copromorpha (Co.pro.mor’pha. Gr. fem. n. kopros dung; Gr. fem. n. morphe a form, shape; N.L. fem. n. Copromorpha a microbe associated with faeces)	
A bacterial genus identified by metagenomic analyses. The genus includes all bacteria with genomes that show ≥60% average amino acid identity (AAI) to the type genome from the type species Candidatus Copromorpha excrementavium. This is a name for the alphanumeric GTDB genus UBA1191. This genus has been assigned by GTDB-Tk v1.3.0 working on GTDB Release 05-RS95 (Chaumeil et al., 2019; Parks et al., 2020) to the order Peptostreptococcales and to the family Anaerovoracaceae.	
Description of Candidatus Copromorpha excrementavium sp. nov.	
Candidatus Copromorpha excrementavium (ex.cre.ment.a’vi.um. L. neut. n. excrementum excrement; L. fem. n. avis bird; N.L. gen. n. excrementavium of bird excrement)	
A bacterial species identified by metagenomic analyses. This species includes all bacteria with genomes that show ≥95% average nucleotide identity (ANI) to the type genome, which has been assigned the MAG ID CHK176-22527 and which is available via NCBI BioSample SAMN15817193. This is a new name for the alphanumeric GTDB species sp900542385. The GC content of the type genome is 42.88% and the genome length is 1.8 Mbp.	
Description of Candidatus Copromorpha excrementigallinarum sp. nov.	
Candidatus Copromorpha excrementigallinarum (ex.cre.men.ti.gal.li.na’rum. L. neut. n. excrementum excrement; L. fem. n. gallina hen; N.L. gen. n. excrementigallinarum of hen excrement)	
A bacterial species identified by metagenomic analyses. This species includes all bacteria with genomes that show ≥95% average nucleotide identity (ANI) to the type genome, which has been assigned the MAG ID ChiHcec3-6078 and which is available via NCBI BioSample SAMN15817131. The GC content of the type genome is 48.15% and the genome length is 2.0 Mbp.	

Description of Candidatus Copromorpha excrementipullorum sp. nov.	
Candidatus Copromorpha excrementipullorum (ex.cre.men.ti.pul.lo’rum. L. neut. n. excrementum excrement; L. masc. n. pullus a young chicken; N.L. gen. n. excrementipullorum of young chicken excrement)	
A bacterial species identified by metagenomic analyses. This species includes all bacteria with genomes that show ≥95% average nucleotide identity (ANI) to the type genome, which has been assigned the MAG ID ChiSjej4B22-8349 and which is available via NCBI BioSample SAMN15817205. This is a new name for the alphanumeric GTDB species sp900543485. The GC content of the type genome is 49.77% and the genome length is 1.9 Mbp.	
Description of Candidatus Coproplasma gen. nov.	
Candidatus Coproplasma (Co.pro.plas’ma. Gr. fem. n. kopros dung; Gr. neut. n. plasma a form; N.L. neut. n. Coproplasma a microbe associated with faeces)	
A bacterial genus identified by metagenomic analyses. The genus includes all bacteria with genomes that show ≥60% average amino acid identity (AAI) to the type genome from the type species Candidatus Coproplasma stercoravium. This is a name for the alphanumeric GTDB genus UBA11940. This genus has been assigned by GTDB-Tk v1.3.0 working on GTDB Release 05-RS95 (Chaumeil et al., 2019; Parks et al., 2020) to the order Christensenellales and to the family Borkfalkiaceae.	
Description of Candidatus Coproplasma avicola sp. nov.	
Candidatus Coproplasma avicola (a.vi’co.la. L. fem. n. avis bird; L. suff. -cola inhabitant of; N.L. n. avicola inhabitant of birds)	
A bacterial species identified by metagenomic analyses. This species includes all bacteria with genomes that show ≥95% average nucleotide identity (ANI) to the type genome, which has been assigned the MAG ID ChiW16-3235 and which is available via NCBI BioSample SAMN15817075. The GC content of the type genome is 51.11% and the genome length is 1.5 Mbp.	
Description of Candidatus Coproplasma avistercoris sp. nov.	
Candidatus Coproplasma avistercoris (a.vi.ster’co.ris. L. fem. n. avis bird; L. neut. n. stercus dung; N.L. gen. n. avistercoris of bird faeces)	
A bacterial species identified by metagenomic analyses. This species includes all bacteria with genomes that show ≥95% average nucleotide identity (ANI) to the type genome, which has been assigned the MAG ID ChiW7-3743 and which is available via NCBI BioSample SAMN15817023. The GC content of the type genome is 56.32% and the genome length is 1.4 Mbp.	
Description of Candidatus Coproplasma excrementavium sp. nov.	
Candidatus Coproplasma excrementavium (ex.cre.ment.a’vi.um. L. neut. n. excrementum excrement; L. fem. n. avis bird; N.L. gen. n. excrementavium of bird excrement)	
A bacterial species identified by metagenomic analyses. This species includes all bacteria with genomes that show ≥95% average nucleotide identity (ANI) to the type genome, which has been assigned the MAG ID CHK179-18245 and which is available via NCBI BioSample SAMN15817045. The GC content of the type genome is 50.26% and the genome length is 1.4 Mbp.	
Description of Candidatus Coproplasma excrementigallinarum sp. nov.	
Candidatus Coproplasma excrementigallinarum (ex.cre.men.ti.gal.li.na’rum. L. neut. n. excrementum excrement; L. fem. n. gallina hen; N.L. gen. n. excrementigallinarum of hen excrement)	
A bacterial species identified by metagenomic analyses. This species includes all bacteria with genomes that show ≥95% average nucleotide identity (ANI) to the type genome, which has been assigned the MAG ID CHK195-12923 and which is available via NCBI BioSample SAMN15817050. The GC content of the type genome is 50.00% and the genome length is 1.4 Mbp.	
Description of Candidatus Coproplasma excrementipullorum sp. nov.	
Candidatus Coproplasma excrementipullorum (ex.cre.men.ti.pul.lo’rum. L. neut. n. excrementum excrement; L. masc. n. pullus a young chicken; N.L. gen. n. excrementipullorum of young chicken excrement)	
A bacterial species identified by metagenomic analyses. This species includes all bacteria with genomes that show ≥95% average nucleotide identity (ANI) to the type genome, which has been assigned the MAG ID 10570 and which is available via NCBI BioSample SAMN15817148. The GC content of the type genome is 51.24% and the genome length is 1.7 Mbp.	
Description of Candidatus Coproplasma stercoravium sp. nov.	
Candidatus Coproplasma stercoravium (ster.cor.a’vi.um. L. neut. n. stercus dung; L. fem. n. avis bird; N.L. gen. n. stercoravium of bird faeces)	
A bacterial species identified by metagenomic analyses. This species includes all bacteria with genomes that show ≥95% average nucleotide identity (ANI) to the type genome, which has been assigned the MAG ID CHK180-19203 and which is available via NCBI BioSample SAMN15817006. The GC content of the type genome is 51.86% and the genome length is 1.6 Mbp.	

Description of Candidatus Coproplasma stercorigallinarum sp. nov.	
Candidatus Coproplasma stercorigallinarum (ster.co.ri.gal.li.na’rum. L. neut. n. stercus dung; L. fem. n. gallina hen; N.L. gen. n. stercorigallinarum of hen faeces)	
A bacterial species identified by metagenomic analyses. This species includes all bacteria with genomes that show ≥95% average nucleotide identity (ANI) to the type genome, which has been assigned the MAG ID CHK191-24566 and which is available via NCBI BioSample SAMN15817171. This is a new name for the alphanumeric GTDB species sp900549005. The GC content of the type genome is 51.83% and the genome length is 1.6 Mbp.	
Description of Candidatus Coproplasma stercoripullorum sp. nov.	
Candidatus Coproplasma stercoripullorum (ster.co.ri.pul.lo’rum. L. neut. n. stercus dung; L. masc. n. pullus a young chicken; N.L. gen. n. stercoripullorum of the faceces of young chickens)	
A bacterial species identified by metagenomic analyses. This species includes all bacteria with genomes that show ≥95% average nucleotide identity (ANI) to the type genome, which has been assigned the MAG ID ChiW25-3613 and which is available via NCBI BioSample SAMN15817071. The GC content of the type genome is 51.17% and the genome length is 1.7 Mbp.	
Description of Candidatus Coprosoma gen. nov.	
Candidatus Coprosoma (Co.pro.so’ma. Gr. fem. n. kopros dung; Gr. neut. n. soma a body; N.L. neut. n. Coprosoma a microbe associated with faeces)	
A bacterial genus identified by metagenomic analyses. The genus includes all bacteria with genomes that show ≥60% average amino acid identity (AAI) to the type genome from the type species Candidatus Coprosoma intestinipullorum. This is a name for the alphanumeric GTDB genus CAG-822. This genus has been assigned by GTDB-Tk v1.3.0 working on GTDB Release 05-RS95 (Chaumeil et al., 2019; Parks et al., 2020) to the order RF39 and to the family CAG-822.	
Description of Candidatus Coprosoma intestinipullorum sp. nov.	
Candidatus Coprosoma intestinipullorum (in.tes.ti.ni.pul.lo’rum. L. neut. n. intestinum gut; L. masc. n. pullus a young chicken; N.L. gen. n. intestinipullorum of the gut of young chickens)	
A bacterial species identified by metagenomic analyses. This species includes all bacteria with genomes that show ≥95% average nucleotide identity (ANI) to the type genome, which has been assigned the MAG ID CHK147-3167 and which is available via NCBI BioSample SAMN15817008. The GC content of the type genome is 31.20% and the genome length is 1.3 Mbp.	
Description of Candidatus Coprousia gen. nov.	
Candidatus Coprousia (Copr.ou’si.a. Gr. fem. n. kopros dung; Gr. fem. n. ousia an essence; N.L. fem. n. Coprousia a microbe associated with faeces)	
A bacterial genus identified by metagenomic analyses. The genus includes all bacteria with genomes that show ≥60% average amino acid identity (AAI) to the type genome from the type species Candidatus Coprousia avicola. This is a name for the alphanumeric GTDB genus An7. This genus has been assigned by GTDB-Tk v1.3.0 working on GTDB Release 05-RS95 (Chaumeil et al., 2019; Parks et al., 2020) to the order Coriobacteriales and to the family Coriobacteriaceae.	
Description of Candidatus Coprousia avicola sp. nov.	
Candidatus Coprousia avicola (a.vi’co.la. L. fem. n. avis bird; L. suff. -cola inhabitant of; N.L. n. avicola inhabitant of birds)	
A bacterial species identified by metagenomic analyses. This species includes all bacteria with genomes that show ≥95% average nucleotide identity (ANI) to the type genome, which has been assigned the MAG ID ChiHjej11B10-5566 and which is available via NCBI BioSample SAMN15817195. This is a new name for the alphanumeric GTDB species sp002159765. The GC content of the type genome is 65.47% and the genome length is 2.4 Mbp.	
Description of Candidatus Coprovicinus gen. nov.	
Candidatus Coprovicinus (Co.pro.vi.ci’nus. Gr. fem. n. kopros dung; L. masc. n. vicinus a neighbour; N.L. masc. n. Coprovicinus a microbe associated with faeces)	
A bacterial genus identified by metagenomic analyses. The genus includes all bacteria with genomes that show ≥60% average amino acid identity (AAI) to the type genome from the type species Candidatus Coprovicinus avistercoris. This is a name for the alphanumeric GTDB genus UMGS1418. This genus has been assigned by GTDB-Tk v1.3.0 working on GTDB Release 05-RS95 (Chaumeil et al., 2019; Parks et al., 2020) to the order Coriobacteriales and to the family Atopobiaceae.	
Description of Candidatus Coprovicinus avistercoris sp. nov.	
Candidatus Coprovicinus avistercoris (a.vi.ster’co.ris. L. fem. n. avis bird; L. neut. n. stercus dung; N.L. gen. n. avistercoris of bird faeces)	
A bacterial species identified by metagenomic analyses. This species includes all bacteria with genomes that show ≥95% average nucleotide identity (ANI) to the type genome, which has been assigned the MAG ID ChiHjej12B11-29160 and which is available via NCBI BioSample SAMN15817198. This is a new name for the alphanumeric GTDB species sp900551595. The GC content of the type genome is 49.51% and the genome length is 1.7 Mbp.	

Description of Candidatus Coprovivens gen. nov.	
Candidatus Coprovivens (Co.pro.vi’vens. Gr. fem. n. kopros dung; N.L. pres. part. vivens living; N.L. fem. n. Coprovivens a microbe associated with faeces)	
A bacterial genus identified by metagenomic analyses. The genus includes all bacteria with genomes that show ≥60% average amino acid identity (AAI) to the type genome from the type species Candidatus Coprovivens excrementavium. This is a name for the alphanumeric GTDB genus UBA11963. This genus has been assigned by GTDB-Tk v1.3.0 working on GTDB Release 05-RS95 (Chaumeil et al., 2019; Parks et al., 2020) to the order RF39 and to the family CAG-1000.	
Description of Candidatus Coprovivens excrementavium sp. nov.	
Candidatus Coprovivens excrementavium (ex.cre.ment.a’vi.um. L. neut. n. excrementum excrement; L. fem. n. avis bird; N.L. gen. n. excrementavium of bird excrement)	
A bacterial species identified by metagenomic analyses. This species includes all bacteria with genomes that show ≥95% average nucleotide identity (ANI) to the type genome, which has been assigned the MAG ID 3297 and which is available via NCBI BioSample SAMN15817083. The GC content of the type genome is 28.25% and the genome length is 2.2 Mbp.	
Description of Candidatus Corynebacterium avicola sp. nov.	
Candidatus Corynebacterium avicola (a.vi’co.la. L. fem. n. avis bird; L. suff. -cola inhabitant of; N.L. n. avicola inhabitant of birds)	
A bacterial species identified by metagenomic analyses. This species includes all bacteria with genomes that show ≥95% average nucleotide identity (ANI) to the type genome, which has been assigned the MAG ID CHK32-1732 and which is available via NCBI BioSample SAMN15816750. The GC content of the type genome is 66.95% and the genome length is 3.1 Mbp.	
Description of Candidatus Corynebacterium faecigallinarum sp. nov.	
Candidatus Corynebacterium faecigallinarum (fae.ci.gal.li.na’rum. L. fem. n. faex, faecis excrement; L. fem. n. gallina hen; N.L. gen. n. faecigallinarum of hen faeces)	
A bacterial species identified by metagenomic analyses. This species includes all bacteria with genomes that show ≥95% average nucleotide identity (ANI) to the type genome, which has been assigned the MAG ID ChiHjej13B12-4958 and which is available via NCBI BioSample SAMN15816631. The GC content of the type genome is 66.91% and the genome length is 2.8 Mbp.	
Description of Candidatus Corynebacterium faecipullorum sp. nov.	
Candidatus Corynebacterium faecipullorum (fae.ci.pul.lo’rum. L. fem. n. faex, faecis excrement; L. masc. n. pullus a young chicken; N.L. gen. n. faecipullorum of young chicken faeces)	
A bacterial species identified by metagenomic analyses. This species includes all bacteria with genomes that show ≥95% average nucleotide identity (ANI) to the type genome, which has been assigned the MAG ID 913 and which is available via NCBI BioSample SAMN15816858. This is a new name for the alphanumeric GTDB species sp001836165. The GC content of the type genome is 61.16% and the genome length is 2.1 Mbp.	
Description of Candidatus Corynebacterium gallistercoris sp. nov.	
Candidatus Corynebacterium gallistercoris (gal.li.ster’co.ris. L. masc. n. gallus chicken; L. neut. n. stercus dung; N.L. gen. n. gallistercoris of chicken faeces)	
A bacterial species identified by metagenomic analyses. This species includes all bacteria with genomes that show ≥95% average nucleotide identity (ANI) to the type genome, which has been assigned the MAG ID 4376 and which is available via NCBI BioSample SAMN15816747. The GC content of the type genome is 62.96% and the genome length is 2.0 Mbp.	
Description of Candidatus Corynebacterium intestinavium sp. nov.	
Candidatus Corynebacterium intestinavium (in.tes.tin.a’vi.um. L. neut. n. intestinum gut; L. fem. n. avis bird; N.L. gen. n. intestinavium of the gut of birds)	
A bacterial species identified by metagenomic analyses. This species includes all bacteria with genomes that show ≥95% average nucleotide identity (ANI) to the type genome, which has been assigned the MAG ID 5925 and which is available via NCBI BioSample SAMN15816787. The GC content of the type genome is 65.59% and the genome length is 1.9 Mbp.	
Description of Candidatus Cottocaccomicrobium gen. nov.	
Candidatus Cottocaccomicrobium (Cot.to.cac.co.mi.cro’bi.um. Gr. masc. n. kottos chicken; Gr. fem. n. kakke faeces; N.L. neut. n. microbium a microbe; N.L. neut. n. Cottocaccomicrobium a microbe asociated with chicken faeces)	
A bacterial genus identified by metagenomic analyses. The genus includes all bacteria with genomes that show ≥60% average amino acid identity (AAI) to the type genome from the type species Candidatus Cottocaccamicrobium excrementipullorum. This genus was identified but not named by Glendinning et al. (2020). This genus has been assigned by GTDB-Tk v1.3.0 working on GTDB Release 05-RS95 (Chaumeil et al., 2019; Parks et al., 2020) to the order Lachnospirales and to the family Lachnospiraceae.	

Description of Candidatus Cottocaccomicrobium excrementipullorum sp. nov.	
Candidatus Cottocaccomicrobium excrementipullorum (ex.cre.men.ti.pul.lo’rum. L. neut. n. excrementum excrement; L. masc. n. pullus a young chicken; N.L. gen. n. excrementipullorum of young chicken excrement)	
A bacterial species identified by metagenomic analyses. This species includes all bacteria with genomes that show ≥95% average nucleotide identity (ANI) to the type genome, which has been assigned the MAG ID CHK179-5732 and which is available via NCBI BioSample SAMN15816932. The GC content of the type genome is 47.54% and the genome length is 3.4 Mbp.	
Description of Candidatus Cryptobacteroides gen. nov.	
Candidatus Cryptobacteroides (Cryp.to.bac.te.ro’i.des. Gr. masc. adj. kryptos hidden; N.L. masc. n. Bacteroides a genus name; N.L. masc. n. Cryptobacteroides a genus related to the genus Bacteroides but distinct from it)	
A bacterial genus identified by metagenomic analyses. The genus includes all bacteria with genomes that show ≥60% average amino acid identity (AAI) to the type genome from the type species Candidatus Cryptobacteroides avicola. This is a name for the alphanumeric GTDB genus RC9. This genus has been assigned by GTDB-Tk v1.3.0 working on GTDB Release 05-RS95 (Chaumeil et al., 2019; Parks et al., 2020) to the order Bacteroidales and to the family UBA932.	
Description of Candidatus Cryptobacteroides avicola sp. nov.	
Candidatus Cryptobacteroides avicola (a.vi’co.la. L. fem. n. avis bird; L. suff. -cola inhabitant of; N.L. n. avicola inhabitant of birds)	
A bacterial species identified by metagenomic analyses. This species includes all bacteria with genomes that show ≥95% average nucleotide identity (ANI) to the type genome, which has been assigned the MAG ID G3-8215 and which is available via NCBI BioSample SAMN15817056. The GC content of the type genome is 49.81% and the genome length is 2.6 Mbp.	
Description of Candidatus Cryptobacteroides avistercoris sp. nov.	
Candidatus Cryptobacteroides avistercoris (a.vi.ster’co.ris. L. fem. n. avis bird; L. neut. n. stercus dung; N.L. gen. n. avistercoris of bird faeces)	
A bacterial species identified by metagenomic analyses. This species includes all bacteria with genomes that show ≥95% average nucleotide identity (ANI) to the type genome, which has been assigned the MAG ID B3-1481 and which is available via NCBI BioSample SAMN15817057. The GC content of the type genome is 59.28% and the genome length is 1.6 Mbp.	
Description of Candidatus Cryptobacteroides excrementavium sp. nov.	
Candidatus Cryptobacteroides excrementavium (ex.cre.ment.a’vi.um. L. neut. n. excrementum excrement; L. fem. n. avis bird; N.L. gen. n. excrementavium of bird excrement)	
A bacterial species identified by metagenomic analyses. This species includes all bacteria with genomes that show ≥95% average nucleotide identity (ANI) to the type genome, which has been assigned the MAG ID B2-16538 and which is available via NCBI BioSample SAMN15817059. The GC content of the type genome is 50.29% and the genome length is 2.1 Mbp.	
Description of Candidatus Cryptobacteroides excrementigallinarum sp. nov.	
Candidatus Cryptobacteroides excrementigallinarum (ex.cre.men.ti.gal.li.na’rum. L. neut. n. excrementum excrement; L. fem. n. gallina hen; N.L. gen. n. excrementigallinarum of hen excrement)	
A bacterial species identified by metagenomic analyses. This species includes all bacteria with genomes that show ≥95% average nucleotide identity (ANI) to the type genome, which has been assigned the MAG ID ChiHecolR1B25-7735 and which is available via NCBI BioSample SAMN15817167. This is a new name for the alphanumeric GTDB species sp900543205. The GC content of the type genome is 57.80% and the genome length is 1.8 Mbp.	
Description of Candidatus Cryptobacteroides excrementipullorum sp. nov.	
Candidatus Cryptobacteroides excrementipullorum (ex.cre.men.ti.pul.lo’rum. L. neut. n. excrementum excrement; L. masc. gen. pl. n. pullorum of young chickens; N.L. gen. n. excrementipullorum of young chicken excrement)	
A bacterial species identified by metagenomic analyses. This species includes all bacteria with genomes that show ≥95% average nucleotide identity (ANI) to the type genome, which has been assigned the MAG ID 2478 and which is available via NCBI BioSample SAMN15817061. The GC content of the type genome is 52.35% and the genome length is 2.5 Mbp.	
Description of Candidatus Cryptobacteroides faecavium sp. nov.	
Candidatus Cryptobacteroides faecavium (faec.a’vi.um. L. fem. n. faex, faecis excrement; L. fem. n. avis bird; N.L. gen. n. faecavium of bird faeces)	
A bacterial species identified by metagenomic analyses. This species includes all bacteria with genomes that show ≥95% average nucleotide identity (ANI) to the type genome, which has been assigned the MAG ID B2-22910 and which is available via NCBI BioSample SAMN15817063. The GC content of the type genome is 52.41% and the genome length is 2.4 Mbp.	

Description of Candidatus Cryptobacteroides faecigallinarum sp. nov.	
Candidatus Cryptobacteroides faecigallinarum (fae.ci.gal.li.na’rum. L. fem. n. faex, faecis excrement; L. fem. n. gallina hen; N.L. gen. n. faecigallinarum of hen faeces)	
A bacterial species identified by metagenomic analyses. This species includes all bacteria with genomes that show ≥95% average nucleotide identity (ANI) to the type genome, which has been assigned the MAG ID B1-13419 and which is available via NCBI BioSample SAMN15817072. The GC content of the type genome is 49.87% and the genome length is 2.0 Mbp.	
Description of Candidatus Cryptobacteroides faecipullorum sp. nov.	
Candidatus Cryptobacteroides faecipullorum (fae.ci.pul.lo’rum. L. fem. n. faex, faecis excrement; L. masc. n. pullus a young chicken; N.L. gen. n. faecipullorum of young chicken faeces)	
A bacterial species identified by metagenomic analyses. This species includes all bacteria with genomes that show ≥95% average nucleotide identity (ANI) to the type genome, which has been assigned the MAG ID B1-15692 and which is available via NCBI BioSample SAMN15817080. The GC content of the type genome is 49.55% and the genome length is 2.3 Mbp.	
Description of Candidatus Cryptobacteroides gallistercoris sp. nov.	
Candidatus Cryptobacteroides gallistercoris (gal.li.ster’co.ris. L. masc. n gallus chicken; L. neut. n. stercus dung; N.L. gen. n. gallistercoris of chicken faeces)	
A bacterial species identified by metagenomic analyses. This species includes all bacteria with genomes that show ≥95% average nucleotide identity (ANI) to the type genome, which has been assigned the MAG ID F1-3629 and which is available via NCBI BioSample SAMN15817088. The GC content of the type genome is 51.79% and the genome length is 2.0 Mbp.	
Description of Candidatus Cryptobacteroides intestinavium sp. nov.	
Candidatus Cryptobacteroides intestinavium (in.tes.tin.a’vi.um. L. neut. n. intestinum gut; L. fem. n. avis bird; N.L. gen. n. intestinavium of the gut of birds)	
A bacterial species identified by metagenomic analyses. This species includes all bacteria with genomes that show ≥95% average nucleotide identity (ANI) to the type genome, which has been assigned the MAG ID B1-20833 and which is available via NCBI BioSample SAMN15817087. The GC content of the type genome is 51.51% and the genome length is 2.4 Mbp.	
Description of Candidatus Cryptobacteroides intestinigallinarum sp. nov.	
Candidatus Cryptobacteroides intestinigallinarum (in.tes.ti.ni.gal.li.na’rum. L. neut. n. intestinum gut; L. fem. n. gallina hen; N.L. gen. n. intestinigallinarum of the gut of the hens)	
A bacterial species identified by metagenomic analyses. This species includes all bacteria with genomes that show ≥95% average nucleotide identity (ANI) to the type genome, which has been assigned the MAG ID B1-3475 and which is available via NCBI BioSample SAMN15817089. The GC content of the type genome is 49.55% and the genome length is 2.3 Mbp.	
Description of Candidatus Cryptobacteroides intestinipullorum sp. nov.	
Candidatus Cryptobacteroides intestinipullorum (in.tes.ti.ni.pul.lo’rum. L. neut. n. intestinum gut; L. masc. n. pullus a young chicken; N.L. gen. n. intestinipullorum of the gut of young chickens)	
A bacterial species identified by metagenomic analyses. This species includes all bacteria with genomes that show ≥95% average nucleotide identity (ANI) to the type genome, which has been assigned the MAG ID 33258 and which is available via NCBI BioSample SAMN15817090. The GC content of the type genome is 50.60% and the genome length is 2.4 Mbp.	
Description of Candidatus Cryptobacteroides merdavium sp. nov.	
Candidatus Cryptobacteroides merdavium (merd.a’vi.um. L. fem. n. merda faeces; L. fem. n. avis bird; N.L. gen. n. merdavium of bird faeces)	
A bacterial species identified by metagenomic analyses. This species includes all bacteria with genomes that show ≥95% average nucleotide identity (ANI) to the type genome, which has been assigned the MAG ID D5-748 and which is available via NCBI BioSample SAMN15817104. The GC content of the type genome is 50.98% and the genome length is 2.5 Mbp.	
Description of Candidatus Cryptobacteroides merdigallinarum sp. nov.	
Candidatus Cryptobacteroides merdigallinarum (mer.di.gal.li.na’rum. L. fem. n. merda faeces; L. fem. n. gallina hen; N.L. gen. n. merdigallinarum of hen faeces)	
A bacterial species identified by metagenomic analyses. This species includes all bacteria with genomes that show ≥95% average nucleotide identity (ANI) to the type genome, which has been assigned the MAG ID 20514 and which is available via NCBI BioSample SAMN15817110. The GC content of the type genome is 54.69% and the genome length is 2.3 Mbp.	

Description of Candidatus Cryptobacteroides merdipullorum sp. nov.	
Candidatus Cryptobacteroides merdipullorum (mer.di.pul.lo’rum. L. fem. n. merda faeces; L. masc. n. pullus a young chicken; N.L. gen. n. merdipullorum of the faeces of young chickens)	
A bacterial species identified by metagenomic analyses. This species includes all bacteria with genomes that show ≥95% average nucleotide identity (ANI) to the type genome, which has been assigned the MAG ID ChiHecec2B26-709 and which is available via NCBI BioSample SAMN15817116. The GC content of the type genome is 57.16% and the genome length is 2.0 Mbp.	
Description of Candidatus Cryptobacteroides pullicola sp. nov.	
Candidatus Cryptobacteroides pullicola (pul.li’co.la. L. masc. n. pullus a young chicken; L. suff. -cola inhabitant of; N.L. n. pullicola an inhabitant of young chickens)	
A bacterial species identified by metagenomic analyses. This species includes all bacteria with genomes that show ≥95% average nucleotide identity (ANI) to the type genome, which has been assigned the MAG ID ChiHecec2B26-3624 and which is available via NCBI BioSample SAMN15817179. This is a new name for the alphanumeric GTDB species sp001915575. The GC content of the type genome is 58.01% and the genome length is 1.7 Mbp.	
Description of Candidatus Desulfovibrio faecigallinarum sp. nov.	
Candidatus Desulfovibrio faecigallinarum (fae.ci.gal.li.na’rum. L. fem. n. faex, faecis excrement; L. fem. n. gallina hen; N.L. gen. n. faecigallinarum of hen faeces)	
A bacterial species identified by metagenomic analyses. This species includes all bacteria with genomes that show ≥95% average nucleotide identity (ANI) to the type genome, which has been assigned the MAG ID 8923 and which is available via NCBI BioSample SAMN15816873. This is a new name for the alphanumeric GTDB species sp002159665. The GC content of the type genome is 57.29% and the genome length is 2.0 Mbp.	
Description of Candidatus Desulfovibrio gallistercoris sp. nov.	
Candidatus Desulfovibrio gallistercoris (gal.li.ster’co.ris. L. masc. n. gallus chicken; L. neut. n. stercus dung; N.L. gen. n. gallistercoris of chicken faeces)	
A bacterial species identified by metagenomic analyses. This species includes all bacteria with genomes that show ≥95% average nucleotide identity (ANI) to the type genome, which has been assigned the MAG ID ChiGjej2B2-32749 and which is available via NCBI BioSample SAMN15816654. The GC content of the type genome is 64.54% and the genome length is 2.8 Mbp.	
Description of Candidatus Desulfovibrio intestinavium sp. nov.	
Candidatus Desulfovibrio intestinavium (in.tes.tin.a’vi.um. L. neut. n. intestinum gut; L. fem. n. avis bird; N.L. gen. n. intestinavium of the gut of birds)	
A bacterial species identified by metagenomic analyses. This species includes all bacteria with genomes that show ≥95% average nucleotide identity (ANI) to the type genome, which has been assigned the MAG ID 5032 and which is available via NCBI BioSample SAMN15816664. The GC content of the type genome is 64.60% and the genome length is 2.5 Mbp.	
Description of Candidatus Desulfovibrio intestinigallinarum sp. nov.	
Candidatus Desulfovibrio intestinigallinarum (in.tes.ti.ni.gal.li.na’rum. L. neut. n. intestinum gut; L. fem. n. gallina hen; N.L. gen. n. intestinigallinarum of the gut of the hens)	
A bacterial species identified by metagenomic analyses. This species includes all bacteria with genomes that show ≥95% average nucleotide identity (ANI) to the type genome, which has been assigned the MAG ID ChiHecec3B27-2601 and which is available via NCBI BioSample SAMN15816737. The GC content of the type genome is 61.03% and the genome length is 2.9 Mbp.	
Description of Candidatus Desulfovibrio intestinipullorum sp. nov.	
Candidatus Desulfovibrio intestinipullorum (in.tes.ti.ni.pul.lo’rum. L. neut. n. intestinum gut; L. masc. n. pullus a young chicken; N.L. gen. n. intestinipullorum of the gut of young chickens)	
A bacterial species identified by metagenomic analyses. This species includes all bacteria with genomes that show ≥95% average nucleotide identity (ANI) to the type genome, which has been assigned the MAG ID ChiHecec2B26-446 and which is available via NCBI BioSample SAMN15816774. The GC content of the type genome is 60.23% and the genome length is 2.8 Mbp.	
Description of Candidatus Dietzia intestinigallinarum sp. nov.	
Candidatus Dietzia intestinigallinarum (in.tes.ti.ni.gal.li.na’rum. L. neut. n. intestinum gut; L. fem. n. gallina hen; N.L. gen. n. intestinigallinarum of the gut of the hens)	
A bacterial species identified by metagenomic analyses. This species includes all bacteria with genomes that show ≥95% average nucleotide identity (ANI) to the type genome, which has been assigned the MAG ID ChiHjej12B11-1528 and which is available via NCBI BioSample SAMN15816635. The GC content of the type genome is 69.86% and the genome length is 3.9 Mbp.	

Description of Candidatus Dietzia intestinipullorum sp. nov.	
Candidatus Dietzia intestinipullorum (in.tes.ti.ni.pul.lo’rum. L. neut. n. intestinum gut; L. masc. n. pullus a young chicken; N.L. gen. n. intestinipullorum of the gut of young chickens)	
A bacterial species identified by metagenomic analyses. This species includes all bacteria with genomes that show ≥95% average nucleotide identity (ANI) to the type genome, which has been assigned the MAG ID ChiHjej13B12-8321 and which is available via NCBI BioSample SAMN15816639. The GC content of the type genome is 71.26% and the genome length is 3.0 Mbp.	
Description of Candidatus Dietzia merdigallinarum sp. nov.	
Candidatus Dietzia merdigallinarum (mer.di.gal.li.na’rum. L. fem. n. merda faeces; L. fem. n. gallina hen; N.L. gen. n. merdigallinarum of hen faeces)	
A bacterial species identified by metagenomic analyses. This species includes all bacteria with genomes that show ≥95% average nucleotide identity (ANI) to the type genome, which has been assigned the MAG ID ChiHjej8B7-16427 and which is available via NCBI BioSample SAMN15816758. The GC content of the type genome is 68.76% and the genome length is 3.6 Mbp.	
Description of Candidatus Dorea faecigallinarum sp. nov.	
Candidatus Dorea faecigallinarum (fae.ci.gal.li.na’rum. L. fem. n. faex, faecis excrement; L. fem. n. gallina hen; N.L. gen. n. faecigallinarum of hen faeces)	
A bacterial species identified by metagenomic analyses. This species includes all bacteria with genomes that show ≥95% average nucleotide identity (ANI) to the type genome, which has been assigned the MAG ID ChiHjej12B11-29902 and which is available via NCBI BioSample SAMN15816646. The GC content of the type genome is 50.13% and the genome length is 2.0 Mbp.	
Description of Candidatus Dorea faecipullorum sp. nov.	
Candidatus Dorea faecipullorum (fae.ci.pul.lo’rum. L. fem. n. faex, faecis excrement; L. masc. n. pullus a young chicken; N.L. gen. n. faecipullorum of young chicken faeces)	
A bacterial species identified by metagenomic analyses. This species includes all bacteria with genomes that show ≥95% average nucleotide identity (ANI) to the type genome, which has been assigned the MAG ID ChiGjej2B2-10896 and which is available via NCBI BioSample SAMN15816847. This is a new name for the alphanumeric GTDB species sp900543315. The GC content of the type genome is 45.40% and the genome length is 2.3 Mbp.	
Description of Candidatus Dorea gallistercoris sp. nov.	
Candidatus Dorea gallistercoris (gal.li.ster’co.ris. L. masc. n. gallus chicken; L. neut. n. stercus dung; N.L. gen. n. gallistercoris of chicken faeces)	
A bacterial species identified by metagenomic analyses. This species includes all bacteria with genomes that show ≥95% average nucleotide identity (ANI) to the type genome, which has been assigned the MAG ID ChiSxjej1B13-11762 and which is available via NCBI BioSample SAMN15816753. The GC content of the type genome is 51.71% and the genome length is 2.2 Mbp.	
Description of Candidatus Dorea intestinavium sp. nov.	
Candidatus Dorea intestinavium (in.tes.tin.a’vi.um. L. neut. n. intestinum gut; L. fem. n. avis bird; N.L. gen. n. intestinavium of the gut of birds)	
A bacterial species identified by metagenomic analyses. This species includes all bacteria with genomes that show ≥95% average nucleotide identity (ANI) to the type genome, which has been assigned the MAG ID CHK160-14747 and which is available via NCBI BioSample SAMN15816767. The GC content of the type genome is 35.91% and the genome length is 1.9 Mbp.	
Description of Candidatus Dorea intestinigallinarum sp. nov.	
Candidatus Dorea intestinigallinarum (in.tes.ti.ni.gal.li.na’rum. L. neut. n. intestinum gut; L. fem. n. gallina hen; N.L. gen. n. intestinigallinarum of the gut of the hens)	
A bacterial species identified by metagenomic analyses. This species includes all bacteria with genomes that show ≥95% average nucleotide identity (ANI) to the type genome, which has been assigned the MAG ID CHK188-17839 and which is available via NCBI BioSample SAMN15816854. This is a new name for the alphanumeric GTDB species sp000765215. The GC content of the type genome is 54.58% and the genome length is 2.5 Mbp.	
Description of Candidatus Dorea merdavium sp. nov.	
Candidatus Dorea merdavium (merd.a’vi.um. L. fem. n. merda faeces; L. fem. n. avis bird; N.L. gen. n. merdavium of bird faeces)	
A bacterial species identified by metagenomic analyses. This species includes all bacteria with genomes that show ≥95% average nucleotide identity (ANI) to the type genome, which has been assigned the MAG ID ChiSxjej1B13-1060 and which is available via NCBI BioSample SAMN15816851. This is a new name for the alphanumeric GTDB species sp900312975. The GC content of the type genome is 53.23% and the genome length is 2.0 Mbp.	

Description of Candidatus Dorea stercoravium sp. nov.	
Candidatus Dorea stercoravium (ster.cor.a’vi.um. L. neut. n. stercus dung; L. fem. n. avis bird; N.L. gen. n. stercoravium of bird faeces)	
A bacterial species identified by metagenomic analyses. This species includes all bacteria with genomes that show ≥95% average nucleotide identity (ANI) to the type genome, which has been assigned the MAG ID ChiSjej1B19-6982 and which is available via NCBI BioSample SAMN15816837. This is a new name for the alphanumeric GTDB species sp002160985. The GC content of the type genome is 55.00% and the genome length is 2.5 Mbp.	
Description of Candidatus Duodenibacillus intestinavium sp. nov.	
Candidatus Duodenibacillus intestinavium (in.tes.tin.a’vi.um. L. neut. n. intestinum gut; L. fem. n. avis bird; N.L. gen. n. intestinavium of the gut of birds)	
A bacterial species identified by metagenomic analyses. This species includes all bacteria with genomes that show ≥95% average nucleotide identity (ANI) to the type genome, which has been assigned the MAG ID 2430 and which is available via NCBI BioSample SAMN15816841. This is a new name for the alphanumeric GTDB species sp900538905. The GC content of the type genome is 55.01% and the genome length is 1.8 Mbp.	
Description of Candidatus Duodenibacillus intestinigallinarum sp. nov.	
Candidatus Duodenibacillus intestinigallinarum (in.tes.ti.ni.gal.li.na’rum. L. neut. n. intestinum gut; L. fem. n. gallina hen; N.L. gen. n. intestinigallinarum of the gut of the hens)	
A bacterial species identified by metagenomic analyses. This species includes all bacteria with genomes that show ≥95% average nucleotide identity (ANI) to the type genome, which has been assigned the MAG ID CHK1-2119 and which is available via NCBI BioSample SAMN15816840. This is a new name for the alphanumeric GTDB species sp003472385. The GC content of the type genome is 56.00% and the genome length is 2.0 Mbp.	
Description of Candidatus Egerieenecus gen. nov.	
Candidatus Egerieenecus (E.ge.ri.e.en.e’cus. L. fem. n. egeries dung; Gr. masc. enoikos inhabitant; N.L. masc. n. Egerieenecus a microbe associated with faeces)	
A bacterial genus identified by metagenomic analyses. The genus includes all bacteria with genomes that show ≥60% average amino acid identity (AAI) to the type genome from the type species Candidatus Egerieenecus merdigallinarum. This is a name for the alphanumeric GTDB genus UMGS1600. This genus has been assigned by GTDB-Tk v1.3.0 working on GTDB Release 05-RS95 (Chaumeil et al., 2019; Parks et al., 2020) to the order Christensenellales and to the family CAG-74.	
Description of Candidatus Egerieenecus merdigallinarum sp. nov.	
Candidatus Egerieenecus merdigallinarum (mer.di.gal.li.na’rum. L. fem. n. merda faeces; L. fem. n. gallina hen; N.L. gen. n. merdigallinarum of hen faeces)	
A bacterial species identified by metagenomic analyses. This species includes all bacteria with genomes that show ≥95% average nucleotide identity (ANI) to the type genome, which has been assigned the MAG ID ChiSxjej2B14-4419 and which is available via NCBI BioSample SAMN15817218. This is a new name for the alphanumeric GTDB species sp900553295. The GC content of the type genome is 60.15% and the genome length is 2.5 Mbp.	
Description of Candidatus Egerieicola gen. nov.	
Candidatus Egerieicola (E.ge.ri.e.i’co.la. L. fem. n. egeries dung; L. suff. -cola inhabitant of; N.L. fem. n. Egerieicola a microbe associated with faeces)	
A bacterial genus identified by metagenomic analyses. The genus includes all bacteria with genomes that show ≥60% average amino acid identity (AAI) to the type genome from the type species Candidatus Egerieicola faecale. This is a name for the alphanumeric GTDB genus UBA1375. This genus has been assigned by GTDB-Tk v1.3.0 working on GTDB Release 05-RS95 (Chaumeil et al., 2019; Parks et al., 2020) to the order Oscillospirales and to the family Ruminococcaceae.	
Description of Candidatus Egerieicola faecalis sp. nov.	
Candidatus Egerieicola faecale (fae.ca’lis. L. fem. adj. faecalis of faeces)	
A bacterial species identified by metagenomic analyses. This species includes all bacteria with genomes that show ≥95% average nucleotide identity (ANI) to the type genome, which has been assigned the MAG ID 4509 and which is available via NCBI BioSample SAMN15817200. This is a new name for the alphanumeric GTDB species sp002305795. The GC content of the type genome is 55.66% and the genome length is 1.8 Mbp.	
Description of Candidatus Egerieicola pullicola sp. nov.	
Candidatus Egerieicola pullicola (pul.li’co.la. L. masc. n. pullus a young chicken; L. suff. -cola inhabitant of; N.L. n. pullicola an inhabitant of young chickens)	
A bacterial species identified by metagenomic analyses. This species includes all bacteria with genomes that show ≥95% average nucleotide identity (ANI) to the type genome, which has been assigned the MAG ID CHK184-25365 and which is available via NCBI BioSample SAMN15817017. The GC content of the type genome is 52.81% and the genome length is 1.9 Mbp.	

Description of Candidatus Egerieimonas gen. nov.	
Candidatus Egerieimonas (E.ge.ri.e.i.mo’nas. L. fem. n. egeries dung; L. fem. n. monas a monad; N.L. fem. n. Egerieimonas a microbe associated with faeces)	
A bacterial genus identified by metagenomic analyses. The genus includes all bacteria with genomes that show ≥60% average amino acid identity (AAI) to the type genome from the type species Candidatus Egerieimonas intestinavium. This is a name for the alphanumeric GTDB genus UMGS1472. This genus has been assigned by GTDB-Tk v1.3.0 working on GTDB Release 05-RS95 (Chaumeil et al., 2019; Parks et al., 2020) to the order Lachnospirales and to the family Lachnospiraceae.	
Description of Candidatus Egerieimonas faecigallinarum sp. nov.	
Candidatus Egerieimonas faecigallinarum (fae.ci.gal.li.na’rum. L. fem. n. faex, faecis excrement; L. fem. n. gallina hen; N.L. gen. n. faecigallinarum of hen faeces)	
A bacterial species identified by metagenomic analyses. This species includes all bacteria with genomes that show ≥95% average nucleotide identity (ANI) to the type genome, which has been assigned the MAG ID CHK180-10209 and which is available via NCBI BioSample SAMN15817015. The GC content of the type genome is 51.75% and the genome length is 2.9 Mbp.	
Description of Candidatus Egerieimonas intestinavium sp. nov.	
Candidatus Egerieimonas intestinavium (in.tes.tin.a’vi.um. L. neut. n. intestinum gut; L. fem. n. avis bird; N.L. gen. n. intestinavium of the gut of birds)	
A bacterial species identified by metagenomic analyses. This species includes all bacteria with genomes that show ≥95% average nucleotide identity (ANI) to the type genome, which has been assigned the MAG ID ChiSxjej1B13-7041 and which is available via NCBI BioSample SAMN15817079. The GC content of the type genome is 55.23% and the genome length is 2.6 Mbp.	
Description of Candidatus Egerieisoma gen. nov.	
Candidatus Egerieisoma (E.ge.ri.e.so’ma. L. fem. n. egeries dung; Gr. neut. n. soma a body; N.L. neut. n. Egerieisoma a microbe associated with faeces)	
A bacterial genus identified by metagenomic analyses. The genus includes all bacteria with genomes that show ≥60% average amino acid identity (AAI) to the type genome from the type species Candidatus Egerieisoma faecipullorum. This is a name for the alphanumeric GTDB genus UMGS1537. This genus has been assigned by GTDB-Tk v1.3.0 working on GTDB Release 05-RS95 (Chaumeil et al., 2019; Parks et al., 2020) to the order UBA1212 and to the family UBA1255.	
Description of Candidatus Egerieisoma faecipullorum sp. nov.	
Candidatus Egerieisoma faecipullorum (fae.ci.pul.lo’rum. L. fem. n. faex, faecis excrement; L. masc. n. pullus a young chicken; N.L. gen. n. faecipullorum of young chicken faeces)	
A bacterial species identified by metagenomic analyses. This species includes all bacteria with genomes that show ≥95% average nucleotide identity (ANI) to the type genome, which has been assigned the MAG ID CHK195-4489 and which is available via NCBI BioSample SAMN15817230. This is a new name for the alphanumeric GTDB species sp900543695. The GC content of the type genome is 50.69% and the genome length is 2.0 Mbp.	
Description of Candidatus Egerieousia gen. nov.	
Candidatus Egerieousia (E.ge.ri.e.ou’si.a. L. fem. n. egeries dung; Gr. fem. n. ousia an essence; N.L. fem. n. Egerieousia a microbe associated with faeces)	
A bacterial genus identified by metagenomic analyses. The genus includes all bacteria with genomes that show ≥60% average amino acid identity (AAI) to the type genome from the type species Candidatus Egerieousia excrementavium. This is a name for the alphanumeric GTDB genus UBA1232. This genus has been assigned by GTDB-Tk v1.3.0 working on GTDB Release 05-RS95 (Chaumeil et al., 2019; Parks et al., 2020) to the order Bacteroidales and to the family UBA932.	
Description of Candidatus Egerieousia excrementavium sp. nov.	
Candidatus Egerieousia excrementavium (ex.cre.ment.a’vi.um. L. neut. n. excrementum excrement; L. fem. n. avis bird; N.L. gen. n. excrementavium of bird excrement)	
A bacterial species identified by metagenomic analyses. This species includes all bacteria with genomes that show ≥95% average nucleotide identity (ANI) to the type genome, which has been assigned the MAG ID 15467 and which is available via NCBI BioSample SAMN15817149. The GC content of the type genome is 46.96% and the genome length is 1.5 Mbp.	
Description of Candidatus Eisenbergiella intestinigallinarum sp. nov.	
Candidatus Eisenbergiella intestinigallinarum (in.tes.ti.ni.gal.li.na’rum. L. neut. n. intestinum gut; L. fem. n. gallina hen; N.L. gen. n. intestinigallinarum of the gut of the hens)	
A bacterial species identified by metagenomic analyses. This species includes all bacteria with genomes that show ≥95% average nucleotide identity (ANI) to the type genome, which has been assigned the MAG ID ChiBcec1-1630 and which is available via NCBI BioSample SAMN15816806. This is a new name for the alphanumeric GTDB species sp900544445. The GC content of the type genome is 53.15% and the genome length is 3.4 Mbp.	

Description of Candidatus Eisenbergiella intestinipullorum sp. nov.	
Candidatus Eisenbergiella intestinipullorum (in.tes.ti.ni.pul.lo’rum. L. neut. n. intestinum gut; L. masc. n. pullus a young chicken; N.L. gen. n. intestinipullorum of the gut of young chickens)	
A bacterial species identified by metagenomic analyses. This species includes all bacteria with genomes that show ≥95% average nucleotide identity (ANI) to the type genome, which has been assigned the MAG ID CHK177-9469 and which is available via NCBI BioSample SAMN15816580. The GC content of the type genome is 54.63% and the genome length is 3.5 Mbp.	
Description of Candidatus Eisenbergiella merdavium sp. nov.	
Candidatus Eisenbergiella merdavium (merd.a’vi.um. L. fem. n. merda faeces; L. fem. n. avis bird; N.L. gen. n. merdavium of bird faeces)	
A bacterial species identified by metagenomic analyses. This species includes all bacteria with genomes that show ≥95% average nucleotide identity (ANI) to the type genome, which has been assigned the MAG ID USAMLcec2-132 and which is available via NCBI BioSample SAMN15816641. The GC content of the type genome is 54.07% and the genome length is 4.2 Mbp.	
Description of Candidatus Eisenbergiella merdigallinarum sp. nov.	
Candidatus Eisenbergiella merdigallinarum (mer.di.gal.li.na’rum. L. fem. n. merda faeces; L. fem. n. gallina hen; N.L. gen. n. merdigallinarum of hen faeces)	
A bacterial species identified by metagenomic analyses. This species includes all bacteria with genomes that show ≥95% average nucleotide identity (ANI) to the type genome, which has been assigned the MAG ID USAMLcec3-2134 and which is available via NCBI BioSample SAMN15816643. The GC content of the type genome is 57.03% and the genome length is 3.2 Mbp.	
Description of Candidatus Eisenbergiella merdipullorum sp. nov.	
Candidatus Eisenbergiella merdipullorum (mer.di.pul.lo’rum. L. fem. n. merda faeces; L. masc. n. pullus a young chicken; N.L. gen. n. merdipullorum of the faeces of young chickens)	
A bacterial species identified by metagenomic analyses. This species includes all bacteria with genomes that show ≥95% average nucleotide identity (ANI) to the type genome, which has been assigned the MAG ID CHK179-7159 and which is available via NCBI BioSample SAMN15816597. The GC content of the type genome is 51.92% and the genome length is 3.5 Mbp.	
Description of Candidatus Eisenbergiella pullicola sp. nov.	
Candidatus Eisenbergiella pullicola (pul.li’co.la. L. masc. n. pullus a young chicken; L. suff. -cola inhabitant of; N.L. n. pullicola an inhabitant of young chickens)	
A bacterial species identified by metagenomic analyses. This species includes all bacteria with genomes that show ≥95% average nucleotide identity (ANI) to the type genome, which has been assigned the MAG ID CHK197-24098 and which is available via NCBI BioSample SAMN15816836. This is a new name for the alphanumeric GTDB species sp003343625. The GC content of the type genome is 54.54% and the genome length is 2.6 Mbp.	
Description of Candidatus Eisenbergiella pullistercoris sp. nov.	
Candidatus Eisenbergiella pullistercoris (pul.li.ster’co.ris. L. masc. n. pullus a young chicken; L. neut. n. stercus dung; N.L. gen. n. pullistercoris of young chicken faeces)	
A bacterial species identified by metagenomic analyses. This species includes all bacteria with genomes that show ≥95% average nucleotide identity (ANI) to the type genome, which has been assigned the MAG ID ChiSxjej3B15-24422 and which is available via NCBI BioSample SAMN15816711. The GC content of the type genome is 56.23% and the genome length is 3.3 Mbp.	
Description of Candidatus Eisenbergiella stercoravium sp. nov.	
Candidatus Eisenbergiella stercoravium (ster.cor.a’vi.um. L. neut. n. stercus dung; L. fem. n. avis bird; N.L. gen. n. stercoravium of bird faeces)	
A bacterial species identified by metagenomic analyses. This species includes all bacteria with genomes that show ≥95% average nucleotide identity (ANI) to the type genome, which has been assigned the MAG ID USAMLcec4-2206 and which is available via NCBI BioSample SAMN15816624. The GC content of the type genome is 51.92% and the genome length is 3.9 Mbp.	
Description of Candidatus Eisenbergiella stercorigallinarum sp. nov.	
Candidatus Eisenbergiella stercorigallinarum (ster.co.ri.gal.li.na’rum. L. neut. n. stercus dung; L. fem. n. gallina hen; N.L. gen. n. stercorigallinarum of hen faeces)	
A bacterial species identified by metagenomic analyses. This species includes all bacteria with genomes that show ≥95% average nucleotide identity (ANI) to the type genome, which has been assigned the MAG ID ChiHjej8B7-25341 and which is available via NCBI BioSample SAMN15816792. The GC content of the type genome is 55.79% and the genome length is 2.8 Mbp.	

Description of Candidatus Enterenecus gen. nov.	
Candidatus Enterenecus (En.ter.en.e’cus. Gr. neut. n. enteron the gut; Gr. masc. enoikos inhabitant; N.L. masc. n. Enterenecus a microbe associated with the intestines)	
A bacterial genus identified by metagenomic analyses. The genus includes all bacteria with genomes that show ≥60% average amino acid identity (AAI) to the type genome from the type species Candidatus Enterenecus merdae. This is a name for the alphanumeric GTDB genus UBA9475. This genus has been assigned by GTDB-Tk v1.3.0 working on GTDB Release 05-RS95 (Chaumeil et al., 2019; Parks et al., 2020) to the order Oscillospirales and to the family Oscillospiraceae.	
Description of Candidatus Enterenecus avicola sp. nov.	
Candidatus Enterenecus avicola (a.vi’co.la. L. fem. n. avis bird; L. suff. -cola inhabitant of; N.L. n. avicola inhabitant of birds)	
A bacterial species identified by metagenomic analyses. This species includes all bacteria with genomes that show ≥95% average nucleotide identity (ANI) to the type genome, which has been assigned the MAG ID 153 and which is available via NCBI BioSample SAMN15817108. The GC content of the type genome is 60.43% and the genome length is 1.9 Mbp.	
Description of Candidatus Enterenecus avistercoris sp. nov.	
Candidatus Enterenecus avistercoris (a.vi.ster’co.ris. L. fem. n. avis bird; L. neut. n. stercus dung; N.L. gen. n. avistercoris of bird faeces)	
A bacterial species identified by metagenomic analyses. This species includes all bacteria with genomes that show ≥95% average nucleotide identity (ANI) to the type genome, which has been assigned the MAG ID ChiSxjej3B15-11837 and which is available via NCBI BioSample SAMN15817165. The GC content of the type genome is 64.30% and the genome length is 1.5 Mbp.	
Description of Candidatus Enterenecus faecium sp. nov.	
Candidatus Enterenecus faecium (fae’ci.um. L. fem. n. faex, faecis excrement; L. masc. gen. pl. n. faecium of faeces)	
A bacterial species identified by metagenomic analyses. This species includes all bacteria with genomes that show ≥95% average nucleotide identity (ANI) to the type genome, which has been assigned the MAG ID ChiGjej2B2-12916 and which is available via NCBI BioSample SAMN15817211. This is a new name for the alphanumeric GTDB species sp002161675. The GC content of the type genome is 60.00% and the genome length is 2.0 Mbp.	
Description of Candidatus Enterenecus merdae sp. nov.	
Candidatus Enterenecus merdae (mer’dae. L. gen. n. merdae of faeces)	
A bacterial species identified by metagenomic analyses. This species includes all bacteria with genomes that show ≥95% average nucleotide identity (ANI) to the type genome, which has been assigned the MAG ID ChiHcolR17-2730 and which is available via NCBI BioSample SAMN15817102. The GC content of the type genome is 63.48% and the genome length is 1.7 Mbp.	
Description of Candidatus Enterenecus stercoripullorum sp. nov.	
Candidatus Enterenecus stercoripullorum (ster.co.ri.pul.lo’rum. L. neut. n. stercus dung; L. masc. n. pullus a young chicken; N.L. gen. n. stercoripullorum of the faceces of young chickens)	
A bacterial species identified by metagenomic analyses. This species includes all bacteria with genomes that show ≥95% average nucleotide identity (ANI) to the type genome, which has been assigned the MAG ID 3668 and which is available via NCBI BioSample SAMN15817106. The GC content of the type genome is 60.97% and the genome length is 1.8 Mbp.	
Description of Candidatus Enterocloster excrementigallinarum sp. nov.	
Candidatus Enterocloster excrementigallinarum (ex.cre.men.ti.gal.li.na’rum. L. neut. n. excrementum excrement; L. fem. n. gallina hen; N.L. gen. n. excrementigallinarum of hen excrement)	
A bacterial species identified by metagenomic analyses. This species includes all bacteria with genomes that show ≥95% average nucleotide identity (ANI) to the type genome, which has been assigned the MAG ID CHK198-12963 and which is available via NCBI BioSample SAMN15816811. This is a new name for the alphanumeric GTDB species sp900547035. The GC content of the type genome is 51.32% and the genome length is 3.1 Mbp.	
Description of Candidatus Enterocloster excrementipullorum sp. nov.	
Candidatus Enterocloster excrementipullorum (ex.cre.men.ti.pul.lo’rum. L. neut. n. excrementum excrement; L. masc. n. pullus a young chicken; N.L. gen. n. excrementipullorum of young chicken excrement)	
A bacterial species identified by metagenomic analyses. This species includes all bacteria with genomes that show ≥95% average nucleotide identity (ANI) to the type genome, which has been assigned the MAG ID CHK180-15479 and which is available via NCBI BioSample SAMN15816584. The GC content of the type genome is 53.85% and the genome length is 2.9 Mbp.	

Description of Candidatus Enterocloster faecavium sp. nov.	
Candidatus Enterocloster faecavium (faec.a’vi.um. L. fem. n. faex, faecis excrement; L. fem. n. avis bird; N.L. gen. n. faecavium of bird faeces)	
A bacterial species identified by metagenomic analyses. This species includes all bacteria with genomes that show ≥95% average nucleotide identity (ANI) to the type genome, which has been assigned the MAG ID CHK188-4685 and which is available via NCBI BioSample SAMN15816596. The GC content of the type genome is 52.23% and the genome length is 2.8 Mbp.	
Description of Candidatus Enterococcus avicola sp. nov.	
Candidatus Enterococcus avicola (a.vi’co.la. L. fem. n. avis bird; L. suff. -cola inhabitant of; N.L. n. avicola inhabitant of birds)	
A bacterial species identified by metagenomic analyses. This species includes all bacteria with genomes that show ≥95% average nucleotide identity (ANI) to the type genome, which has been assigned the MAG ID CHK172-16539 and which is available via NCBI BioSample SAMN15816900. Although GTDB has assigned this species to the genus it calls Enterococcus_I, this genus designation cannot be incorporated into a well-formed binomial, so in naming this species, we have used the current validly published name for the genus. The GC content of the type genome is 36.87% and the genome length is 2.2 Mbp.	
Description of Candidatus Enterococcus stercoravium sp. nov.	
Candidatus Enterococcus stercoravium (ster.cor.a’vi.um. L. neut. n. stercus dung; L. fem. n. avis bird; N.L. gen. n. stercoravium of bird faeces)	
A bacterial species identified by metagenomic analyses. This species includes all bacteria with genomes that show ≥95% average nucleotide identity (ANI) to the type genome, which has been assigned the MAG ID CHK172-14336 and which is available via NCBI BioSample SAMN15816907. Although GTDB has assigned this species to the genus it calls Enterococcus_C, this genus designation cannot be incorporated into a well-formed binomial, so in naming this species, we have used the current validly published name for the genus. The GC content of the type genome is 44.16% and the genome length is 2.3 Mbp.	
Description of Candidatus Enterococcus stercoripullorum sp. nov.	
Candidatus Enterococcus stercoripullorum (ster.co.ri.pul.lo’rum. L. neut. n. stercus dung; L. masc. n. pullus a young chicken; N.L. gen. n. stercoripullorum of the faceces of young chickens)	
A bacterial species identified by metagenomic analyses. This species includes all bacteria with genomes that show ≥95% average nucleotide identity (ANI) to the type genome, which has been assigned the MAG ID ChiHjej12B11-924 and which is available via NCBI BioSample SAMN15816914. Although GTDB has assigned this species to the genus it calls Enterococcus_E, this genus designation cannot be incorporated into a well-formed binomial, so in naming this species, we have used the current validly published name for the genus. The GC content of the type genome is 36.20% and the genome length is 2.3 Mbp.	
Description of Candidatus Enterocola gen. nov.	
Candidatus Enterocola (En.te.ro’co.la. Gr. neut. n. enteron the gut; L. suff. -cola inhabitant of; N.L. fem. n. Enterocola a microbe associated with the gut)	
A bacterial genus identified by metagenomic analyses. The genus includes all bacteria with genomes that show ≥60% average amino acid identity (AAI) to the type genome from the type species Candidatus Enterocola intestinipullorum. This is a name for the alphanumeric GTDB genus RUG163. This genus has been assigned by GTDB-Tk v1.3.0 working on GTDB Release 05-RS95 (Chaumeil et al., 2019; Parks et al., 2020) to the order Bacteroidales and to the family Paludibacteraceae.	
Description of Candidatus Enterocola intestinipullorum sp. nov.	
Candidatus Enterocola intestinipullorum (in.tes.ti.ni.pul.lo’rum. L. neut. n. intestinum gut; L. masc. n. pullus a young chicken; N.L. gen. n. intestinipullorum of the gut of young chickens)	
A bacterial species identified by metagenomic analyses. This species includes all bacteria with genomes that show ≥95% average nucleotide identity (ANI) to the type genome, which has been assigned the MAG ID D3-1215 and which is available via NCBI BioSample SAMN15817113. The GC content of the type genome is 47.46% and the genome length is 1.8 Mbp.	
Description of Candidatus Enteromonas gen. nov.	
Candidatus Enteromonas (En.te.ro.mo’nas. Gr. neut. n. enteron the gut; L. fem. n. monas a monad; N.L. fem. n. Enteromonas a microbe associated with the intestines)	
A bacterial genus identified by metagenomic analyses. The genus includes all bacteria with genomes that show ≥60% average amino acid identity (AAI) to the type genome from the type species Candidatus Enteromonas pullistercoris. This is a name for the alphanumeric GTDB genus UBA733. This genus has been assigned by GTDB-Tk v1.3.0 working on GTDB Release 05-RS95 (Chaumeil et al., 2019; Parks et al., 2020) to the order RFN20 and to the family CAG-826.	

Description of Candidatus Enteromonas pullicola sp. nov.	
Candidatus Enteromonas pullicola (pul.li’co.la. L. masc. n. pullus a young chicken; L. suff. -cola inhabitant of; N.L. n. pullicola an inhabitant of young chickens)	
A bacterial species identified by metagenomic analyses. This species includes all bacteria with genomes that show ≥95% average nucleotide identity (ANI) to the type genome, which has been assigned the MAG ID ChiGjej1B1-22543 and which is available via NCBI BioSample SAMN15817133. The GC content of the type genome is 57.43% and the genome length is 1.2 Mbp.	
Description of Candidatus Enteromonas pullistercoris sp. nov.	
Candidatus Enteromonas pullistercoris (pul.li.ster’co.ris. L. masc. n. pullus a young chicken; L. neut. n. stercus dung; N.L. gen. n. pullistercoris of young chicken faeces)	
A bacterial species identified by metagenomic analyses. This species includes all bacteria with genomes that show ≥95% average nucleotide identity (ANI) to the type genome, which has been assigned the MAG ID 17113 and which is available via NCBI BioSample SAMN15817142. The GC content of the type genome is 53.99% and the genome length is 1.4 Mbp.	
Description of Candidatus Enterosoma gen. nov.	
Candidatus Enterosoma (En.te.ro.so’ma. Gr. neut. n. enteron the gut; Gr. neut. n. soma a body; N.L. neut. n. Enterosoma a microbe associated with the intestines)	
A bacterial genus identified by metagenomic analyses. The genus includes all bacteria with genomes that show ≥60% average amino acid identity (AAI) to the type genome from the type species Candidatus Enterosoma merdigallinarum. This is a name for the alphanumeric GTDB genus UBA7642. This genus has been assigned by GTDB-Tk v1.3.0 working on GTDB Release 05-RS95 (Chaumeil et al., 2019; Parks et al., 2020) to the order RFN20 and to the family CAG-288.	
Description of Candidatus Enterosoma merdigallinarum sp. nov.	
Candidatus Enterosoma merdigallinarum (mer.di.gal.li.na’rum. L. fem. n. merda faeces; L. fem. n. gallina hen; N.L. gen. n. merdigallinarum of hen faeces)	
A bacterial species identified by metagenomic analyses. This species includes all bacteria with genomes that show ≥95% average nucleotide identity (ANI) to the type genome, which has been assigned the MAG ID 33044 and which is available via NCBI BioSample SAMN15817141. The GC content of the type genome is 51.22% and the genome length is 1.4 Mbp.	
Description of Candidatus Enterousia gen. nov.	
Candidatus Enterousia (En.ter.ou’si.a. Gr. neut. n. enteron the gut; Gr. fem. n. ousia an essence; N.L. fem. n. Enterousia a microbe associated with the intestines)	
A bacterial genus identified by metagenomic analyses. The genus includes all bacteria with genomes that show ≥60% average amino acid identity (AAI) to the type genome from the type species Candidatus Enterousia excrementavium. This is a name for the alphanumeric GTDB genus Rs-D84. This genus has been assigned by GTDB-Tk v1.3.0 working on GTDB Release 05-RS95 (Chaumeil et al., 2019; Parks et al., 2020) to the order Rs-D84 and to the family Rs-D84.	
Description of Candidatus Enterousia avicola sp. nov.	
Candidatus Enterousia avicola (a.vi’co.la. L. fem. n. avis bird; L. suff. -cola inhabitant of; N.L. n. avicola inhabitant of birds)	
A bacterial species identified by metagenomic analyses. This species includes all bacteria with genomes that show ≥95% average nucleotide identity (ANI) to the type genome, which has been assigned the MAG ID CHK136-897 and which is available via NCBI BioSample SAMN15817144. The GC content of the type genome is 39.01% and the genome length is 0.9 Mbp.	
Description of Candidatus Enterousia avistercoris sp. nov.	
Candidatus Enterousia avistercoris (a.vi.ster’co.ris. L. fem. n. avis bird; L. neut. n. stercus dung; N.L. gen. n. avistercoris of bird faeces)	
A bacterial species identified by metagenomic analyses. This species includes all bacteria with genomes that show ≥95% average nucleotide identity (ANI) to the type genome, which has been assigned the MAG ID 8207 and which is available via NCBI BioSample SAMN15817150. The GC content of the type genome is 43.79% and the genome length is 0.8 Mbp.	
Description of Candidatus Enterousia excrementavium sp. nov.	
Candidatus Enterousia excrementavium (ex.cre.ment.a’vi.um. L. neut. n. excrementum excrement; L. fem. n. avis bird; N.L. gen. n. excrementavium of bird excrement)	
A bacterial species identified by metagenomic analyses. This species includes all bacteria with genomes that show ≥95% average nucleotide identity (ANI) to the type genome, which has been assigned the MAG ID B1-16210 and which is available via NCBI BioSample SAMN15817158. The GC content of the type genome is 44.26% and the genome length is 0.9 Mbp.	

Description of Candidatus Enterousia intestinigallinarum sp. nov.	
Candidatus Enterousia intestinigallinarum (in.tes.ti.ni.gal.li.na’rum. L. neut. n. intestinum gut; L. fem. n. gallina hen; N.L. gen. n. intestinigallinarum of the gut of the hens)	
A bacterial species identified by metagenomic analyses. This species includes all bacteria with genomes that show ≥95% average nucleotide identity (ANI) to the type genome, which has been assigned the MAG ID ChiGjej3B3-5194 and which is available via NCBI BioSample SAMN15817183. This is a new name for the alphanumeric GTDB species sp900546185. The GC content of the type genome is 45.89% and the genome length is 0.9 Mbp.	
Description of Candidatus Erysipelatoclostridium merdavium sp. nov.	
Candidatus Erysipelatoclostridium merdavium (merd.a’vi.um. L. fem. n. merda faeces; L. fem. n. avis bird; N.L. gen. n. merdavium of bird faeces)	
A bacterial species identified by metagenomic analyses. This species includes all bacteria with genomes that show ≥95% average nucleotide identity (ANI) to the type genome, which has been assigned the MAG ID ChiGjej1B1-14440 and which is available via NCBI BioSample SAMN15816860. This is a new name for the alphanumeric GTDB species sp002160495. The GC content of the type genome is 29.32% and the genome length is 2.6 Mbp.	
Description of Candidatus Eubacterium avistercoris sp. nov.	
Candidatus Eubacterium avistercoris (a.vi.ster’co.ris. L. fem. n. avis bird; L. neut. n. stercus dung; N.L. gen. n. avistercoris of bird faeces)	
A bacterial species identified by metagenomic analyses. This species includes all bacteria with genomes that show ≥95% average nucleotide identity (ANI) to the type genome, which has been assigned the MAG ID CHK192-9172 and which is available via NCBI BioSample SAMN15816888. Although GTDB has assigned this species to the genus it calls Eubacterium_I, this genus designation cannot be incorporated into a well-formed binomial, so in naming this species, we have used the current validly published name for the genus. The GC content of the type genome is 45.90% and the genome length is 2.6 Mbp.	
Description of Candidatus Eubacterium faecale sp. nov.	
Candidatus Eubacterium faecale (fae.ca’le. L. neut. adj. faecale of faeces)	
A bacterial species identified by metagenomic analyses. This species includes all bacteria with genomes that show ≥95% average nucleotide identity (ANI) to the type genome, which has been assigned the MAG ID CHK188-16595 and which is available via NCBI BioSample SAMN15816917. This is a new name for the alphanumeric GTDB species sp000431535. Although GTDB has assigned this species to the genus it calls Eubacterium_R, this genus designation cannot be incorporated into a well-formed binomial, so in naming this species, we have used the current validly published name for the genus. The GC content of the type genome is 46.56% and the genome length is 1.8 Mbp.	
Description of Candidatus Eubacterium faecavium sp. nov.	
Candidatus Eubacterium faecavium (faec.a’vi.um. L. fem. n. faex, faecis excrement; L. fem. n. avis bird; N.L. gen. n. faecavium of bird faeces)	
A bacterial species identified by metagenomic analyses. This species includes all bacteria with genomes that show ≥95% average nucleotide identity (ANI) to the type genome, which has been assigned the MAG ID ChiHecec3B27-3607 and which is available via NCBI BioSample SAMN15816921. This is a new name for the alphanumeric GTDB species sp900539845. Although GTDB has assigned this species to the genus it calls Eubacterium_R, this genus designation cannot be incorporated into a well-formed binomial, so in naming this species, we have used the current validly published name for the genus. The GC content of the type genome is 45.56% and the genome length is 1.9 Mbp.	
Description of Candidatus Eubacterium faecigallinarum sp. nov.	
Candidatus Eubacterium faecigallinarum (fae.ci.gal.li.na’rum. L. fem. n. faex, faecis excrement; L. fem. n. gallina hen; N.L. gen. n. faecigallinarum of hen faeces)	
A bacterial species identified by metagenomic analyses. This species includes all bacteria with genomes that show ≥95% average nucleotide identity (ANI) to the type genome, which has been assigned the MAG ID 8396 and which is available via NCBI BioSample SAMN15816904. Although GTDB has assigned this species to the genus it calls Eubacterium_R, this genus designation cannot be incorporated into a well-formed binomial, so in naming this species, we have used the current validly published name for the genus. The GC content of the type genome is 43.37% and the genome length is 1.6 Mbp.	
Description of Candidatus Eubacterium faecipullorum sp. nov.	
Candidatus Eubacterium faecipullorum (fae.ci.pul.lo’rum. L. fem. n. faex, faecis excrement; L. masc. n. pullus a young chicken; N.L. gen. n. faecipullorum of young chicken faeces)	
A bacterial species identified by metagenomic analyses. This species includes all bacteria with genomes that show ≥95% average nucleotide identity (ANI) to the type genome, which has been assigned the MAG ID 421 and which is available via NCBI BioSample SAMN15816928. This is a new name for the alphanumeric GTDB species sp900546785. Although GTDB has assigned this species to the genus it calls Eubacterium_R, this genus designation cannot be incorporated into a well-formed binomial, so in naming this species, we have used the current validly published name for the genus. The GC content of the type genome is 47.26% and the genome length is 1.9 Mbp.	

Description of Candidatus Eubacterium pullicola sp. nov.	
Candidatus Eubacterium pullicola (pul.li’co.la. L. masc. n. pullus a young chicken; L. suff. -cola inhabitant of; N.L. n. pullicola inhabitant of young chicken)	
A bacterial species identified by metagenomic analyses. This species includes all bacteria with genomes that show ≥95% average nucleotide identity (ANI) to the type genome, which has been assigned the MAG ID ChiHjej12B11-11929 and which is available via NCBI BioSample SAMN15816916. This is a new name for the alphanumeric GTDB species sp900540015. Although GTDB has assigned this species to the genus it calls Eubacterium_M, this genus designation cannot be incorporated into a well-formed binomial, so in naming this species, we have used the current validly published name for the genus. The GC content of the type genome is 41.55% and the genome length is 1.2 Mbp.	
Description of Candidatus Evtepia excrementipullorum sp. nov.	
Candidatus Evtepia excrementipullorum (ex.cre.men.ti.pul.lo’rum. L. neut. n. excrementum excrement; L. masc. n. pullus a young chicken; N.L. gen. n. excrementipullorum of young chicken excrement)	
A bacterial species identified by metagenomic analyses. This species includes all bacteria with genomes that show ≥95% average nucleotide identity (ANI) to the type genome, which has been assigned the MAG ID ChiSjej3B21-3892 and which is available via NCBI BioSample SAMN15816827. This is a new name for the alphanumeric GTDB species sp900546255. The GC content of the type genome is 63.06% and the genome length is 2.0 Mbp.	
Description of Candidatus Evtepia faecavium sp. nov.	
Candidatus Evtepia faecavium (faec.a’vi.um. L. fem. n. faex, faecis excrement; L. fem. n. avis bird; N.L. gen. n. faecavium of bird faeces)	
A bacterial species identified by metagenomic analyses. This species includes all bacteria with genomes that show ≥95% average nucleotide identity (ANI) to the type genome, which has been assigned the MAG ID ChiHecec3B27-8621 and which is available via NCBI BioSample SAMN15816713. The GC content of the type genome is 65.40% and the genome length is 2.0 Mbp.	
Description of Candidatus Evtepia faecigallinarum sp. nov.	
Candidatus Evtepia faecigallinarum (fae.ci.gal.li.na’rum. L. fem. n. faex, faecis excrement; L. fem. n. gallina hen; N.L. gen. n. faecigallinarum of hen faeces)	
A bacterial species identified by metagenomic analyses. This species includes all bacteria with genomes that show ≥95% average nucleotide identity (ANI) to the type genome, which has been assigned the MAG ID ChiHcec3-601 and which is available via NCBI BioSample SAMN15816724. The GC content of the type genome is 63.50% and the genome length is 2.4 Mbp.	
Description of Candidatus Excrementavichristensenella gen. nov.	
Candidatus Excrementavichristensenella (Ex.cre.ment.a.vi.chris.ten.sen.el’la. L. neut. n. excrementum excrement; L. fem. n. avis bird; N.L. fem. n. Christensenella a genus name; N.L. fem n. Excrementavichristensenella a genus related to the genus Christensenella but distinct from it and found in poultry faeces)	
A bacterial genus identified by metagenomic analyses. The genus includes all bacteria with genomes that show ≥60% average amino acid identity (AAI) to the type genome from the type species Candidatus Excrementavichristensenella intestinipullorum. This genus has been assigned by GTDB-Tk v1.3.0 working on GTDB Release 05-RS95 (Chaumeil et al., 2019; Parks et al., 2020) to the order Christensenellales and to the family CAG-74.	
Description of Candidatus Excrementavichristensenella intestinipullorum sp. nov.	
Candidatus Excrementavichristensenella intestinipullorum (in.tes.ti.ni.pul.lo’rum. L. neut. n. intestinum gut; L. masc. n. pullus a young chicken; N.L. gen. n. intestinipullorum of the gut of young chickens)	
A bacterial species identified by metagenomic analyses. This species includes all bacteria with genomes that show ≥95% average nucleotide identity (ANI) to the type genome, which has been assigned the MAG ID ChiGjej2B2-1688 and which is available via NCBI BioSample SAMN15816955. The GC content of the type genome is 62.99% and the genome length is 2.8 Mbp.	
Description of Candidatus Faecalibacterium avium sp. nov.	
Candidatus Faecalibacterium avium (a’vi.um. L. fem. pl. n. avium of birds)	
A bacterial species identified by metagenomic analyses. This species includes all bacteria with genomes that show ≥95% average nucleotide identity (ANI) to the type genome, which has been assigned the MAG ID CHK182-10647 and which is available via NCBI BioSample SAMN15816876. This is a new name for the alphanumeric GTDB species sp002160915. The GC content of the type genome is 62.96% and the genome length is 2.2 Mbp.	
Description of Candidatus Faecalibacterium faecigallinarum sp. nov.	
Candidatus Faecalibacterium faecigallinarum (fae.ci.gal.li.na’rum. L. fem. n. faex, faecis excrement; L. fem. n. gallina hen; N.L. gen. n. faecigallinarum of hen faeces)	
A bacterial species identified by metagenomic analyses. This species includes all bacteria with genomes that show ≥95% average nucleotide identity (ANI) to the type genome, which has been assigned the MAG ID ChiSjej5B23-2810 and which is available via NCBI BioSample SAMN15816583. The GC content of the type genome is 63.38% and the genome length is 2.1 Mbp.	

Description of Candidatus Faecalibacterium faecipullorum sp. nov.	
Candidatus Faecalibacterium faecipullorum (fae.ci.pul.lo’rum. L. fem. n. faex, faecis excrement; L. masc. n. pullus a young chicken; N.L. gen. n. faecipullorum of young chicken faeces)	
A bacterial species identified by metagenomic analyses. This species includes all bacteria with genomes that show ≥95% average nucleotide identity (ANI) to the type genome, which has been assigned the MAG ID ChiHjej9B8-13557 and which is available via NCBI BioSample SAMN15816651. The GC content of the type genome is 65.75% and the genome length is 2.1 Mbp.	
Description of Candidatus Faecalibacterium gallistercoris sp. nov.	
Candidatus Faecalibacterium gallistercoris (gal.li.ster’co.ris. L. masc. n. gallus chicken; L. neut. n. stercus dung; N.L. gen. n. gallistercoris of chicken faeces)	
A bacterial species identified by metagenomic analyses. This species includes all bacteria with genomes that show ≥95% average nucleotide identity (ANI) to the type genome, which has been assigned the MAG ID ChiBcec16-3735 and which is available via NCBI BioSample SAMN15816605. The GC content of the type genome is 64.68% and the genome length is 2.1 Mbp.	
Description of Candidatus Faecalibacterium intestinavium sp. nov.	
Candidatus Faecalibacterium intestinavium (in.tes.tin.a’vi.um. L. neut. n. intestinum gut; L. fem. n. avis bird; N.L. gen. n. intestinavium of the gut of birds)	
A bacterial species identified by metagenomic analyses. This species includes all bacteria with genomes that show ≥95% average nucleotide identity (ANI) to the type genome, which has been assigned the MAG ID 742 and which is available via NCBI BioSample SAMN15816744. The GC content of the type genome is 61.60% and the genome length is 1.8 Mbp.	
Description of Candidatus Faecalibacterium intestinigallinarum sp. nov.	
Candidatus Faecalibacterium intestinigallinarum (in.tes.ti.ni.gal.li.na’rum. L. neut. n. intestinum gut; L. fem. n. gallina hen; N.L. gen. n. intestinigallinarum of the gut of the hens)	
A bacterial species identified by metagenomic analyses. This species includes all bacteria with genomes that show ≥95% average nucleotide identity (ANI) to the type genome, which has been assigned the MAG ID ChiHcolR34-3080 and which is available via NCBI BioSample SAMN15816770. The GC content of the type genome is 64.34% and the genome length is 2.1 Mbp.	
Description of Candidatus Faecalibacterium intestinipullorum sp. nov.	
Candidatus Faecalibacterium intestinipullorum (in.tes.ti.ni.pul.lo’rum. L. neut. n. intestinum gut; L. masc. n. pullus a young chicken; N.L. gen. n. intestinipullorum of the gut of young chickens)	
A bacterial species identified by metagenomic analyses. This species includes all bacteria with genomes that show ≥95% average nucleotide identity (ANI) to the type genome, which has been assigned the MAG ID ChiHcolR21-11242 and which is available via NCBI BioSample SAMN15816785. The GC content of the type genome is 61.39% and the genome length is 2.1 Mbp.	
Description of Candidatus Faecalicoccus intestinipullorum sp. nov.	
Candidatus Faecalicoccus intestinipullorum (in.tes.ti.ni.pul.lo’rum. L. neut. n. intestinum gut; L. masc. n. pullus a young chicken; N.L. gen. n. intestinipullorum of the gut of young chickens)	
A bacterial species identified by metagenomic analyses. This species includes all bacteria with genomes that show ≥95% average nucleotide identity (ANI) to the type genome, which has been assigned the MAG ID ChiHjej8B7-5959 and which is available via NCBI BioSample SAMN15816766. The GC content of the type genome is 40.89% and the genome length is 1.4 Mbp.	
Description of Candidatus Faecaligallichristensenella gen. nov.	
Candidatus Faecaligallichristensenella (Fae.ca.li.gal.li.chris.ten.sen.el’la. N.L. masc. adj. faecalis pertaining to faeces; L. masc. n. gallus chicken; N.L. fem. n. Christensenella a genus name; N.L. fem. n. Faecaligallichristensenella a genus related to the genus Christensenella but distinct from it and found in poultry faeces)	
A bacterial genus identified by metagenomic analyses. The genus includes all bacteria with genomes that show ≥60% average amino acid identity (AAI) to the type genome from the type species Candidatus Faecaligallichristensenella faecipullorum. This genus was identified but not named by Glendinning et al. (2020). This genus has been assigned by GTDB-Tk v1.3.0 working on GTDB Release 05-RS95 (Chaumeil et al., 2019; Parks et al., 2020) to the order Christensenellales and to the family CAG-74.	
Description of Candidatus Faecaligallichristensenella faecipullorum sp. nov.	
Candidatus Faecaligallichristensenella faecipullorum (fae.ci.pul.lo’rum. L. fem. n. faex, faecis excrement; L. masc. n. pullus a young chicken; N.L. gen. n. faecipullorum of young chicken faeces)	
A bacterial species identified by metagenomic analyses. This species includes all bacteria with genomes that show ≥95% average nucleotide identity (ANI) to the type genome, which has been assigned the MAG ID ChiSjej6B24-5839 and which is available via NCBI BioSample SAMN15816940. The GC content of the type genome is 58.49% and the genome length is 2.6 Mbp.	

Description of Candidatus Faecenecus gen. nov.	
Candidatus Faecenecus (Faec.en.e’cus. L. fem. n. faex dregs; Gr. masc. enoikos inhabitant; N.L. masc. n. Faecenecus a microbe associated with faeces)	
A bacterial genus identified by metagenomic analyses. The genus includes all bacteria with genomes that show ≥60% average amino acid identity (AAI) to the type genome from the type species Candidatus Faecenecus gallistercoris. This is a name for the alphanumeric GTDB genus CAG-988. This genus has been assigned by GTDB-Tk v1.3.0 working on GTDB Release 05-RS95 (Chaumeil et al., 2019; Parks et al., 2020) to the order RF39 and to the family CAG-611.	
Description of Candidatus Faecenecus gallistercoris sp. nov.	
Candidatus Faecenecus gallistercoris (gal.li.ster’co.ris. L. masc. n gallus chicken; L. neut. n. stercus dung; N.L. gen. n. gallistercoris of chicken faeces)	
A bacterial species identified by metagenomic analyses. This species includes all bacteria with genomes that show ≥95% average nucleotide identity (ANI) to the type genome, which has been assigned the MAG ID CHK165-10780 and which is available via NCBI BioSample SAMN15817166. This is a new name for the alphanumeric GTDB species sp003149915. The GC content of the type genome is 34.49% and the genome length is 1.2 Mbp.	
Description of Candidatus Faecicola gen. nov.	
Candidatus Faecicola (Fae.ci’co.la. L. fem. n. faex dregs; L. suff. -cola inhabitant of; N.L. fem. n. Faecicola a microbe associated with faeces)	
A bacterial genus identified by metagenomic analyses. The genus includes all bacteria with genomes that show ≥60% average amino acid identity (AAI) to the type genome from the type species Candidatus Faecicola pullistercoris. This is a name for the alphanumeric GTDB genus CAG-1138. This genus has been assigned by GTDB-Tk v1.3.0 working on GTDB Release 05-RS95 (Chaumeil et al., 2019; Parks et al., 2020) to the order 4C28d-15 and to the family CAG-917.	
Description of Candidatus Faecicola pullistercoris sp. nov.	
Candidatus Faecicola pullistercoris (pul.li.ster’co.ris. L. masc. n. pullus a young chicken; L. fem. n. avis bird; N.L. gen. n. pullistercoris of young chicken faeces)	
A bacterial species identified by metagenomic analyses. This species includes all bacteria with genomes that show ≥95% average nucleotide identity (ANI) to the type genome, which has been assigned the MAG ID 5944 and which is available via NCBI BioSample SAMN15817151. The GC content of the type genome is 48.54% and the genome length is 1.6 Mbp.	
Description of Candidatus Faecimonas gen. nov.	
Candidatus Faecimonas (Fae.ci.mo’nas. L. fem. n. faex dregs; L. fem. n. monas a monad; N.L. fem. n. Faecimonas a microbe associated with faeces)	
A bacterial genus identified by metagenomic analyses. The genus includes all bacteria with genomes that show ≥60% average amino acid identity (AAI) to the type genome from the type species Candidatus Faecimonas intestinavium. This is a name for the alphanumeric GTDB genus CAG-877. This genus has been assigned by GTDB-Tk v1.3.0 working on GTDB Release 05-RS95 (Chaumeil et al., 2019; Parks et al., 2020) to the order RF39 and to the family CAG-611.	
Description of Candidatus Faecimonas gallistercoris sp. nov.	
Candidatus Faecimonas gallistercoris (gal.li.ster’co.ris. L. masc. n gallus chicken; L. neut. n. stercus dung; N.L. gen. n. gallistercoris of chicken faeces)	
A bacterial species identified by metagenomic analyses. This species includes all bacteria with genomes that show ≥95% average nucleotide identity (ANI) to the type genome, which has been assigned the MAG ID CHK189-3136 and which is available via NCBI BioSample SAMN15817016. The GC content of the type genome is 28.18% and the genome length is 1.4 Mbp.	
Description of Candidatus Faecimonas intestinavium sp. nov.	
Candidatus Faecimonas intestinavium (in.tes.tin.a’vi.um. L. neut. n. intestinum gut; L. fem. n. avis bird; N.L. gen. n. intestinavium of the gut of birds)	
A bacterial species identified by metagenomic analyses. This species includes all bacteria with genomes that show ≥95% average nucleotide identity (ANI) to the type genome, which has been assigned the MAG ID USAMLcec2-12447 and which is available via NCBI BioSample SAMN15817225. This is a new name for the alphanumeric GTDB species sp900554305. The GC content of the type genome is 29.22% and the genome length is 1.8 Mbp.	
Description of Candidatus Faecimorpha gen. nov.	
Candidatus Faecimorpha (Fae.ci.mor’pha. L. fem. n. faex dregs; Gr. fem. n. morphe a form, shape; N.L. fem. n. Faecimorpha a microbe associated with faeces)	
A bacterial genus identified by metagenomic analyses. The genus includes all bacteria with genomes that show ≥60% average amino acid identity (AAI) to the type genome from the type species Candidatus Faecimorpha stercoravium. This is a name for the alphanumeric GTDB genus UBA1390. This genus has been assigned by GTDB-Tk v1.3.0 working on GTDB Release 05-RS95 (Chaumeil et al., 2019; Parks et al., 2020) to the order Lachnospirales and to the family UBA1390.	

Description of Candidatus Faecimorpha stercoravium sp. nov.	
Candidatus Faecimorpha stercoravium (ster.cor.a’vi.um. L. neut. n. stercus dung; L. fem. n. avis bird; N.L. gen. n. stercoravium of bird faeces)	
A bacterial species identified by metagenomic analyses. This species includes all bacteria with genomes that show ≥95% average nucleotide identity (ANI) to the type genome, which has been assigned the MAG ID CHK195-9767 and which is available via NCBI BioSample SAMN15817172. This is a new name for the alphanumeric GTDB species sp002305315. The GC content of the type genome is 49.81% and the genome length is 2.4 Mbp.	
Description of Candidatus Faeciplasma gen. nov.	
Candidatus Faeciplasma (Fae.ci.plas’ma. L. fem. n. faex dregs; Gr. neut. n. plasma a form; N.L. neut. n. Faeciplasma a microbe associated with faeces)	
A bacterial genus identified by metagenomic analyses. The genus includes all bacteria with genomes that show ≥60% average amino acid identity (AAI) to the type genome from the type species Candidatus Faeciplasma avium. This is a name for the alphanumeric GTDB genus UBA1409. This genus has been assigned by GTDB-Tk v1.3.0 working on GTDB Release 05-RS95 (Chaumeil et al., 2019; Parks et al., 2020) to the order Oscillospirales and to the family Ruminococcaceae.	
Description of Candidatus Faeciplasma avium sp. nov.	
Candidatus Faeciplasma avium (a’vi.um. L. fem. pl. n. avium of birds)	
A bacterial species identified by metagenomic analyses. This species includes all bacteria with genomes that show ≥95% average nucleotide identity (ANI) to the type genome, which has been assigned the MAG ID 1370 and which is available via NCBI BioSample SAMN15817208. This is a new name for the alphanumeric GTDB species sp002305045. The GC content of the type genome is 51.56% and the genome length is 1.6 Mbp.	
Description of Candidatus Faeciplasma gallinarum sp. nov.	
Candidatus Faeciplasma gallinarum (gal.li.na’rum. L. fem. n. gallina a hen; L. gen. fem. pl. n. gallinarum of hens)	
A bacterial species identified by metagenomic analyses. This species includes all bacteria with genomes that show ≥95% average nucleotide identity (ANI) to the type genome, which has been assigned the MAG ID CHK157-1446 and which is available via NCBI BioSample SAMN15817182. This is a new name for the alphanumeric GTDB species sp002338885. The GC content of the type genome is 49.59% and the genome length is 1.6 Mbp.	
Description of Candidatus Faeciplasma pullistercoris sp. nov.	
Candidatus Faeciplasma pullistercoris (pul.li.ster’co.ris. L. masc. n. pullus a young chicken; L. neut. n. stercus dung; N.L. gen. n. pullistercoris of young chicken faeces)	
A bacterial species identified by metagenomic analyses. This species includes all bacteria with genomes that show ≥95% average nucleotide identity (ANI) to the type genome, which has been assigned the MAG ID CHK33-4379 and which is available via NCBI BioSample SAMN15817120. The GC content of the type genome is 49.45% and the genome length is 1.5 Mbp.	
Description of Candidatus Faecisoma gen. nov.	
Candidatus Faecisoma (Fae.ci.so’ma. L. fem. n. faex dregs; Gr. neut. n. soma a body; N.L. neut. n. Faecisoma a microbe associated with faeces)	
A bacterial genus identified by metagenomic analyses. The genus includes all bacteria with genomes that show ≥60% average amino acid identity (AAI) to the type genome from the type species Candidatus Faecisoma merdavium. This is a name for the alphanumeric GTDB genus CAG-878. This genus has been assigned by GTDB-Tk v1.3.0 working on GTDB Release 05-RS95 (Chaumeil et al., 2019; Parks et al., 2020) to the order RF39 and to the family CAG-822.	
Description of Candidatus Faecisoma merdavium sp. nov.	
Candidatus Faecisoma merdavium (merd.a’vi.um. L. fem. n. merda faeces; L. fem. n. avis bird; N.L. gen. n. merdavium of bird faeces)	
A bacterial species identified by metagenomic analyses. This species includes all bacteria with genomes that show ≥95% average nucleotide identity (ANI) to the type genome, which has been assigned the MAG ID 6595 and which is available via NCBI BioSample SAMN15817101. The GC content of the type genome is 24.63% and the genome length is 1.3 Mbp.	
Description of Candidatus Faecivicinus gen. nov.	
Candidatus Faecivicinus (Fae.ci.vi.ci’nus. L. fem. n. faex dregs; L. masc. n. vicinus a neighbour; N.L. masc. n. Faecivicinus a microbe associated with faeces)	
A bacterial genus identified by metagenomic analyses. The genus includes all bacteria with genomes that show ≥60% average amino acid identity (AAI) to the type genome from the type species Candidatus Faecivicinus avistercoris. This is a name for the alphanumeric GTDB genus UMGS1603. This genus has been assigned by GTDB-Tk v1.3.0 working on GTDB Release 05-RS95 (Chaumeil et al., 2019; Parks et al., 2020) to the order Christensenellales and to the family CAG-74.	

Description of Candidatus Faecivicinus avistercoris sp. nov.	
Candidatus Faecivicinus avistercoris (a.vi.ster’co.ris. L. fem. n. avis bird; L. neut. n. stercus dung; N.L. gen. n. avistercoris of bird faeces)	
A bacterial species identified by metagenomic analyses. This species includes all bacteria with genomes that show ≥95% average nucleotide identity (ANI) to the type genome, which has been assigned the MAG ID 905 and which is available via NCBI BioSample SAMN15817031. The GC content of the type genome is 63.46% and the genome length is 2.7 Mbp.	
Description of Candidatus Faecivivens gen. nov.	
Candidatus Faecivivens (Fae.ci.vi’vens. L. fem. n. faex dregs; N.L. pres. part. vivens living; N.L. fem. n. Faecivivens a microbe associated with faeces)	
A bacterial genus identified by metagenomic analyses. The genus includes all bacteria with genomes that show ≥60% average amino acid identity (AAI) to the type genome from the type species Candidatus Faecivivens stercorigallinarum. This is a name for the alphanumeric GTDB genus UBA1448. This genus has been assigned by GTDB-Tk v1.3.0 working on GTDB Release 05-RS95 (Chaumeil et al., 2019; Parks et al., 2020) to the order Oscillospirales and to the family Ruminococcaceae.	
Description of Candidatus Faecivivens stercoravium sp. nov.	
Candidatus Faecivivens stercoravium (ster.cor.a’vi.um. L. neut. n. stercus dung; L. fem. n. avis bird; N.L. gen. n. stercoravium of bird faeces)	
A bacterial species identified by metagenomic analyses. This species includes all bacteria with genomes that show ≥95% average nucleotide identity (ANI) to the type genome, which has been assigned the MAG ID CHK189-12415 and which is available via NCBI BioSample SAMN15817018. The GC content of the type genome is 59.75% and the genome length is 2.3 Mbp.	
Description of Candidatus Faecivivens stercorigallinarum sp. nov.	
Candidatus Faecivivens stercorigallinarum (ster.co.ri.gal.li.na’rum. L. neut. n. stercus dung; L. fem. n. gallina hen; N.L. gen. n. stercorigallinarum of hen faeces)	
A bacterial species identified by metagenomic analyses. This species includes all bacteria with genomes that show ≥95% average nucleotide identity (ANI) to the type genome, which has been assigned the MAG ID 4960 and which is available via NCBI BioSample SAMN15817121. The GC content of the type genome is 52.99% and the genome length is 2.2 Mbp.	
Description of Candidatus Faecivivens stercoripullorum sp. nov.	
Candidatus Faecivivens stercoripullorum (ster.co.ri.pul.lo’rum. L. neut. n. stercus dung; L. masc. n. pullus a young chicken; N.L. gen. n. stercoripullorum of the faceces of young chickens)	
A bacterial species identified by metagenomic analyses. This species includes all bacteria with genomes that show ≥95% average nucleotide identity (ANI) to the type genome, which has been assigned the MAG ID ChiBcec7-5410 and which is available via NCBI BioSample SAMN15817124. The GC content of the type genome is 50.38% and the genome length is 2.0 Mbp.	
Description of Candidatus Faecousia gen. nov.	
Candidatus Faecousia (Faec.ou’si.a. L. fem. n. faex dregs; Gr. fem. n. ousia an essence; N.L. fem. n. Faecousia a microbe associated with faeces)	
A bacterial genus identified by metagenomic analyses. The genus includes all bacteria with genomes that show ≥60% average amino acid identity (AAI) to the type genome from the type species Candidatus Faecousia intestinigallinarum. This is a name for the alphanumeric GTDB genus CAG-110. This genus has been assigned by GTDB-Tk v1.3.0 working on GTDB Release 05-RS95 (Chaumeil et al., 2019; Parks et al., 2020) to the order Oscillospirales and to the family Oscillospiraceae.	
Description of Candidatus Faecousia excrementigallinarum sp. nov.	
Candidatus Faecousia excrementigallinarum (ex.cre.men.ti.gal.li.na’rum. L. neut. n. excrementum excrement; L. fem. n. gallina hen; N.L. gen. n. excrementigallinarum of hen excrement)	
A bacterial species identified by metagenomic analyses. This species includes all bacteria with genomes that show ≥95% average nucleotide identity (ANI) to the type genome, which has been assigned the MAG ID 13361 and which is available via NCBI BioSample SAMN15817055. The GC content of the type genome is 56.40% and the genome length is 1.9 Mbp.	
Description of Candidatus Faecousia excrementipullorum sp. nov.	
Candidatus Faecousia excrementipullorum (ex.cre.men.ti.pul.lo’rum. L. neut. n. excrementum excrement; L. masc. n. pullus a young chicken; N.L. gen. n. excrementipullorum of young chicken excrement)	
A bacterial species identified by metagenomic analyses. This species includes all bacteria with genomes that show ≥95% average nucleotide identity (ANI) to the type genome, which has been assigned the MAG ID ChiSxjej6B18-3616 and which is available via NCBI BioSample SAMN15817060. The GC content of the type genome is 56.19% and the genome length is 1.7 Mbp.	

Description of Candidatus Faecousia faecavium sp. nov.	
Candidatus Faecousia faecavium (faec.a’vi.um. L. fem. n. faex, faecis excrement; L. fem. n. avis bird; N.L. gen. n. faecavium of bird faeces)	
A bacterial species identified by metagenomic analyses. This species includes all bacteria with genomes that show ≥95% average nucleotide identity (ANI) to the type genome, which has been assigned the MAG ID ChiBcec21-2751 and which is available via NCBI BioSample SAMN15817064. The GC content of the type genome is 53.76% and the genome length is 2.4 Mbp.	
Description of Candidatus Faecousia faecigallinarum sp. nov.	
Candidatus Faecousia faecigallinarum (fae.ci.gal.li.na’rum. L. fem. n. faex, faecis excrement; L. fem. n. gallina hen; N.L. gen. n. faecigallinarum of hen faeces)	
A bacterial species identified by metagenomic analyses. This species includes all bacteria with genomes that show ≥95% average nucleotide identity (ANI) to the type genome, which has been assigned the MAG ID ChiHcolR29-948 and which is available via NCBI BioSample SAMN15817073. The GC content of the type genome is 58.86% and the genome length is 1.9 Mbp.	
Description of Candidatus Faecousia faecipullorum sp. nov.	
Candidatus Faecousia faecipullorum (fae.ci.pul.lo’rum. L. fem. n. faex, faecis excrement; L. masc. n. pullus a young chicken; N.L. gen. n. faecipullorum of young chicken faeces)	
A bacterial species identified by metagenomic analyses. This species includes all bacteria with genomes that show ≥95% average nucleotide identity (ANI) to the type genome, which has been assigned the MAG ID ChiHecec2B26-1122 and which is available via NCBI BioSample SAMN15817098. The GC content of the type genome is 55.50% and the genome length is 1.9 Mbp.	
Description of Candidatus Faecousia gallistercoris sp. nov.	
Candidatus Faecousia gallistercoris (gal.li.ster’co.ris. L. masc. n gallus chicken; L. neut. n. stercus dung; N.L. gen. n. gallistercoris of chicken faeces)	
A bacterial species identified by metagenomic analyses. This species includes all bacteria with genomes that show ≥95% average nucleotide identity (ANI) to the type genome, which has been assigned the MAG ID 7739 and which is available via NCBI BioSample SAMN15817186. This is a new name for the alphanumeric GTDB species sp900546915. The GC content of the type genome is 58.30% and the genome length is 1.8 Mbp.	
Description of Candidatus Faecousia intestinavium sp. nov.	
Candidatus Faecousia intestinavium (in.tes.tin.a’vi.um. L. neut. n. intestinum gut; L. fem. n. avis bird; N.L. gen. n. intestinavium of the gut of birds)	
A bacterial species identified by metagenomic analyses. This species includes all bacteria with genomes that show ≥95% average nucleotide identity (ANI) to the type genome, which has been assigned the MAG ID ChiHcec3-9842 and which is available via NCBI BioSample SAMN15817111. The GC content of the type genome is 57.16% and the genome length is 2.1 Mbp.	
Description of Candidatus Faecousia intestinigallinarum sp. nov.	
Candidatus Faecousia intestinigallinarum (in.tes.ti.ni.gal.li.na’rum. L. neut. n. intestinum gut; L. fem. n. gallina hen; N.L. gen. n. intestinigallinarum of the gut of the hens)	
A bacterial species identified by metagenomic analyses. This species includes all bacteria with genomes that show ≥95% average nucleotide identity (ANI) to the type genome, which has been assigned the MAG ID ChiSxjej3B15-29383 and which is available via NCBI BioSample SAMN15817112. The GC content of the type genome is 55.74% and the genome length is 2.1 Mbp.	
Description of Candidatus Fimadaptatus gen. nov.	
Candidatus Fimadaptatus (Fim.a.dap.ta’tus. L. neut. n. fimum dung; L. past part. masc. adaptatus adapted to; N.L. masc. n. Fimadaptatus a microbe associated with faeces)	
A bacterial genus identified by metagenomic analyses. The genus includes all bacteria with genomes that show ≥60% average amino acid identity (AAI) to the type genome from the type species Candidatus Fimadaptatus faecigallinarum. This is a name for the alphanumeric GTDB genus UMGS1633. This genus has been assigned by GTDB-Tk v1.3.0 working on GTDB Release 05-RS95 (Chaumeil et al., 2019; Parks et al., 2020) to the order Christensenellales and to the family CAG-74.	
Description of Candidatus Fimadaptatus faecigallinarum sp. nov.	
Candidatus Fimadaptatus faecigallinarum (fae.ci.gal.li.na’rum. L. fem. n. faex, faecis excrement; L. fem. n. gallina hen; N.L. gen. n. faecigallinarum of hen faeces)	
A bacterial species identified by metagenomic analyses. This species includes all bacteria with genomes that show ≥95% average nucleotide identity (ANI) to the type genome, which has been assigned the MAG ID ChiSxjej2B14-8506 and which is available via NCBI BioSample SAMN15817140. The GC content of the type genome is 60.44% and the genome length is 2.8 Mbp.	

Description of Candidatus Fimenecus gen. nov.	
Candidatus Fimenecus (Fim.en.e’cus. L. neut. n. fimum dung; Gr. masc. enoikos inhabitant; N.L. masc. n. Fimenecus a microbe associated with faeces)	
A bacterial genus identified by metagenomic analyses. The genus includes all bacteria with genomes that show ≥60% average amino acid identity (AAI) to the type genome from the type species Candidatus Fimenecus excrementigallinarum. This is a name for the alphanumeric GTDB genus CAG-180. This genus has been assigned by GTDB-Tk v1.3.0 working on GTDB Release 05-RS95 (Chaumeil et al., 2019; Parks et al., 2020) to the order Oscillospirales and to the family Acutalibacteraceae.	
Description of Candidatus Fimenecus excrementavium sp. nov.	
Candidatus Fimenecus excrementavium (ex.cre.ment.a’vi.um. L. neut. n. excrementum excrement; L. fem. n. avis bird; N.L. gen. n. excrementavium of bird excrement)	
A bacterial species identified by metagenomic analyses. This species includes all bacteria with genomes that show ≥95% average nucleotide identity (ANI) to the type genome, which has been assigned the MAG ID ChiSjej1B19-6168 and which is available via NCBI BioSample SAMN15817011. The GC content of the type genome is 50.61% and the genome length is 1.8 Mbp.	
Description of Candidatus Fimenecus excrementigallinarum sp. nov.	
Candidatus Fimenecus excrementigallinarum (ex.cre.men.ti.gal.li.na’rum. L. neut. n. excrementum excrement; L. fem. n. gallina hen; N.L. gen. n. excrementigallinarum of hen excrement)	
A bacterial species identified by metagenomic analyses. This species includes all bacteria with genomes that show ≥95% average nucleotide identity (ANI) to the type genome, which has been assigned the MAG ID ChiGjej1B1-19959 and which is available via NCBI BioSample SAMN15817134. The GC content of the type genome is 60.29% and the genome length is 1.8 Mbp.	
Description of Candidatus Fimenecus stercoravium sp. nov.	
Candidatus Fimenecus stercoravium (ster.cor.a’vi.um. L. neut. n. stercus dung; L. fem. n. avis bird; N.L. gen. n. stercoravium of bird faeces)	
A bacterial species identified by metagenomic analyses. This species includes all bacteria with genomes that show ≥95% average nucleotide identity (ANI) to the type genome, which has been assigned the MAG ID ChiHcolR13-3023 and which is available via NCBI BioSample SAMN15817181. This is a new name for the alphanumeric GTDB species sp002314305. The GC content of the type genome is 55.43% and the genome length is 1.9 Mbp.	
Description of Candidatus Fimicola gen. nov.	
Candidatus Fimicola (Fi.mi’co.la. L. neut. n. fimum dung; L. suff. -cola inhabitant of; N.L. fem. n. Fimicola a microbe associated with faeces)	
A bacterial genus identified by metagenomic analyses. The genus includes all bacteria with genomes that show ≥60% average amino acid identity (AAI) to the type genome from the type species Candidatus Fimicola merdigallinarum. This is a name for the alphanumeric GTDB genus An114. This genus has been assigned by GTDB-Tk v1.3.0 working on GTDB Release 05-RS95 (Chaumeil et al., 2019; Parks et al., 2020) to the order Lachnospirales and to the family Anaerotignaceae.	
Description of Candidatus Fimicola cottocaccae sp. nov.	
Candidatus Fimicola cottocaccae (cot.to.cac’cae. Gr. masc. n. kottos chicken Gr. fem. n. kakke faeces; N.L. gen. n. cottocaccae of chicken faeces)	
A bacterial species identified by metagenomic analyses. This species includes all bacteria with genomes that show ≥95% average nucleotide identity (ANI) to the type genome, which has been assigned the MAG ID ChiW9-1577 and which is available via NCBI BioSample SAMN15817191. This is a new name for the alphanumeric GTDB species sp002161055. The GC content of the type genome is 31.91% and the genome length is 1.8 Mbp.	
Description of Candidatus Fimicola merdigallinarum sp. nov.	
Candidatus Fimicola merdigallinarum (mer.di.gal.li.na’rum. L. fem. n. merda faeces; L. fem. n. gallina hen; N.L. gen. n. merdigallinarum of hen faeces)	
A bacterial species identified by metagenomic analyses. This species includes all bacteria with genomes that show ≥95% average nucleotide identity (ANI) to the type genome, which has been assigned the MAG ID F6-4510 and which is available via NCBI BioSample SAMN15817136. The GC content of the type genome is 32.46% and the genome length is 1.8 Mbp.	
Description of Candidatus Fimihabitans gen. nov.	
Candidatus Fimihabitans (Fi.mi.ha’bi.tans. L. neut. n. fimum dung; L. pres. part. habitans an inhabitant; N.L. fem. n. Fimihabitans a microbe associated with faeces)	
A bacterial genus identified by metagenomic analyses. The genus includes all bacteria with genomes that show ≥60% average amino acid identity (AAI) to the type genome from the type species Candidatus Fimihabitans intestinipullorum. This is a name for the alphanumeric GTDB genus UMGS1648. This genus has been assigned by GTDB-Tk v1.3.0 working on GTDB Release 05-RS95 (Chaumeil et al., 2019; Parks et al., 2020) to the order RF39 and to the family CAG-822.	

Description of Candidatus Fimihabitans intestinipullorum sp. nov.	
Candidatus Fimihabitans intestinipullorum (in.tes.ti.ni.pul.lo’rum. L. neut. n. intestinum gut; L. masc. n. pullus a young chicken; N.L. gen. n. intestinipullorum of the gut of young chickens)	
A bacterial species identified by metagenomic analyses. This species includes all bacteria with genomes that show ≥95% average nucleotide identity (ANI) to the type genome, which has been assigned the MAG ID CHK197-8231 and which is available via NCBI BioSample SAMN15817229. This is a new name for the alphanumeric GTDB species sp900553765. The GC content of the type genome is 33.44% and the genome length is 1.3 Mbp.	
Description of Candidatus Fimimonas gen. nov.	
Candidatus Fimimonas (Fi.mi.mo’nas. L. neut. n. fimum dung; L. fem. n. monas a monad; N.L. fem. n. Fimimonas a microbe associated with faeces)	
A bacterial genus identified by metagenomic analyses. The genus includes all bacteria with genomes that show ≥60% average amino acid identity (AAI) to the type genome from the type species Candidatus Fimimonas gallinarum. This is a name for the alphanumeric GTDB genus CAG-1435. This genus has been assigned by GTDB-Tk v1.3.0 working on GTDB Release 05-RS95 (Chaumeil et al., 2019; Parks et al., 2020) to the order Christensenellales and to the family CAG-314.	
Description of Candidatus Fimimonas gallinarum sp. nov.	
Candidatus Fimimonas gallinarum (gal.li.na’rum. L. fem. n. gallina a hen; L. gen. fem. pl. n. gallinarum of hens)	
A bacterial species identified by metagenomic analyses. This species includes all bacteria with genomes that show ≥95% average nucleotide identity (ANI) to the type genome, which has been assigned the MAG ID CHK121-14286 and which is available via NCBI BioSample SAMN15817176. This is a new name for the alphanumeric GTDB species sp000433775. The GC content of the type genome is 45.96% and the genome length is 1.4 Mbp.	
Description of Candidatus Fimimonas merdipullorum sp. nov.	
Candidatus Fimimonas merdipullorum (mer.di.pul.lo’rum. L. fem. n. merda faeces; L. masc. n. pullus a young chicken; N.L. gen. n. merdipullorum of the faeces of young chickens)	
A bacterial species identified by metagenomic analyses. This species includes all bacteria with genomes that show ≥95% average nucleotide identity (ANI) to the type genome, which has been assigned the MAG ID ChiHjej12B11-7776 and which is available via NCBI BioSample SAMN15817153. The GC content of the type genome is 53.17% and the genome length is 1.3 Mbp.	
Description of Candidatus Fimimorpha gen. nov.	
Candidatus Fimimorpha (Fi.mi.mor’pha. L. neut. n. fimum dung; Gr. fem. n. morphe a form, shape; N.L. fem. n. Fimimorpha a microbe associated with faeces)	
A bacterial genus identified by metagenomic analyses. The genus includes all bacteria with genomes that show ≥60% average amino acid identity (AAI) to the type genome from the type species Candidatus Fimimorpha faecalis. This is a name for the alphanumeric GTDB genus CHKCI001. This genus has been assigned by GTDB-Tk v1.3.0 working on GTDB Release 05-RS95 (Chaumeil et al., 2019; Parks et al., 2020) to the order Lachnospirales and to the family Lachnospiraceae.	
Description of Candidatus Fimimorpha excrementavium sp. nov.	
Candidatus Fimimorpha excrementavium (ex.cre.ment.a’vi.um. L. neut. n. excrementum excrement; L. fem. n. avis bird; N.L. gen. n. excrementavium of bird excrement)	
A bacterial species identified by metagenomic analyses. This species includes all bacteria with genomes that show ≥95% average nucleotide identity (ANI) to the type genome, which has been assigned the MAG ID CHK193-21555 and which is available via NCBI BioSample SAMN15817029. The GC content of the type genome is 48.70% and the genome length is 3.1 Mbp.	
Description of Candidatus Fimimorpha faecalis sp. nov.	
Candidatus Fimimorpha faecalis (fae.ca’lis. L. fem. adj. faecalis of faeces)	
A bacterial species identified by metagenomic analyses. This species includes all bacteria with genomes that show ≥95% average nucleotide identity (ANI) to the type genome, which has been assigned the MAG ID ChiW13-3771 and which is available via NCBI BioSample SAMN15817177. This is a new name for the alphanumeric GTDB species sp900045905. The GC content of the type genome is 36.24% and the genome length is 2.9 Mbp.	
Description of Candidatus Fimiplasma gen. nov.	
Candidatus Fimiplasma (Fi.mi.plas’ma. L. neut. n. fimum dung; Gr. neut. n. plasma a form; N.L. neut. n. Fimiplasma a microbe associated with faeces)	
A bacterial genus identified by metagenomic analyses. The genus includes all bacteria with genomes that show ≥60% average amino acid identity (AAI) to the type genome from the type species Candidatus Fimiplasma intestinipullorum. This is a name for the alphanumeric GTDB genus CHKCI006. This genus has been assigned by GTDB-Tk v1.3.0 working on GTDB Release 05-RS95 (Chaumeil et al., 2019; Parks et al., 2020) to the order Erysipelotrichales and to the family Erysipelatoclostridiaceae.	

Description of Candidatus Fimiplasma intestinipullorum sp. nov.	
Candidatus Fimiplasma intestinipullorum (in.tes.ti.ni.pul.lo’rum. L. neut. n. intestinum gut; L. masc. n. pullus a young chicken; N.L. gen. n. intestinipullorum of the gut of young chickens)	
A bacterial species identified by metagenomic analyses. This species includes all bacteria with genomes that show ≥95% average nucleotide identity (ANI) to the type genome, which has been assigned the MAG ID CHK195-11698 and which is available via NCBI BioSample SAMN15817196. This is a new name for the alphanumeric GTDB species sp900018345. The GC content of the type genome is 43.31% and the genome length is 2.5 Mbp.	
Description of Candidatus Fimisoma gen. nov.	
Candidatus Fimisoma (Fi.mi.so’ma. L. neut. n. fimum dung; Gr. neut. n. soma a body; N.L. neut. n. Fimisoma a microbe associated with faeces)	
A bacterial genus identified by metagenomic analyses. The genus includes all bacteria with genomes that show ≥60% average amino acid identity (AAI) to the type genome from the type species Candidatus Fimisoma avicola. This is a name for the alphanumeric GTDB genus CAG-145. This genus has been assigned by GTDB-Tk v1.3.0 working on GTDB Release 05-RS95 (Chaumeil et al., 2019; Parks et al., 2020) to the order Peptostreptococcales and to the family Anaerovoracaceae.	
Description of Candidatus Fimisoma avicola sp. nov.	
Candidatus Fimisoma avicola (a.vi’co.la. L. fem. n. avis bird; L. suff. -cola inhabitant of; N.L. n. avicola inhabitant of birds)	
A bacterial species identified by metagenomic analyses. This species includes all bacteria with genomes that show ≥95% average nucleotide identity (ANI) to the type genome, which has been assigned the MAG ID 11300 and which is available via NCBI BioSample SAMN15817197. This is a new name for the alphanumeric GTDB species sp900542565. The GC content of the type genome is 47.90% and the genome length is 2.0 Mbp.	
Description of Candidatus Fimivicinus gen. nov.	
Candidatus Fimivicinus (Fi.mi.vi.ci’nus. L. neut. n. fimum dung; L. masc. n. vicinus a neighbour; N.L. masc. n. Fimivicinus a microbe associated with faeces)	
A bacterial genus identified by metagenomic analyses. The genus includes all bacteria with genomes that show ≥60% average amino acid identity (AAI) to the type genome from the type species Candidatus Fimivicinus intestinavium. This is a name for the alphanumeric GTDB genus UBA1691. This genus has been assigned by GTDB-Tk v1.3.0 working on GTDB Release 05-RS95 (Chaumeil et al., 2019; Parks et al., 2020) to the order Oscillospirales and to the family Acutalibacteraceae.	
Description of Candidatus Fimivicinus intestinavium sp. nov.	
Candidatus Fimivicinus intestinavium (in.tes.tin.a’vi.um. L. neut. n. intestinum gut; L. fem. n. avis bird; N.L. gen. n. intestinavium of the gut of birds)	
A bacterial species identified by metagenomic analyses. This species includes all bacteria with genomes that show ≥95% average nucleotide identity (ANI) to the type genome, which has been assigned the MAG ID 2526 and which is available via NCBI BioSample SAMN15817188. This is a new name for the alphanumeric GTDB species sp900552985. The GC content of the type genome is 55.20% and the genome length is 2.5 Mbp.	
Description of Candidatus Fimivivens gen. nov.	
Candidatus Fimivivens (Fi.mi.vi’vens. L. neut. n. fimum dung; N.L. pres. part. vivens living; N.L. fem. n. Fimivivens a microbe associated with faeces)	
A bacterial genus identified by metagenomic analyses. The genus includes all bacteria with genomes that show ≥60% average amino acid identity (AAI) to the type genome from the type species Candidatus Fimivivens faecavium. This is a name for the alphanumeric GTDB genus D5. This genus has been assigned by GTDB-Tk v1.3.0 working on GTDB Release 05-RS95 (Chaumeil et al., 2019; Parks et al., 2020) to the order Oscillospirales and to the family Ruminococcaceae.	
Description of Candidatus Fimivivens faecavium sp. nov.	
Candidatus Fimivivens faecavium (faec.a’vi.um. L. fem. n. faex, faecis excrement; L. fem. n. avis bird; N.L. gen. n. faecavium of bird faeces)	
A bacterial species identified by metagenomic analyses. This species includes all bacteria with genomes that show ≥95% average nucleotide identity (ANI) to the type genome, which has been assigned the MAG ID CHK195-35099 and which is available via NCBI BioSample SAMN15817038. The GC content of the type genome is 58.86% and the genome length is 2.0 Mbp.	
Description of Candidatus Fimousia gen. nov.	
Candidatus Fimousia (Fim.ou’si.a. L. neut. n. fimum dung; Gr. fem. n. ousia an essence; N.L. fem. n. Fimousia a microbe associated with faeces)	
A bacterial genus identified by metagenomic analyses. The genus includes all bacteria with genomes that show ≥60% average amino acid identity (AAI) to the type genome from the type species Candidatus Fimousia stercorigallinarum. This is a name for the alphanumeric GTDB genus 992a. This genus has been assigned by GTDB-Tk v1.3.0 working on GTDB Release 05-RS95 (Chaumeil et al., 2019; Parks et al., 2020) to the order Lachnospirales and to the family Lachnospiraceae.	

Description of Candidatus Fimousia stercorigallinarum sp. nov.	
Candidatus Fimousia stercorigallinarum (ster.co.ri.gal.li.na’rum. L. neut. n. stercus dung; L. fem. n. gallina hen; N.L. gen. n. stercorigallinarum of hen faeces)	
A bacterial species identified by metagenomic analyses. This species includes all bacteria with genomes that show ≥95% average nucleotide identity (ANI) to the type genome, which has been assigned the MAG ID ChiSxjej3B15-1827 and which is available via NCBI BioSample SAMN15817114. The GC content of the type genome is 41.52% and the genome length is 2.3 Mbp.	
Description of Candidatus Flavonifractor avicola sp. nov.	
Candidatus Flavonifractor avicola (a.vi’co.la. L. fem. n. avis bird; L. suff. -cola inhabitant of; N.L. n. avicola inhabitant of birds)	
A bacterial species identified by metagenomic analyses. This species includes all bacteria with genomes that show ≥95% average nucleotide identity (ANI) to the type genome, which has been assigned the MAG ID CHK178-4001 and which is available via NCBI BioSample SAMN15816843. This is a new name for the alphanumeric GTDB species sp002161085. The GC content of the type genome is 60.44% and the genome length is 2.5 Mbp.	
Description of Candidatus Flavonifractor avistercoris sp. nov.	
Candidatus Flavonifractor avistercoris (a.vi.ster’co.ris. L. fem. n. avis bird; L. neut. n. stercus dung; N.L. gen. n. avistercoris of bird faeces)	
A bacterial species identified by metagenomic analyses. This species includes all bacteria with genomes that show ≥95% average nucleotide identity (ANI) to the type genome, which has been assigned the MAG ID 6084 and which is available via NCBI BioSample SAMN15816821. This is a new name for the alphanumeric GTDB species sp002161215. The GC content of the type genome is 65.16% and the genome length is 2.4 Mbp.	
Description of Candidatus Flavonifractor intestinigallinarum sp. nov.	
Candidatus Flavonifractor intestinigallinarum (in.tes.ti.ni.gal.li.na’rum. L. neut. n. intestinum gut; L. fem. n. gallina hen; N.L. gen. n. intestinigallinarum of the gut of the hens)	
A bacterial species identified by metagenomic analyses. This species includes all bacteria with genomes that show ≥95% average nucleotide identity (ANI) to the type genome, which has been assigned the MAG ID CHK192-8294 and which is available via NCBI BioSample SAMN15816592. The GC content of the type genome is 61.61% and the genome length is 2.5 Mbp.	
Description of Candidatus Flavonifractor intestinipullorum sp. nov.	
Candidatus Flavonifractor intestinipullorum (in.tes.ti.ni.pul.lo’rum. L. neut. n. intestinum gut; L. masc. n. pullus a young chicken; N.L. gen. n. intestinipullorum of the gut of young chickens)	
A bacterial species identified by metagenomic analyses. This species includes all bacteria with genomes that show ≥95% average nucleotide identity (ANI) to the type genome, which has been assigned the MAG ID CHK189-11263 and which is available via NCBI BioSample SAMN15816594. The GC content of the type genome is 63.72% and the genome length is 2.2 Mbp.	
Description of Candidatus Flavonifractor merdavium sp. nov.	
Candidatus Flavonifractor merdavium (merd.a’vi.um. L. fem. n. merda faeces; L. fem. n. avis bird; N.L. gen. n. merdavium of bird faeces)	
A bacterial species identified by metagenomic analyses. This species includes all bacteria with genomes that show ≥95% average nucleotide identity (ANI) to the type genome, which has been assigned the MAG ID 3313 and which is available via NCBI BioSample SAMN15816644. The GC content of the type genome is 62.46% and the genome length is 2.7 Mbp.	
Description of Candidatus Flavonifractor merdigallinarum sp. nov.	
Candidatus Flavonifractor merdigallinarum (mer.di.gal.li.na’rum. L. fem. n. merda faeces; L. fem. n. gallina hen; N.L. gen. n. merdigallinarum of hen faeces)	
A bacterial species identified by metagenomic analyses. This species includes all bacteria with genomes that show ≥95% average nucleotide identity (ANI) to the type genome, which has been assigned the MAG ID ChiBcec16-6824 and which is available via NCBI BioSample SAMN15816721. The GC content of the type genome is 61.09% and the genome length is 2.6 Mbp.	
Description of Candidatus Flavonifractor merdipullorum sp. nov.	
Candidatus Flavonifractor merdipullorum (mer.di.pul.lo’rum. L. fem. n. merda faeces; L. masc. n. pullus a young chicken; N.L. gen. n. merdipullorum of the faeces of young chickens)	
A bacterial species identified by metagenomic analyses. This species includes all bacteria with genomes that show ≥95% average nucleotide identity (ANI) to the type genome, which has been assigned the MAG ID ChiGjej6B6-1540 and which is available via NCBI BioSample SAMN15816748. The GC content of the type genome is 61.06% and the genome length is 2.1 Mbp.	

Description of Candidatus Fournierella excrementavium sp. nov.	
Candidatus Fournierella excrementavium (ex.cre.ment.a’vi.um. L. neut. n. excrementum excrement; L. fem. n. avis bird; N.L. gen. n. excrementavium of bird excrement)	
A bacterial species identified by metagenomic analyses. This species includes all bacteria with genomes that show ≥95% average nucleotide identity (ANI) to the type genome, which has been assigned the MAG ID ChiHcec27-1717 and which is available via NCBI BioSample SAMN15816881. This is a new name for the alphanumeric GTDB species sp004558145. The GC content of the type genome is 63.90% and the genome length is 2.4 Mbp.	
Description of Candidatus Fournierella excrementigallinarum sp. nov.	
Candidatus Fournierella excrementigallinarum (ex.cre.men.ti.gal.li.na’rum. L. neut. n. excrementum excrement; L. fem. n. gallina hen; N.L. gen. n. excrementigallinarum of hen excrement)	
A bacterial species identified by metagenomic analyses. This species includes all bacteria with genomes that show ≥95% average nucleotide identity (ANI) to the type genome, which has been assigned the MAG ID 1136 and which is available via NCBI BioSample SAMN15816650. The GC content of the type genome is 64.27% and the genome length is 2.1 Mbp.	
Description of Candidatus Fournierella merdavium sp. nov.	
Candidatus Fournierella merdavium (merd.a’vi.um. L. fem. n. merda faeces; L. fem. n. avis bird; N.L. gen. n. merdavium of bird faeces)	
A bacterial species identified by metagenomic analyses. This species includes all bacteria with genomes that show ≥95% average nucleotide identity (ANI) to the type genome, which has been assigned the MAG ID ChiBcec4-1730 and which is available via NCBI BioSample SAMN15816653. The GC content of the type genome is 64.33% and the genome length is 2.6 Mbp.	
Description of Candidatus Fournierella merdigallinarum sp. nov.	
Candidatus Fournierella merdigallinarum (mer.di.gal.li.na’rum. L. fem. n. merda faeces; L. fem. n. gallina hen; N.L. gen. n. merdigallinarum of hen faeces)	
A bacterial species identified by metagenomic analyses. This species includes all bacteria with genomes that show ≥95% average nucleotide identity (ANI) to the type genome, which has been assigned the MAG ID 6296 and which is available via NCBI BioSample SAMN15816675. The GC content of the type genome is 65.05% and the genome length is 2.4 Mbp.	
Description of Candidatus Fournierella merdipullorum sp. nov.	
Candidatus Fournierella merdipullorum (mer.di.pul.lo’rum. L. fem. n. merda faeces; L. masc. n. pullus a young chicken; N.L. gen. n. merdipullorum of the faeces of young chickens)	
A bacterial species identified by metagenomic analyses. This species includes all bacteria with genomes that show ≥95% average nucleotide identity (ANI) to the type genome, which has been assigned the MAG ID ChiGjej4B4-18154 and which is available via NCBI BioSample SAMN15816693. The GC content of the type genome is 62.57% and the genome length is 2.5 Mbp.	
Description of Candidatus Fournierella pullicola sp. nov.	
Candidatus Fournierella pullicola (pul.li’co.la. L. masc. n. pullus a young chicken; L. suff. -cola inhabitant of; N.L. n. pullicola an inhabitant of young chickens)	
A bacterial species identified by metagenomic analyses. This species includes all bacteria with genomes that show ≥95% average nucleotide identity (ANI) to the type genome, which has been assigned the MAG ID 2239 and which is available via NCBI BioSample SAMN15816745. The GC content of the type genome is 62.57% and the genome length is 2.4 Mbp.	
Description of Candidatus Fournierella pullistercoris sp. nov.	
Candidatus Fournierella pullistercoris (pul.li.ster’co.ris. L. masc. n. pullus a young chicken; L. neut. n. stercus dung; N.L. gen. n. pullistercoris of young chicken faeces)	
A bacterial species identified by metagenomic analyses. This species includes all bacteria with genomes that show ≥95% average nucleotide identity (ANI) to the type genome, which has been assigned the MAG ID B5-2728 and which is available via NCBI BioSample SAMN15816762. The GC content of the type genome is 52.45% and the genome length is 1.7 Mbp.	
Description of Candidatus Fusicatenibacter intestinigallinarum sp. nov.	
Candidatus Fusicatenibacter intestinigallinarum (in.tes.ti.ni.gal.li.na’rum. L. neut. n. intestinum gut; L. fem. n. gallina hen; N.L. gen. n. intestinigallinarum of the gut of the hens)	
A bacterial species identified by metagenomic analyses. This species includes all bacteria with genomes that show ≥95% average nucleotide identity (ANI) to the type genome, which has been assigned the MAG ID CHK185-5351 and which is available via NCBI BioSample SAMN15816585. The GC content of the type genome is 51.22% and the genome length is 2.9 Mbp.	

Description of Candidatus Fusicatenibacter intestinipullorum sp. nov.	
Candidatus Fusicatenibacter intestinipullorum (in.tes.ti.ni.pul.lo’rum. L. neut. n. intestinum gut; L. masc. n. pullus a young chicken; N.L. gen. n. intestinipullorum of the gut of young chickens)	
A bacterial species identified by metagenomic analyses. This species includes all bacteria with genomes that show ≥95% average nucleotide identity (ANI) to the type genome, which has been assigned the MAG ID ChiBcec11-5794 and which is available via NCBI BioSample SAMN15816833. This is a new name for the alphanumeric GTDB species sp900543115. The GC content of the type genome is 49.70% and the genome length is 2.6 Mbp.	
Description of Candidatus Fusicatenibacter merdavium sp. nov.	
Candidatus Fusicatenibacter merdavium (merd.a’vi.um. L. fem. n. merda faeces; L. fem. n. avis bird; N.L. gen. n. merdavium of bird faeces)	
A bacterial species identified by metagenomic analyses. This species includes all bacteria with genomes that show ≥95% average nucleotide identity (ANI) to the type genome, which has been assigned the MAG ID CHK183-1962 and which is available via NCBI BioSample SAMN15816614. The GC content of the type genome is 51.02% and the genome length is 2.7 Mbp.	
Description of Candidatus Fusobacterium pullicola sp. nov.	
Candidatus Fusobacterium pullicola (pul.li’co.la. L. masc. n. pullus a young chicken; L. suff. -cola inhabitant of; N.L. n. pullicola inhabitant of a young chicken)	
A bacterial species identified by metagenomic analyses. This species includes all bacteria with genomes that show ≥95% average nucleotide identity (ANI) to the type genome, which has been assigned the MAG ID A6-441 and which is available via NCBI BioSample SAMN15816927. This is a new name for the alphanumeric GTDB species sp900549465. Although GTDB has assigned this species to the genus it calls Fusobacterium_A, this genus designation cannot be incorporated into a well-formed binomial, so in naming this species, we have used the current validly published name for the genus. The GC content of the type genome is 29.86% and the genome length is 1.8 Mbp.	
Description of Candidatus Gallacutalibacter gen. nov.	
Candidatus Gallacutalibacter (Gall.a.cu.ta.li.bac’ter. L. masc. n. gallus chicken; N.L. masc. n. Acutalibacter a genus name; N.L. masc. n. Gallacutalibacter a genus related to the genus Acutalibacter but distinct from it and found in poultry)	
A bacterial genus identified by metagenomic analyses. The genus includes all bacteria with genomes that show ≥60% average amino acid identity (AAI) to the type genome from the type species Candidatus Gallacutalibacter pullicola. This genus was identified but not named by Glendinning et al. (2020). This genus has been assigned by GTDB-Tk v1.3.0 working on GTDB Release 05-RS95 (Chaumeil et al., 2019; Parks et al., 2020) to the order Oscillospirales and to the family Acutalibacteraceae.	
Description of Candidatus Gallacutalibacter pullicola sp. nov.	
Candidatus Gallacutalibacter pullicola (pul.li’co.la. L. masc. n. pullus a young chicken; L. suff. -cola inhabitant of; N.L. n. pullicola an inhabitant of young chickens)	
A bacterial species identified by metagenomic analyses. This species includes all bacteria with genomes that show ≥95% average nucleotide identity (ANI) to the type genome, which has been assigned the MAG ID ChiSjej1B19-7085 and which is available via NCBI BioSample SAMN15816935. The GC content of the type genome is 56.02% and the genome length is 2.5 Mbp.	
Description of Candidatus Gallacutalibacter pullistercoris sp. nov.	
Candidatus Gallacutalibacter pullistercoris (pul.li.ster’co.ris. L. masc. n. pullus a young chicken; L. neut. n. stercus dung; N.L. gen. n. pullistercoris of young chicken faeces)	
A bacterial species identified by metagenomic analyses. This species includes all bacteria with genomes that show ≥95% average nucleotide identity (ANI) to the type genome, which has been assigned the MAG ID 13869 and which is available via NCBI BioSample SAMN15816961. The GC content of the type genome is 51.21% and the genome length is 2.4 Mbp.	
Description of Candidatus Gallacutalibacter stercoravium sp. nov.	
Candidatus Gallacutalibacter stercoravium (ster.cor.a’vi.um. L. neut. n. stercus dung; L. fem. n. avis bird; N.L. gen. n. stercoravium of bird faeces)	
A bacterial species identified by metagenomic analyses. This species includes all bacteria with genomes that show ≥95% average nucleotide identity (ANI) to the type genome, which has been assigned the MAG ID CHK176-13069 and which is available via NCBI BioSample SAMN15816939. The GC content of the type genome is 51.38% and the genome length is 2.7 Mbp.	
Description of Candidatus Gallibacteroides gen. nov.	
Candidatus Gallibacteroides (Gal.li.bac.te.ro’i.des. L. masc. n. gallus chicken; N.L. masc. n. Bacteroides a genus name; N.L. masc. n. Gallibacteroides a genus related to the genus Bacteroides but distinct from it and found in poultry)	
A bacterial genus identified by metagenomic analyses. The genus includes all bacteria with genomes that show ≥60% average amino acid identity (AAI) to the type genome from the type species Candidatus Gallibacteroides avistercoris. This genus has been assigned by GTDB-Tk v1.3.0 working on GTDB Release 05-RS95 (Chaumeil et al., 2019; Parks et al., 2020) to the order Bacteroidales and to the family Barnesiellaceae.	

Description of Candidatus Gallibacteroides avistercoris sp. nov.	
Candidatus Gallibacteroides avistercoris (a.vi.ster’co.ris. L. fem. n. avis bird; L. neut. n. stercus dung; N.L. gen. n. avistercoris of bird faeces)	
A bacterial species identified by metagenomic analyses. This species includes all bacteria with genomes that show ≥95% average nucleotide identity (ANI) to the type genome, which has been assigned the MAG ID CHK158-818 and which is available via NCBI BioSample SAMN15816984. The GC content of the type genome is 46.12% and the genome length is 2.3 Mbp.	
Description of Candidatus Galligastranaerophilus gen. nov.	
Candidatus Galligastranaerophilus (Gal.li.gastr.an.a.e.ro’phi.lus. L. masc. n. gallus chicken; N.L. masc. n. Gastranaerophilus a genus name; N.L. masc. n. Galligastranaerophilus a genus related to the genus Gastranaerophilus but distinct from it and found in poultry)	
A bacterial genus identified by metagenomic analyses. The genus includes all bacteria with genomes that show ≥60% average amino acid identity (AAI) to the type genome from the type species Candidatus Galligastranaerophilus faecipullorum. This genus has been assigned by GTDB-Tk v1.3.0 working on GTDB Release 05-RS95 (Chaumeil et al., 2019; Parks et al., 2020) to the order Gastranaerophilales and to the family Gastranaerophilaceae.	
Description of Candidatus Galligastranaerophilus faecipullorum sp. nov.	
Candidatus Galligastranaerophilus faecipullorum (fae.ci.pul.lo’rum. L. fem. n. faex, faecis excrement; L. masc. n. pullus a young chicken; N.L. gen. n. faecipullorum of young chicken faeces)	
A bacterial species identified by metagenomic analyses. This species includes all bacteria with genomes that show ≥95% average nucleotide identity (ANI) to the type genome, which has been assigned the MAG ID ChiW23-1657 and which is available via NCBI BioSample SAMN15816949. The GC content of the type genome is 39.52% and the genome length is 1.7 Mbp.	
Description of Candidatus Galligastranaerophilus gallistercoris sp. nov.	
Candidatus Galligastranaerophilus gallistercoris (gal.li.ster’co.ris. L. masc. n gallus chicken; L. neut. n. stercus dung; N.L. gen. n. gallistercoris of chicken faeces)	
A bacterial species identified by metagenomic analyses. This species includes all bacteria with genomes that show ≥95% average nucleotide identity (ANI) to the type genome, which has been assigned the MAG ID CHK123-4750 and which is available via NCBI BioSample SAMN15816963. The GC content of the type genome is 35.45% and the genome length is 1.8 Mbp.	
Description of Candidatus Galligastranaerophilus intestinavium sp. nov.	
Candidatus Galligastranaerophilus intestinavium (in.tes.tin.a’vi.um. L. neut. n. intestinum gut; L. fem. n. avis bird; N.L. gen. n. intestinavium of the gut of birds)	
A bacterial species identified by metagenomic analyses. This species includes all bacteria with genomes that show ≥95% average nucleotide identity (ANI) to the type genome, which has been assigned the MAG ID CHK152-2871 and which is available via NCBI BioSample SAMN15816967. The GC content of the type genome is 35.80% and the genome length is 1.6 Mbp.	
Description of Candidatus Galligastranaerophilus intestinigallinarum sp. nov.	
Candidatus Galligastranaerophilus intestinigallinarum (in.tes.ti.ni.gal.li.na’rum. L. neut. n. intestinum gut; L. fem. n. gallina hen; N.L. gen. n. intestinigallinarum of the gut of the hens)	
A bacterial species identified by metagenomic analyses. This species includes all bacteria with genomes that show ≥95% average nucleotide identity (ANI) to the type genome, which has been assigned the MAG ID CHK123-3438 and which is available via NCBI BioSample SAMN15816973. The GC content of the type genome is 32.50% and the genome length is 1.7 Mbp.	
Description of Candidatus Gallilactobacillus gen. nov.	
Candidatus Gallilactobacillus (Gal.li.lac.to.ba.cil’lus. L. masc. n. gallus chicken; N.L. masc. n. Lactobacillus a genus name; N.L. masc. n. Gallilactobacillus a genus related to the genus Lactobacillus but distinct from it and found in poultry)	
A bacterial genus identified by metagenomic analyses. The genus includes all bacteria with genomes that show ≥60% average amino acid identity (AAI) to the type genome from the type species Candidatus Gallilactobacillus intestinavium. This genus has been assigned by GTDB-Tk v1.3.0 working on GTDB Release 05-RS95 (Chaumeil et al., 2019; Parks et al., 2020) to the order Lactobacillales and to the family Lactobacillaceae.	
Description of Candidatus Gallilactobacillus intestinavium sp. nov.	
Candidatus Gallilactobacillus intestinavium (in.tes.tin.a’vi.um. L. neut. n. intestinum gut; L. fem. n. avis bird; N.L. gen. n. intestinavium of the gut of birds)	
A bacterial species identified by metagenomic analyses. This species includes all bacteria with genomes that show ≥95% average nucleotide identity (ANI) to the type genome, which has been assigned the MAG ID C6-149 and which is available via NCBI BioSample SAMN15816970. The GC content of the type genome is 29.69% and the genome length is 1.2 Mbp.	

Description of Candidatus Gallimonas gallistercoris sp. nov.	
Candidatus Gallimonas gallistercoris (gal.li.ster’co.ris. L. masc. n gallus chicken; L. neut. n. stercus dung; N.L. gen. n. gallistercoris of chicken faeces)	
A bacterial species identified by metagenomic analyses. This species includes all bacteria with genomes that show ≥95% average nucleotide identity (ANI) to the type genome, which has been assigned the MAG ID CHK156-179 and which is available via NCBI BioSample SAMN15816677. The GC content of the type genome is 58.55% and the genome length is 1.6 Mbp.	
Description of Candidatus Gallimonas intestinavium sp. nov.	
Candidatus Gallimonas intestinavium (in.tes.tin.a’vi.um. L. neut. n. intestinum gut; L. fem. n. avis bird; N.L. gen. n. intestinavium of the gut of birds)	
A bacterial species identified by metagenomic analyses. This species includes all bacteria with genomes that show ≥95% average nucleotide identity (ANI) to the type genome, which has been assigned the MAG ID ChiW7-2402 and which is available via NCBI BioSample SAMN15816844. This is a new name for the alphanumeric GTDB species sp003343805. The GC content of the type genome is 58.63% and the genome length is 1.8 Mbp.	
Description of Candidatus Gallimonas intestinigallinarum sp. nov.	
Candidatus Gallimonas intestinigallinarum (in.tes.ti.ni.gal.li.na’rum. L. neut. n. intestinum gut; L. fem. n. gallina hen; N.L. gen. n. intestinigallinarum of the gut of the hens)	
A bacterial species identified by metagenomic analyses. This species includes all bacteria with genomes that show ≥95% average nucleotide identity (ANI) to the type genome, which has been assigned the MAG ID CHK33-5263 and which is available via NCBI BioSample SAMN15816692. The GC content of the type genome is 57.82% and the genome length is 1.6 Mbp.	
Description of Candidatus Gallipaludibacter gen. nov.	
Candidatus Gallipaludibacter (Gal.li.pa.lu.di.bac’ter. L. masc. n. gallus chicken; N.L. masc. n. Paludibacter a genus name; N.L. masc. n. Gallipaludibacter a genus related to the genus Paludibacter but distinct from it and found in poultry)	
A bacterial genus identified by metagenomic analyses. The genus includes all bacteria with genomes that show ≥60% average amino acid identity (AAI) to the type genome from the type species Candidatus Gallipaludibacter merdavium. This genus has been assigned by GTDB-Tk v1.3.0 working on GTDB Release 05-RS95 (Chaumeil et al., 2019; Parks et al., 2020) to the order Bacteroidales and to the family Paludibacteraceae.	
Description of Candidatus Gallipaludibacter merdavium sp. nov.	
Candidatus Gallipaludibacter merdavium (merd.a’vi.um. L. fem. n. merda faeces; L. fem. n. avis bird; N.L. gen. n. merdavium of bird faeces)	
A bacterial species identified by metagenomic analyses. This species includes all bacteria with genomes that show ≥95% average nucleotide identity (ANI) to the type genome, which has been assigned the MAG ID G3-3990 and which is available via NCBI BioSample SAMN15816954. The GC content of the type genome is 41.91% and the genome length is 2.9 Mbp.	
Description of Candidatus Gallitreponema gen. nov.	
Candidatus Gallitreponema (Gal.li.tre.po.ne’ma. L. masc. n. gallus chicken; N.L. neut. n. Treponema a genus name; N.L. neut. n. Gallitreponema a genus related to the genus Treponema but distinct from it and found in poultry)	
A bacterial genus identified by metagenomic analyses. The genus includes all bacteria with genomes that show ≥60% average amino acid identity (AAI) to the type genome from the type species Candidatus Gallitreponema excrementavium. This genus has been assigned by GTDB-Tk v1.3.0 working on GTDB Release 05-RS95 (Chaumeil et al., 2019; Parks et al., 2020) to the order Treponematales and to the family Treponemataceae.	
Description of Candidatus Gallitreponema excrementavium sp. nov.	
Candidatus Gallitreponema excrementavium (ex.cre.ment.a’vi.um. L. neut. n. excrementum excrement; L. fem. n. avis bird; N.L. gen. n. excrementavium of bird excrement)	
A bacterial species identified by metagenomic analyses. This species includes all bacteria with genomes that show ≥95% average nucleotide identity (ANI) to the type genome, which has been assigned the MAG ID 10532 and which is available via NCBI BioSample SAMN15816962. The GC content of the type genome is 40.13% and the genome length is 2.4 Mbp.	
Description of Candidatus Galloscillospira gen. nov.	
Candidatus Galloscillospira (Gall.os.cil.lo.spi’ra. L. masc. n. gallus chicken; N.L. fem. n. Oscillospira a genus name; N.L. fem. n. Galloscillospira. a genus related to the genus Oscillospira but distinct from it and found in poultry)	
A bacterial genus identified by metagenomic analyses. The genus includes all bacteria with genomes that show ≥60% average amino acid identity (AAI) to the type genome from the type species Candidatus Galloscillospira excrementipullorum. This genus belongs to the new family Candidatus Galloscillospiraceae.	

Description of Candidatus Galloscillospira excrementavium sp. nov.	
Candidatus Galloscillospira excrementavium (ex.cre.ment.a’vi.um. L. neut. n. excrementum excrement; L. fem. n. avis bird; N.L. gen. n. excrementavium of bird excrement)	
A bacterial species identified by metagenomic analyses. This species includes all bacteria with genomes that show ≥95% average nucleotide identity (ANI) to the type genome, which has been assigned the MAG ID ChiSjej1B19-13426 and which is available via NCBI BioSample SAMN15816937. The GC content of the type genome is 65.18% and the genome length is 2.1 Mbp.	
Description of Candidatus Galloscillospira excrementipullorum sp. nov.	
Candidatus Galloscillospira excrementipullorum (ex.cre.men.ti.pul.lo’rum. L. neut. n. excrementum excrement; L. masc. n. pullus a young chicken; N.L. gen. n. excrementipullorum of young chicken excrement)	
A bacterial species identified by metagenomic analyses. This species includes all bacteria with genomes that show ≥95% average nucleotide identity (ANI) to the type genome, which has been assigned the MAG ID ChiHjej8B7-10251 and which is available via NCBI BioSample SAMN15816946. The GC content of the type genome is 60.78% and the genome length is 1.6 Mbp.	
Description of Candidatus Galloscillospira stercoripullorum sp. nov.	
Candidatus Galloscillospira stercoripullorum (ster.co.ri.pul.lo’rum. L. neut. n. stercus dung; L. masc. n. pullus a young chicken; N.L. gen. n. stercoripullorum of the faceces of young chickens)	
A bacterial species identified by metagenomic analyses. This species includes all bacteria with genomes that show ≥95% average nucleotide identity (ANI) to the type genome, which has been assigned the MAG ID CHK33-6455 and which is available via NCBI BioSample SAMN15816975. The GC content of the type genome is 62.96% and the genome length is 1.9 Mbp.	
Description of Candidatus Galloscillospiraceae fam. nov.	
Candidatus Galloscillospiraceae (Gall.os.cil.lo.spi.ra.ce’ae. N.L. fem. n. Galloscillospira. type genus of the family genus; N.L. suff. –ceae to denote a family; N.L. fem. pl. n. Galloscillospiraceae, the family of the genus Galloscillospira)	
A bacterial family identified by metagenomic analyses. This family has been defined by the absence of a family assignment for the type species when GTDB-Tk v1.3.0 is applied to GTDB Release 05-RS95 (Chaumeil et al., 2019; Parks et al., 2020). GTDB assigns the type species and thus the family to the order Oscillospirales.	
Description of Candidatus Gemmiger avicola sp. nov.	
Candidatus Gemmiger avicola (a.vi’co.la. L. fem. n. avis bird; L. suff. -cola inhabitant of; N.L. n. avicola inhabitant of birds)	
A bacterial species identified by metagenomic analyses. This species includes all bacteria with genomes that show ≥95% average nucleotide identity (ANI) to the type genome, which has been assigned the MAG ID ChiBcec8-13705 and which is available via NCBI BioSample SAMN15816825. This is a new name for the alphanumeric GTDB species sp900548355. The GC content of the type genome is 61.69% and the genome length is 2.3 Mbp.	
Description of Candidatus Gemmiger avistercoris sp. nov.	
Candidatus Gemmiger avistercoris (a.vi.ster’co.ris. L. fem. n. avis bird; L. neut. n. stercus dung; N.L. gen. n. avistercoris of bird faeces)	
A bacterial species identified by metagenomic analyses. This species includes all bacteria with genomes that show ≥95% average nucleotide identity (ANI) to the type genome, which has been assigned the MAG ID CHK188-11489 and which is available via NCBI BioSample SAMN15816604. The GC content of the type genome is 63.30% and the genome length is 2.2 Mbp.	
Description of Candidatus Gemmiger avium sp. nov.	
Candidatus Gemmiger avium (a’vi.um. L. fem. pl. n. avium of birds)	
A bacterial species identified by metagenomic analyses. This species includes all bacteria with genomes that show ≥95% average nucleotide identity (ANI) to the type genome, which has been assigned the MAG ID ChiGjej4B4-15321 and which is available via NCBI BioSample SAMN15816926. This is a new name for the alphanumeric GTDB species sp002160955. Although GTDB has assigned this species to the genus it calls Gemmiger_A, this genus designation cannot be incorporated into a well-formed binomial, so in naming this species, we have used the current validly published name for the genus. The GC content of the type genome is 62.71% and the genome length is 2.6 Mbp.	
Description of Candidatus Gemmiger excrementavium sp. nov.	
Candidatus Gemmiger excrementavium (ex.cre.ment.a’vi.um. L. neut. n. excrementum excrement; L. fem. n. avis bird; N.L. gen. n. excrementavium of bird excrement)	
A bacterial species identified by metagenomic analyses. This species includes all bacteria with genomes that show ≥95% average nucleotide identity (ANI) to the type genome, which has been assigned the MAG ID 3436 and which is available via NCBI BioSample SAMN15816690. The GC content of the type genome is 60.89% and the genome length is 2.5 Mbp.	

Description of Candidatus Gemmiger excrementigallinarum sp. nov.	
Candidatus Gemmiger excrementigallinarum (ex.cre.men.ti.gal.li.na’rum. L. neut. n. excrementum excrement; L. fem. n. avis bird; N.L. gen. n. excrementigallinarum of hen excrement)	
A bacterial species identified by metagenomic analyses. This species includes all bacteria with genomes that show ≥95% average nucleotide identity (ANI) to the type genome, which has been assigned the MAG ID ChiSxjej1B13-11774 and which is available via NCBI BioSample SAMN15816691. The GC content of the type genome is 58.73% and the genome length is 2.4 Mbp.	
Description of Candidatus Gemmiger excrementipullorum sp. nov.	
Candidatus Gemmiger excrementipullorum (ex.cre.men.ti.pul.lo’rum. L. neut. n. excrementum excrement; L. masc. n. pullus a young chicken; N.L. gen. n. excrementipullorum of young chicken excrement)	
A bacterial species identified by metagenomic analyses. This species includes all bacteria with genomes that show ≥95% average nucleotide identity (ANI) to the type genome, which has been assigned the MAG ID ChiHecec2B26-7398 and which is available via NCBI BioSample SAMN15816726. The GC content of the type genome is 64.11% and the genome length is 2.1 Mbp.	
Description of Candidatus Gemmiger faecavium sp. nov.	
Candidatus Gemmiger faecavium (faec.a’vi.um. L. fem. n. faex, faecis excrement; L. fem. n. avis bird; N.L. gen. n. faecavium of bird faeces)	
A bacterial species identified by metagenomic analyses. This species includes all bacteria with genomes that show ≥95% average nucleotide identity (ANI) to the type genome, which has been assigned the MAG ID ChiSxjej1B13-1558 and which is available via NCBI BioSample SAMN15816740. The GC content of the type genome is 60.04% and the genome length is 2.5 Mbp.	
Description of Candidatus Gemmiger faecigallinarum sp. nov.	
Candidatus Gemmiger faecigallinarum (fae.ci.gal.li.na’rum. L. fem. n. faex, faecis excrement; L. fem. n. gallina hen; N.L. gen. n. faecigallinarum of hen faeces)	
A bacterial species identified by metagenomic analyses. This species includes all bacteria with genomes that show ≥95% average nucleotide identity (ANI) to the type genome, which has been assigned the MAG ID 14795 and which is available via NCBI BioSample SAMN15816795. The GC content of the type genome is 63.90% and the genome length is 2.6 Mbp.	
Description of Candidatus Gemmiger stercoravium sp. nov.	
Candidatus Gemmiger stercoravium (ster.cor.a’vi.um. L. neut. n. stercus dung; L. fem. n. avis bird; N.L. gen. n. stercoravium of bird faeces)	
A bacterial species identified by metagenomic analyses. This species includes all bacteria with genomes that show ≥95% average nucleotide identity (ANI) to the type genome, which has been assigned the MAG ID ChiBcolR8-2160 and which is available via NCBI BioSample SAMN15816579. The GC content of the type genome is 65.46% and the genome length is 2.5 Mbp.	
Description of Candidatus Gemmiger stercorigallinarum sp. nov.	
Candidatus Gemmiger stercorigallinarum (ster.co.ri.gal.li.na’rum. L. neut. n. stercus dung; L. fem. n. gallina hen; N.L. gen. n. stercorigallinarum of hen faeces)	
A bacterial species identified by metagenomic analyses. This species includes all bacteria with genomes that show ≥95% average nucleotide identity (ANI) to the type genome, which has been assigned the MAG ID CHK183-27628 and which is available via NCBI BioSample SAMN15816586. The GC content of the type genome is 64.08% and the genome length is 2.5 Mbp.	
Description of Candidatus Gemmiger stercoripullorum sp. nov.	
Candidatus Gemmiger stercoripullorum (ster.co.ri.pul.lo’rum. L. neut. n. stercus dung; L. masc. n. pullus a young chicken; N.L. gen. n. stercoripullorum of the faceces of young chickens)	
A bacterial species identified by metagenomic analyses. This species includes all bacteria with genomes that show ≥95% average nucleotide identity (ANI) to the type genome, which has been assigned the MAG ID ChiSjej4B22-15101 and which is available via NCBI BioSample SAMN15816589. The GC content of the type genome is 64.37% and the genome length is 2.2 Mbp.	
Description of Candidatus Gordonibacter avicola sp. nov.	
Candidatus Gordonibacter avicola (a.vi’co.la. L. fem. n. avis bird; L. suff. -cola inhabitant of; N.L. n. avicola inhabitant of birds)	
A bacterial species identified by metagenomic analyses. This species includes all bacteria with genomes that show ≥95% average nucleotide identity (ANI) to the type genome, which has been assigned the MAG ID ChiSxjej6B18-1472 and which is available via NCBI BioSample SAMN15816757. The GC content of the type genome is 59.91% and the genome length is 3.1 Mbp.	

Description of Candidatus Halomonas stercoripullorum sp. nov.	
Candidatus Halomonas stercoripullorum (ster.co.ri.pul.lo’rum. L. neut. n. stercus dung; L. masc. n. pullus a young chicken; N.L. gen. n. stercoripullorum of the faceces of young chickens)	
A bacterial species identified by metagenomic analyses. This species includes all bacteria with genomes that show ≥95% average nucleotide identity (ANI) to the type genome, which has been assigned the MAG ID 1193 and which is available via NCBI BioSample SAMN15816734. The GC content of the type genome is 59.74% and the genome length is 2.1 Mbp.	
Description of Candidatus Helicobacter avicola sp. nov.	
Candidatus Helicobacter avicola (a.vi’co.la. L. fem. n. avis bird; L. suff. -cola inhabitant of; N.L. n. avicola inhabitant of birds)	
A bacterial species identified by metagenomic analyses. This species includes all bacteria with genomes that show ≥95% average nucleotide identity (ANI) to the type genome, which has been assigned the MAG ID 14449 and which is available via NCBI BioSample SAMN15816913. Although GTDB has assigned this species to the genus it calls Helicobacter_F, this genus designation cannot be incorporated into a well-formed binomial, so in naming this species, we have used the current validly published name for the genus. The GC content of the type genome is 42.81% and the genome length is 1.7 Mbp.	
Description of Candidatus Helicobacter avistercoris sp. nov.	
Candidatus Helicobacter avistercoris (a.vi.ster’co.ris. L. fem. n. avis bird; L. neut. n. stercus dung; N.L. gen. n. avistercoris of bird faeces)	
A bacterial species identified by metagenomic analyses. This species includes all bacteria with genomes that show ≥95% average nucleotide identity (ANI) to the type genome, which has been assigned the MAG ID CHK158-8274 and which is available via NCBI BioSample SAMN15816903. Although GTDB has assigned this species to the genus it calls Helicobacter_G, this genus designation cannot be incorporated into a well-formed binomial, so in naming this species, we have used the current validly published name for the genus. The GC content of the type genome is 38.34% and the genome length is 1.4 Mbp.	
Description of Candidatus Hungatella pullicola sp. nov.	
Candidatus Hungatella pullicola (pul.li’co.la. L. masc. n. pullus a young chicken; L. suff. -cola inhabitant of; N.L. n. pullicola an inhabitant of young chickens)	
A bacterial species identified by metagenomic analyses. This species includes all bacteria with genomes that show ≥95% average nucleotide identity (ANI) to the type genome, which has been assigned the MAG ID CHK186-17716 and which is available via NCBI BioSample SAMN15816620. The GC content of the type genome is 44.57% and the genome length is 2.9 Mbp.	
Description of Candidatus Ignatzschineria merdigallinarum sp. nov.	
Candidatus Ignatzschineria merdigallinarum (mer.di.gal.li.na’rum. L. fem. n. merda faeces; L. fem. n. gallina hen; N.L. gen. n. merdigallinarum of hen faeces)	
A bacterial species identified by metagenomic analyses. This species includes all bacteria with genomes that show ≥95% average nucleotide identity (ANI) to the type genome, which has been assigned the MAG ID CHK160-9182 and which is available via NCBI BioSample SAMN15816769. The GC content of the type genome is 39.66% and the genome length is 2.3 Mbp.	
Description of Candidatus Intestinimonas merdavium sp. nov.	
Candidatus Intestinimonas merdavium (merd.a’vi.um. L. fem. n. merda faeces; L. fem. n. avis bird; N.L. gen. n. merdavium of bird faeces)	
A bacterial species identified by metagenomic analyses. This species includes all bacteria with genomes that show ≥95% average nucleotide identity (ANI) to the type genome, which has been assigned the MAG ID CHK33-7979 and which is available via NCBI BioSample SAMN15816706. The GC content of the type genome is 61.44% and the genome length is 2.4 Mbp.	
Description of Candidatus Intestinimonas pullistercoris sp. nov.	
Candidatus Intestinimonas pullistercoris (pul.li.ster’co.ris. L. masc. n. pullus a young chicken; L. neut. n. stercus dung; N.L. gen. n. pullistercoris of young chicken faeces)	
A bacterial species identified by metagenomic analyses. This species includes all bacteria with genomes that show ≥95% average nucleotide identity (ANI) to the type genome, which has been assigned the MAG ID CHK186-1790 and which is available via NCBI BioSample SAMN15816581. The GC content of the type genome is 64.76% and the genome length is 2.4 Mbp.	
Description of Candidatus Intestinimonas stercoravium sp. nov.	
Candidatus Intestinimonas stercoravium (ster.cor.a’vi.um. L. neut. n. stercus dung; L. fem. n. avis bird; N.L. gen. n. stercoravium of bird faeces)	
A bacterial species identified by metagenomic analyses. This species includes all bacteria with genomes that show ≥95% average nucleotide identity (ANI) to the type genome, which has been assigned the MAG ID ChiBcolR2-622 and which is available via NCBI BioSample SAMN15816599. The GC content of the type genome is 65.33% and the genome length is 2.2 Mbp.	

Description of Candidatus Intestinimonas stercorigallinarum sp. nov.	
Candidatus Intestinimonas stercorigallinarum (ster.co.ri.gal.li.na’rum. L. neut. n. stercus dung; L. fem. n. gallina hen; N.L. gen. n. stercorigallinarum of hen faeces)	
A bacterial species identified by metagenomic analyses. This species includes all bacteria with genomes that show ≥95% average nucleotide identity (ANI) to the type genome, which has been assigned the MAG ID ChiSxjej2B14-1745 and which is available via NCBI BioSample SAMN15816719. The GC content of the type genome is 65.69% and the genome length is 2.1 Mbp.	
Description of Candidatus Janibacter merdipullorum sp. nov.	
Candidatus Janibacter merdipullorum (mer.di.pul.lo’rum. L. fem. n. merda faeces; L. masc. n. pullus a young chicken; N.L. gen. n. merdipullorum of the faeces of young chickens)	
A bacterial species identified by metagenomic analyses. This species includes all bacteria with genomes that show ≥95% average nucleotide identity (ANI) to the type genome, which has been assigned the MAG ID ChiHjej13B12-21492 and which is available via NCBI BioSample SAMN15816685. The GC content of the type genome is 71.49% and the genome length is 2.8 Mbp.	
Description of Candidatus Jeotgalibaca merdavium sp. nov.	
Candidatus Jeotgalibaca merdavium (merd.a’vi.um. L. fem. n. merda faeces; L. fem. n. avis bird; N.L. gen. n. merdavium of bird faeces)	
A bacterial species identified by metagenomic analyses. This species includes all bacteria with genomes that show ≥95% average nucleotide identity (ANI) to the type genome, which has been assigned the MAG ID CHK171-505 and which is available via NCBI BioSample SAMN15816828. This is a new name for the alphanumeric GTDB species sp001975685. The GC content of the type genome is 38.39% and the genome length is 2.0 Mbp.	
Description of Candidatus Jeotgalibaca pullicola sp. nov.	
Candidatus Jeotgalibaca pullicola (pul.li’co.la. L. masc. n. pullus a young chicken; L. suff. -cola inhabitant of; N.L. n. pullicola an inhabitant of young chickens)	
A bacterial species identified by metagenomic analyses. This species includes all bacteria with genomes that show ≥95% average nucleotide identity (ANI) to the type genome, which has been assigned the MAG ID CHK172-9797 and which is available via NCBI BioSample SAMN15816826. This is a new name for the alphanumeric GTDB species sp003955755. The GC content of the type genome is 36.93% and the genome length is 2.6 Mbp.	
Description of Candidatus Jeotgalicoccus stercoravium sp. nov.	
Candidatus Jeotgalicoccus stercoravium (ster.cor.a’vi.um. L. neut. n. stercus dung; L. fem. n. avis bird; N.L. gen. n. stercoravium of bird faeces)	
A bacterial species identified by metagenomic analyses. This species includes all bacteria with genomes that show ≥95% average nucleotide identity (ANI) to the type genome, which has been assigned the MAG ID CHK148-7025 and which is available via NCBI BioSample SAMN15816765. The GC content of the type genome is 36.17% and the genome length is 1.7 Mbp.	
Description of Candidatus Kurthia intestinigallinarum sp. nov.	
Candidatus Kurthia intestinigallinarum (in.tes.ti.ni.gal.li.na’rum. L. neut. n. intestinum gut; L. fem. n. gallina hen; N.L. gen. n. intestinigallinarum of the gut of the hens)	
A bacterial species identified by metagenomic analyses. This species includes all bacteria with genomes that show ≥95% average nucleotide identity (ANI) to the type genome, which has been assigned the MAG ID CHK171-3164 and which is available via NCBI BioSample SAMN15816863. This is a new name for the alphanumeric GTDB species sp002418445. The GC content of the type genome is 39.62% and the genome length is 2.9 Mbp.	
Description of Candidatus Lachnoclostridium avicola sp. nov.	
Candidatus Lachnoclostridium avicola (a.vi’co.la. L. fem. n. avis bird; L. suff. -cola inhabitant of; N.L. n. avicola inhabitant of birds)	
A bacterial species identified by metagenomic analyses. This species includes all bacteria with genomes that show ≥95% average nucleotide identity (ANI) to the type genome, which has been assigned the MAG ID CHK190-19777 and which is available via NCBI BioSample SAMN15816889. Although GTDB has assigned this species to the genus it calls Lachnoclostridium_A, this genus designation cannot be incorporated into a well-formed binomial, so in naming this species, we have used the current validly published name for the genus. The GC content of the type genome is 54.75% and the genome length is 2.9 Mbp.	
Description of Candidatus Lachnoclostridium pullistercoris sp. nov.	
Candidatus Lachnoclostridium pullistercoris (pul.li.ster’co.ris. L. masc. n. pullus a young chicken; L. neut. n. stercus dung; N.L. gen. n. pullistercoris of young chicken faeces)	
A bacterial species identified by metagenomic analyses. This species includes all bacteria with genomes that show ≥95% average nucleotide identity (ANI) to the type genome, which has been assigned the MAG ID CHK183-5548 and which is available via NCBI BioSample SAMN15816884. Although GTDB has assigned this species to the genus it calls Lachnoclostridium_A, this genus designation cannot be incorporated into a well-formed binomial, so in naming this species, we have used the current validly published name for the genus. The GC content of the type genome is 54.35% and the genome length is 2.8 Mbp.	

Description of Candidatus Lachnoclostridium stercoravium sp. nov.	
Candidatus Lachnoclostridium stercoravium (ster.cor.a’vi.um. L. neut. n. stercus dung; L. fem. n. avis bird; N.L. gen. n. stercoravium of bird faeces)	
A bacterial species identified by metagenomic analyses. This species includes all bacteria with genomes that show ≥95% average nucleotide identity (ANI) to the type genome, which has been assigned the MAG ID CHK178-16964 and which is available via NCBI BioSample SAMN15816887. Although GTDB has assigned this species to the genus it calls Lachnoclostridium_A, this genus designation cannot be incorporated into a well-formed binomial, so in naming this species, we have used the current validly published name for the genus. The GC content of the type genome is 49.55% and the genome length is 3.0 Mbp.	
Description of Candidatus Lachnoclostridium stercorigallinarum sp. nov.	
Candidatus Lachnoclostridium stercorigallinarum (ster.co.ri.gal.li.na’rum. L. neut. n. stercus dung; L. fem. n. gallina hen; N.L. gen. n. stercorigallinarum of hen faeces)	
A bacterial species identified by metagenomic analyses. This species includes all bacteria with genomes that show ≥95% average nucleotide identity (ANI) to the type genome, which has been assigned the MAG ID ChiBcec1-1093 and which is available via NCBI BioSample SAMN15816897. Although GTDB has assigned this species to the genus it calls Lachnoclostridium_A, this genus designation cannot be incorporated into a well-formed binomial, so in naming this species, we have used the current validly published name for the genus. The GC content of the type genome is 54.29% and the genome length is 2.4 Mbp.	
Description of Candidatus Lachnoclostridium stercoripullorum sp. nov.	
Candidatus Lachnoclostridium stercoripullorum (ster.co.ri.pul.lo’rum. L. neut. n. stercus dung; L. masc. n. pullus a young chicken; N.L. gen. n. stercoripullorum of the faceces of young chickens)	
A bacterial species identified by metagenomic analyses. This species includes all bacteria with genomes that show ≥95% average nucleotide identity (ANI) to the type genome, which has been assigned the MAG ID ChiGjej4B4-12881 and which is available via NCBI BioSample SAMN15816908. Although GTDB has assigned this species to the genus it calls Lachnoclostridium_A, this genus designation cannot be incorporated into a well-formed binomial, so in naming this species, we have used the current validly published name for the genus. The GC content of the type genome is 59.38% and the genome length is 2.3 Mbp.	
Description of Candidatus Lactobacillus pullistercoris sp. nov.	
Candidatus Lactobacillus pullistercoris (pul.li.ster’co.ris. L. masc. n. pullus a young chicken; L. neut. n. stercus dung; N.L. gen. n. pullistercoris of young chicken faeces)	
A bacterial species identified by metagenomic analyses. This species includes all bacteria with genomes that show ≥95% average nucleotide identity (ANI) to the type genome, which has been assigned the MAG ID F6-686 and which is available via NCBI BioSample SAMN15816686. The GC content of the type genome is 34.54% and the genome length is 1.7 Mbp.	
Description of Candidatus Lawsonibacter pullicola sp. nov.	
Candidatus Lawsonibacter pullicola (pul.li’co.la. L. masc. n. pullus a young chicken; L. suff. -cola inhabitant of; N.L. n. pullicola an inhabitant of young chickens)	
A bacterial species identified by metagenomic analyses. This species includes all bacteria with genomes that show ≥95% average nucleotide identity (ANI) to the type genome, which has been assigned the MAG ID CHK178-3907 and which is available via NCBI BioSample SAMN15816869. This is a new name for the alphanumeric GTDB species sp002160305. The GC content of the type genome is 62.98% and the genome length is 2.3 Mbp.	
Description of Candidatus Levilactobacillus faecigallinarum sp. nov.	
Candidatus Levilactobacillus faecigallinarum (fae.ci.gal.li.na’rum. L. fem. n. faex, faecis excrement; L. fem. n. gallina hen; N.L. gen. n. faecigallinarum of hen faeces)	
A bacterial species identified by metagenomic analyses. This species includes all bacteria with genomes that show ≥95% average nucleotide identity (ANI) to the type genome, which has been assigned the MAG ID CHK173-259 and which is available via NCBI BioSample SAMN15816755. The GC content of the type genome is 52.18% and the genome length is 1.8 Mbp.	
Description of Candidatus Ligilactobacillus avistercoris sp. nov.	
Candidatus Ligilactobacillus avistercoris (a.vi.ster’co.ris. L. fem. n. avis bird; L. neut. n. stercus dung; N.L. gen. n. avistercoris of bird faeces)	
A bacterial species identified by metagenomic analyses. This species includes all bacteria with genomes that show ≥95% average nucleotide identity (ANI) to the type genome, which has been assigned the MAG ID ChiBile7-59 and which is available via NCBI BioSample SAMN15816642. The GC content of the type genome is 51.08% and the genome length is 1.2 Mbp.	

Description of Candidatus Ligilactobacillus excrementavium sp. nov.	
Candidatus Ligilactobacillus excrementavium (ex.cre.ment.a’vi.um. L. neut. n. excrementum excrement; L. fem. n. avis bird; N.L. gen. n. excrementavium of bird excrement)	
A bacterial species identified by metagenomic analyses. This species includes all bacteria with genomes that show ≥95% average nucleotide identity (ANI) to the type genome, which has been assigned the MAG ID 2259 and which is available via NCBI BioSample SAMN15816683. The GC content of the type genome is 38.00% and the genome length is 1.9 Mbp.	
Description of Candidatus Ligilactobacillus excrementigallinarum sp. nov.	
Candidatus Ligilactobacillus excrementigallinarum (ex.cre.men.ti.gal.li.na’rum. L. neut. n. excrementum excrement; L. fem. n. gallina hen; N.L. gen. n. excrementigallinarum of hen excrement)	
A bacterial species identified by metagenomic analyses. This species includes all bacteria with genomes that show ≥95% average nucleotide identity (ANI) to the type genome, which has been assigned the MAG ID 6627 and which is available via NCBI BioSample SAMN15816741. The GC content of the type genome is 34.16% and the genome length is 1.2 Mbp.	
Description of Candidatus Ligilactobacillus excrementipullorum sp. nov.	
Candidatus Ligilactobacillus excrementipullorum (ex.cre.men.ti.pul.lo’rum. L. neut. n. excrementum excrement; L. masc. n. pullus a young chicken; N.L. gen. n. excrementipullorum of young chicken excrement)	
A bacterial species identified by metagenomic analyses. This species includes all bacteria with genomes that show ≥95% average nucleotide identity (ANI) to the type genome, which has been assigned the MAG ID CHK171-2193 and which is available via NCBI BioSample SAMN15816749. The GC content of the type genome is 42.06% and the genome length is 2.0 Mbp.	
Description of Candidatus Ligilactobacillus faecavium sp. nov.	
Candidatus Ligilactobacillus faecavium (faec.a’vi.um. L. fem. n. faex, faecis excrement; L. fem. n. avis bird; N.L. gen. n. faecavium of bird faeces)	
A bacterial species identified by metagenomic analyses. This species includes all bacteria with genomes that show ≥95% average nucleotide identity (ANI) to the type genome, which has been assigned the MAG ID 3439 and which is available via NCBI BioSample SAMN15816798. The GC content of the type genome is 40.05% and the genome length is 1.3 Mbp.	
Description of Candidatus Limadaptatus gen. nov.	
Candidatus Limadaptatus (Lim.a.dap.ta’tus. L. masc. n. limus dung; L. past part. masc. adaptatus adapted to; N.L. masc. n. Limadaptatus a microbe associated with faeces)	
A bacterial genus identified by metagenomic analyses. The genus includes all bacteria with genomes that show ≥60% average amino acid identity (AAI) to the type genome from the type species Candidatus Limadaptatus stercoripullorum. This is a name for the alphanumeric GTDB genus UMGS1688. This genus has been assigned by GTDB-Tk v1.3.0 working on GTDB Release 05-RS95 (Chaumeil et al., 2019; Parks et al., 2020) to the order Christensenellales and to the family CAG-917.	
Description of Candidatus Limadaptatus stercoravium sp. nov.	
Candidatus Limadaptatus stercoravium (ster.cor.a’vi.um. L. neut. n. stercus dung; L. fem. n. avis bird; N.L. gen. n. stercoravium of bird faeces)	
A bacterial species identified by metagenomic analyses. This species includes all bacteria with genomes that show ≥95% average nucleotide identity (ANI) to the type genome, which has been assigned the MAG ID CHK154-227 and which is available via NCBI BioSample SAMN15817097. The GC content of the type genome is 58.03% and the genome length is 1.4 Mbp.	
Description of Candidatus Limadaptatus stercorigallinarum sp. nov.	
Candidatus Limadaptatus stercorigallinarum (ster.co.ri.gal.li.na’rum. L. neut. n. stercus dung; L. fem. n. gallina hen; N.L. gen. n. stercorigallinarum of hen faeces)	
A bacterial species identified by metagenomic analyses. This species includes all bacteria with genomes that show ≥95% average nucleotide identity (ANI) to the type genome, which has been assigned the MAG ID 1063 and which is available via NCBI BioSample SAMN15817231. This is a new name for the alphanumeric GTDB species sp900544575. The GC content of the type genome is 56.41% and the genome length is 1.5 Mbp.	
Description of Candidatus Limadaptatus stercoripullorum sp. nov.	
Candidatus Limadaptatus stercoripullorum (ster.co.ri.pul.lo’rum. L. neut. n. stercus dung; L. masc. n. pullus a young chicken; N.L. gen. n. stercoripullorum of the faceces of young chickens)	
A bacterial species identified by metagenomic analyses. This species includes all bacteria with genomes that show ≥95% average nucleotide identity (ANI) to the type genome, which has been assigned the MAG ID 10406 and which is available via NCBI BioSample SAMN15817154. The GC content of the type genome is 61.19% and the genome length is 1.4 Mbp.	

Description of Candidatus Limenecus gen. nov.	
Candidatus Limenecus (Lim.en.e’cus. L. masc. n. limus dung; Gr. masc. enoikos inhabitant; N.L. masc. n. Limenecus a microbe associated with faeces)	
A bacterial genus identified by metagenomic analyses. The genus includes all bacteria with genomes that show ≥60% average amino acid identity (AAI) to the type genome from the type species Candidatus Limenecus avicola. This is a name for the alphanumeric GTDB genus CAG-306. This genus has been assigned by GTDB-Tk v1.3.0 working on GTDB Release 05-RS95 (Chaumeil et al., 2019; Parks et al., 2020) to the order Gastranaerophilales and to the family Gastranaerophilaceae.	
Description of Candidatus Limenecus avicola sp. nov.	
Candidatus Limenecus avicola (a.vi’co.la. L. fem. n. avis bird; L. suff. -cola inhabitant of; N.L. n. avicola inhabitant of birds)	
A bacterial species identified by metagenomic analyses. This species includes all bacteria with genomes that show ≥95% average nucleotide identity (ANI) to the type genome, which has been assigned the MAG ID CHK154-7741 and which is available via NCBI BioSample SAMN15817206. This is a new name for the alphanumeric GTDB species sp000980375. The GC content of the type genome is 36.63% and the genome length is 2.1 Mbp.	
Description of Candidatus Limicola gen. nov.	
Candidatus Limicola (Li.mi’co.la. L. masc. n. limus dung; L. suff. -cola inhabitant of; N.L. fem. n. Limicola a microbe associated with faeces)	
A bacterial genus identified by metagenomic analyses. The genus includes all bacteria with genomes that show ≥60% average amino acid identity (AAI) to the type genome from the type species Candidatus Limicola stercorigallinarum. This is a name for the alphanumeric GTDB genus An2-A. This genus has been assigned by GTDB-Tk v1.3.0 working on GTDB Release 05-RS95 (Chaumeil et al., 2019; Parks et al., 2020) to the order Coriobacteriales and to the family Coriobacteriaceae.	
Description of Candidatus Limicola stercorigallinarum sp. nov.	
Candidatus Limicola stercorigallinarum (ster.co.ri.gal.li.na’rum. L. neut. n. stercus dung; L. fem. n. gallina hen; N.L. gen. n. stercorigallinarum of hen faeces)	
A bacterial species identified by metagenomic analyses. This species includes all bacteria with genomes that show ≥95% average nucleotide identity (ANI) to the type genome, which has been assigned the MAG ID ChiGjej1B1-16188 and which is available via NCBI BioSample SAMN15817082. The GC content of the type genome is 60.48% and the genome length is 2.0 Mbp.	
Description of Candidatus Limihabitans gen. nov.	
Candidatus Limihabitans (Li.mi.ha’bi.tans. L. masc. n. limus dung; L. pres. part. habitans an inhabitant; N.L. fem. n. Limihabitans a microbe associated with faeces)	
A bacterial genus identified by metagenomic analyses. The genus includes all bacteria with genomes that show ≥60% average amino acid identity (AAI) to the type genome from the type species Candidatus Limihabitans stercoravium. This is a name for the alphanumeric GTDB genus UMGS1707. This genus has been assigned by GTDB-Tk v1.3.0 working on GTDB Release 05-RS95 (Chaumeil et al., 2019; Parks et al., 2020) to the order 4C28d-15 and to the family CAG-314.	
Description of Candidatus Limihabitans stercoravium sp. nov.	
Candidatus Limihabitans stercoravium (ster.cor.a’vi.um. L. neut. n. stercus dung; L. fem. n. avis bird; N.L. gen. n. stercoravium of bird faeces)	
A bacterial species identified by metagenomic analyses. This species includes all bacteria with genomes that show ≥95% average nucleotide identity (ANI) to the type genome, which has been assigned the MAG ID 3394 and which is available via NCBI BioSample SAMN15817227. This is a new name for the alphanumeric GTDB species sp900547645. The GC content of the type genome is 43.23% and the genome length is 1.6 Mbp.	
Description of Candidatus Limimorpha gen. nov.	
Candidatus Limimorpha (Li.mi.mor’pha. L. masc. n. limus dung; Gr. fem. n. morphe a form, shape; N.L. fem. n. Limimorpha a microbe associated with faeces)	
A bacterial genus identified by metagenomic analyses. The genus includes all bacteria with genomes that show ≥60% average amino acid identity (AAI) to the type genome from the type species Candidatus Limimorpha avicola. This is a name for the alphanumeric GTDB genus F082. This genus has been assigned by GTDB-Tk v1.3.0 working on GTDB Release 05-RS95 (Chaumeil et al., 2019; Parks et al., 2020) to the order Bacteroidales and to the family F082.	
Description of Candidatus Limimorpha avicola sp. nov.	
Candidatus Limimorpha avicola (a.vi’co.la. L. fem. n. avis bird; L. suff. -cola inhabitant of; N.L. n. avicola inhabitant of birds)	
A bacterial species identified by metagenomic analyses. This species includes all bacteria with genomes that show ≥95% average nucleotide identity (ANI) to the type genome, which has been assigned the MAG ID Gambia15-481 and which is available via NCBI BioSample SAMN15817194. This is a new name for the alphanumeric GTDB species sp002633315. The GC content of the type genome is 38.09% and the genome length is 2.5 Mbp.	

Description of Candidatus Limiplasma gen. nov.	
Candidatus Limiplasma (Li.mi.plas’ma. L. masc. n. limus dung; Gr. neut. n. plasma a form; N.L. neut. n. Limiplasma a microbe associated with faeces)	
A bacterial genus identified by metagenomic analyses. The genus includes all bacteria with genomes that show ≥60% average amino acid identity (AAI) to the type genome from the type species Candidatus Limiplasma pullicola. This is a name for the alphanumeric GTDB genus Firm-11. This genus has been assigned by GTDB-Tk v1.3.0 working on GTDB Release 05-RS95 (Chaumeil et al., 2019; Parks et al., 2020) to the order Christensenellales and to the family CAG-74.	
Description of Candidatus Limiplasma merdipullorum sp. nov.	
Candidatus Limiplasma merdipullorum (mer.di.pul.lo’rum. L. fem. n. merda faeces; L. masc. n. pullus a young chicken; N.L. gen. n. merdipullorum of the faeces of young chickens)	
A bacterial species identified by metagenomic analyses. This species includes all bacteria with genomes that show ≥95% average nucleotide identity (ANI) to the type genome, which has been assigned the MAG ID ChiBcec16-3123 and which is available via NCBI BioSample SAMN15817217. This is a new name for the alphanumeric GTDB species sp900540045. The GC content of the type genome is 61.78% and the genome length is 2.6 Mbp.	
Description of Candidatus Limiplasma pullicola sp. nov.	
Candidatus Limiplasma pullicola (pul.li’co.la. L. masc. n. pullus a young chicken; L. suff. -cola inhabitant of; N.L. n. pullicola an inhabitant of young chickens)	
A bacterial species identified by metagenomic analyses. This species includes all bacteria with genomes that show ≥95% average nucleotide identity (ANI) to the type genome, which has been assigned the MAG ID 2223 and which is available via NCBI BioSample SAMN15817092. The GC content of the type genome is 62.99% and the genome length is 2.7 Mbp.	
Description of Candidatus Limiplasma pullistercoris sp. nov.	
Candidatus Limiplasma pullistercoris (pul.li.ster’co.ris. L. masc. n. pullus a young chicken; L. neut. n. stercus dung; N.L. gen. n. pullistercoris of young chicken faeces)	
A bacterial species identified by metagenomic analyses. This species includes all bacteria with genomes that show ≥95% average nucleotide identity (ANI) to the type genome, which has been assigned the MAG ID ChiBcec15-2748 and which is available via NCBI BioSample SAMN15817228. This is a new name for the alphanumeric GTDB species sp900553905. The GC content of the type genome is 62.75% and the genome length is 2.8 Mbp.	
Description of Candidatus Limiplasma stercoravium sp. nov.	
Candidatus Limiplasma stercoravium (ster.cor.a’vi.um. L. neut. n. stercus dung; L. fem. n. avis bird; N.L. gen. n. stercoravium of bird faeces)	
A bacterial species identified by metagenomic analyses. This species includes all bacteria with genomes that show ≥95% average nucleotide identity (ANI) to the type genome, which has been assigned the MAG ID CHK169-20388 and which is available via NCBI BioSample SAMN15817129. The GC content of the type genome is 62.11% and the genome length is 2.4 Mbp.	
Description of Candidatus Limisoma gen. nov.	
Candidatus Limisoma (Li.mi.so’ma. L. masc. n. limus dung; Gr. neut. n. soma a body; N.L. neut. n. Limisoma a microbe associated with faeces)	
A bacterial genus identified by metagenomic analyses. The genus includes all bacteria with genomes that show ≥60% average amino acid identity (AAI) to the type genome from the type species Candidatus Limisoma faecipullorum. This is a name for the alphanumeric GTDB genus CAG-279. This genus has been assigned by GTDB-Tk v1.3.0 working on GTDB Release 05-RS95 (Chaumeil et al., 2019; Parks et al., 2020) to the order Bacteroidales and to the family Muribaculaceae.	
Description of Candidatus Limisoma faecipullorum sp. nov.	
Candidatus Limisoma faecipullorum (fae.ci.pul.lo’rum. L. fem. n. faex, faecis excrement; L. masc. n. pullus a young chicken; N.L. gen. n. faecipullorum of young chicken faeces)	
A bacterial species identified by metagenomic analyses. This species includes all bacteria with genomes that show ≥95% average nucleotide identity (ANI) to the type genome, which has been assigned the MAG ID 6919 and which is available via NCBI BioSample SAMN15817069. The GC content of the type genome is 45.95% and the genome length is 2.2 Mbp.	
Description of Candidatus Limisoma gallistercoris sp. nov.	
Candidatus Limisoma gallistercoris (gal.li.ster’co.ris. L. masc. n gallus chicken; L. neut. n. stercus dung; N.L. gen. n. gallistercoris of chicken faeces)	
A bacterial species identified by metagenomic analyses. This species includes all bacteria with genomes that show ≥95% average nucleotide identity (ANI) to the type genome, which has been assigned the MAG ID CHK136-1475 and which is available via NCBI BioSample SAMN15817174. This is a new name for the alphanumeric GTDB species sp900550025. The GC content of the type genome is 48.31% and the genome length is 2.2 Mbp.	

Description of Candidatus Limisoma intestinavium sp. nov.	
Candidatus Limisoma intestinavium (in.tes.tin.a’vi.um. L. neut. n. intestinum gut; L. fem. n. avis bird; N.L. gen. n. intestinavium of the gut of birds)	
A bacterial species identified by metagenomic analyses. This species includes all bacteria with genomes that show ≥95% average nucleotide identity (ANI) to the type genome, which has been assigned the MAG ID 17073 and which is available via NCBI BioSample SAMN15817199. This is a new name for the alphanumeric GTDB species sp900541555. The GC content of the type genome is 48.25% and the genome length is 1.8 Mbp.	
Description of Candidatus Limivicinus gen. nov.	
Candidatus Limivicinus (Li.mi.vi.ci’nus. L. masc. n. limus dung; L. masc. n. vicinus a neighbour; N.L. masc. n. Limivicinus a microbe associated with faeces)	
A bacterial genus identified by metagenomic analyses. The genus includes all bacteria with genomes that show ≥60% average amino acid identity (AAI) to the type genome from the type species Candidatus Limivicinus faecipullorum. This is a name for the alphanumeric GTDB genus UBA1777. This genus has been assigned by GTDB-Tk v1.3.0 working on GTDB Release 05-RS95 (Chaumeil et al., 2019; Parks et al., 2020) to the order Oscillospirales and to the family Oscillospiraceae.	
Description of Candidatus Limivicinus faecipullorum sp. nov.	
Candidatus Limivicinus faecipullorum (fae.ci.pul.lo’rum. L. fem. n. faex, faecis excrement; L. masc. n. pullus a young chicken; N.L. gen. n. faecipullorum of young chicken faeces)	
A bacterial species identified by metagenomic analyses. This species includes all bacteria with genomes that show ≥95% average nucleotide identity (ANI) to the type genome, which has been assigned the MAG ID ChiHcec3-8852 and which is available via NCBI BioSample SAMN15817081. The GC content of the type genome is 58.52% and the genome length is 1.9 Mbp.	
Description of Candidatus Limivivens gen. nov.	
Candidatus Limivivens (Li.mi.vi’vens. L. masc. n. limus dung; N.L. pres. part. vivens living; N.L. fem. n. Limivivens a microbe associated with faeces)	
A bacterial genus identified by metagenomic analyses. The genus includes all bacteria with genomes that show ≥60% average amino acid identity (AAI) to the type genome from the type species Candidatus Limivivens intestinipullorum. This is a name for the alphanumeric GTDB genus GCA-900066135. This genus has been assigned by GTDB-Tk v1.3.0 working on GTDB Release 05-RS95 (Chaumeil et al., 2019; Parks et al., 2020) to the order Lachnospirales and to the family Lachnospiraceae.	
Description of Candidatus Limivivens intestinipullorum sp. nov.	
Candidatus Limivivens intestinipullorum (in.tes.ti.ni.pul.lo’rum. L. neut. n. intestinum gut; L. masc. n. pullus a young chicken; N.L. gen. n. intestinipullorum of the gut of young chickens)	
A bacterial species identified by metagenomic analyses. This species includes all bacteria with genomes that show ≥95% average nucleotide identity (ANI) to the type genome, which has been assigned the MAG ID CHK190-19873 and which is available via NCBI BioSample SAMN15817025. The GC content of the type genome is 52.58% and the genome length is 3.4 Mbp.	
Description of Candidatus Limivivens merdigallinarum sp. nov.	
Candidatus Limivivens merdigallinarum (mer.di.gal.li.na’rum. L. fem. n. merda faeces; L. fem. n. gallina hen; N.L. gen. n. merdigallinarum of hen faeces)	
A bacterial species identified by metagenomic analyses. This species includes all bacteria with genomes that show ≥95% average nucleotide identity (ANI) to the type genome, which has been assigned the MAG ID ChiSjej3B21-11622 and which is available via NCBI BioSample SAMN15817007. The GC content of the type genome is 50.09% and the genome length is 3.3 Mbp.	
Description of Candidatus Limosilactobacillus excrementigallinarum sp. nov.	
Candidatus Limosilactobacillus excrementigallinarum (ex.cre.men.ti.gal.li.na’rum. L. neut. n. excrementum excrement; L. fem. n. gallina hen; N.L. gen. n. excrementigallinarum of hen excrement)	
A bacterial species identified by metagenomic analyses. This species includes all bacteria with genomes that show ≥95% average nucleotide identity (ANI) to the type genome, which has been assigned the MAG ID 2685 and which is available via NCBI BioSample SAMN15816672. The GC content of the type genome is 41.67% and the genome length is 1.4 Mbp.	
Description of Candidatus Limosilactobacillus faecipullorum sp. nov.	
Candidatus Limosilactobacillus faecipullorum (fae.ci.pul.lo’rum. L. fem. n. faex, faecis excrement; L. masc. n. pullus a young chicken; N.L. gen. n. faecipullorum of young chicken faeces)	
A bacterial species identified by metagenomic analyses. This species includes all bacteria with genomes that show ≥95% average nucleotide identity (ANI) to the type genome, which has been assigned the MAG ID 7774 and which is available via NCBI BioSample SAMN15816663. The GC content of the type genome is 43.03% and the genome length is 1.5 Mbp.	

Description of Candidatus Limosilactobacillus gallistercoris sp. nov.	
Candidatus Limosilactobacillus gallistercoris (gal.li.ster’co.ris. L. masc. n gallus chicken; L. neut. n. stercus dung; N.L. gen. n. gallistercoris of chicken faeces)	
A bacterial species identified by metagenomic analyses. This species includes all bacteria with genomes that show ≥95% average nucleotide identity (ANI) to the type genome, which has been assigned the MAG ID CHK158-2993 and which is available via NCBI BioSample SAMN15816598. The GC content of the type genome is 52.74% and the genome length is 1.2 Mbp.	
Description of Candidatus Limosilactobacillus intestinavium sp. nov.	
Candidatus Limosilactobacillus intestinavium (in.tes.tin.a’vi.um. L. neut. n. intestinum gut; L. fem. n. avis bird; N.L. gen. n. intestinavium of the gut of birds)	
A bacterial species identified by metagenomic analyses. This species includes all bacteria with genomes that show ≥95% average nucleotide identity (ANI) to the type genome, which has been assigned the MAG ID 2331 and which is available via NCBI BioSample SAMN15816838. This is a new name for the alphanumeric GTDB species sp900557215. The GC content of the type genome is 38.77% and the genome length is 1.5 Mbp.	
Description of Candidatus Limosilactobacillus intestinigallinarum sp. nov.	
Candidatus Limosilactobacillus intestinigallinarum (in.tes.ti.ni.gal.li.na’rum. L. neut. n. intestinum gut; L. fem. n. gallina hen; N.L. gen. n. intestinigallinarum of the gut of the hens)	
A bacterial species identified by metagenomic analyses. This species includes all bacteria with genomes that show ≥95% average nucleotide identity (ANI) to the type genome, which has been assigned the MAG ID CHK176-5070 and which is available via NCBI BioSample SAMN15816600. The GC content of the type genome is 54.91% and the genome length is 1.5 Mbp.	
Description of Candidatus Limosilactobacillus intestinipullorum sp. nov.	
Candidatus Limosilactobacillus intestinipullorum (in.tes.ti.ni.pul.lo’rum. L. neut. n. intestinum gut; L. masc. n. pullus a young chicken; N.L. gen. n. intestinipullorum of the gut of young chickens)	
A bacterial species identified by metagenomic analyses. This species includes all bacteria with genomes that show ≥95% average nucleotide identity (ANI) to the type genome, which has been assigned the MAG ID ChiHecolR2B26-165 and which is available via NCBI BioSample SAMN15816601. The GC content of the type genome is 49.12% and the genome length is 1.6 Mbp.	
Description of Candidatus Limosilactobacillus merdavium sp. nov.	
Candidatus Limosilactobacillus merdavium (merd.a’vi.um. L. fem. n. merda faeces; L. fem. n. avis bird; N.L. gen. n. merdavium of bird faeces)	
A bacterial species identified by metagenomic analyses. This species includes all bacteria with genomes that show ≥95% average nucleotide identity (ANI) to the type genome, which has been assigned the MAG ID 876 and which is available via NCBI BioSample SAMN15816723. The GC content of the type genome is 39.60% and the genome length is 1.4 Mbp.	
Description of Candidatus Limosilactobacillus merdigallinarum sp. nov.	
Candidatus Limosilactobacillus merdigallinarum (mer.di.gal.li.na’rum. L. fem. n. merda faeces; L. fem. n. gallina hen; N.L. gen. n. merdigallinarum of hen faeces)	
A bacterial species identified by metagenomic analyses. This species includes all bacteria with genomes that show ≥95% average nucleotide identity (ANI) to the type genome, which has been assigned the MAG ID ChiSxjej3B15-572 and which is available via NCBI BioSample SAMN15816736. The GC content of the type genome is 44.36% and the genome length is 1.4 Mbp.	
Description of Candidatus Limosilactobacillus merdipullorum sp. nov.	
Candidatus Limosilactobacillus merdipullorum (mer.di.pul.lo’rum. L. fem. n. merda faeces; L. masc. n. pullus a young chicken; N.L. gen. n. merdipullorum of the faeces of young chickens)	
A bacterial species identified by metagenomic analyses. This species includes all bacteria with genomes that show ≥95% average nucleotide identity (ANI) to the type genome, which has been assigned the MAG ID ChiHejej3B27-2180 and which is available via NCBI BioSample SAMN15816756. The GC content of the type genome is 49.90% and the genome length is 1.3 Mbp.	
Description of Candidatus Limousia gen. nov.	
Candidatus Limousia (Lim.ou’si.a. L. masc. n. limus dung; Gr. fem. n. ousia an essence; N.L. fem. n. Limousia a microbe associated with faeces)	
A bacterial genus identified by metagenomic analyses. The genus includes all bacteria with genomes that show ≥60% average amino acid identity (AAI) to the type genome from the type species Candidatus Limousia pullorum. This is a name for the alphanumeric GTDB genus An172. This genus has been assigned by GTDB-Tk v1.3.0 working on GTDB Release 05-RS95 (Chaumeil et al., 2019; Parks et al., 2020) to the order Oscillospirales and to the family Acutalibacteraceae.	

Description of Candidatus Limousia pullorum sp. nov.	
Candidatus Limousia pullorum (pul.lo’rum. L. gen. pl. n. pullorum of young chickens)	
A bacterial species identified by metagenomic analyses. This species includes all bacteria with genomes that show ≥95% average nucleotide identity (ANI) to the type genome, which has been assigned the MAG ID ChiGjej1B1-1684 and which is available via NCBI BioSample SAMN15817202. This is a new name for the alphanumeric GTDB species sp002160515. The GC content of the type genome is 40.91% and the genome length is 1.7 Mbp.	
Description of Candidatus Luteimonas excrementigallinarum sp. nov.	
Candidatus Luteimonas excrementigallinarum (ex.cre.men.ti.gal.li.na’rum. L. neut. n. excrementum excrement; L. fem. n. gallina hen; N.L. gen. n. excrementigallinarum of hen excrement)	
A bacterial species identified by metagenomic analyses. This species includes all bacteria with genomes that show ≥95% average nucleotide identity (ANI) to the type genome, which has been assigned the MAG ID CHK165-14161 and which is available via NCBI BioSample SAMN15816707. The GC content of the type genome is 68.39% and the genome length is 2.5 Mbp.	
Description of Candidatus Luteococcus avicola sp. nov.	
Candidatus Luteococcus avicola (a.vi’co.la. L. fem. n. avis bird; L. suff. -cola inhabitant of; N.L. n. avicola inhabitant of birds)	
A bacterial species identified by metagenomic analyses. This species includes all bacteria with genomes that show ≥95% average nucleotide identity (ANI) to the type genome, which has been assigned the MAG ID 4979 and which is available via NCBI BioSample SAMN15816867. This is a new name for the alphanumeric GTDB species sp002387005. The GC content of the type genome is 68.14% and the genome length is 2.9 Mbp.	
Description of Candidatus Mailhella excrementigallinarum sp. nov.	
Candidatus Mailhella excrementigallinarum (ex.cre.men.ti.gal.li.na’rum. L. neut. n. excrementum excrement; L. fem. n. gallina hen; N.L. gen. n. excrementigallinarum of hen excrement)	
A bacterial species identified by metagenomic analyses. This species includes all bacteria with genomes that show ≥95% average nucleotide identity (ANI) to the type genome, which has been assigned the MAG ID 708 and which is available via NCBI BioSample SAMN15816871. This is a new name for the alphanumeric GTDB species sp003150275. The GC content of the type genome is 60.07% and the genome length is 3.0 Mbp.	
Description of Candidatus Mailhella merdavium sp. nov.	
Candidatus Mailhella merdavium (merd.a’vi.um. L. fem. n. merda faeces; L. fem. n. avis bird; N.L. gen. n. merdavium of bird faeces)	
A bacterial species identified by metagenomic analyses. This species includes all bacteria with genomes that show ≥95% average nucleotide identity (ANI) to the type genome, which has been assigned the MAG ID ChiBcec6-11642 and which is available via NCBI BioSample SAMN15816648. The GC content of the type genome is 56.49% and the genome length is 2.7 Mbp.	
Description of Candidatus Mailhella merdigallinarum sp. nov.	
Candidatus Mailhella merdigallinarum (mer.di.gal.li.na’rum. L. fem. n. merda faeces; L. fem. n. gallina hen; N.L. gen. n. merdigallinarum of hen faeces)	
A bacterial species identified by metagenomic analyses. This species includes all bacteria with genomes that show ≥95% average nucleotide identity (ANI) to the type genome, which has been assigned the MAG ID CHK186-16707 and which is available via NCBI BioSample SAMN15816842. This is a new name for the alphanumeric GTDB species sp900541395. The GC content of the type genome is 61.96% and the genome length is 2.4 Mbp.	
Description of Candidatus Massiliomicrobiota merdigallinarum sp. nov.	
Candidatus Massiliomicrobiota merdigallinarum (mer.di.gal.li.na’rum. L. fem. n. merda faeces; L. fem. n. gallina hen; N.L. gen. n. merdigallinarum of hen faeces)	
A bacterial species identified by metagenomic analyses. This species includes all bacteria with genomes that show ≥95% average nucleotide identity (ANI) to the type genome, which has been assigned the MAG ID CHK183-8118 and which is available via NCBI BioSample SAMN15816832. This is a new name for the alphanumeric GTDB species sp002160815. The GC content of the type genome is 31.33% and the genome length is 2.4 Mbp.	
Description of Candidatus Mediterraneibacter avicola sp. nov.	
Candidatus Mediterraneibacter avicola (a.vi’co.la. L. fem. n. avis bird; L. suff. -cola inhabitant of; N.L. n. avicola inhabitant of birds)	
A bacterial species identified by metagenomic analyses. This species includes all bacteria with genomes that show ≥95% average nucleotide identity (ANI) to the type genome, which has been assigned the MAG ID ChiGjej3B3-8055 and which is available via NCBI BioSample SAMN15816612. The GC content of the type genome is 48.65% and the genome length is 2.4 Mbp.	

Description of Candidatus Mediterraneibacter caccavium sp. nov.	
Candidatus Mediterraneibacter caccavium (cacc.a’vi.um. Gr. fem. n. kakke faeces; L. fem. n. avis bird; N.L. gen. n. caccavium of bird faeces)	
A bacterial species identified by metagenomic analyses. This species includes all bacteria with genomes that show ≥95% average nucleotide identity (ANI) to the type genome, which has been assigned the MAG ID ChiSjej5B23-15282 and which is available via NCBI BioSample SAMN15816861. This is a new name for the alphanumeric GTDB species sp002161355. The GC content of the type genome is 51.40% and the genome length is 2.6 Mbp.	
Description of Candidatus Mediterraneibacter caccogallinarum sp. nov.	
Candidatus Mediterraneibacter caccogallinarum (cac.co.gal.li.na’rum. Gr. fem. n. kakke faeces; L. fem. n. gallina hen; N.L. gen. n. caccogallinarum of hen faeces)	
A bacterial species identified by metagenomic analyses. This species includes all bacteria with genomes that show ≥95% average nucleotide identity (ANI) to the type genome, which has been assigned the MAG ID ChiHcolR18-251 and which is available via NCBI BioSample SAMN15816801. This is a new name for the alphanumeric GTDB species sp002314255. The GC content of the type genome is 50.75% and the genome length is 2.6 Mbp.	
Description of Candidatus Mediterraneibacter colneyensis sp. nov.	
Candidatus Mediterraneibacter colneyensis (col.ney.en’sis. N.L. fem. adj. colneyensis pertaining to Colney, the Norfolk village which is home to the Quadram Institute where the species was first described)	
A bacterial species identified by metagenomic analyses. This species includes all bacteria with genomes that show ≥95% average nucleotide identity (ANI) to the type genome, which has been assigned the MAG ID ChiGjej5B5-19924 and which is available via NCBI BioSample SAMN15816732. The GC content of the type genome is 50.96% and the genome length is 1.9 Mbp.	
Description of Candidatus Mediterraneibacter cottocaccae sp. nov.	
Candidatus Mediterraneibacter cottocaccae (cot.to.cac’cae. Gr. masc. n. kottos chicken Gr. fem. n. kakke faeces; N.L. gen. n. cottocaccae of chicken faeces)	
A bacterial species identified by metagenomic analyses. This species includes all bacteria with genomes that show ≥95% average nucleotide identity (ANI) to the type genome, which has been assigned the MAG ID CHK192-87 and which is available via NCBI BioSample SAMN15816835. This is a new name for the alphanumeric GTDB species sp002160525. The GC content of the type genome is 50.07% and the genome length is 4.0 Mbp.	
Description of Candidatus Mediterraneibacter excrementavium sp. nov.	
Candidatus Mediterraneibacter excrementavium (ex.cre.ment.a’vi.um. L. neut. n. excrementum excrement; L. fem. n. avis bird; N.L. gen. n. excrementavium of bird excrement)	
A bacterial species identified by metagenomic analyses. This species includes all bacteria with genomes that show ≥95% average nucleotide identity (ANI) to the type genome, which has been assigned the MAG ID ChiGjej2B2-38138 and which is available via NCBI BioSample SAMN15816630. The GC content of the type genome is 51.42% and the genome length is 2.1 Mbp.	
Description of Candidatus Mediterraneibacter excrementigallinarum sp. nov.	
Candidatus Mediterraneibacter excrementigallinarum (ex.cre.men.ti.gal.li.na’rum. L. neut. n. excrementum excrement; L. fem. n. gallina hen; N.L. gen. n. excrementigallinarum of hen excrement)	
A bacterial species identified by metagenomic analyses. This species includes all bacteria with genomes that show ≥95% average nucleotide identity (ANI) to the type genome, which has been assigned the MAG ID CHK143-6153 and which is available via NCBI BioSample SAMN15816575. The GC content of the type genome is 48.97% and the genome length is 3.1 Mbp.	
Description of Candidatus Mediterraneibacter excrementipullorum sp. nov.	
Candidatus Mediterraneibacter excrementipullorum (ex.cre.men.ti.pul.lo’rum. L. neut. n. excrementum excrement; L. masc. n. pullus a young chicken; N.L. gen. n. excrementipullorum of young chicken excrement)	
A bacterial species identified by metagenomic analyses. This species includes all bacteria with genomes that show ≥95% average nucleotide identity (ANI) to the type genome, which has been assigned the MAG ID ChiSjej6B24-3024 and which is available via NCBI BioSample SAMN15816810. This is a new name for the alphanumeric GTDB species sp9005552. The GC content of the type genome is 48.88% and the genome length is 2.4 Mbp.	
Description of Candidatus Mediterraneibacter faecavium sp. nov.	
Candidatus Mediterraneibacter faecavium (faec.a’vi.um. L. fem. n. faex, faecis excrement; L. fem. n. avis bird; N.L. gen. n. faecavium of bird faeces)	
A bacterial species identified by metagenomic analyses. This species includes all bacteria with genomes that show ≥95% average nucleotide identity (ANI) to the type genome, which has been assigned the MAG ID CHK196-7946 and which is available via NCBI BioSample SAMN15816577. The GC content of the type genome is 49.45% and the genome length is 2.8 Mbp.	

Description of Candidatus Mediterraneibacter faecigallinarum sp. nov.	
Candidatus Mediterraneibacter faecigallinarum (fae.ci.gal.li.na’rum. L. fem. n. faex, faecis excrement; L. fem. n. gallina hen; N.L. gen. n. faecigallinarum of hen faeces)	
A bacterial species identified by metagenomic analyses. This species includes all bacteria with genomes that show ≥95% average nucleotide identity (ANI) to the type genome, which has been assigned the MAG ID ChiGjej1B1-1692 and which is available via NCBI BioSample SAMN15816637. The GC content of the type genome is 51.63% and the genome length is 2.6 Mbp.	
Description of Candidatus Mediterraneibacter faecipullorum sp. nov.	
Candidatus Mediterraneibacter faecipullorum (fae.ci.pul.lo’rum. L. fem. n. faex, faecis excrement; L. masc. n. pullus a young chicken; N.L. gen. n. faecipullorum of young chicken faeces)	
A bacterial species identified by metagenomic analyses. This species includes all bacteria with genomes that show ≥95% average nucleotide identity (ANI) to the type genome, which has been assigned the MAG ID ChiW19-954 and which is available via NCBI BioSample SAMN15816638. The GC content of the type genome is 47.71% and the genome length is 2.7 Mbp.	
Description of Candidatus Mediterraneibacter gallistercoris sp. nov.	
Candidatus Mediterraneibacter gallistercoris (gal.li.ster’co.ris. L. masc. n. gallus chicken; L. neut. n. stercus dung; N.L. gen. n. gallistercoris of chicken faeces)	
A bacterial species identified by metagenomic analyses. This species includes all bacteria with genomes that show ≥95% average nucleotide identity (ANI) to the type genome, which has been assigned the MAG ID CHK165-2605 and which is available via NCBI BioSample SAMN15816636. The GC content of the type genome is 47.20% and the genome length is 2.5 Mbp.	
Description of Candidatus Mediterraneibacter guildfordensis sp. nov.	
Candidatus Mediterraneibacter guildfordensis (guild.ford.en’sis. N.L. masc. adj. guildfordensis pertaining to Guildford, English town that is home to the University of Surrey)	
A bacterial species identified by metagenomic analyses. This species includes all bacteria with genomes that show ≥95% average nucleotide identity (ANI) to the type genome, which has been assigned the MAG ID ChiHcec3-18395 and which is available via NCBI BioSample SAMN15816784. The GC content of the type genome is 52.47% and the genome length is 2.2 Mbp.	
Description of Candidatus Mediterraneibacter intestinavium sp. nov.	
Candidatus Mediterraneibacter intestinavium (in.tes.tin.a’vi.um. L. neut. n. intestinum gut; L. fem. n. avis bird; N.L. gen. n. intestinavium of the gut of birds)	
A bacterial species identified by metagenomic analyses. This species includes all bacteria with genomes that show ≥95% average nucleotide identity (ANI) to the type genome, which has been assigned the MAG ID ChiBcec12-2655 and which is available via NCBI BioSample SAMN15816591. The GC content of the type genome is 50.53% and the genome length is 2.9 Mbp.	
Description of Candidatus Mediterraneibacter intestinigallinarum sp. nov.	
Candidatus Mediterraneibacter intestinigallinarum (in.tes.ti.ni.gal.li.na’rum. L. neut. n. intestinum gut; L. fem. n. gallina hen; N.L. gen. n. intestinigallinarum of the gut of the hens)	
A bacterial species identified by metagenomic analyses. This species includes all bacteria with genomes that show ≥95% average nucleotide identity (ANI) to the type genome, which has been assigned the MAG ID ChiBcec15-2237 and which is available via NCBI BioSample SAMN15816645. The GC content of the type genome is 46.81% and the genome length is 3.1 Mbp.	
Description of Candidatus Mediterraneibacter intestinipullorum sp. nov.	
Candidatus Mediterraneibacter intestinipullorum (in.tes.ti.ni.pul.lo’rum. L. neut. n. intestinum gut; L. masc. n. pullus a young chicken; N.L. gen. n. intestinipullorum of the gut of young chickens)	
A bacterial species identified by metagenomic analyses. This species includes all bacteria with genomes that show ≥95% average nucleotide identity (ANI) to the type genome, which has been assigned the MAG ID CHK161-4361 and which is available via NCBI BioSample SAMN15816656. The GC content of the type genome is 49.75% and the genome length is 2.5 Mbp.	
Description of Candidatus Mediterraneibacter merdavium sp. nov.	
Candidatus Mediterraneibacter merdavium (merd.a’vi.um. L. fem. n. merda faeces; L. fem. n. avis bird; N.L. gen. n. merdavium of bird faeces)	
A bacterial species identified by metagenomic analyses. This species includes all bacteria with genomes that show ≥95% average nucleotide identity (ANI) to the type genome, which has been assigned the MAG ID ChiBcolR7-8672 and which is available via NCBI BioSample SAMN15816660. The GC content of the type genome is 49.88% and the genome length is 2.6 Mbp.	

Description of Candidatus Mediterraneibacter merdigallinarum sp. nov.	
Candidatus Mediterraneibacter merdigallinarum (mer.di.gal.li.na’rum. L. fem. n. merda faeces; L. fem. n. gallina hen; N.L. gen. n. merdigallinarum of hen faeces)	
A bacterial species identified by metagenomic analyses. This species includes all bacteria with genomes that show ≥95% average nucleotide identity (ANI) to the type genome, which has been assigned the MAG ID ChiW16-1363 and which is available via NCBI BioSample SAMN15816674. The GC content of the type genome is 46.95% and the genome length is 2.4 Mbp.	
Description of Candidatus Mediterraneibacter merdipullorum sp. nov.	
Candidatus Mediterraneibacter merdipullorum (mer.di.pul.lo’rum. L. fem. n. merda faeces; L. masc. n. pullus a young chicken; N.L. gen. n. merdipullorum of the faeces of young chickens)	
A bacterial species identified by metagenomic analyses. This species includes all bacteria with genomes that show ≥95% average nucleotide identity (ANI) to the type genome, which has been assigned the MAG ID ChiSxjej4B16-6421 and which is available via NCBI BioSample SAMN15816679. The GC content of the type genome is 54.17% and the genome length is 2.2 Mbp.	
Description of Candidatus Mediterraneibacter norfolkensis sp. nov.	
Candidatus Mediterraneibacter norfolkensis (nor.folk.en’sis. N.L. masc. adj. norfolkensis pertaining to the English county of Norfolk)	
A bacterial species identified by metagenomic analyses. This species includes all bacteria with genomes that show ≥95% average nucleotide identity (ANI) to the type genome, which has been assigned the MAG ID ChiW9-3490 and which is available via NCBI BioSample SAMN15816789. The GC content of the type genome is 48.85% and the genome length is 3.5 Mbp.	
Description of Candidatus Mediterraneibacter norwichensis sp. nov.	
Candidatus Mediterraneibacter norwichensis (nor.wich.en’sis. N.L. masc. adj. norwichensis pertaining to English city of Norwich)	
A bacterial species identified by metagenomic analyses. This species includes all bacteria with genomes that show ≥95% average nucleotide identity (ANI) to the type genome, which has been assigned the MAG ID CHK180-4461 and which is available via NCBI BioSample SAMN15816628. The GC content of the type genome is 47.61% and the genome length is 2.6 Mbp.	
Description of Candidatus Mediterraneibacter ornithocaccae sp. nov.	
Candidatus Mediterraneibacter ornithocaccae (or.ni.tho.cac’cae. Gr. masc. or fem. n. ornis, ornithos bird Gr. fem. n. kakke faeces; N.L. gen. n. ornithocaccae of bird faeces)	
A bacterial species identified by metagenomic analyses. This species includes all bacteria with genomes that show ≥95% average nucleotide identity (ANI) to the type genome, which has been assigned the MAG ID ChiGjej1B1-20579 and which is available via NCBI BioSample SAMN15816839. This is a new name for the alphanumeric GTDB species sp002159505. The GC content of the type genome is 47.31% and the genome length is 2.6 Mbp.	
Description of Candidatus Mediterraneibacter pullicola sp. nov.	
Candidatus Mediterraneibacter pullicola (pul.li’co.la. L. masc. n. pullus a young chicken; L. suff. -cola inhabitant of; N.L. n. pullicola an inhabitant of young chickens)	
A bacterial species identified by metagenomic analyses. This species includes all bacteria with genomes that show ≥95% average nucleotide identity (ANI) to the type genome, which has been assigned the MAG ID ChiSjej2B20-11307 and which is available via NCBI BioSample SAMN15816678. The GC content of the type genome is 47.35% and the genome length is 2.1 Mbp.	
Description of Candidatus Mediterraneibacter pullistercoris sp. nov.	
Candidatus Mediterraneibacter pullistercoris (pul.li.ster’co.ris. L. masc. n. pullus a young chicken; L. neut. n. stercus dung; N.L. gen. n. pullistercoris of young chicken faeces)	
A bacterial species identified by metagenomic analyses. This species includes all bacteria with genomes that show ≥95% average nucleotide identity (ANI) to the type genome, which has been assigned the MAG ID ChiHjej8B7-7219 and which is available via NCBI BioSample SAMN15816602. The GC content of the type genome is 48.73% and the genome length is 2.3 Mbp.	
Description of Candidatus Mediterraneibacter quadrami sp. nov.	
Candidatus Mediterraneibacter quadrami (quad.ra’mi. N.L. gen. n. quadrami of the Quadram Institute)	
A bacterial species identified by metagenomic analyses. This species includes all bacteria with genomes that show ≥95% average nucleotide identity (ANI) to the type genome, which has been assigned the MAG ID ChiBcec15-3976 and which is available via NCBI BioSample SAMN15816790. The GC content of the type genome is 52.68% and the genome length is 2.0 Mbp.	

Description of Candidatus Mediterraneibacter stercoravium sp. nov.	
Candidatus Mediterraneibacter stercoravium (ster.cor.a’vi.um. L. neut. n. stercus dung; L. fem. n. avis bird; N.L. gen. n. stercoravium of bird faeces)	
A bacterial species identified by metagenomic analyses. This species includes all bacteria with genomes that show ≥95% average nucleotide identity (ANI) to the type genome, which has been assigned the MAG ID CHK196-3914 and which is available via NCBI BioSample SAMN15816603. The GC content of the type genome is 48.72% and the genome length is 2.5 Mbp.	
Description of Candidatus Mediterraneibacter stercorigallinarum sp. nov.	
Candidatus Mediterraneibacter stercorigallinarum (ster.co.ri.gal.li.na’rum. L. neut. n. stercus dung; L. fem. n. gallina hen; N.L. gen. n. stercorigallinarum of hen faeces)	
A bacterial species identified by metagenomic analyses. This species includes all bacteria with genomes that show ≥95% average nucleotide identity (ANI) to the type genome, which has been assigned the MAG ID ChiGjej1B1-13045 and which is available via NCBI BioSample SAMN15816697. The GC content of the type genome is 50.04% and the genome length is 2.3 Mbp.	
Description of Candidatus Mediterraneibacter stercoripullorum sp. nov.	
Candidatus Mediterraneibacter stercoripullorum (ster.co.ri.pul.lo’rum. L. neut. n. stercus dung; L. masc. n. pullus a young chicken; N.L. gen. n. stercoripullorum of the faceces of young chickens)	
A bacterial species identified by metagenomic analyses. This species includes all bacteria with genomes that show ≥95% average nucleotide identity (ANI) to the type genome, which has been assigned the MAG ID CHK195-1396 and which is available via NCBI BioSample SAMN15816610. The GC content of the type genome is 48.12% and the genome length is 3.2 Mbp.	
Description of Candidatus Mediterraneibacter surreyensis sp. nov.	
Candidatus Mediterraneibacter surreyensis (sur.rey.en’sis. N.L. masc. adj. surreyensis pertaining to the English county of Surrey where the samples in the study were collected)	
A bacterial species identified by metagenomic analyses. This species includes all bacteria with genomes that show ≥95% average nucleotide identity (ANI) to the type genome, which has been assigned the MAG ID CHK177-12742 and which is available via NCBI BioSample SAMN15816593. The GC content of the type genome is 46.80% and the genome length is 2.9 Mbp.	
Description of Candidatus Mediterraneibacter tabaqchaliae sp. nov.	
Candidatus Mediterraneibacter tabaqchaliae (ta.baq.cha’li.ae. N.L. fem. gen. n. tabaqchaliae named in honour of British microbiologist Soad Tabaqchali)	
A bacterial species identified by metagenomic analyses. This species includes all bacteria with genomes that show ≥95% average nucleotide identity (ANI) to the type genome, which has been assigned the MAG ID ChiGjej3B3-11674 and which is available via NCBI BioSample SAMN15816791. The GC content of the type genome is 51.91% and the genome length is 2.7 Mbp.	
Description of Candidatus Mediterraneibacter vanvlietii sp. nov.	
Candidatus Mediterraneibacter vanvlietii (van.vliet’i.i. N.L. gen. n. vanvlietii named in honour of Dutch microbiologist Arnoud van Vliet)	
A bacterial species identified by metagenomic analyses. This species includes all bacteria with genomes that show ≥95% average nucleotide identity (ANI) to the type genome, which has been assigned the MAG ID ChiBcec1-362 and which is available via NCBI BioSample SAMN15816623. The GC content of the type genome is 48.47% and the genome length is 3.0 Mbp.	
Description of Candidatus Megamonas gallistercoris sp. nov.	
Candidatus Megamonas gallistercoris (gal.li.ster’co.ris. L. masc. n. gallus chicken; L. neut. n. stercus dung; N.L. gen. n. gallistercoris of chicken faeces)	
A bacterial species identified by metagenomic analyses. This species includes all bacteria with genomes that show ≥95% average nucleotide identity (ANI) to the type genome, which has been assigned the MAG ID ChiGjej6B6-7947 and which is available via NCBI BioSample SAMN15816859. This is a new name for the alphanumeric GTDB species sp900554895. The GC content of the type genome is 40.34% and the genome length is 2.2 Mbp.	
Description of Candidatus Merdenecus gen. nov.	
Candidatus Merdenecus (Merd.en.e’cus. L. fem. n. merda dung; Gr. masc. enoikos inhabitant; N.L. masc. n. Merdenecus a microbe associated with faeces)	
A bacterial genus identified by metagenomic analyses. The genus includes all bacteria with genomes that show ≥60% average amino acid identity (AAI) to the type genome from the type species Candidatus Merdenecus merdavium. This is a name for the alphanumeric GTDB genus MCWD5. This genus has been assigned by GTDB-Tk v1.3.0 working on GTDB Release 05-RS95 (Chaumeil et al., 2019; Parks et al., 2020) to the order Lachnospirales and to the family Lachnospiraceae.	

Description of Candidatus Merdenecus pullicola sp. nov.	
Candidatus Merdenecus pullicola (pul.li’co.la. L. masc. n. pullus a young chicken; L. suff. -cola inhabitant of; N.L. n. pullicola an inhabitant of young chickens)	
A bacterial species identified by metagenomic analyses. This species includes all bacteria with genomes that show ≥95% average nucleotide identity (ANI) to the type genome, which has been assigned the MAG ID CHK160-2840 and which is available via NCBI BioSample SAMN15817122. The GC content of the type genome is 35.46% and the genome length is 2.6 Mbp.	
Description of Candidatus Merdibacter merdavium sp. nov.	
Candidatus Merdibacter merdavium (merd.a’vi.um. L. fem. n. merda faeces; L. fem. n. avis bird; N.L. gen. n. merdavium of bird faeces)	
A bacterial species identified by metagenomic analyses. This species includes all bacteria with genomes that show ≥95% average nucleotide identity (ANI) to the type genome, which has been assigned the MAG ID CHK187-11901 and which is available via NCBI BioSample SAMN15816582. The GC content of the type genome is 53.13% and the genome length is 2.1 Mbp.	
Description of Candidatus Merdibacter merdigallinarum sp. nov.	
Candidatus Merdibacter merdigallinarum (mer.di.gal.li.na’rum. L. fem. n. merda faeces; L. fem. n. gallina hen; N.L. gen. n. merdigallinarum of hen faeces)	
A bacterial species identified by metagenomic analyses. This species includes all bacteria with genomes that show ≥95% average nucleotide identity (ANI) to the type genome, which has been assigned the MAG ID ChiGjej6B6-453 and which is available via NCBI BioSample SAMN15816595. The GC content of the type genome is 53.92% and the genome length is 1.8 Mbp.	
Description of Candidatus Merdibacter merdipullorum sp. nov.	
Candidatus Merdibacter merdipullorum (mer.di.pul.lo’rum. L. fem. n. merda faeces; L. masc. n. pullus a young chicken; N.L. gen. n. merdipullorum of the faeces of young chickens)	
A bacterial species identified by metagenomic analyses. This species includes all bacteria with genomes that show ≥95% average nucleotide identity (ANI) to the type genome, which has been assigned the MAG ID ChiGjej1B1-19782 and which is available via NCBI BioSample SAMN15816850. This is a new name for the alphanumeric GTDB species sp900543035. The GC content of the type genome is 55.41% and the genome length is 1.9 Mbp.	
Description of Candidatus Merdicola gen. nov.	
Candidatus Merdicola (Mer.di’co.la. L. fem. n. merda dung; L. suff. -cola inhabitant of; N.L. fem. n. Merdicola a microbe associated with faeces)	
A bacterial genus identified by metagenomic analyses. The genus includes all bacteria with genomes that show ≥60% average amino acid identity (AAI) to the type genome from the type species Candidatus Merdicola faecigallinarum. This is a name for the alphanumeric GTDB genus CAG-354. This genus has been assigned by GTDB-Tk v1.3.0 working on GTDB Release 05-RS95 (Chaumeil et al., 2019; Parks et al., 2020) to the order TANB77 and to the family CAG-508.	
Description of Candidatus Merdicola faecigallinarum sp. nov.	
Candidatus Merdicola faecigallinarum (fae.ci.gal.li.na’rum. L. fem. n. faex, faecis excrement; L. fem. n. gallina hen; N.L. gen. n. faecigallinarum of hen faeces)	
A bacterial species identified by metagenomic analyses. This species includes all bacteria with genomes that show ≥95% average nucleotide identity (ANI) to the type genome, which has been assigned the MAG ID CHK195-15760 and which is available via NCBI BioSample SAMN15817051. The GC content of the type genome is 28.83% and the genome length is 1.5 Mbp.	
Description of Candidatus Merdimorpha gen. nov.	
Candidatus Merdimorpha (Mer.di.mor’pha. L. fem. n. merda dung; Gr. fem. n. morphe a form, shape; N.L. fem. n. Merdimorpha a microbe associated with faeces)	
A bacterial genus identified by metagenomic analyses. The genus includes all bacteria with genomes that show ≥60% average amino acid identity (AAI) to the type genome from the type species Candidatus Merdimorpha intestinavium. This is a name for the alphanumeric GTDB genus UBA1820. This genus has been assigned by GTDB-Tk v1.3.0 working on GTDB Release 05-RS95 (Chaumeil et al., 2019; Parks et al., 2020) to the order Flavobacteriales and to the family UBA1820.	
Description of Candidatus Merdimorpha intestinavium sp. nov.	
Candidatus Merdimorpha intestinavium (in.tes.tin.a’vi.um. L. neut. n. intestinum gut; L. fem. n. avis bird; N.L. gen. n. intestinavium of the gut of birds)	
A bacterial species identified by metagenomic analyses. This species includes all bacteria with genomes that show ≥95% average nucleotide identity (ANI) to the type genome, which has been assigned the MAG ID CHK1-7158 and which is available via NCBI BioSample SAMN15817210. This is a new name for the alphanumeric GTDB species sp002314265. The GC content of the type genome is 56.50% and the genome length is 1.8 Mbp.	

Description of Candidatus Merdimorpha stercoravium sp. nov.	
Candidatus Merdimorpha stercoravium (ster.cor.a’vi.um. L. neut. n. stercus dung; L. fem. n. avis bird; N.L. gen. n. stercoravium of bird faeces)	
A bacterial species identified by metagenomic analyses. This species includes all bacteria with genomes that show ≥95% average nucleotide identity (ANI) to the type genome, which has been assigned the MAG ID 1383 and which is available via NCBI BioSample SAMN15817125. The GC content of the type genome is 57.82% and the genome length is 1.7 Mbp.	
Description of Candidatus Merdiplasma gen. nov.	
Candidatus Merdiplasma (Mer.di.plas’ma. L. fem. n. merda dung; Gr. neut. n. plasma a form; N.L. neut. n. Merdiplasma a microbe associated with faeces)	
A bacterial genus identified by metagenomic analyses. The genus includes all bacteria with genomes that show ≥60% average amino acid identity (AAI) to the type genome from the type species Candidatus Merdiplasma excrementigallinarum. This is a name for the alphanumeric GTDB genus UBA2856. This genus has been assigned by GTDB-Tk v1.3.0 working on GTDB Release 05-RS95 (Chaumeil et al., 2019; Parks et al., 2020) to the order Lachnospirales and to the family Lachnospiraceae.	
Description of Candidatus Merdiplasma excrementigallinarum sp. nov.	
Candidatus Merdiplasma excrementigallinarum (ex.cre.men.ti.gal.li.na’rum. L. neut. n. excrementum excrement; L. fem. n. gallina hen; N.L. gen. n. excrementigallinarum of hen excrement)	
A bacterial species identified by metagenomic analyses. This species includes all bacteria with genomes that show ≥95% average nucleotide identity (ANI) to the type genome, which has been assigned the MAG ID ChiBcec6-7307 and which is available via NCBI BioSample SAMN15817161. The GC content of the type genome is 52.97% and the genome length is 2.4 Mbp.	
Description of Candidatus Merdisoma gen. nov.	
Candidatus Merdisoma (Mer.di.so’ma. L. fem. n. merda dung; Gr. neut. n. soma a body; N.L. neut. n. Merdisoma a microbe associated with faeces)	
A bacterial genus identified by metagenomic analyses. The genus includes all bacteria with genomes that show ≥60% average amino acid identity (AAI) to the type genome from the type species Candidatus Merdisoma merdipullorum. This is a name for the alphanumeric GTDB genus GCA-900066575. This genus has been assigned by GTDB-Tk v1.3.0 working on GTDB Release 05-RS95 (Chaumeil et al., 2019; Parks et al., 2020) to the order Lachnospirales and to the family Lachnospiraceae.	
Description of Candidatus Merdisoma faecale sp. nov.	
Candidatus Merdisoma faecalis (fae.ca’le. L. neut. adj. faecale of faeces)	
A bacterial species identified by metagenomic analyses. This species includes all bacteria with genomes that show ≥95% average nucleotide identity (ANI) to the type genome, which has been assigned the MAG ID ChiBcolR2-1241 and which is available via NCBI BioSample SAMN15817219. This is a new name for the alphanumeric GTDB species sp002160765. The GC content of the type genome is 51.56% and the genome length is 2.7 Mbp.	
Description of Candidatus Merdisoma merdipullorum sp. nov.	
Candidatus Merdisoma merdipullorum (mer.di.pul.lo’rum. L. fem. n. merda faeces; L. masc. n. pullus a young chicken; N.L. gen. n. merdipullorum of the faeces of young chickens)	
A bacterial species identified by metagenomic analyses. This species includes all bacteria with genomes that show ≥95% average nucleotide identity (ANI) to the type genome, which has been assigned the MAG ID CHK197-19677 and which is available via NCBI BioSample SAMN15817042. The GC content of the type genome is 50.06% and the genome length is 2.9 Mbp.	
Description of Candidatus Merdivicinus gen. nov.	
Candidatus Merdivicinus (Mer.di.vi.ci’nus. L. fem. n. merda dung; L. masc. n. vicinus a neighbour; N.L. masc. n. Merdivicinus a microbe associated with faeces)	
A bacterial genus identified by metagenomic analyses. The genus includes all bacteria with genomes that show ≥60% average amino acid identity (AAI) to the type genome from the type species Candidatus Merdivicinus faecavium. This is a name for the alphanumeric GTDB genus UMGS1826. This genus has been assigned by GTDB-Tk v1.3.0 working on GTDB Release 05-RS95 (Chaumeil et al., 2019; Parks et al., 2020) to the order Oscillospirales and to the family Ruminococcaceae.	
Description of Candidatus Merdivicinus excrementipullorum sp. nov.	
Candidatus Merdivicinus excrementipullorum (ex.cre.men.ti.pul.lo’rum. L. neut. n. excrementum excrement; L. masc. n. pullus a young chicken; N.L. gen. n. excrementipullorum of young chicken excrement)	
A bacterial species identified by metagenomic analyses. This species includes all bacteria with genomes that show ≥95% average nucleotide identity (ANI) to the type genome, which has been assigned the MAG ID CHK199-13235 and which is available via NCBI BioSample SAMN15817032. The GC content of the type genome is 54.67% and the genome length is 2.5 Mbp.	

Description of Candidatus Merdivicinus faecavium sp. nov.	
Candidatus Merdivicinus faecavium (faec.a’vi.um. L. fem. n. faex, faecis excrement; L. fem. n. avis bird; N.L. gen. n. faecavium of bird faeces)	
A bacterial species identified by metagenomic analyses. This species includes all bacteria with genomes that show ≥95% average nucleotide identity (ANI) to the type genome, which has been assigned the MAG ID CHK186-19003 and which is available via NCBI BioSample SAMN15817036. The GC content of the type genome is 61.26% and the genome length is 2.5 Mbp.	
Description of Candidatus Merdivicinus intestinavium sp. nov.	
Candidatus Merdivicinus intestinavium (in.tes.tin.a’vi.um. L. neut. n. intestinum gut; L. fem. n. avis bird; N.L. gen. n. intestinavium of the gut of birds)	
A bacterial species identified by metagenomic analyses. This species includes all bacteria with genomes that show ≥95% average nucleotide identity (ANI) to the type genome, which has been assigned the MAG ID CHK188-1901 and which is available via NCBI BioSample SAMN15817003. The GC content of the type genome is 59.41% and the genome length is 2.3 Mbp.	
Description of Candidatus Merdivicinus intestinigallinarum sp. nov.	
Candidatus Merdivicinus intestinigallinarum (in.tes.ti.ni.gal.li.na’rum. L. neut. n. intestinum gut; L. fem. n. gallina hen; N.L. gen. n. intestinigallinarum of the gut of the hens)	
A bacterial species identified by metagenomic analyses. This species includes all bacteria with genomes that show ≥95% average nucleotide identity (ANI) to the type genome, which has been assigned the MAG ID ChiBcec18-2170 and which is available via NCBI BioSample SAMN15817163. The GC content of the type genome is 56.27% and the genome length is 2.5 Mbp.	
Description of Candidatus Merdivivens gen. nov.	
Candidatus Merdivivens (Mer.di.vi’vens. L. fem. n. merda dung; N.L. pres. part. vivens living; N.L. fem. n. Merdivivens a microbe associated with faeces)	
A bacterial genus identified by metagenomic analyses. The genus includes all bacteria with genomes that show ≥60% average amino acid identity (AAI) to the type genome from the type species Candidatus Merdivivens pullistercoris. This is a name for the alphanumeric GTDB genus UBA3382. This genus has been assigned by GTDB-Tk v1.3.0 working on GTDB Release 05-RS95 (Chaumeil et al., 2019; Parks et al., 2020) to the order Bacteroidales and to the family UBA932.	
Description of Candidatus Merdivivens faecigallinarum sp. nov.	
Candidatus Merdivivens faecigallinarum (fae.ci.gal.li.na’rum. L. fem. n. faex, faecis excrement; L. fem. n. gallina hen; N.L. gen. n. faecigallinarum of hen faeces)	
A bacterial species identified by metagenomic analyses. This species includes all bacteria with genomes that show ≥95% average nucleotide identity (ANI) to the type genome, which has been assigned the MAG ID B3-2255 and which is available via NCBI BioSample SAMN15817168. This is a new name for the alphanumeric GTDB species sp002159555. The GC content of the type genome is 49.87% and the genome length is 1.9 Mbp.	
Description of Candidatus Merdivivens pullicola sp. nov.	
Candidatus Merdivivens pullicola (pul.li’co.la. L. masc. n. pullus a young chicken; L. suff. -cola inhabitant of; N.L. n. pullicola an inhabitant of young chickens)	
A bacterial species identified by metagenomic analyses. This species includes all bacteria with genomes that show ≥95% average nucleotide identity (ANI) to the type genome, which has been assigned the MAG ID B1-8020 and which is available via NCBI BioSample SAMN15817062. The GC content of the type genome is 48.22% and the genome length is 2.0 Mbp.	
Description of Candidatus Merdivivens pullistercoris sp. nov.	
Candidatus Merdivivens pullistercoris (pul.li.ster’co.ris. L. masc. n. pullus a young chicken; L. neut. n. stercus dung; N.L. gen. n. pullistercoris of young chicken faeces)	
A bacterial species identified by metagenomic analyses. This species includes all bacteria with genomes that show ≥95% average nucleotide identity (ANI) to the type genome, which has been assigned the MAG ID 10037 and which is available via NCBI BioSample SAMN15817074. The GC content of the type genome is 48.95% and the genome length is 2.1 Mbp.	
Description of Candidatus Merdousia gen. nov.	
Candidatus Merdousia (Merd.ou’si.a. L. fem. n. merda dung; Gr. fem. n. ousia an essence; N.L. fem. n. Merdousia a microbe associated with faeces)	
A bacterial genus identified by metagenomic analyses. The genus includes all bacteria with genomes that show ≥60% average amino acid identity (AAI) to the type genome from the type species Candidatus Merdousia gallistercoris. This is a name for the alphanumeric GTDB genus CAG-312. This genus has been assigned by GTDB-Tk v1.3.0 working on GTDB Release 05-RS95 (Chaumeil et al., 2019; Parks et al., 2020) to the order Opitutales and to the family CAG-312.	

Description of Candidatus Merdousia gallistercoris sp. nov.	
Candidatus Merdousia gallistercoris (gal.li.ster’co.ris. L. masc. n gallus chicken; L. neut. n. stercus dung; N.L. gen. n. gallistercoris of chicken faeces)	
A bacterial species identified by metagenomic analyses. This species includes all bacteria with genomes that show ≥95% average nucleotide identity (ANI) to the type genome, which has been assigned the MAG ID CHK197-16368 and which is available via NCBI BioSample SAMN15817207. This is a new name for the alphanumeric GTDB species sp900545715. The GC content of the type genome is 49.37% and the genome length is 2.4 Mbp.	
Description of Candidatus Methanocorpusculum faecipullorum sp. nov.	
Candidatus Methanocorpusculum faecipullorum (fae.ci.pul.lo’rum. L. fem. n. faex, faecis excrement; L. masc. n. pullus a young chicken; N.L. gen. n. faecipullorum of young chicken faeces)	
An archaeal species identified by metagenomic analyses. This species includes all archaea with genomes that show ≥95% average nucleotide identity (ANI) to the type genome, which has been assigned the MAG ID E1-3281 and which is available via NCBI BioSample SAMN15816796. The GC content of the type genome is 50.72% and the genome length is 1.2 Mbp.	
Description of Candidatus Methanospyradousia gen. nov.	
Candidatus Methanospyradousia (Meth.an.o.spy.rad.ou’si.a. N.L. neut. n. methanum, methane; N.L. pref. methano-, pertaining to methane; Gr. fem. n. spyras ball of dung; Gr. fem. n. ousia an essence; N.L. fem. n. Methanospyradousia a methanogenic microbe associated with the intestines)	
An archaeal genus identified by metagenomic analyses. The genus includes all archaea with genomes that show ≥60% average amino acid identity (AAI) to the type genome from the type species Candidatus Methanospyradousia avicola. This is a name for the alphanumeric GTDB genus UBA71. This genus has been assigned by GTDB-Tk v1.3.0 working on GTDB Release 05-RS95 (Chaumeil et al., 2019; Parks et al., 2020) to the order Methanomassiliicoccales and to the family Methanomethylophilaceae.	
Description of Candidatus Methanospyradousia avicola sp. nov.	
Candidatus Methanospyradousia avicola (a.vi’co.la. L. fem. n. avis bird; L. suff. -cola inhabitant of; N.L. n. avicola inhabitant of birds)	
An archaeal species identified by metagenomic analyses. This species includes all archaea with genomes that show ≥95% average nucleotide identity (ANI) to the type genome, which has been assigned the MAG ID 6227 and which is available via NCBI BioSample SAMN15817164. The GC content of the type genome is 60.22% and the genome length is 1.5 Mbp.	
Description of Candidatus Microbacterium pullistercoris sp. nov.	
Candidatus Microbacterium pullistercoris (pul.li.ster’co.ris. L. masc. n. pullus a young chicken; L. neut. n. stercus dung; N.L. gen. n. pullistercoris of young chicken faeces)	
A bacterial species identified by metagenomic analyses. This species includes all bacteria with genomes that show ≥95% average nucleotide identity (ANI) to the type genome, which has been assigned the MAG ID ChiGjej1B1-5908 and which is available via NCBI BioSample SAMN15816649. The GC content of the type genome is 68.74% and the genome length is 2.5 Mbp.	
Description of Candidatus Microbacterium stercoravium sp. nov.	
Candidatus Microbacterium stercoravium (ster.cor.a’vi.um. L. neut. n. stercus dung; L. fem. n. avis bird; N.L. gen. n. stercoravium of bird faeces)	
A bacterial species identified by metagenomic analyses. This species includes all bacteria with genomes that show ≥95% average nucleotide identity (ANI) to the type genome, which has been assigned the MAG ID ChiHjej8B7-3636 and which is available via NCBI BioSample SAMN15816680. The GC content of the type genome is 69.44% and the genome length is 2.5 Mbp.	
Description of Candidatus Monoglobus merdigallinarum sp. nov.	
Candidatus Monoglobus merdigallinarum (mer.di.gal.li.na’rum. L. fem. n. merda faeces; L. fem. n. gallina hen; N.L. gen. n. merdigallinarum of hen faeces)	
A bacterial species identified by metagenomic analyses. This species includes all bacteria with genomes that show ≥95% average nucleotide identity (ANI) to the type genome, which has been assigned the MAG ID 5790 and which is available via NCBI BioSample SAMN15816780. The GC content of the type genome is 48.24% and the genome length is 1.5 Mbp.	
Description of Candidatus Mucispirillum faecigallinarum sp. nov.	
Candidatus Mucispirillum faecigallinarum (fae.ci.gal.li.na’rum. L. fem. n. faex, faecis excrement; L. fem. n. gallina hen; N.L. gen. n. faecigallinarum of hen faeces)	
A bacterial species identified by metagenomic analyses. This species includes all bacteria with genomes that show ≥95% average nucleotide identity (ANI) to the type genome, which has been assigned the MAG ID ChiW4-1371 and which is available via NCBI BioSample SAMN15816684. The GC content of the type genome is 31.75% and the genome length is 2.2 Mbp.	

Description of Candidatus Negativibacillus faecipullorum sp. nov.	
Candidatus Negativibacillus faecipullorum (fae.ci.pul.lo’rum. L. fem. n. faex, faecis excrement; L. masc. n. pullus a young chicken; N.L. gen. n. faecipullorum of young chicken faeces)	
A bacterial species identified by metagenomic analyses. This species includes all bacteria with genomes that show ≥95% average nucleotide identity (ANI) to the type genome, which has been assigned the MAG ID ChiBcec6-1156 and which is available via NCBI BioSample SAMN15816879. This is a new name for the alphanumeric GTDB species sp900547455. The GC content of the type genome is 57.54% and the genome length is 2.0 Mbp.	
Description of Candidatus Nesterenkonia stercoripullorum sp. nov.	
Candidatus Nesterenkonia stercoripullorum (ster.co.ri.pul.lo’rum. L. neut. n. stercus dung; L. masc. n. pullus a young chicken; N.L. gen. n. stercoripullorum of the faceces of young chickens)	
A bacterial species identified by metagenomic analyses. This species includes all bacteria with genomes that show ≥95% average nucleotide identity (ANI) to the type genome, which has been assigned the MAG ID ChiHejej3B27-3195 and which is available via NCBI BioSample SAMN15816751. The GC content of the type genome is 65.88% and the genome length is 2.6 Mbp.	
Description of Candidatus Niameybacter stercoravium sp. nov.	
Candidatus Niameybacter stercoravium (ster.cor.a’vi.um. L. neut. n. stercus dung; L. fem. n. avis bird; N.L. gen. n. stercoravium of bird faeces)	
A bacterial species identified by metagenomic analyses. This species includes all bacteria with genomes that show ≥95% average nucleotide identity (ANI) to the type genome, which has been assigned the MAG ID 2467 and which is available via NCBI BioSample SAMN15816773. The GC content of the type genome is 35.14% and the genome length is 2.9 Mbp.	
Description of Candidatus Nocardiopsis merdipullorum sp. nov.	
Candidatus Nocardiopsis merdipullorum (mer.di.pul.lo’rum. L. fem. n. merda faeces; L. masc. n. pullus a young chicken; N.L. gen. n. merdipullorum of the faeces of young chickens)	
A bacterial species identified by metagenomic analyses. This species includes all bacteria with genomes that show ≥95% average nucleotide identity (ANI) to the type genome, which has been assigned the MAG ID ChiHjej10B9-18110 and which is available via NCBI BioSample SAMN15816716. The GC content of the type genome is 65.81% and the genome length is 4.2 Mbp.	
Description of Candidatus Nosocomiicoccus stercorigallinarum sp. nov.	
Candidatus Nosocomiicoccus stercorigallinarum (ster.co.ri.gal.li.na’rum. L. neut. n. stercus dung; L. fem. n. gallina hen; N.L. gen. n. stercorigallinarum of hen faeces)	
A bacterial species identified by metagenomic analyses. This species includes all bacteria with genomes that show ≥95% average nucleotide identity (ANI) to the type genome, which has been assigned the MAG ID CHK169-14505 and which is available via NCBI BioSample SAMN15816647. The GC content of the type genome is 34.64% and the genome length is 1.3 Mbp.	
Description of Candidatus Oceanisphaera merdipullorum sp. nov.	
Candidatus Oceanisphaera merdipullorum (mer.di.pul.lo’rum. L. fem. n. merda faeces; L. masc. n. pullus a young chicken; N.L. gen. n. merdipullorum of the faeces of young chickens)	
A bacterial species identified by metagenomic analyses. This species includes all bacteria with genomes that show ≥95% average nucleotide identity (ANI) to the type genome, which has been assigned the MAG ID 819 and which is available via NCBI BioSample SAMN15816797. The GC content of the type genome is 50.19% and the genome length is 2.9 Mbp.	
Description of Candidatus Odoribacter faecigallinarum sp. nov.	
Candidatus Odoribacter faecigallinarum (fae.ci.gal.li.na’rum. L. fem. n. faex, faecis excrement; L. fem. n. gallina hen; N.L. gen. n. faecigallinarum of hen faeces)	
A bacterial species identified by metagenomic analyses. This species includes all bacteria with genomes that show ≥95% average nucleotide identity (ANI) to the type genome, which has been assigned the MAG ID 23274 and which is available via NCBI BioSample SAMN15816743. The GC content of the type genome is 48.17% and the genome length is 2.2 Mbp.	
Description of Candidatus Olsenella avicola sp. nov.	
Candidatus Olsenella avicola (a.vi’co.la. L. fem. n. avis bird; L. suff. -cola inhabitant of; N.L. n. avicola inhabitant of birds)	
A bacterial species identified by metagenomic analyses. This species includes all bacteria with genomes that show ≥95% average nucleotide identity (ANI) to the type genome, which has been assigned the MAG ID CHK1-7693 and which is available via NCBI BioSample SAMN15816923. This is a new name for the alphanumeric GTDB species sp002159625. Although GTDB has assigned this species to the genus it calls Olsenella_E, this genus designation cannot be incorporated into a well-formed binomial, so in naming this species, we have used the current validly published name for the genus. The GC content of the type genome is 67.59% and the genome length is 2.2 Mbp.	

Description of Candidatus Olsenella avistercoris sp. nov.	
Candidatus Olsenella avistercoris (a.vi.ster’co.ris. L. fem. n. avis bird; L. neut. n. stercus dung; N.L. gen. n. avistercoris of bird faeces)	
A bacterial species identified by metagenomic analyses. This species includes all bacteria with genomes that show ≥95% average nucleotide identity (ANI) to the type genome, which has been assigned the MAG ID CHK136-6238 and which is available via NCBI BioSample SAMN15816919. This is a new name for the alphanumeric GTDB species sp002160255. Although GTDB has assigned this species to the genus it calls Olsenella_E, this genus designation cannot be incorporated into a well-formed binomial, so in naming this species, we have used the current validly published name for the genus. The GC content of the type genome is 69.05% and the genome length is 2.0 Mbp.	
Description of Candidatus Olsenella excrementavium sp. nov.	
Candidatus Olsenella excrementavium (ex.cre.ment.a’vi.um. L. neut. n. excrementum excrement; L. fem. n. avis bird; N.L. gen. n. excrementavium of bird excrement)	
A bacterial species identified by metagenomic analyses. This species includes all bacteria with genomes that show ≥95% average nucleotide identity (ANI) to the type genome, which has been assigned the MAG ID ChiHjej10B9-743 and which is available via NCBI BioSample SAMN15816922. This is a new name for the alphanumeric GTDB species sp002305805. Although GTDB has assigned this species to the genus it calls Olsenella_E, this genus designation cannot be incorporated into a well-formed binomial, so in naming this species, we have used the current validly published name for the genus. The GC content of the type genome is 66.50% and the genome length is 1.8 Mbp.	
Description of Candidatus Olsenella excrementigallinarum sp. nov.	
Candidatus Olsenella excrementigallinarum (ex.cre.men.ti.gal.li.na’rum. L. neut. n. excrementum excrement; L. fem. n. gallina hen; N.L. gen. n. excrementigallinarum of hen excrement)	
A bacterial species identified by metagenomic analyses. This species includes all bacteria with genomes that show ≥95% average nucleotide identity (ANI) to the type genome, which has been assigned the MAG ID ChiHjej12B11-23512 and which is available via NCBI BioSample SAMN15816920. This is a new name for the alphanumeric GTDB species sp900119915. Although GTDB has assigned this species to the genus it calls Olsenella_E, this genus designation cannot be incorporated into a well-formed binomial, so in naming this species, we have used the current validly published name for the genus. The GC content of the type genome is 68.67% and the genome length is 1.8 Mbp.	
Description of Candidatus Olsenella pullicola sp. nov.	
Candidatus Olsenella pullicola (pul.li’co.la. L. masc. n. pullus a young chicken; L. suff. -cola inhabitant of; N.L. n. pullicola an inhabitant of young chickens)	
A bacterial species identified by metagenomic analyses. This species includes all bacteria with genomes that show ≥95% average nucleotide identity (ANI) to the type genome, which has been assigned the MAG ID ChiHecec1B25-7792 and which is available via NCBI BioSample SAMN15816895. Although GTDB has assigned this species to the genus it calls Olsenella_E, this genus designation cannot be incorporated into a well-formed binomial, so in naming this species, we have used the current validly published name for the genus. The GC content of the type genome is 65.74% and the genome length is 2.3 Mbp.	
Description of Candidatus Olsenella pullistercoris sp. nov.	
Candidatus Olsenella pullistercoris (pul.li.ster’co.ris. L. masc. n. pullus a young chicken; L. neut. n. stercus dung; N.L. gen. n. pullistercoris of young chicken faeces)	
A bacterial species identified by metagenomic analyses. This species includes all bacteria with genomes that show ≥95% average nucleotide identity (ANI) to the type genome, which has been assigned the MAG ID ChiHjej12B11-14209 and which is available via NCBI BioSample SAMN15816899. Although GTDB has assigned this species to the genus it calls Olsenella_E, this genus designation cannot be incorporated into a well-formed binomial, so in naming this species, we have used the current validly published name for the genus. The GC content of the type genome is 67.21% and the genome length is 1.9 Mbp.	
Description of Candidatus Olsenella stercoravium sp. nov.	
Candidatus Olsenella stercoravium (ster.cor.a’vi.um. L. neut. n. stercus dung; L. fem. n. avis bird; N.L. gen. n. stercoravium of bird faeces)	
A bacterial species identified by metagenomic analyses. This species includes all bacteria with genomes that show ≥95% average nucleotide identity (ANI) to the type genome, which has been assigned the MAG ID ChiHecolR3B27-1887 and which is available via NCBI BioSample SAMN15816902. Although GTDB has assigned this species to the genus it calls Olsenella_E, this genus designation cannot be incorporated into a well-formed binomial, so in naming this species, we have used the current validly published name for the genus. The GC content of the type genome is 67.26% and the genome length is 1.8 Mbp.	

Description of Candidatus Onthenecus gen. nov.	
Candidatus Onthenecus (Onth.en.e’cus. Gr. masc. n. onthos dung; Gr. masc. enoikos inhabitant; N.L. masc. n. Onthenecus a microbe associated with faeces)	
A bacterial genus identified by metagenomic analyses. The genus includes all bacteria with genomes that show ≥60% average amino acid identity (AAI) to the type genome from the type species Candidatus Onthenecus intestinigallinarum. This is a name for the alphanumeric GTDB genus OEMS01. This genus has been assigned by GTDB-Tk v1.3.0 working on GTDB Release 05-RS95 (Chaumeil et al., 2019; Parks et al., 2020) to the order Christensenellales and to the family CAG-74.	
Description of Candidatus Onthenecus intestinigallinarum sp. nov.	
Candidatus Onthenecus intestinigallinarum (in.tes.ti.ni.gal.li.na’rum. L. neut. n. intestinum gut; L. fem. n. gallina hen; N.L. gen. n. intestinigallinarum of the gut of the hens)	
A bacterial species identified by metagenomic analyses. This species includes all bacteria with genomes that show ≥95% average nucleotide identity (ANI) to the type genome, which has been assigned the MAG ID ChiSxjej2B14-6234 and which is available via NCBI BioSample SAMN15817054. The GC content of the type genome is 66.20% and the genome length is 2.4 Mbp.	
Description of Candidatus Onthocola gen. nov.	
Candidatus Onthocola (On.tho’co.la. Gr. masc. n. onthos dung; L. suff. -cola inhabitant of; N.L. fem. n. Onthocola a microbe associated with faeces)	
A bacterial genus identified by metagenomic analyses. The genus includes all bacteria with genomes that show ≥60% average amino acid identity (AAI) to the type genome from the type species Candidatus Onthocola gallistercoris. This is a name for the alphanumeric GTDB genus. This genus has been assigned by GTDB-Tk v1.3.0 working on GTDB Release 05-RS95 (Chaumeil et al., 2019; Parks et al., 2020) to the order Lachnospirales and to the family Lachnospiraceae.	
Description of Candidatus Onthocola gallistercoris sp. nov.	
Candidatus Onthocola gallistercoris (gal.li.ster’co.ris. L. masc. n gallus chicken; L. neut. n. stercus dung; N.L. gen. n. gallistercoris of chicken faeces)	
A bacterial species identified by metagenomic analyses. This species includes all bacteria with genomes that show ≥95% average nucleotide identity (ANI) to the type genome, which has been assigned the MAG ID CHK187-14744 and which is available via NCBI BioSample SAMN15817044. The GC content of the type genome is 48.32% and the genome length is 2.4 Mbp.	
Description of Candidatus Onthocola stercoravium sp. nov.	
Candidatus Onthocola stercoravium (ster.cor.a’vi.um. L. neut. n. stercus dung; L. fem. n. avis bird; N.L. gen. n. stercoravium of bird faeces)	
A bacterial species identified by metagenomic analyses. This species includes all bacteria with genomes that show ≥95% average nucleotide identity (ANI) to the type genome, which has been assigned the MAG ID ChiW5-5982 and which is available via NCBI BioSample SAMN15817021. The GC content of the type genome is 28.08% and the genome length is 1.4 Mbp.	
Description of Candidatus Onthocola stercorigallinarum sp. nov.	
Candidatus Onthocola stercorigallinarum (ster.co.ri.gal.li.na’rum. L. neut. n. stercus dung; L. fem. n. gallina hen; N.L. gen. n. stercorigallinarum of hen faeces)	
A bacterial species identified by metagenomic analyses. This species includes all bacteria with genomes that show ≥95% average nucleotide identity (ANI) to the type genome, which has been assigned the MAG ID CHK195-3072 and which is available via NCBI BioSample SAMN15817046. The GC content of the type genome is 27.21% and the genome length is 1.3 Mbp.	
Description of Candidatus Onthomonas gen. nov.	
Candidatus Onthomonas (On.tho.mo’nas. Gr. masc. n. onthos dung; L. fem. n. monas a monad; N.L. fem. n. Onthomonas a microbe associated with faeces)	
A bacterial genus identified by metagenomic analyses. The genus includes all bacteria with genomes that show ≥60% average amino acid identity (AAI) to the type genome from the type species Candidatus Onthomonas avicola. This is a name for the alphanumeric GTDB genus NK3B98. This genus has been assigned by GTDB-Tk v1.3.0 working on GTDB Release 05-RS95 (Chaumeil et al., 2019; Parks et al., 2020) to the order Oscillospirales and to the family Oscillospiraceae.	
Description of Candidatus Onthomonas avicola sp. nov.	
Candidatus Onthomonas avicola (a.vi’co.la. L. fem. n. avis bird; L. suff. -cola inhabitant of; N.L. n. avicola inhabitant of birds)	
A bacterial species identified by metagenomic analyses. This species includes all bacteria with genomes that show ≥95% average nucleotide identity (ANI) to the type genome, which has been assigned the MAG ID ChiGjej6B6-14002 and which is available via NCBI BioSample SAMN15817096. The GC content of the type genome is 63.01% and the genome length is 2.4 Mbp.	

Description of Candidatus Onthomorpha gen. nov.	
Candidatus Onthomorpha (On.tho.mor’pha. Gr. masc. n. onthos dung; Gr. fem. n. morphe a form, shape; N.L. fem. n. Onthomorpha a microbe associated with faeces)	
A bacterial genus identified by metagenomic analyses. The genus includes all bacteria with genomes that show ≥60% average amino acid identity (AAI) to the type genome from the type species Candidatus Onthomorpha intestinigallinarum. This is a name for the alphanumeric GTDB genus UBA3388. This genus has been assigned by GTDB-Tk v1.3.0 working on GTDB Release 05-RS95 (Chaumeil et al., 2019; Parks et al., 2020) to the order Bacteroidales and to the family P3.	
Description of Candidatus Onthomorpha intestinigallinarum sp. nov.	
Candidatus Onthomorpha intestinigallinarum (in.tes.ti.ni.gal.li.na’rum. L. neut. n. intestinum gut; L. fem. n. gallina hen; N.L. gen. n. intestinigallinarum of the gut of the hens)	
A bacterial species identified by metagenomic analyses. This species includes all bacteria with genomes that show ≥95% average nucleotide identity (ANI) to the type genome, which has been assigned the MAG ID Gambia16-930 and which is available via NCBI BioSample SAMN15817128. The GC content of the type genome is 42.76% and the genome length is 1.7 Mbp.	
Description of Candidatus Onthoplasma gen. nov.	
Candidatus Onthoplasma (On.tho.plas’ma. Gr. masc. n. onthos dung; Gr. neut. n. plasma a form; N.L. neut. n. Onthoplasma a microbe associated with faeces)	
A bacterial genus identified by metagenomic analyses. The genus includes all bacteria with genomes that show ≥60% average amino acid identity (AAI) to the type genome from the type species Candidatus Onthoplasma faecipullorum. This is a name for the alphanumeric GTDB genus UBA4626. This genus has been assigned by GTDB-Tk v1.3.0 working on GTDB Release 05-RS95 (Chaumeil et al., 2019; Parks et al., 2020) to the order 4C28d-15 and to the family UBA1242.	
Description of Candidatus Onthoplasma faecigallinarum sp. nov.	
Candidatus Onthoplasma faecigallinarum (fae.ci.gal.li.na’rum. L. fem. n. faex, faecis excrement; L. fem. n. gallina hen; N.L. gen. n. faecigallinarum of hen faeces)	
A bacterial species identified by metagenomic analyses. This species includes all bacteria with genomes that show ≥95% average nucleotide identity (ANI) to the type genome, which has been assigned the MAG ID 5992 and which is available via NCBI BioSample SAMN15817127. The GC content of the type genome is 34.71% and the genome length is 1.0 Mbp.	
Description of Candidatus Onthoplasma faecipullorum sp. nov.	
Candidatus Onthoplasma faecipullorum (fae.ci.pul.lo’rum. L. fem. n. faex, faecis excrement; L. masc. n. pullus a young chicken; N.L. gen. n. faecipullorum of young chicken faeces)	
A bacterial species identified by metagenomic analyses. This species includes all bacteria with genomes that show ≥95% average nucleotide identity (ANI) to the type genome, which has been assigned the MAG ID CHK191-42317 and which is available via NCBI BioSample SAMN15817053. The GC content of the type genome is 31.88% and the genome length is 1.1 Mbp.	
Description of Candidatus Onthosoma gen. nov.	
Candidatus Onthosoma (On.tho.so’ma. Gr. masc. n. onthos dung; Gr. neut. n. soma a body; N.L. neut. n. Onthosoma a microbe associated with faeces)	
A bacterial genus identified by metagenomic analyses. The genus includes all bacteria with genomes that show ≥60% average amino acid identity (AAI) to the type genome from the type species Candidatus Onthosoma merdavium. This is a name for the alphanumeric GTDB genus OEMR01. This genus has been assigned by GTDB-Tk v1.3.0 working on GTDB Release 05-RS95 (Chaumeil et al., 2019; Parks et al., 2020) to the order Erysipelotrichales and to the family Erysipelotrichaceae.	
Description of Candidatus Onthosoma merdavium sp. nov.	
Candidatus Onthosoma merdavium (merd.a’vi.um. L. fem. n. merda faeces; L. fem. n. avis bird; N.L. gen. n. merdavium of bird faeces)	
A bacterial species identified by metagenomic analyses. This species includes all bacteria with genomes that show ≥95% average nucleotide identity (ANI) to the type genome, which has been assigned the MAG ID ChiBcec15-4520 and which is available via NCBI BioSample SAMN15817169. This is a new name for the alphanumeric GTDB species sp900199515. The GC content of the type genome is 45.30% and the genome length is 1.6 Mbp.	
Description of Candidatus Onthousia gen. nov.	
Candidatus Onthousia (Onth.ou’si.a. Gr. masc. n. onthos dung; Gr. fem. n. ousia an essence; N.L. fem. n. Onthousia a microbe associated with faeces)	
A bacterial genus identified by metagenomic analyses. The genus includes all bacteria with genomes that show ≥60% average amino acid identity (AAI) to the type genome from the type species Candidatus Onthousia faecavium. This is a name for the alphanumeric GTDB genus CAG-451. This genus has been assigned by GTDB-Tk v1.3.0 working on GTDB Release 05-RS95 (Chaumeil et al., 2019; Parks et al., 2020) to the order RF39 and to the family CAG-611.	

Description of Candidatus Onthousia excrementipullorum sp. nov.	
Candidatus Onthousia excrementipullorum (ex.cre.men.ti.pul.lo’rum. L. neut. n. excrementum excrement; L. masc. n. pullus a young chicken; N.L. gen. n. excrementipullorum of young chicken excrement)	
A bacterial species identified by metagenomic analyses. This species includes all bacteria with genomes that show ≥95% average nucleotide identity (ANI) to the type genome, which has been assigned the MAG ID CHK184-20233 and which is available via NCBI BioSample SAMN15817019. The GC content of the type genome is 27.72% and the genome length is 1.3 Mbp.	
Description of Candidatus Onthousia faecavium sp. nov.	
Candidatus Onthousia faecavium (faec.a’vi.um. L. fem. n. faex, faecis excrement; L. fem. n. avis bird; N.L. gen. n. faecavium of bird faeces)	
A bacterial species identified by metagenomic analyses. This species includes all bacteria with genomes that show ≥95% average nucleotide identity (ANI) to the type genome, which has been assigned the MAG ID CHK195-6217 and which is available via NCBI BioSample SAMN15817026. The GC content of the type genome is 28.40% and the genome length is 1.3 Mbp.	
Description of Candidatus Onthousia faecigallinarum sp. nov.	
Candidatus Onthousia faecigallinarum (fae.ci.gal.li.na’rum. L. fem. n. faex, faecis excrement; L. fem. n. gallina hen; N.L. gen. n. faecigallinarum of hen faeces)	
A bacterial species identified by metagenomic analyses. This species includes all bacteria with genomes that show ≥95% average nucleotide identity (ANI) to the type genome, which has been assigned the MAG ID CHK135-1819 and which is available via NCBI BioSample SAMN15817105. The GC content of the type genome is 32.62% and the genome length is 1.1 Mbp.	
Description of Candidatus Onthousia faecipullorum sp. nov.	
Candidatus Onthousia faecipullorum (fae.ci.pul.lo’rum. L. fem. n. faex, faecis excrement; L. masc. n. pullus a young chicken; N.L. gen. n. faecipullorum of young chicken faeces)	
A bacterial species identified by metagenomic analyses. This species includes all bacteria with genomes that show ≥95% average nucleotide identity (ANI) to the type genome, which has been assigned the MAG ID CHK195-26880 and which is available via NCBI BioSample SAMN15817040. The GC content of the type genome is 27.41% and the genome length is 1.4 Mbp.	
Description of Candidatus Onthovicinus gen. nov.	
Candidatus Onthovicinus (On.tho.vi.ci’nus. Gr. masc. n. onthos dung; L. masc. n. vicinus a neighbour; N.L. masc. n. Onthovicinus a microbe associated with faeces)	
A bacterial genus identified by metagenomic analyses. The genus includes all bacteria with genomes that show ≥60% average amino acid identity (AAI) to the type genome from the type species Candidatus Onthovicinus excrementipullorum. This is a name for the alphanumeric GTDB genus UMGS1839. This genus has been assigned by GTDB-Tk v1.3.0 working on GTDB Release 05-RS95 (Chaumeil et al., 2019; Parks et al., 2020) to the order Oscillospirales and to the family Acutalibacteraceae.	
Description of Candidatus Onthovicinus excrementipullorum sp. nov.	
Candidatus Onthovicinus excrementipullorum (ex.cre.men.ti.pul.lo’rum. L. neut. n. excrementum excrement; L. masc. n. pullus a young chicken; N.L. gen. n. excrementipullorum of young chicken excrement)	
A bacterial species identified by metagenomic analyses. This species includes all bacteria with genomes that show ≥95% average nucleotide identity (ANI) to the type genome, which has been assigned the MAG ID CHK185-12131 and which is available via NCBI BioSample SAMN15817020. The GC content of the type genome is 55.55% and the genome length is 2.4 Mbp.	
Description of Candidatus Onthovivens gen. nov.	
Candidatus Onthovivens (On.tho.vi’vens. Gr. masc. n. onthos dung; N.L. pres. part. vivens living; N.L. fem. n. Onthovivens a microbe associated with faeces)	
A bacterial genus identified by metagenomic analyses. The genus includes all bacteria with genomes that show ≥60% average amino acid identity (AAI) to the type genome from the type species Candidatus Onthovivens merdipullorum. This is a name for the alphanumeric GTDB genus UBA4855. This genus has been assigned by GTDB-Tk v1.3.0 working on GTDB Release 05-RS95 (Chaumeil et al., 2019; Parks et al., 2020) to the order RFN20 and to the family CAG-826.	
Description of Candidatus Onthovivens merdipullorum sp. nov.	
Candidatus Onthovivens merdipullorum (mer.di.pul.lo’rum. L. fem. n. merda faeces; L. masc. n. pullus a young chicken; N.L. gen. n. merdipullorum of the faeces of young chickens)	
A bacterial species identified by metagenomic analyses. This species includes all bacteria with genomes that show ≥95% average nucleotide identity (ANI) to the type genome, which has been assigned the MAG ID 11159 and which is available via NCBI BioSample SAMN15817139. The GC content of the type genome is 27.05% and the genome length is 1.5 Mbp.	

Description of Candidatus Ornithocaccomicrobium gen. nov.	
Candidatus Ornithocaccomicrobium (Or.ni.tho.cac.co.mi.cro’bi.um. Gr. masc. or fem. n. ornis, ornithos bird; Gr. fem. n. kakke faeces; N.L. neut. n. microbium a microbe; N.L. neut. n. Ornithocaccomicrobium A microbe found in chicken faceces)	
A bacterial genus identified by metagenomic analyses. The genus includes all bacteria with genomes that show ≥60% average amino acid identity (AAI) to the type genome from the type species Candidatus Ornithocaccomicrobium faecavium. This genus was identified but not named by Glendinning et al. (2020). This genus has been assigned by GTDB-Tk v1.3.0 working on GTDB Release 05-RS95 (Chaumeil et al., 2019; Parks et al., 2020) to the order Christensenellales and to the family CAG-74.	
Description of Candidatus Ornithocaccomicrobium faecavium sp. nov.	
Candidatus Ornithocaccomicrobium faecavium (faec.a’vi.um. L. fem. n. faex, faecis excrement; L. fem. n. avis bird; N.L. gen. n. faecavium of bird faeces)	
A bacterial species identified by metagenomic analyses. This species includes all bacteria with genomes that show ≥95% average nucleotide identity (ANI) to the type genome, which has been assigned the MAG ID CHK183-6373 and which is available via NCBI BioSample SAMN15816945. The GC content of the type genome is 59.48% and the genome length is 2.9 Mbp.	
Description of Candidatus Ornithoclostridium gen. nov.	
Candidatus Ornithoclostridium (Or.ni.tho.clos.tri’di.um. Gr. masc. or fem. n. ornis, ornithos bird; N.L. neut. n. Clostridium a genus name; N.L. neut. n. Ornithoclostridium a genus related to the genus Clostridium but distinct from it and found in poultry)	
A bacterial genus identified by metagenomic analyses. The genus includes all bacteria with genomes that show ≥60% average amino acid identity (AAI) to the type genome from the type species Candidatus Ornithoclostridium excrementipullorum. This genus has been assigned by GTDB-Tk v1.3.0 working on GTDB Release 05-RS95 (Chaumeil et al., 2019; Parks et al., 2020) to the order Christensenellales and to the family UBA3700.	
Description of Candidatus Ornithoclostridium excrementipullorum sp. nov.	
Candidatus Ornithoclostridium excrementipullorum (ex.cre.men.ti.pul.lo’rum. L. neut. n. excrementum excrement; L. masc. n. pullus a young chicken; N.L. gen. n. excrementipullorum of young chicken excrement)	
A bacterial species identified by metagenomic analyses. This species includes all bacteria with genomes that show ≥95% average nucleotide identity (ANI) to the type genome, which has been assigned the MAG ID ChiW5-1639 and which is available via NCBI BioSample SAMN15816971. The GC content of the type genome is 54.62% and the genome length is 1.6 Mbp.	
Description of Candidatus Ornithoclostridium faecavium sp. nov.	
Candidatus Ornithoclostridium faecavium (faec.a’vi.um. L. fem. n. faex, faecis excrement; L. fem. n. avis bird; N.L. gen. n. faecavium of bird faeces)	
A bacterial species identified by metagenomic analyses. This species includes all bacteria with genomes that show ≥95% average nucleotide identity (ANI) to the type genome, which has been assigned the MAG ID 63 and which is available via NCBI BioSample SAMN15816992. The GC content of the type genome is 48.09% and the genome length is 1.8 Mbp.	
Description of Candidatus Ornithoclostridium faecigallinarum sp. nov.	
Candidatus Ornithoclostridium faecigallinarum (fae.ci.gal.li.na’rum. L. fem. n. faex, faecis excrement; L. fem. n. gallina hen; N.L. gen. n. faecigallinarum of hen faeces)	
A bacterial species identified by metagenomic analyses. This species includes all bacteria with genomes that show ≥95% average nucleotide identity (ANI) to the type genome, which has been assigned the MAG ID ChiHcolR4-3946 and which is available via NCBI BioSample SAMN15816996. The GC content of the type genome is 57.12% and the genome length is 1.6 Mbp.	
Description of Candidatus Ornithomonoglobus gen. nov.	
Candidatus Ornithomonoglobus (Or.ni.tho.mo.no.glo’bus. Gr. masc. or fem. n. ornis, ornithos bird; N.L. masc. n. Monoglobus a genus name; N.L. masc. n. Ornithomonoglobus a genus related to the genus Monoglobus but distinct from it and found in poultry)	
A bacterial genus identified by metagenomic analyses. The genus includes all bacteria with genomes that show ≥60% average amino acid identity (AAI) to the type genome from the type species Candidatus Ornithomonoglobus merdipullorum. This genus was identified but not named by Glendinning et al. (2020). This genus has been assigned by GTDB-Tk v1.3.0 working on GTDB Release 05-RS95 (Chaumeil et al., 2019; Parks et al., 2020) to the order Monoglobales and to the family UBA1381.	
Description of Candidatus Ornithomonoglobus intestinigallinarum sp. nov.	
Candidatus Ornithomonoglobus intestinigallinarum (in.tes.ti.ni.gal.li.na’rum. L. neut. n. intestinum gut; L. fem. n. gallina hen; N.L. gen. n. intestinigallinarum of the gut of the hens)	
A bacterial species identified by metagenomic analyses. This species includes all bacteria with genomes that show ≥95% average nucleotide identity (ANI) to the type genome, which has been assigned the MAG ID CHK181-108 and which is available via NCBI BioSample SAMN15816941. The GC content of the type genome is 49.36% and the genome length is 2.2 Mbp.	

Description of Candidatus Ornithomonoglobus merdipullorum sp. nov.	
Candidatus Ornithomonoglobus merdipullorum (mer.di.pul.lo’rum. L. fem. n. merda faeces; L. masc. n. pullus a young chicken; N.L. gen. n. merdipullorum of the faeces of young chickens)	
A bacterial species identified by metagenomic analyses. This species includes all bacteria with genomes that show ≥95% average nucleotide identity (ANI) to the type genome, which has been assigned the MAG ID USAMLcec3-3695 and which is available via NCBI BioSample SAMN15816942. The GC content of the type genome is 48.52% and the genome length is 2.7 Mbp.	
Description of Candidatus Ornithospirochaeta gen. nov.	
Candidatus Ornithospirochaeta (Or.ni.tho.spi.ro.chae’ta. Gr. masc. or fem. n. ornis, ornithos bird; N.L. fem. n. Spirochaeta a genus name; N.L. fem. n. Ornithospirochaeta a genus related to the genus Spirochaeta but distinct from it and found in poultry)	
A bacterial genus identified by metagenomic analyses. The genus includes all bacteria with genomes that show ≥60% average amino acid identity (AAI) to the type genome from the type species Candidatus Ornithospirochaeta stercoravium. This genus has been assigned by GTDB-Tk v1.3.0 working on GTDB Release 05-RS95 (Chaumeil et al., 2019; Parks et al., 2020) to the order Sphaerochaetales and to the family Sphaerochaetaceae.	
Description of Candidatus Ornithospirochaeta avicola sp. nov.	
Candidatus Ornithospirochaeta avicola (a.vi’co.la. L. fem. n. avis bird; L. suff. -cola inhabitant of; N.L. n. avicola inhabitant of birds)	
A bacterial species identified by metagenomic analyses. This species includes all bacteria with genomes that show ≥95% average nucleotide identity (ANI) to the type genome, which has been assigned the MAG ID Gambia11-129 and which is available via NCBI BioSample SAMN15816993. The GC content of the type genome is 42.81% and the genome length is 1.5 Mbp.	
Description of Candidatus Ornithospirochaeta stercoravium sp. nov.	
Candidatus Ornithospirochaeta stercoravium (ster.cor.a’vi.um. L. neut. n. stercus dung; L. fem. n. avis bird; N.L. gen. n. stercoravium of bird faeces)	
A bacterial species identified by metagenomic analyses. This species includes all bacteria with genomes that show ≥95% average nucleotide identity (ANI) to the type genome, which has been assigned the MAG ID 14700 and which is available via NCBI BioSample SAMN15816953. The GC content of the type genome is 46.26% and the genome length is 2.0 Mbp.	
Description of Candidatus Ornithospirochaeta stercorigallinarum sp. nov.	
Candidatus Ornithospirochaeta stercorigallinarum (ster.co.ri.gal.li.na’rum. L. neut. n. stercus dung; L. fem. n. gallina hen; N.L. gen. n. stercorigallinarum of hen faeces)	
A bacterial species identified by metagenomic analyses. This species includes all bacteria with genomes that show ≥95% average nucleotide identity (ANI) to the type genome, which has been assigned the MAG ID ChiHecec3B27-9561 and which is available via NCBI BioSample SAMN15816957. The GC content of the type genome is 46.77% and the genome length is 1.9 Mbp.	
Description of Candidatus Ornithospirochaeta stercoripullorum sp. nov.	
Candidatus Ornithospirochaeta stercoripullorum (ster.co.ri.pul.lo’rum. L. neut. n. stercus dung; L. masc. n. pullus a young chicken; N.L. gen. n. stercoripullorum of the faceces of young chickens)	
A bacterial species identified by metagenomic analyses. This species includes all bacteria with genomes that show ≥95% average nucleotide identity (ANI) to the type genome, which has been assigned the MAG ID 7293 and which is available via NCBI BioSample SAMN15816978. The GC content of the type genome is 45.57% and the genome length is 2.0 Mbp.	
Description of Candidatus Oscillibacter avistercoris sp. nov.	
Candidatus Oscillibacter avistercoris (a.vi.ster’co.ris. L. fem. n. avis bird; L. neut. n. stercus dung; N.L. gen. n. avistercoris of bird faeces)	
A bacterial species identified by metagenomic analyses. This species includes all bacteria with genomes that show ≥95% average nucleotide identity (ANI) to the type genome, which has been assigned the MAG ID CHK176-14096 and which is available via NCBI BioSample SAMN15816820. This is a new name for the alphanumeric GTDB species sp900556925. The GC content of the type genome is 63.55% and the genome length is 2.3 Mbp.	
Description of Candidatus Oscillibacter excrementavium sp. nov.	
Candidatus Oscillibacter excrementavium (ex.cre.ment.a’vi.um. L. neut. n. excrementum excrement; L. fem. n. avis bird; N.L. gen. n. excrementavium of bird excrement)	
A bacterial species identified by metagenomic analyses. This species includes all bacteria with genomes that show ≥95% average nucleotide identity (ANI) to the type genome, which has been assigned the MAG ID 5302 and which is available via NCBI BioSample SAMN15816661. The GC content of the type genome is 63.73% and the genome length is 2.5 Mbp.	

Description of Candidatus Oscillibacter excrementigallinarum sp. nov.	
Candidatus Oscillibacter excrementigallinarum (ex.cre.men.ti.gal.li.na’rum. L. neut. n. excrementum excrement; L. fem. n. gallina hen; N.L. gen. n. excrementigallinarum of hen excrement)	
A bacterial species identified by metagenomic analyses. This species includes all bacteria with genomes that show ≥95% average nucleotide identity (ANI) to the type genome, which has been assigned the MAG ID ChiBcec18-1249 and which is available via NCBI BioSample SAMN15816667. The GC content of the type genome is 64.01% and the genome length is 2.4 Mbp.	
Description of Candidatus Oscillibacter pullicola sp. nov.	
Candidatus Oscillibacter pullicola (pul.li’co.la. L. masc. n. pullus a young chicken; L. suff. -cola inhabitant of; N.L. n. pullicola an inhabitant of young chickens)	
A bacterial species identified by metagenomic analyses. This species includes all bacteria with genomes that show ≥95% average nucleotide identity (ANI) to the type genome, which has been assigned the MAG ID ChiBcolR2-4535 and which is available via NCBI BioSample SAMN15816652. The GC content of the type genome is 63.62% and the genome length is 2.4 Mbp.	
Description of Candidatus Paenalcaligenes intestinipullorum sp. nov.	
Candidatus Paenalcaligenes intestinipullorum (in.tes.ti.ni.pul.lo’rum. L. neut. n. intestinum gut; L. masc. n. pullus a young chicken; N.L. gen. n. intestinipullorum of the gut of young chickens)	
A bacterial species identified by metagenomic analyses. This species includes all bacteria with genomes that show ≥95% average nucleotide identity (ANI) to the type genome, which has been assigned the MAG ID 9264 and which is available via NCBI BioSample SAMN15816786. The GC content of the type genome is 51.92% and the genome length is 1.8 Mbp.	
Description of Candidatus Paenibacillus intestinavium sp. nov.	
Candidatus Paenibacillus intestinavium (in.tes.tin.a’vi.um. L. neut. n. intestinum gut; L. fem. n. avis bird; N.L. gen. n. intestinavium of the gut of birds)	
A bacterial species identified by metagenomic analyses. This species includes all bacteria with genomes that show ≥95% average nucleotide identity (ANI) to the type genome, which has been assigned the MAG ID CHK172-12487 and which is available via NCBI BioSample SAMN15816909. Although GTDB has assigned this species to the genus it calls Paenibacillus_C, this genus designation cannot be incorporated into a well-formed binomial, so in naming this species, we have used the current validly published name for the genus. The GC content of the type genome is 39.31% and the genome length is 4.6 Mbp.	
Description of Candidatus Parabacteroides faecavium sp. nov.	
Candidatus Parabacteroides faecavium (faec.a’vi.um. L. fem. n. faex, faecis excrement; L. fem. n. avis bird; N.L. gen. n. faecavium of bird faeces)	
A bacterial species identified by metagenomic analyses. This species includes all bacteria with genomes that show ≥95% average nucleotide identity (ANI) to the type genome, which has been assigned the MAG ID CHK152-2511 and which is available via NCBI BioSample SAMN15816864. This is a new name for the alphanumeric GTDB species sp000436495. The GC content of the type genome is 42.40% and the genome length is 3.4 Mbp.	
Description of Candidatus Parabacteroides intestinavium sp. nov.	
Candidatus Parabacteroides intestinavium (in.tes.tin.a’vi.um. L. neut. n. intestinum gut; L. fem. n. avis bird; N.L. gen. n. intestinavium of the gut of birds)	
A bacterial species identified by metagenomic analyses. This species includes all bacteria with genomes that show ≥95% average nucleotide identity (ANI) to the type genome, which has been assigned the MAG ID ChiHjej11B10-3189 and which is available via NCBI BioSample SAMN15816658. The GC content of the type genome is 44.93% and the genome length is 2.8 Mbp.	
Description of Candidatus Parabacteroides intestinigallinarum sp. nov.	
Candidatus Parabacteroides intestinigallinarum (in.tes.ti.ni.gal.li.na’rum. L. neut. n. intestinum gut; L. fem. n. gallina hen; N.L. gen. n. intestinigallinarum of the gut of the hens)	
A bacterial species identified by metagenomic analyses. This species includes all bacteria with genomes that show ≥95% average nucleotide identity (ANI) to the type genome, which has been assigned the MAG ID ChiHecec2B26-12326 and which is available via NCBI BioSample SAMN15816728. The GC content of the type genome is 52.90% and the genome length is 2.9 Mbp.	
Description of Candidatus Parabacteroides intestinipullorum sp. nov.	
Candidatus Parabacteroides intestinipullorum (in.tes.ti.ni.pul.lo’rum. L. neut. n. intestinum gut; L. masc. n. pullus a young chicken; N.L. gen. n. intestinipullorum of the gut of young chickens)	
A bacterial species identified by metagenomic analyses. This species includes all bacteria with genomes that show ≥95% average nucleotide identity (ANI) to the type genome, which has been assigned the MAG ID ChiGjej6B6-14162 and which is available via NCBI BioSample SAMN15816857. This is a new name for the alphanumeric GTDB species sp900552415. The GC content of the type genome is 50.53% and the genome length is 3.2 Mbp.	

Description of Candidatus Paralactobacillus gallistercoris sp. nov.	
Candidatus Paralactobacillus gallistercoris (gal.li.ster’co.ris. L. masc. n gallus chicken; L. neut. n. stercus dung; N.L. gen. n. gallistercoris of chicken faeces)	
A bacterial species identified by metagenomic analyses. This species includes all bacteria with genomes that show ≥95% average nucleotide identity (ANI) to the type genome, which has been assigned the MAG ID F6-6636 and which is available via NCBI BioSample SAMN15816781. The GC content of the type genome is 35.69% and the genome length is 1.2 Mbp.	
Description of Candidatus Paraprevotella stercoravium sp. nov.	
Candidatus Paraprevotella stercoravium (ster.cor.a’vi.um. L. neut. n. stercus dung; L. fem. n. avis bird; N.L. gen. n. stercoravium of bird faeces)	
A bacterial species identified by metagenomic analyses. This species includes all bacteria with genomes that show ≥95% average nucleotide identity (ANI) to the type genome, which has been assigned the MAG ID G3-2149 and which is available via NCBI BioSample SAMN15816669. The GC content of the type genome is 45.06% and the genome length is 3.2 Mbp.	
Description of Candidatus Paraprevotella stercorigallinarum sp. nov.	
Candidatus Paraprevotella stercorigallinarum (ster.co.ri.gal.li.na’rum. L. neut. n. stercus dung; L. fem. n. gallina hen; N.L. gen. n. stercorigallinarum of hen faeces)	
A bacterial species identified by metagenomic analyses. This species includes all bacteria with genomes that show ≥95% average nucleotide identity (ANI) to the type genome, which has been assigned the MAG ID 11093 and which is available via NCBI BioSample 3SAMN15816852. This is a new name for the alphanumeric GTDB species sp900546665. The GC content of the type genome is 43.79% and the genome length is 2.9 Mbp.	
Description of Candidatus Parasutterella gallistercoris sp. nov.	
Candidatus Parasutterella gallistercoris (gal.li.ster’co.ris. L. masc. n gallus chicken; L. neut. n. stercus dung; N.L. gen. n. gallistercoris of chicken faeces)	
A bacterial species identified by metagenomic analyses. This species includes all bacteria with genomes that show ≥95% average nucleotide identity (ANI) to the type genome, which has been assigned the MAG ID 21611 and which is available via NCBI BioSample SAMN15816870. This is a new name for the alphanumeric GTDB species sp000980495. The GC content of the type genome is 49.58% and the genome length is 1.9 Mbp.	
Description of Candidatus Pelethenecus gen. nov.	
Candidatus Pelethenecus (Pe.leth.en.e’cus. Gr. masc. n. pelethos dung; Gr. masc. enoikos inhabitant; N.L. masc. n. Pelethenecus a microbe associated with faeces)	
A bacterial genus identified by metagenomic analyses. The genus includes all bacteria with genomes that show ≥60% average amino acid identity (AAI) to the type genome from the type species Candidatus Pelethenecus faecipullorum. This is a name for the alphanumeric GTDB genus UMGS268. This genus has been assigned by GTDB-Tk v1.3.0 working on GTDB Release 05-RS95 (Chaumeil et al., 2019; Parks et al., 2020) to the order Acholeplasmatales and to the family Anaeroplasmataceae.	
Description of Candidatus Pelethenecus faecipullorum sp. nov.	
Candidatus Pelethenecus faecipullorum (fae.ci.pul.lo’rum. L. fem. n. faex, faecis excrement; L. masc. n. pullus a young chicken; N.L. gen. n. faecipullorum of young chicken faeces)	
A bacterial species identified by metagenomic analyses. This species includes all bacteria with genomes that show ≥95% average nucleotide identity (ANI) to the type genome, which has been assigned the MAG ID ChiW17-6978 and which is available via NCBI BioSample SAMN15817226. This is a new name for the alphanumeric GTDB species sp900540175. The GC content of the type genome is 39.85% and the genome length is 1.3 Mbp.	
Description of Candidatus Pelethocola gen. nov.	
Candidatus Pelethocola (Pe.le.tho’co.la. Gr. masc. n. pelethos dung; L. suff. -cola inhabitant of; N.L. fem. n. Pelethocola a microbe associated with faeces)	
A bacterial genus identified by metagenomic analyses. The genus includes all bacteria with genomes that show ≥60% average amino acid identity (AAI) to the type genome from the type species Candidatus Pelethocola excrementipullorum. This is a name for the alphanumeric GTDB genus UBA5416. This genus has been assigned by GTDB-Tk v1.3.0 working on GTDB Release 05-RS95 (Chaumeil et al., 2019; Parks et al., 2020) to the order Lachnospirales and to the family Lachnospiraceae.	
Description of Candidatus Pelethocola excrementipullorum sp. nov.	
Candidatus Pelethocola excrementipullorum (ex.cre.men.ti.pul.lo’rum. L. neut. n. excrementum excrement; L. masc. n. pullus a young chicken; N.L. gen. n. excrementipullorum of young chicken excrement)	
A bacterial species identified by metagenomic analyses. This species includes all bacteria with genomes that show ≥95% average nucleotide identity (ANI) to the type genome, which has been assigned the MAG ID CHK160-5124 and which is available via NCBI BioSample SAMN15817143. The GC content of the type genome is 43.72% and the genome length is 3.9 Mbp.	

Description of Candidatus Pelethomonas gen. nov.	
Candidatus Pelethomonas (Pe.le.tho.mo’nas. Gr. masc. n. pelethos dung; L. fem. n. monas a monad; N.L. fem. n. Pelethomonas a microbe associated with faeces)	
A bacterial genus identified by metagenomic analyses. The genus includes all bacteria with genomes that show ≥60% average amino acid identity (AAI) to the type genome from the type species Candidatus Pelethomonas intestinigallinarum. This is a name for the alphanumeric GTDB genus UMGS1872. This genus has been assigned by GTDB-Tk v1.3.0 working on GTDB Release 05-RS95 (Chaumeil et al., 2019; Parks et al., 2020) to the order Oscillospirales and to the family Oscillospiraceae.	
Description of Candidatus Pelethomonas intestinigallinarum sp. nov.	
Candidatus Pelethomonas intestinigallinarum (in.tes.ti.ni.gal.li.na’rum. L. neut. n. intestinum gut; L. fem. n. gallina hen; N.L. gen. n. intestinigallinarum of the gut of the hens)	
A bacterial species identified by metagenomic analyses. This species includes all bacteria with genomes that show ≥95% average nucleotide identity (ANI) to the type genome, which has been assigned the MAG ID ChiSjej2B20-3600 and which is available via NCBI BioSample SAMN15817014. The GC content of the type genome is 63.89% and the genome length is 2.2 Mbp.	
Description of Candidatus Pelethosoma gen. nov.	
Candidatus Pelethosoma (Pe.le.tho.so’ma. Gr. masc. n. pelethos dung; Gr. neut. n. soma a body; N.L. neut. n. Pelethosoma a microbe associated with faeces)	
A bacterial genus identified by metagenomic analyses. The genus includes all bacteria with genomes that show ≥60% average amino acid identity (AAI) to the type genome from the type species Candidatus Pelethosoma merdigallinarum. This is a name for the alphanumeric GTDB genus UMGS2016. This genus has been assigned by GTDB-Tk v1.3.0 working on GTDB Release 05-RS95 (Chaumeil et al., 2019; Parks et al., 2020) to the order RF39 and to the family CAG-822.	
Description of Candidatus Pelethosoma merdigallinarum sp. nov.	
Candidatus Pelethosoma merdigallinarum (mer.di.gal.li.na’rum. L. fem. n. merda faeces; L. fem. n. gallina hen; N.L. gen. n. merdigallinarum of hen faeces)	
A bacterial species identified by metagenomic analyses. This species includes all bacteria with genomes that show ≥95% average nucleotide identity (ANI) to the type genome, which has been assigned the MAG ID CHK195-5794 and which is available via NCBI BioSample SAMN15817039. The GC content of the type genome is 30.57% and the genome length is 1.3 Mbp.	
Description of Candidatus Pelethousia gen. nov.	
Candidatus Pelethousia (Pe.leth.ou’si.a. Gr. masc. n. pelethos dung; Gr. fem. n. ousia an essence; N.L. fem. n. Pelethousia a microbe associated with faeces)	
A bacterial genus identified by metagenomic analyses. The genus includes all bacteria with genomes that show ≥60% average amino acid identity (AAI) to the type genome from the type species Candidatus Pelethousia gallinarum. This is a name for the alphanumeric GTDB genus UBA5394. This genus has been assigned by GTDB-Tk v1.3.0 working on GTDB Release 05-RS95 (Chaumeil et al., 2019; Parks et al., 2020) to the order Christensenellales and to the family CAG-138.	
Description of Candidatus Pelethousia gallinarum sp. nov.	
Candidatus Pelethousia gallinarum (gal.li.na’rum. L. fem. n. gallina a hen; L. fem. gen. pl. n. gallinarum of hens)	
A bacterial species identified by metagenomic analyses. This species includes all bacteria with genomes that show ≥95% average nucleotide identity (ANI) to the type genome, which has been assigned the MAG ID ChiHcec27-1353 and which is available via NCBI BioSample SAMN15817178. This is a new name for the alphanumeric GTDB species sp003150565. The GC content of the type genome is 59.40% and the genome length is 2.0 Mbp.	
Description of Candidatus Phascolarctobacterium stercoravium sp. nov.	
Candidatus Phascolarctobacterium stercoravium (ster.cor.a’vi.um. L. neut. n. stercus dung; L. fem. n. avis bird; N.L. gen. n. stercoravium of bird faeces)	
A bacterial species identified by metagenomic analyses. This species includes all bacteria with genomes that show ≥95% average nucleotide identity (ANI) to the type genome, which has been assigned the MAG ID ChiBcec14-732 and which is available via NCBI BioSample SAMN15816834. This is a new name for the alphanumeric GTDB species sp000436095. The GC content of the type genome is 46.50% and the genome length is 1.7 Mbp.	

Description of Candidatus Phocaeicola caecigallinarum sp. nov.	
Candidatus Phocaeicola caecigallinarum (cae.ci.gal.li.na’rum. L. neut. n. caecum the caecum; L. fem. n. gallina a hen; N.L. gen. n. caecigallinarum of the caecum of hens)	
A bacterial species identified by metagenomic analyses. This species includes all bacteria with genomes that show ≥95% average nucleotide identity (ANI) to the type genome, which has been assigned the MAG ID ChiHjej11B10-3694 and which is available via NCBI BioSample SAMN15816802. This is a new name for the alphanumeric GTDB species sp002161565. The GC content of the type genome is 46.18% and the genome length is 3.2 Mbp.	
Description of Candidatus Phocaeicola excrementigallinarum sp. nov.	
Candidatus Phocaeicola excrementigallinarum (ex.cre.men.ti.gal.li.na’rum. L. neut. n. excrementum excrement; L. fem. n. gallina hen; N.L. gen. n. excrementigallinarum of hen excrement)	
A bacterial species identified by metagenomic analyses. This species includes all bacteria with genomes that show ≥95% average nucleotide identity (ANI) to the type genome, which has been assigned the MAG ID 12279 and which is available via NCBI BioSample SAMN15816632. The GC content of the type genome is 50.57% and the genome length is 2.5 Mbp.	
Description of Candidatus Phocaeicola excrementipullorum sp. nov.	
Candidatus Phocaeicola excrementipullorum (ex.cre.men.ti.pul.lo’rum. L. neut. n. excrementum excrement; L. masc. n. pullus a young chicken; N.L. gen. n. excrementipullorum of young chicken excrement)	
A bacterial species identified by metagenomic analyses. This species includes all bacteria with genomes that show ≥95% average nucleotide identity (ANI) to the type genome, which has been assigned the MAG ID 8470 and which is available via NCBI BioSample SAMN15816808. This is a new name for the alphanumeric GTDB species sp900546095. The GC content of the type genome is 49.09% and the genome length is 3.1 Mbp.	
Description of Candidatus Phocaeicola faecigallinarum sp. nov.	
Candidatus Phocaeicola faecigallinarum (fae.ci.gal.li.na’rum. L. fem. n. faex, faecis excrement; L. fem. n. gallina hen; N.L. gen. n. faecigallinarum of hen faeces)	
A bacterial species identified by metagenomic analyses. This species includes all bacteria with genomes that show ≥95% average nucleotide identity (ANI) to the type genome, which has been assigned the MAG ID G4-2901 and which is available via NCBI BioSample SAMN15816657. The GC content of the type genome is 40.33% and the genome length is 3.3 Mbp.	
Description of Candidatus Phocaeicola faecipullorum sp. nov.	
Candidatus Phocaeicola faecipullorum (fae.ci.pul.lo’rum. L. fem. n. faex, faecis excrement; L. masc. n. pullus a young chicken; N.L. gen. n. faecipullorum of young chicken faeces)	
A bacterial species identified by metagenomic analyses. This species includes all bacteria with genomes that show ≥95% average nucleotide identity (ANI) to the type genome, which has been assigned the MAG ID 17637 and which is available via NCBI BioSample SAMN15816682. The GC content of the type genome is 39.94% and the genome length is 3.9 Mbp.	
Description of Candidatus Phocaeicola gallinarum sp. nov.	
Candidatus Phocaeicola gallinarum (gal.li.na’rum. L. fem. n. gallina a hen; L. gen. pl. n. gallinarum of hens)	
A bacterial species identified by metagenomic analyses. This species includes all bacteria with genomes that show ≥95% average nucleotide identity (ANI) to the type genome, which has been assigned the MAG ID ChiGjej6B6-595 and which is available via NCBI BioSample SAMN15816805. This is a new name for the alphanumeric GTDB species sp900540105. The GC content of the type genome is 45.79% and the genome length is 2.8 Mbp.	
Description of Candidatus Phocaeicola gallistercoris sp. nov.	
Candidatus Phocaeicola gallistercoris (gal.li.ster’co.ris. L. masc. n gallus chicken; L. neut. n. stercus dung; N.L. gen. n. gallistercoris of chicken faeces)	
A bacterial species identified by metagenomic analyses. This species includes all bacteria with genomes that show ≥95% average nucleotide identity (ANI) to the type genome, which has been assigned the MAG ID Gambia9-593 and which is available via NCBI BioSample SAMN15816698. The GC content of the type genome is 38.34% and the genome length is 2.3 Mbp.	
Description of Candidatus Phocaeicola merdavium sp. nov.	
Candidatus Phocaeicola merdavium (merd.a’vi.um. L. fem. n. merda faeces; L. fem. n. avis bird; N.L. gen. n. merdavium of bird faeces)	
A bacterial species identified by metagenomic analyses. This species includes all bacteria with genomes that show ≥95% average nucleotide identity (ANI) to the type genome, which has been assigned the MAG ID CHK136-5299 and which is available via NCBI BioSample SAMN15816804. This is a new name for the alphanumeric GTDB species sp002161765. The GC content of the type genome is 44.54% and the genome length is 2.6 Mbp.	

Description of Candidatus Phocaeicola merdigallinarum sp. nov.	
Candidatus Phocaeicola merdigallinarum (mer.di.gal.li.na’rum. L. fem. n. merda faeces; L. fem. n. gallina hen; N.L. gen. n. merdigallinarum of hen faeces)	
A bacterial species identified by metagenomic analyses. This species includes all bacteria with genomes that show ≥95% average nucleotide identity (ANI) to the type genome, which has been assigned the MAG ID 17689 and which is available via NCBI BioSample SAMN15816829. This is a new name for the alphanumeric GTDB species sp900066455. The GC content of the type genome is 46.19% and the genome length is 3.3 Mbp.	
Description of Candidatus Prevotella avicola sp. nov.	
Candidatus Prevotella avicola (a.vi’co.la. L. fem. n. avis bird; L. suff. -cola inhabitant of; N.L. n. avicola inhabitant of birds)	
A bacterial species identified by metagenomic analyses. This species includes all bacteria with genomes that show ≥95% average nucleotide identity (ANI) to the type genome, which has been assigned the MAG ID ChiHecec3B27-8219 and which is available via NCBI BioSample SAMN15816846. This is a new name for the alphanumeric GTDB species sp000435635. The GC content of the type genome is 51.22% and the genome length is 1.9 Mbp.	
Description of Candidatus Prevotella intestinigallinarum sp. nov.	
Candidatus Prevotella intestinigallinarum (in.tes.ti.ni.gal.li.na’rum. L. neut. n. intestinum gut; L. fem. n. gallina hen; N.L. gen. n. intestinigallinarum of the gut of the hens)	
A bacterial species identified by metagenomic analyses. This species includes all bacteria with genomes that show ≥95% average nucleotide identity (ANI) to the type genome, which has been assigned the MAG ID 146 and which is available via NCBI BioSample SAMN15816872. This is a new name for the alphanumeric GTDB species sp900540415. The GC content of the type genome is 56.37% and the genome length is 2.9 Mbp.	
Description of Candidatus Prevotella stercoripullorum sp. nov.	
Candidatus Prevotella stercoripullorum (ster.co.ri.pul.lo’rum. L. neut. n. stercus dung; L. masc. n. pullus a young chicken; N.L. gen. n. stercoripullorum of the faceces of young chickens)	
A bacterial species identified by metagenomic analyses. This species includes all bacteria with genomes that show ≥95% average nucleotide identity (ANI) to the type genome, which has been assigned the MAG ID USASDec6-549 and which is available via NCBI BioSample SAMN15816866. This is a new name for the alphanumeric GTDB species sp900554045. The GC content of the type genome is 53.35% and the genome length is 2.5 Mbp.	
Description of Candidatus Protoclostridium stercorigallinarum sp. nov.	
Candidatus Protoclostridium stercorigallinarum (ster.co.ri.gal.li.na’rum. L. neut. n. stercus dung; L. fem. n. gallina hen; N.L. gen. n. stercorigallinarum of hen faeces)	
A bacterial species identified by metagenomic analyses. This species includes all bacteria with genomes that show ≥95% average nucleotide identity (ANI) to the type genome, which has been assigned the MAG ID 12435 and which is available via NCBI BioSample SAMN15816772. The GC content of the type genome is 56.70% and the genome length is 1.7 Mbp.	
Description of Candidatus Pseudogracilibacillus intestinigallinarum sp. nov.	
Candidatus Pseudogracilibacillus intestinigallinarum (in.tes.ti.ni.gal.li.na’rum. L. neut. n. intestinum gut; L. fem. n. gallina hen; N.L. gen. n. intestinigallinarum of the gut of the hens)	
A bacterial species identified by metagenomic analyses. This species includes all bacteria with genomes that show ≥95% average nucleotide identity (ANI) to the type genome, which has been assigned the MAG ID CHK169-2315 and which is available via NCBI BioSample SAMN15816775. The GC content of the type genome is 35.03% and the genome length is 2.5 Mbp.	
Description of Candidatus Pseudomonas excrementavium sp. nov.	
Candidatus Pseudomonas excrementavium (ex.cre.ment.a’vi.um. L. neut. n. excrementum excrement; L. fem. n. avis bird; N.L. gen. n. excrementavium of bird excrement)	
A bacterial species identified by metagenomic analyses. This species includes all bacteria with genomes that show ≥95% average nucleotide identity (ANI) to the type genome, which has been assigned the MAG ID CHK174-787 and which is available via NCBI BioSample SAMN15816898. Although GTDB has assigned this species to the genus it calls Pseudomonas_D, this genus designation cannot be incorporated into a well-formed binomial, so in naming this species, we have used the current validly published name for the genus. The GC content of the type genome is 61.84% and the genome length is 3.0 Mbp.	

Description of Candidatus Pullibacteroides gen. nov.	
Candidatus Pullibacteroides (Pul.li.bac.te.ro’i.des. L. masc. n. pullus a young chicken; N.L. masc. n. Bacteroides a genus name; N.L. masc. n. Pullibacteroides a genus related to the genus Bacteroides but distinct from it and found in poultry)	
A bacterial genus identified by metagenomic analyses. The genus includes all bacteria with genomes that show ≥60% average amino acid identity (AAI) to the type genome from the type species Candidatus Pullibacteroides excrementavium. This genus has been assigned by GTDB-Tk v1.3.0 working on GTDB Release 05-RS95 (Chaumeil et al., 2019; Parks et al., 2020) to the order Bacteroidales and to the family P3.	
Description of Candidatus Pullibacteroides excrementavium sp. nov.	
Candidatus Pullibacteroides excrementavium (ex.cre.ment.a’vi.um. L. neut. n. excrementum excrement; L. fem. n. avis bird; N.L. gen. n. excrementavium of bird excrement)	
A bacterial species identified by metagenomic analyses. This species includes all bacteria with genomes that show ≥95% average nucleotide identity (ANI) to the type genome, which has been assigned the MAG ID 2889 and which is available via NCBI BioSample SAMN15816989. The GC content of the type genome is 51.34% and the genome length is 2.4 Mbp.	
Description of Candidatus Pullichristensenella gen. nov.	
Candidatus Pullichristensenella (Pul.li.chris.ten.sen.el’la. L. masc. n. pullus a young chicken; N.L. fem. n. Christensenella a genus name; N.L. fem. n. Pullichristensenella a genus related to the genus Christensenella but distinct from it and found in poultry)	
A bacterial genus identified by metagenomic analyses. The genus includes all bacteria with genomes that show ≥60% average amino acid identity (AAI) to the type genome from the type species Candidatus Pullichristensenella avicola. This genus has been assigned by GTDB-Tk v1.3.0 working on GTDB Release 05-RS95 (Chaumeil et al., 2019; Parks et al., 2020) to the order Christensenellales and to the family CAG-74.	
Description of Candidatus Pullichristensenella avicola sp. nov.	
Candidatus Pullichristensenella avicola (a.vi’co.la. L. fem. n. avis bird; L. suff. -cola inhabitant of; N.L. n. avicola inhabitant of birds)	
A bacterial species identified by metagenomic analyses. This species includes all bacteria with genomes that show ≥95% average nucleotide identity (ANI) to the type genome, which has been assigned the MAG ID 10205 and which is available via NCBI BioSample SAMN15816956. The GC content of the type genome is 63.02% and the genome length is 2.3 Mbp.	
Description of Candidatus Pullichristensenella excrementigallinarum sp. nov.	
Candidatus Pullichristensenella excrementigallinarum (ex.cre.men.ti.gal.li.na’rum. L. neut. n. excrementum excrement; L. fem. n. gallina hen; N.L. gen. n. excrementigallinarum of hen excrement)	
A bacterial species identified by metagenomic analyses. This species includes all bacteria with genomes that show ≥95% average nucleotide identity (ANI) to the type genome, which has been assigned the MAG ID ChiHcec3-11533 and which is available via NCBI BioSample SAMN15816983. The GC content of the type genome is 57.19% and the genome length is 2.2 Mbp.	
Description of Candidatus Pullichristensenella excrementipullorum sp. nov.	
Candidatus Pullichristensenella excrementipullorum (ex.cre.men.ti.pul.lo’rum. L. neut. n. excrementum excrement; L. masc. n. pullus a young chicken; N.L. gen. n. excrementipullorum of young chicken excrement)	
A bacterial species identified by metagenomic analyses. This species includes all bacteria with genomes that show ≥95% average nucleotide identity (ANI) to the type genome, which has been assigned the MAG ID 1279 and which is available via NCBI BioSample SAMN15817001. The GC content of the type genome is 63.33% and the genome length is 2.7 Mbp.	
Description of Candidatus Pullichristensenella stercorigallinarum sp. nov.	
Candidatus Pullichristensenella stercorigallinarum (ster.co.ri.gal.li.na’rum. L. neut. n. stercus dung; L. fem. n. gallina hen; N.L. gen. n. stercorigallinarum of hen faeces)	
A bacterial species identified by metagenomic analyses. This species includes all bacteria with genomes that show ≥95% average nucleotide identity (ANI) to the type genome, which has been assigned the MAG ID ChiSjej6B24-2974 and which is available via NCBI BioSample SAMN15816933. The GC content of the type genome is 60.06% and the genome length is 2.7 Mbp.	
Description of Candidatus Pullichristensenella stercoripullorum sp. nov.	
Candidatus Pullichristensenella stercoripullorum (ster.co.ri.pul.lo’rum. L. neut. n. stercus dung; L. masc. n. pullus a young chicken; N.L. gen. n. stercoripullorum of the faceces of young chickens)	
A bacterial species identified by metagenomic analyses. This species includes all bacteria with genomes that show ≥95% average nucleotide identity (ANI) to the type genome, which has been assigned the MAG ID 5266 and which is available via NCBI BioSample SAMN15816952. The GC content of the type genome is 64.02% and the genome length is 2.3 Mbp.	

Description of Candidatus Pullilachnospira gen. nov.	
Candidatus Pullilachnospira (Pul.li.lach.no.spi’ra. L. masc. n. pullus a young chicken; N.L. fem. n. Lachnospira a genus name; N.L. fem. n. Pullilachnospira a genus related to the genus Lachnospira but distinct from it and found in poultry)	
A bacterial genus identified by metagenomic analyses. The genus includes all bacteria with genomes that show ≥60% average amino acid identity (AAI) to the type genome from the type species Candidatus Pullilachnospira stercoravium. This genus was identified but not named by Glendinning et al. (2020). This genus has been assigned by GTDB-Tk v1.3.0 working on GTDB Release 05-RS95 (Chaumeil et al., 2019; Parks et al., 2020) to the order Lachnospirales and to the family Lachnospiraceae.	
Description of Candidatus Pullilachnospira gallistercoris sp. nov.	
Candidatus Pullilachnospira gallistercoris (gal.li.ster’co.ris. L. masc. n gallus chicken; L. neut. n. stercus dung; N.L. gen. n. gallistercoris of chicken faeces)	
A bacterial species identified by metagenomic analyses. This species includes all bacteria with genomes that show ≥95% average nucleotide identity (ANI) to the type genome, which has been assigned the MAG ID ChiSjej5B23-6657 and which is available via NCBI BioSample SAMN15816936. The GC content of the type genome is 53.36% and the genome length is 2.5 Mbp.	
Description of Candidatus Pullilachnospira intestinigallinarum sp. nov.	
Candidatus Pullilachnospira intestinigallinarum (in.tes.ti.ni.gal.li.na’rum. L. neut. n. intestinum gut; L. fem. n. gallina hen; N.L. gen. n. intestinigallinarum of the gut of the hens)	
A bacterial species identified by metagenomic analyses. This species includes all bacteria with genomes that show ≥95% average nucleotide identity (ANI) to the type genome, which has been assigned the MAG ID CHK192-16996 and which is available via NCBI BioSample SAMN15816938. The GC content of the type genome is 51.71% and the genome length is 2.8 Mbp.	
Description of Candidatus Pullilachnospira stercoravium sp. nov.	
Candidatus Pullilachnospira stercoravium (ster.cor.a’vi.um. L. neut. n. stercus dung; L. fem. n. avis bird; N.L. gen. n. stercoravium of bird faeces)	
A bacterial species identified by metagenomic analyses. This species includes all bacteria with genomes that show ≥95% average nucleotide identity (ANI) to the type genome, which has been assigned the MAG ID ChiBcec2-4451 and which is available via NCBI BioSample SAMN15816944. The GC content of the type genome is 53.14% and the genome length is 2.8 Mbp.	
Description of Candidatus Pygmaiobacter gallistercoris sp. nov.	
Candidatus Pygmaiobacter gallistercoris (gal.li.ster’co.ris. L. masc. n. gallus chicken; L. neut. n. stercus dung; N.L. gen. n. gallistercoris of chicken faeces)	
A bacterial species identified by metagenomic analyses. This species includes all bacteria with genomes that show ≥95% average nucleotide identity (ANI) to the type genome, which has been assigned the MAG ID ChiGjej6B6-17065 and which is available via NCBI BioSample SAMN15816776. The GC content of the type genome is 61.74% and the genome length is 1.6 Mbp.	
Description of Candidatus Rikenella faecigallinarum sp. nov.	
Candidatus Rikenella faecigallinarum (fae.ci.gal.li.na’rum. L. fem. n. faex, faecis excrement; L. fem. n. gallina hen; N.L. gen. n. faecigallinarum of hen faeces)	
A bacterial species identified by metagenomic analyses. This species includes all bacteria with genomes that show ≥95% average nucleotide identity (ANI) to the type genome, which has been assigned the MAG ID ChiBcec15-1070 and which is available via NCBI BioSample SAMN15816768. The GC content of the type genome is 56.15% and the genome length is 1.8 Mbp.	
Description of Candidatus Rothia avicola sp. nov.	
Candidatus Rothia avicola (a.vi’co.la. L. fem. n. avis bird; L. suff. -cola inhabitant of; N.L. n. avicola inhabitant of birds)	
A bacterial species identified by metagenomic analyses. This species includes all bacteria with genomes that show ≥95% average nucleotide identity (ANI) to the type genome, which has been assigned the MAG ID ChiHjej12B11-9195 and which is available via NCBI BioSample SAMN15816701. The GC content of the type genome is 60.05% and the genome length is 2.1 Mbp.	
Description of Candidatus Rothia avistercoris sp. nov.	
Candidatus Rothia avistercoris (a.vi.ster’co.ris. L. fem. n. avis bird; L. neut. n. stercus dung; N.L. gen. n. avistercoris of bird faeces)	
A bacterial species identified by metagenomic analyses. This species includes all bacteria with genomes that show ≥95% average nucleotide identity (ANI) to the type genome, which has been assigned the MAG ID ChiHjej10B9-4811 and which is available via NCBI BioSample SAMN15816788. The GC content of the type genome is 59.79% and the genome length is 2.0 Mbp.	

Description of Candidatus Ruania gallistercoris sp. nov.	
Candidatus Ruania gallistercoris (gal.li.ster’co.ris. L. masc. n. gallus chicken; L. neut. n. stercus dung; N.L. gen. n. gallistercoris of chicken faeces)	
A bacterial species identified by metagenomic analyses. This species includes all bacteria with genomes that show ≥95% average nucleotide identity (ANI) to the type genome, which has been assigned the MAG ID ChiGjej4B4-7305 and which is available via NCBI BioSample SAMN15816695. The GC content of the type genome is 69.62% and the genome length is 4.4 Mbp.	
Description of Candidatus Rubneribacter avistercoris sp. nov.	
Candidatus Rubneribacter avistercoris (a.vi.ster’co.ris. L. fem. n. avis bird; L. neut. n. stercus dung; N.L. gen. n. avistercoris of bird faeces)	
A bacterial species identified by metagenomic analyses. This species includes all bacteria with genomes that show ≥95% average nucleotide identity (ANI) to the type genome, which has been assigned the MAG ID ChiGjej6B6-20359 and which is available via NCBI BioSample SAMN15816703. The GC content of the type genome is 65.17% and the genome length is 3.2 Mbp.	
Description of Candidatus Ruminococcus avistercoris sp. nov.	
Candidatus Ruminococcus avistercoris (a.vi.ster’co.ris. L. fem. n. avis bird; L. neut. n. stercus dung; N.L. gen. n. avistercoris of bird faeces)	
A bacterial species identified by metagenomic analyses. This species includes all bacteria with genomes that show ≥95% average nucleotide identity (ANI) to the type genome, which has been assigned the MAG ID CHK186-6582 and which is available via NCBI BioSample SAMN15816883. Although GTDB has assigned this species to the genus it calls Ruminococcus_G, this genus designation cannot be incorporated into a well-formed binomial, so in naming this species, we have used the current validly published name for the genus. The GC content of the type genome is 50.07% and the genome length is 2.3 Mbp.	
Description of Candidatus Ruminococcus gallistercoris sp. nov.	
Candidatus Ruminococcus gallistercoris (gal.li.ster’co.ris. L. masc. n gallus chicken; L. neut. n. stercus dung; N.L. gen. n. gallistercoris of chicken faeces)	
A bacterial species identified by metagenomic analyses. This species includes all bacteria with genomes that show ≥95% average nucleotide identity (ANI) to the type genome, which has been assigned the MAG ID ChiBcec12-341 and which is available via NCBI BioSample SAMN15816918. This is a new name for the alphanumeric GTDB species sp900552925. Although GTDB has assigned this species to the genus it calls Ruminococcus_H, this genus designation cannot be incorporated into a well-formed binomial, so in naming this species, we have used the current validly published name for the genus. The GC content of the type genome is 62.03% and the genome length is 2.1 Mbp.	
Description of Candidatus Ruminococcus intestinipullorum sp. nov.	
Candidatus Ruminococcus intestinipullorum (in.tes.ti.ni.pul.lo’rum. L. neut. n. intestinum gut; L. masc. n. pullus a young chicken; N.L. gen. n. intestinipullorum of the gut of young chickens)	
A bacterial species identified by metagenomic analyses. This species includes all bacteria with genomes that show ≥95% average nucleotide identity (ANI) to the type genome, which has been assigned the MAG ID 1485 and which is available via NCBI BioSample SAMN15816905. Although GTDB has assigned this species to the genus it calls Ruminococcus_B, this genus designation cannot be incorporated into a well-formed binomial, so in naming this species, we have used the current validly published name for the genus. The GC content of the type genome is 35.77% and the genome length is 2.1 Mbp.	
Description of Candidatus Ruthenibacterium avium sp. nov.	
Candidatus Ruthenibacterium avium (a’vi.um. L. fem. pl. n. avium of birds)	
A bacterial species identified by metagenomic analyses. This species includes all bacteria with genomes that show ≥95% average nucleotide identity (ANI) to the type genome, which has been assigned the MAG ID ChiBcec8-14828 and which is available via NCBI BioSample SAMN15816823. This is a new name for the alphanumeric GTDB species sp002315015. The GC content of the type genome is 51.27% and the genome length is 2.2 Mbp.	
Description of Candidatus Ruthenibacterium merdavium sp. nov.	
Candidatus Ruthenibacterium merdavium (merd.a’vi.um. L. fem. n. merda faeces; L. fem. n. avis bird; N.L. gen. n. merdavium of bird faeces)	
A bacterial species identified by metagenomic analyses. This species includes all bacteria with genomes that show ≥95% average nucleotide identity (ANI) to the type genome, which has been assigned the MAG ID 5933 and which is available via NCBI BioSample SAMN15816578. The GC content of the type genome is 51.15% and the genome length is 2.0 Mbp.	
Description of Candidatus Ruthenibacterium merdigallinarum sp. nov.	
Candidatus Ruthenibacterium merdigallinarum (mer.di.gal.li.na’rum. L. fem. n. merda faeces; L. fem. n. gallina hen; N.L. gen. n. merdigallinarum of hen faeces)	
A bacterial species identified by metagenomic analyses. This species includes all bacteria with genomes that show ≥95% average nucleotide identity (ANI) to the type genome, which has been assigned the MAG ID ChiSjej6B24-7098 and which is available via NCBI BioSample SAMN15816763. The GC content of the type genome is 65.28% and the genome length is 2.2 Mbp.	

Description of Candidatus Ruthenibacterium merdipullorum sp. nov.	
Candidatus Ruthenibacterium merdipullorum (mer.di.pul.lo’rum. L. fem. n. merda faeces; L. masc. n. pullus a young chicken; N.L. gen. n. merdipullorum of the faeces of young chickens)	
A bacterial species identified by metagenomic analyses. This species includes all bacteria with genomes that show ≥95% average nucleotide identity (ANI) to the type genome, which has been assigned the MAG ID ChiSxjej5B17-15602 and which is available via NCBI BioSample SAMN15816878. This is a new name for the alphanumeric GTDB species sp900546885. The GC content of the type genome is 59.69% and the genome length is 2.1 Mbp.	
Description of Candidatus Salinicoccus merdavium sp. nov.	
Candidatus Salinicoccus merdavium (merd.a’vi.um. L. fem. n. merda faeces; L. fem. n. avis bird; N.L. gen. n. merdavium of bird faeces)	
A bacterial species identified by metagenomic analyses. This species includes all bacteria with genomes that show ≥95% average nucleotide identity (ANI) to the type genome, which has been assigned the MAG ID ChiHjej12B11-20095 and which is available via NCBI BioSample SAMN15816874. This is a new name for the alphanumeric GTDB species sp002360325. The GC content of the type genome is 44.20% and the genome length is 1.8 Mbp.	
Description of Candidatus Salinicoccus stercoripullorum sp. nov.	
Candidatus Salinicoccus stercoripullorum (ster.co.ri.pul.lo’rum. L. neut. n. stercus dung; L. masc. n. pullus a young chicken; N.L. gen. n. stercoripullorum of the faceces of young chickens)	
A bacterial species identified by metagenomic analyses. This species includes all bacteria with genomes that show ≥95% average nucleotide identity (ANI) to the type genome, which has been assigned the MAG ID ChiHjej13B12-752 and which is available via NCBI BioSample SAMN15816771. The GC content of the type genome is 48.14% and the genome length is 2.3 Mbp.	
Description of Candidatus Savagella gallinarum sp. nov.	
Candidatus Savagella gallinarum (gal.li.na’rum. L. fem. n. gallina a hen; L. gen. pl. n. gallinarum of hens)	
A bacterial species identified by metagenomic analyses. This species includes all bacteria with genomes that show ≥95% average nucleotide identity (ANI) to the type genome, which has been assigned the MAG ID CHK166-5537 and which is available via NCBI BioSample SAMN15816803. This is a new name for the alphanumeric GTDB species sp001655775. The GC content of the type genome is 26.42% and the genome length is 1.8 Mbp.	
Description of Candidatus Scatavimonas gen. nov.	
Candidatus Scatavimonas (Scat.a.vi.mon’as Gr. neut. n. skor, skatos dung; L. fem. n. avis bird; L. fem. n. monas a monad; N.L. fem. n. N.L. neut. n. Scatavimonas a microbe associated with bird faeces)	
A bacterial genus identified by metagenomic analyses. The genus includes all bacteria with genomes that show ≥60% average amino acid identity (AAI) to the type genome from the type species Candidatus Scatavimonas merdigallinarum. This is a name for the alphanumeric GTDB genus UMGS403. This genus has been assigned by GTDB-Tk v1.3.0 working on GTDB Release 05-RS95 (Chaumeil et al., 2019; Parks et al., 2020) to the order Oscillospirales and to the family Acutalibacteraceae.	
Description of Candidatus Scatavimonas merdigallinarum sp. nov.	
Candidatus Scatavimonas merdigallinarum (mer.di.gal.li.na’rum. L. fem. n. merda faeces; L. fem. n. gallina hen; N.L. gen. n. merdigallinarum of hen faeces)	
A bacterial species identified by metagenomic analyses. This species includes all bacteria with genomes that show ≥95% average nucleotide identity (ANI) to the type genome, which has been assigned the MAG ID ChiSjej1B19-3389 and which is available via NCBI BioSample SAMN15817212. This is a new name for the alphanumeric GTDB species sp900541975. The GC content of the type genome is 46.67% and the genome length is 1.7 Mbp.	
Description of Candidatus Scatenecus gen. nov.	
Candidatus Scatenecus (Scat.en.e’cus. Gr. neut. n. skor, skatos dung; Gr. masc. enoikos inhabitant; N.L. masc. n. Scatenecus a microbe associated with the intestines)	
A bacterial genus identified by metagenomic analyses. The genus includes all bacteria with genomes that show ≥60% average amino acid identity (AAI) to the type genome from the type species Candidatus Scatenecus faecavium. This is a name for the alphanumeric GTDB genus QAMI01. This genus has been assigned by GTDB-Tk v1.3.0 working on GTDB Release 05-RS95 (Chaumeil et al., 2019; Parks et al., 2020) to the order Gastranaerophilales and to the family Gastranaerophilaceae.	

Description of Candidatus Scatenecus faecavium sp. nov.	
Candidatus Scatenecus faecavium (faec.a’vi.um. L. fem. n. faex, faecis excrement; L. fem. n. avis bird; N.L. gen. n. faecavium of bird faeces)	
A bacterial species identified by metagenomic analyses. This species includes all bacteria with genomes that show ≥95% average nucleotide identity (ANI) to the type genome, which has been assigned the MAG ID CHK152-2994 and which is available via NCBI BioSample SAMN15817221. This is a new name for the alphanumeric GTDB species sp900551915. The GC content of the type genome is 37.15% and the genome length is 1.9 Mbp.	
Description of Candidatus Scatocola gen. nov.	
Candidatus Scatocola (Sca.to’co.la. Gr. neut. n. skor, skatos dung; L. suff. -cola inhabitant of; N.L. fem. n. Scatocola a microbe associated with faeces)	
A bacterial genus identified by metagenomic analyses. The genus includes all bacteria with genomes that show ≥60% average amino acid identity (AAI) to the type genome from the type species Candidatus Scatocola faecipullorum. This is a name for the alphanumeric GTDB genus CAG-495. This genus has been assigned by GTDB-Tk v1.3.0 working on GTDB Release 05-RS95 (Chaumeil et al., 2019; Parks et al., 2020) to the order RF32 and to the family CAG-239.	
Description of Candidatus Scatocola faecigallinarum sp. nov.	
Candidatus Scatocola faecigallinarum (fae.ci.gal.li.na’rum. L. fem. n. faex, faecis excrement; L. fem. n. gallina hen; N.L. gen. n. faecigallinarum of hen faeces)	
A bacterial species identified by metagenomic analyses. This species includes all bacteria with genomes that show ≥95% average nucleotide identity (ANI) to the type genome, which has been assigned the MAG ID 2846 and which is available via NCBI BioSample SAMN15817209. This is a new name for the alphanumeric GTDB species sp000436375. The GC content of the type genome is 49.31% and the genome length is 1.7 Mbp.	
Description of Candidatus Scatocola faecipullorum sp. nov.	
Candidatus Scatocola faecipullorum (fae.ci.pul.lo’rum. L. fem. n. faex, faecis excrement; L. masc. n. pullus a young chicken; N.L. gen. n. faecipullorum of young chicken faeces)	
A bacterial species identified by metagenomic analyses. This species includes all bacteria with genomes that show ≥95% average nucleotide identity (ANI) to the type genome, which has been assigned the MAG ID ChiW3-316 and which is available via NCBI BioSample SAMN15817201. This is a new name for the alphanumeric GTDB species sp001917125. The GC content of the type genome is 47.24% and the genome length is 1.7 Mbp.	
Description of Candidatus Scatomonas gen. nov.	
Candidatus Scatomonas (Sca.to.mo’nas. Gr. neut. n. skor, skatos dung; L. fem. n. monas a monad; N.L. fem. n. Scatomonas a microbe associated with the intestines)	
A bacterial genus identified by metagenomic analyses. The genus includes all bacteria with genomes that show ≥60% average amino acid identity (AAI) to the type genome from the type species Candidatus Scatomonas merdigallinarum. This is a name for the alphanumeric GTDB genus OF09-33XD. This genus has been assigned by GTDB-Tk v1.3.0 working on GTDB Release 05-RS95 (Chaumeil et al., 2019; Parks et al., 2020) to the order Lachnospirales and to the family Lachnospiraceae.	
Description of Candidatus Scatomonas merdavium sp. nov.	
Candidatus Scatomonas merdavium (merd.a’vi.um. L. fem. n. merda faeces; L. fem. n. avis bird; N.L. gen. n. merdavium of bird faeces)	
A bacterial species identified by metagenomic analyses. This species includes all bacteria with genomes that show ≥95% average nucleotide identity (ANI) to the type genome, which has been assigned the MAG ID ChiSjej5B23-9500 and which is available via NCBI BioSample SAMN15817033. The GC content of the type genome is 53.14% and the genome length is 2.3 Mbp.	
Description of Candidatus Scatomonas merdigallinarum sp. nov.	
Candidatus Scatomonas merdigallinarum (mer.di.gal.li.na’rum. L. fem. n. merda faeces; L. fem. n. gallina hen; N.L. gen. n. merdigallinarum of hen faeces)	
A bacterial species identified by metagenomic analyses. This species includes all bacteria with genomes that show ≥95% average nucleotide identity (ANI) to the type genome, which has been assigned the MAG ID CHK191-20366 and which is available via NCBI BioSample SAMN15817035. The GC content of the type genome is 53.29% and the genome length is 2.3 Mbp.	
Description of Candidatus Scatomonas pullistercoris sp. nov.	
Candidatus Scatomonas pullistercoris (pul.li.ster’co.ris. L. masc. n. pullus a young chicken; L. neut. n. stercus dung; N.L. gen. n. pullistercoris of young chicken faeces)	
A bacterial species identified by metagenomic analyses. This species includes all bacteria with genomes that show ≥95% average nucleotide identity (ANI) to the type genome, which has been assigned the MAG ID CHK188-20938 and which is available via NCBI BioSample SAMN15817052. The GC content of the type genome is 53.16% and the genome length is 2.3 Mbp.	

Description of Candidatus Scatomorpha gen. nov.	
Candidatus Scatomorpha (Sca.to.mor’pha. Gr. neut. n. skor, skatos dung; Gr. fem. n. morphe a form, shape; N.L. fem. n. Scatomorpha a microbe associated with faeces)	
A bacterial genus identified by metagenomic analyses. The genus includes all bacteria with genomes that show ≥60% average amino acid identity (AAI) to the type genome from the type species Candidatus Scatomorpha merdavium. This is a name for the alphanumeric GTDB genus UBA5446. This genus has been assigned by GTDB-Tk v1.3.0 working on GTDB Release 05-RS95 (Chaumeil et al., 2019; Parks et al., 2020) to the order Oscillospirales and to the family Oscillospiraceae.	
Description of Candidatus Scatomorpha gallistercoris sp. nov.	
Candidatus Scatomorpha gallistercoris (gal.li.ster’co.ris. L. masc. n gallus chicken; L. neut. n. stercus dung; N.L. gen. n. gallistercoris of chicken faeces)	
A bacterial species identified by metagenomic analyses. This species includes all bacteria with genomes that show ≥95% average nucleotide identity (ANI) to the type genome, which has been assigned the MAG ID ChiHjej12B11-5383 and which is available via NCBI BioSample SAMN15817213. This is a new name for the alphanumeric GTDB species sp900544765. The GC content of the type genome is 60.13% and the genome length is 2.4 Mbp.	
Description of Candidatus Scatomorpha intestinavium sp. nov.	
Candidatus Scatomorpha intestinavium (in.tes.tin.a’vi.um. L. neut. n. intestinum gut; L. fem. n. avis bird; N.L. gen. n. intestinavium of the gut of birds)	
A bacterial species identified by metagenomic analyses. This species includes all bacteria with genomes that show ≥95% average nucleotide identity (ANI) to the type genome, which has been assigned the MAG ID ChiBcolR7-354 and which is available via NCBI BioSample SAMN15817058. The GC content of the type genome is 62.47% and the genome length is 2.0 Mbp.	
Description of Candidatus Scatomorpha intestinigallinarum sp. nov.	
Candidatus Scatomorpha intestinigallinarum (in.tes.ti.ni.gal.li.na’rum. L. neut. n. intestinum gut; L. fem. n. gallina hen; N.L. gen. n. intestinigallinarum of the gut of the hens)	
A bacterial species identified by metagenomic analyses. This species includes all bacteria with genomes that show ≥95% average nucleotide identity (ANI) to the type genome, which has been assigned the MAG ID ChiGjej3B3-7149 and which is available via NCBI BioSample SAMN15817216. This is a new name for the alphanumeric GTDB species sp900544295. The GC content of the type genome is 63.01% and the genome length is 2.3 Mbp.	
Description of Candidatus Scatomorpha intestinipullorum sp. nov.	
Candidatus Scatomorpha intestinipullorum (in.tes.ti.ni.pul.lo’rum. L. neut. n. intestinum gut; L. masc. n. pullus a young chicken; N.L. gen. n. intestinipullorum of the gut of young chickens)	
A bacterial species identified by metagenomic analyses. This species includes all bacteria with genomes that show ≥95% average nucleotide identity (ANI) to the type genome, which has been assigned the MAG ID CHK1-1240 and which is available via NCBI BioSample SAMN15817022. The GC content of the type genome is 61.56% and the genome length is 2.5 Mbp.	
Description of Candidatus Scatomorpha merdavium sp. nov.	
Candidatus Scatomorpha merdavium (merd.a’vi.um. L. fem. n. merda faeces; L. fem. n. avis bird; N.L. gen. n. merdavium of bird faeces)	
A bacterial species identified by metagenomic analyses. This species includes all bacteria with genomes that show ≥95% average nucleotide identity (ANI) to the type genome, which has been assigned the MAG ID ChiSxjej3B15-13231 and which is available via NCBI BioSample SAMN15817220. This is a new name for the alphanumeric GTDB species sp004553625. The GC content of the type genome is 61.38% and the genome length is 2.3 Mbp.	
Description of Candidatus Scatomorpha merdigallinarum sp. nov.	
Candidatus Scatomorpha merdigallinarum (mer.di.gal.li.na’rum. L. fem. n. merda faeces; L. fem. n. gallina hen; N.L. gen. n. merdigallinarum of hen faeces)	
A bacterial species identified by metagenomic analyses. This species includes all bacteria with genomes that show ≥95% average nucleotide identity (ANI) to the type genome, which has been assigned the MAG ID CHK187-5235 and which is available via NCBI BioSample SAMN15817028. The GC content of the type genome is 58.23% and the genome length is 2.3 Mbp.	
Description of Candidatus Scatomorpha merdipullorum sp. nov.	
Candidatus Scatomorpha merdipullorum (mer.di.pul.lo’rum. L. fem. n. merda faeces; L. masc. n. pullus a young chicken; N.L. gen. n. merdipullorum of the faeces of young chickens)	
A bacterial species identified by metagenomic analyses. This species includes all bacteria with genomes that show ≥95% average nucleotide identity (ANI) to the type genome, which has been assigned the MAG ID ChiHjej10B9-9673 and which is available via NCBI BioSample SAMN15817099. The GC content of the type genome is 64.45% and the genome length is 1.8 Mbp.	

Description of Candidatus Scatomorpha pullicola sp. nov.	
Candidatus Scatomorpha pullicola (pul.li’co.la. L. masc. n. pullus a young chicken; L. suff. -cola inhabitant of; N.L. n. pullicola an inhabitant of young chickens)	
A bacterial species identified by metagenomic analyses. This species includes all bacteria with genomes that show ≥95% average nucleotide identity (ANI) to the type genome, which has been assigned the MAG ID ChiSjej5B23-7677 and which is available via NCBI BioSample SAMN15817223. This is a new name for the alphanumeric GTDB species sp900543085. The GC content of the type genome is 62.30% and the genome length is 2.1 Mbp.	
Description of Candidatus Scatomorpha pullistercoris sp. nov.	
Candidatus Scatomorpha pullistercoris (pul.li.ster’co.ris. L. masc. n. pullus a young chicken; L. neut. n. stercus dung; N.L. gen. n. pullistercoris of young chicken faeces)	
A bacterial species identified by metagenomic analyses. This species includes all bacteria with genomes that show ≥95% average nucleotide identity (ANI) to the type genome, which has been assigned the MAG ID ChiHecec3B27-6122 and which is available via NCBI BioSample SAMN15817222. This is a new name for the alphanumeric GTDB species sp900546615. The GC content of the type genome is 61.20% and the genome length is 2.2 Mbp.	
Description of Candidatus Scatomorpha stercoravium sp. nov.	
Candidatus Scatomorpha stercoravium (ster.cor.a’vi.um. L. neut. n. stercus dung; L. fem. n. avis bird; N.L. gen. n. stercoravium of bird faeces)	
A bacterial species identified by metagenomic analyses. This species includes all bacteria with genomes that show ≥95% average nucleotide identity (ANI) to the type genome, which has been assigned the MAG ID ChiHecec3B27-8609 and which is available via NCBI BioSample SAMN15817107. The GC content of the type genome is 64.50% and the genome length is 2.0 Mbp.	
Description of Candidatus Scatomorpha stercorigallinarum sp. nov.	
Candidatus Scatomorpha stercorigallinarum (ster.co.ri.gal.li.na’rum. L. neut. n. stercus dung; L. fem. n. gallina hen; N.L. gen. n. stercorigallinarum of hen faeces)	
A bacterial species identified by metagenomic analyses. This species includes all bacteria with genomes that show ≥95% average nucleotide identity (ANI) to the type genome, which has been assigned the MAG ID ChiHjej9B8-2268 and which is available via NCBI BioSample SAMN15817162. The GC content of the type genome is 64.82% and the genome length is 2.0 Mbp.	
Description of Candidatus Scatoplasma gen. nov.	
Candidatus Scatoplasma (Sca.to.plas’ma. Gr. neut. n. skor, skatos dung; Gr. neut. n. plasma a form; N.L. neut. n. Scatoplasma a microbe associated with faeces)	
A bacterial genus identified by metagenomic analyses. The genus includes all bacteria with genomes that show ≥60% average amino acid identity (AAI) to the type genome from the type species Candidatus Scatoplasma merdavium. This is a name for the alphanumeric GTDB genus UBA6879. This genus has been assigned by GTDB-Tk v1.3.0 working on GTDB Release 05-RS95 (Chaumeil et al., 2019; Parks et al., 2020) to the order RFN20 and to the family CAG-288.	
Description of Candidatus Scatoplasma merdavium sp. nov.	
Candidatus Scatoplasma merdavium (merd.a’vi.um. L. fem. n. merda faeces; L. fem. n. avis bird; N.L. gen. n. merdavium of bird faeces)	
A bacterial species identified by metagenomic analyses. This species includes all bacteria with genomes that show ≥95% average nucleotide identity (ANI) to the type genome, which has been assigned the MAG ID 1748 and which is available via NCBI BioSample SAMN15817159. The GC content of the type genome is 36.99% and the genome length is 1.0 Mbp.	
Description of Candidatus Scatosoma gen. nov.	
Candidatus Scatosoma (Sca.to.so’ma. Gr. neut. n. skor, skatos dung; Gr. neut. n. soma a body; N.L. neut. n. Scatosoma a microbe associated with the intestines)	
A bacterial genus identified by metagenomic analyses. The genus includes all bacteria with genomes that show ≥60% average amino acid identity (AAI) to the type genome from the type species Candidatus Scatosoma pullicola. This is a name for the alphanumeric GTDB genus QALS01. This genus has been assigned by GTDB-Tk v1.3.0 working on GTDB Release 05-RS95 (Chaumeil et al., 2019; Parks et al., 2020) to the order Christensenellales and to the family Borkfalkiaceae.	
Description of Candidatus Scatosoma pullicola sp. nov.	
Candidatus Scatosoma pullicola (pul.li’co.la. L. masc. n. pullus a young chicken; L. suff. -cola inhabitant of; N.L. n. pullicola an inhabitant of young chickens)	
A bacterial species identified by metagenomic analyses. This species includes all bacteria with genomes that show ≥95% average nucleotide identity (ANI) to the type genome, which has been assigned the MAG ID CHK183-20193 and which is available via NCBI BioSample SAMN15817013. The GC content of the type genome is 57.19% and the genome length is 2.0 Mbp.	

Description of Candidatus Scatosoma pullistercoris sp. nov.	
Candidatus Scatosoma pullistercoris (pul.li.ster’co.ris. L. masc. n. pullus a young chicken; L. neut. n. stercus dung; N.L. gen. n. pullistercoris of young chicken faeces)	
A bacterial species identified by metagenomic analyses. This species includes all bacteria with genomes that show ≥95% average nucleotide identity (ANI) to the type genome, which has been assigned the MAG ID 11687 and which is available via NCBI BioSample SAMN15817138. The GC content of the type genome is 55.35% and the genome length is 1.5 Mbp.	
Description of Candidatus Scatousia gen. nov.	
Candidatus Scatousia (Scat.ou’si.a. Gr. neut. n. skor, skatos dung; Gr. fem. n. ousia an essence; N.L. fem. n. Scatousia a microbe associated with faeces)	
A bacterial genus identified by metagenomic analyses. The genus includes all bacteria with genomes that show ≥60% average amino acid identity (AAI) to the type genome from the type species Candidatus Scatousia excrementipullorum. This is a name for the alphanumeric GTDB genus CAG-484. This genus has been assigned by GTDB-Tk v1.3.0 working on GTDB Release 05-RS95 (Chaumeil et al., 2019; Parks et al., 2020) to the order Gastranaerophilales and to the family Gastranaerophilaceae.	
Description of Candidatus Scatousia excrementigallinarum sp. nov.	
Candidatus Scatousia excrementigallinarum (ex.cre.men.ti.gal.li.na’rum. L. neut. n. excrementum excrement; L. fem. n. gallina hen; N.L. gen. n. excrementigallinarum of hen excrement)	
A bacterial species identified by metagenomic analyses. This species includes all bacteria with genomes that show ≥95% average nucleotide identity (ANI) to the type genome, which has been assigned the MAG ID 6276 and which is available via NCBI BioSample SAMN15817091. The GC content of the type genome is 36.46% and the genome length is 2.8 Mbp.	
Description of Candidatus Scatousia excrementipullorum sp. nov.	
Candidatus Scatousia excrementipullorum (ex.cre.men.ti.pul.lo’rum. L. neut. n. excrementum excrement; L. masc. n. pullus a young chicken; N.L. gen. n. excrementipullorum of young chicken excrement)	
A bacterial species identified by metagenomic analyses. This species includes all bacteria with genomes that show ≥95% average nucleotide identity (ANI) to the type genome, which has been assigned the MAG ID 10192 and which is available via NCBI BioSample SAMN15817157. The GC content of the type genome is 36.06% and the genome length is 1.8 Mbp.	
Description of Candidatus Scatovicinus gen. nov.	
Candidatus Scatovicinus (Sca.to.vi.ci’nus. Gr. neut. n. skor, skatos dung; L. masc. n. vicinus a neighbour; N.L. masc. n. Scatovicinus a microbe associated with faeces)	
A bacterial genus identified by metagenomic analyses. The genus includes all bacteria with genomes that show ≥60% average amino acid identity (AAI) to the type genome from the type species Candidatus Scatovicinus merdipullorum. This is a name for the alphanumeric GTDB genus UMGS403. This genus has been assigned by GTDB-Tk v1.3.0 working on GTDB Release 05-RS95 (Chaumeil et al., 2019; Parks et al., 2020) to the order Oscillospirales and to the family Acutalibacteraceae.	
Description of Candidatus Scatovicinus merdipullorum sp. nov.	
Candidatus Scatovicinus merdipullorum (mer.di.pul.lo’rum. L. fem. n. merda faeces; L. masc. n. pullus a young chicken; N.L. gen. n. merdipullorum of the faeces of young chickens)	
A bacterial species identified by metagenomic analyses. This species includes all bacteria with genomes that show ≥95% average nucleotide identity (ANI) to the type genome, which has been assigned the MAG ID CHK181-9830 and which is available via NCBI BioSample SAMN15817214. This is a new name for the alphanumeric GTDB species sp900541565. The GC content of the type genome is 46.80% and the genome length is 1.9 Mbp.	
Description of Candidatus Scatovivens gen. nov.	
Candidatus Scatovivens (Sca.to.vi’vens. Gr. neut. n. skor, skatos dung; N.L. pres. part. vivens living; N.L. fem. n. Scatovivens a microbe associated with faeces)	
A bacterial genus identified by metagenomic analyses. The genus includes all bacteria with genomes that show ≥60% average amino acid identity (AAI) to the type genome from the type species Candidatus Scatovivens faecipullorum. This is a name for the alphanumeric GTDB genus UBA7001. This genus has been assigned by GTDB-Tk v1.3.0 working on GTDB Release 05-RS95 (Chaumeil et al., 2019; Parks et al., 2020) to the order TANB77 and to the family CAG-508.	

Description of Candidatus Scatovivens faecipullorum sp. nov.	
Candidatus Scatovivens faecipullorum (fae.ci.pul.lo’rum. L. fem. n. faex, faecis excrement; L. masc. n. pullus a young chicken; N.L. gen. n. faecipullorum of young chicken faeces)	
A bacterial species identified by metagenomic analyses. This species includes all bacteria with genomes that show ≥95% average nucleotide identity (ANI) to the type genome, which has been assigned the MAG ID ChiSxjej5B17-16517 and which is available via NCBI BioSample SAMN15817189. This is a new name for the alphanumeric GTDB species sp900553685. The GC content of the type genome is 25.63% and the genome length is 1.9 Mbp.	
Description of Candidatus Scybalenecus merdavium sp. nov.	
Candidatus Scybalenecus merdavium (merd.a’vi.um. L. fem. n. merda faeces; L. fem. n. avis bird; N.L. gen. n. merdavium of bird faeces)	
A bacterial species identified by metagenomic analyses. This species includes all bacteria with genomes that show ≥95% average nucleotide identity (ANI) to the type genome, which has been assigned the MAG ID CHK176-6737 and which is available via NCBI BioSample SAMN15817203. This is a new name for the alphanumeric GTDB species sp900546735. The GC content of the type genome is 52.55% and the genome length is 1.8 Mbp.	
Description of Candidatus Scybalocola faecavium sp. nov.	
Candidatus Scybalocola faecavium (faec.a’vi.um. L. fem. n. faex, faecis excrement; L. fem. n. avis bird; N.L. gen. n. faecavium of bird faeces)	
A bacterial species identified by metagenomic analyses. This species includes all bacteria with genomes that show ≥95% average nucleotide identity (ANI) to the type genome, which has been assigned the MAG ID CHK196-3395 and which is available via NCBI BioSample SAMN15817004. The GC content of the type genome is 45.55% and the genome length is 3.6 Mbp.	
Description of Candidatus Scybalocola faecigallinarum sp. nov.	
Candidatus Scybalocola faecigallinarum (fae.ci.gal.li.na’rum. L. fem. n. faex, faecis excrement; L. fem. n. gallina hen; N.L. gen. n. faecigallinarum of hen faeces)	
A bacterial species identified by metagenomic analyses. This species includes all bacteria with genomes that show ≥95% average nucleotide identity (ANI) to the type genome, which has been assigned the MAG ID CHK178-757 and which is available via NCBI BioSample SAMN15817027. The GC content of the type genome is 46.78% and the genome length is 3.5 Mbp.	
Description of Candidatus Scybalocola faecipullorum sp. nov.	
Candidatus Scybalocola faecipullorum (fae.ci.pul.lo’rum. L. fem. n. faex, faecis excrement; L. masc. n. pullus a young chicken; N.L. gen. n. faecipullorum of young chicken faeces)	
A bacterial species identified by metagenomic analyses. This species includes all bacteria with genomes that show ≥95% average nucleotide identity (ANI) to the type genome, which has been assigned the MAG ID CHK194-7924 and which is available via NCBI BioSample SAMN15817034. The GC content of the type genome is 44.36% and the genome length is 2.5 Mbp.	
Description of Candidatus Scybalomonas excrementavium sp. nov.	
Candidatus Scybalomonas excrementavium (ex.cre.ment.a’vi.um. L. neut. n. excrementum excrement; L. fem. n. avis bird; N.L. gen. n. excrementavium of bird excrement)	
A bacterial species identified by metagenomic analyses. This species includes all bacteria with genomes that show ≥95% average nucleotide identity (ANI) to the type genome, which has been assigned the MAG ID E3-2379 and which is available via NCBI BioSample SAMN15817094. The GC content of the type genome is 32.78% and the genome length is 2.5 Mbp.	
Description of Candidatus Scybalomonas excrementigallinarum sp. nov.	
Candidatus Scybalomonas excrementigallinarum (ex.cre.men.ti.gal.li.na’rum. L. neut. n. excrementum excrement; L. fem. n. gallina hen; N.L. gen. n. excrementigallinarum of hen excrement)	
A bacterial species identified by metagenomic analyses. This species includes all bacteria with genomes that show ≥95% average nucleotide identity (ANI) to the type genome, which has been assigned the MAG ID 3201 and which is available via NCBI BioSample SAMN15817095. The GC content of the type genome is 34.32% and the genome length is 3.2 Mbp.	
Description of Candidatus Scybalosoma faecavium sp. nov.	
Candidatus Scybalosoma faecavium (faec.a’vi.um. L. fem. n. faex, faecis excrement; L. fem. n. avis bird; N.L. gen. n. faecavium of bird faeces)	
A bacterial species identified by metagenomic analyses. This species includes all bacteria with genomes that show ≥95% average nucleotide identity (ANI) to the type genome, which has been assigned the MAG ID ChiSxjej1B13-2233 and which is available via NCBI BioSample SAMN15817224. This is a new name for the alphanumeric GTDB species sp900545085. The GC content of the type genome is 47.85% and the genome length is 1.6 Mbp.	

Description of Candidatus Scybalousia intestinigallinarum sp. nov.	
Candidatus Scybalousia intestinigallinarum (in.tes.ti.ni.gal.li.na’rum. L. neut. n. intestinum gut; L. fem. n. gallina hen; N.L. gen. n. intestinigallinarum of the gut of the hens)	
A bacterial species identified by metagenomic analyses. This species includes all bacteria with genomes that show ≥95% average nucleotide identity (ANI) to the type genome, which has been assigned the MAG ID CHK193-12526 and which is available via NCBI BioSample SAMN15817037. The GC content of the type genome is 31.54% and the genome length is 1.5 Mbp.	
Description of Candidatus Scybalenecus gen. nov.	
Candidatus Scybalenecus (Scy.bal.en.e’cus. Gr. neut. n. skybalon dung; Gr. masc. enoikos inhabitant; N.L. masc. n. Scybalenecus a microbe associated with faeces)	
A bacterial genus identified by metagenomic analyses. The genus includes all bacteria with genomes that show ≥60% average amino acid identity (AAI) to the type genome from the type species Candidatus Scybalenecus merdavium. This is a name for the alphanumeric GTDB genus UMGS905. This genus has been assigned by GTDB-Tk v1.3.0 working on GTDB Release 05-RS95 (Chaumeil et al., 2019; Parks et al., 2020) to the order Oscillospirales and to the family Acutalibacteraceae.	
Description of Candidatus Scybalocola gen. nov.	
Candidatus Scybalocola (Scy.ba.lo’co.la. Gr. neut. n. skybalon dung; L. suff. -cola inhabitant of; N.L. fem. n. Scybalocola a microbe associated with faeces)	
A bacterial genus identified by metagenomic analyses. The genus includes all bacteria with genomes that show ≥60% average amino acid identity (AAI) to the type genome from the type species Candidatus Scybalocola faecigallinarum. This is a name for the alphanumeric GTDB genus UBA7096. This genus has been assigned by GTDB-Tk v1.3.0 working on GTDB Release 05-RS95 (Chaumeil et al., 2019; Parks et al., 2020) to the order Lachnospirales and to the family Lachnospiraceae.	
Description of Candidatus Scybalomonas gen. nov.	
Candidatus Scybalomonas (Scy.ba.lo.mo’nas. Gr. neut. n. skybalon dung; L. fem. n. monas a monad; N.L. fem. n. Scybalomonas a microbe associated with faeces)	
A bacterial genus identified by metagenomic analyses. The genus includes all bacteria with genomes that show ≥60% average amino acid identity (AAI) to the type genome from the type species Candidatus Scybalomonas excrementigallinarum. This is a name for the alphanumeric GTDB genus UMGS680. This genus has been assigned by GTDB-Tk v1.3.0 working on GTDB Release 05-RS95 (Chaumeil et al., 2019; Parks et al., 2020) to the order Lachnospirales and to the family Lachnospiraceae.	
Description of Candidatus Scybalosoma gen. nov.	
Candidatus Scybalosoma (Scy.ba.lo.so’ma. Gr. neut. n. skybalon dung; Gr. neut. n. soma a body; N.L. neut. n. Scybalosoma a microbe associated with faeces)	
A bacterial genus identified by metagenomic analyses. The genus includes all bacteria with genomes that show ≥60% average amino acid identity (AAI) to the type genome from the type species Candidatus Scybalosoma faecavium. This is a name for the alphanumeric GTDB genus UMGS743. This genus has been assigned by GTDB-Tk v1.3.0 working on GTDB Release 05-RS95 (Chaumeil et al., 2019; Parks et al., 2020) to the order Christensenellales and to the family Christensenellaceae.	
Description of Candidatus Scybalousia gen. nov.	
Candidatus Scybalousia (Scy.bal.ou’si.a. Gr. neut. n. skybalon dung; Gr. fem. n. ousia an essence; N.L. fem. n. Scybalousia a microbe associated with faeces)	
A bacterial genus identified by metagenomic analyses. The genus includes all bacteria with genomes that show ≥60% average amino acid identity (AAI) to the type genome from the type species Candidatus Scybalousia intestinigallinarum. This is a name for the alphanumeric GTDB genus UBA7057. This genus has been assigned by GTDB-Tk v1.3.0 working on GTDB Release 05-RS95 (Chaumeil et al., 2019; Parks et al., 2020) to the order RF39 and to the family CAG-611.	
Description of Candidatus Sellimonas avistercoris sp. nov.	
Candidatus Sellimonas avistercoris (a.vi.ster’co.ris. L. fem. n. avis bird; L. neut. n. stercus dung; N.L. gen. n. avistercoris of bird faeces)	
A bacterial species identified by metagenomic analyses. This species includes all bacteria with genomes that show ≥95% average nucleotide identity (ANI) to the type genome, which has been assigned the MAG ID ChiBcec13-3606 and which is available via NCBI BioSample SAMN15816877. This is a new name for the alphanumeric GTDB species sp002161525. The GC content of the type genome is 46.79% and the genome length is 2.6 Mbp.	
Description of Candidatus Sphingobacterium stercorigallinarum sp. nov.	
Candidatus Sphingobacterium stercorigallinarum (ster.co.ri.gal.li.na’rum. L. neut. n. stercus dung; L. fem. n. gallina hen; N.L. gen. n. stercorigallinarum of hen faeces)	
A bacterial species identified by metagenomic analyses. This species includes all bacteria with genomes that show ≥95% average nucleotide identity (ANI) to the type genome, which has been assigned the MAG ID CHK174-1108 and which is available via NCBI BioSample SAMN15816705. The GC content of the type genome is 44.57% and the genome length is 2.9 Mbp.	

Description of Candidatus Sphingobacterium stercoripullorum sp. nov.	
Candidatus Sphingobacterium stercoripullorum (ster.co.ri.pul.lo’rum. L. neut. n. stercus dung; L. masc. n. pullus a young chicken; N.L. gen. n. stercoripullorum of the faceces of young chickens)	
A bacterial species identified by metagenomic analyses. This species includes all bacteria with genomes that show ≥95% average nucleotide identity (ANI) to the type genome, which has been assigned the MAG ID 1719 and which is available via NCBI BioSample SAMN15816733. The GC content of the type genome is 39.36% and the genome length is 2.7 Mbp.	
Description of Candidatus Sphingomonas excrementigallinarum sp. nov.	
Candidatus Sphingomonas excrementigallinarum (ex.cre.men.ti.gal.li.na’rum. L. neut. n. excrementum excrement; L. fem. n. gallina hen; N.L. gen. n. excrementigallinarum of hen excrement)	
A bacterial species identified by metagenomic analyses. This species includes all bacteria with genomes that show ≥95% average nucleotide identity (ANI) to the type genome, which has been assigned the MAG ID 1562 and which is available via NCBI BioSample SAMN15816779. The GC content of the type genome is 66.49% and the genome length is 3.7 Mbp.	
Description of Candidatus Spyradenecus gen. nov.	
Candidatus Spyradenecus (Spy.rad.en.e’cus. Gr. fem. n. spyras ball of dung; Gr. masc. enoikos inhabitant; N.L. masc. n. Spyradenecus a microbe associated with the intestines)	
A bacterial genus identified by metagenomic analyses. The genus includes all bacteria with genomes that show ≥60% average amino acid identity (AAI) to the type genome from the type species Candidatus Spyradenecus faecavium. This is a name for the alphanumeric GTDB genus W1P29-020. This genus has been assigned by GTDB-Tk v1.3.0 working on GTDB Release 05-RS95 (Chaumeil et al., 2019; Parks et al., 2020) to the order RFP12 and to the family W1P29-020.	
Description of Candidatus Spyradenecus faecavium sp. nov.	
Candidatus Spyradenecus faecavium (faec.a’vi.um. L. fem. n. faex, faecis excrement; L. fem. n. avis bird; N.L. gen. n. faecavium of bird faeces)	
A bacterial species identified by metagenomic analyses. This species includes all bacteria with genomes that show ≥95% average nucleotide identity (ANI) to the type genome, which has been assigned the MAG ID 35461 and which is available via NCBI BioSample SAMN15817160. The GC content of the type genome is 68.44% and the genome length is 1.8 Mbp.	
Description of Candidatus Spyradocola gen. nov.	
Candidatus Spyradocola (Spy.ra.do’co.la. Gr. fem. n. spyras ball of dung; L. suff. -cola inhabitant of; N.L. fem. n. Spyradocola a microbe associated with the intestines)	
A bacterial genus identified by metagenomic analyses. The genus includes all bacteria with genomes that show ≥60% average amino acid identity (AAI) to the type genome from the type species Candidatus Spyradocola merdavium. This is a name for the alphanumeric GTDB genus UBA7102. This genus has been assigned by GTDB-Tk v1.3.0 working on GTDB Release 05-RS95 (Chaumeil et al., 2019; Parks et al., 2020) to the order Christensenellales and to the family UBA1750.	
Description of Candidatus Spyradocola merdavium sp. nov.	
Candidatus Spyradocola merdavium (merd.a’vi.um. L. fem. n. merda faeces; L. fem. n. avis bird; N.L. gen. n. merdavium of bird faeces)	
A bacterial species identified by metagenomic analyses. This species includes all bacteria with genomes that show ≥95% average nucleotide identity (ANI) to the type genome, which has been assigned the MAG ID CHK191-18038 and which is available via NCBI BioSample SAMN15817024. The GC content of the type genome is 66.20% and the genome length is 2.6 Mbp.	
Description of Candidatus Spyradomonas gen. nov.	
Candidatus Spyradomonas (Spy.ra.do.mo’nas. Gr. fem. n. spyras ball of dung; L. fem. n. monas a monad; N.L. fem. n. Spyradomonas a microbe associated with faeces)	
A bacterial genus identified by metagenomic analyses. The genus includes all bacteria with genomes that show ≥60% average amino acid identity (AAI) to the type genome from the type species Candidatus Spyradomonas excrementavium. This is a name for the alphanumeric GTDB genus UMGS951. This genus has been assigned by GTDB-Tk v1.3.0 working on GTDB Release 05-RS95 (Chaumeil et al., 2019; Parks et al., 2020) to the order Gastranaerophilales and to the family Gastranaerophilaceae.	
Description of Candidatus Spyradomonas excrementavium sp. nov.	
Candidatus Spyradomonas excrementavium (ex.cre.ment.a’vi.um. L. neut. n. excrementum excrement; L. fem. n. avis bird; N.L. gen. n. excrementavium of bird excrement)	
A bacterial species identified by metagenomic analyses. This species includes all bacteria with genomes that show ≥95% average nucleotide identity (ANI) to the type genome, which has been assigned the MAG ID CHK149-2741 and which is available via NCBI BioSample SAMN15817232. This is a new name for the alphanumeric GTDB species sp900547155. The GC content of the type genome is 41.54% and the genome length is 2.0 Mbp.	

Description of Candidatus Spyradosoma gen. nov.	
Candidatus Spyradosoma (Spy.ra.do.so’ma. Gr. fem. n. spyras ball of dung; Gr. neut. n. soma a body; N.L. neut. n. Spyradosoma a microbe associated with faeces)	
A bacterial genus identified by metagenomic analyses. The genus includes all bacteria with genomes that show ≥60% average amino acid identity (AAI) to the type genome from the type species Candidatus Spyradosoma merdigallinarum. This is a name for the alphanumeric GTDB genus W0P29-029. This genus has been assigned by GTDB-Tk v1.3.0 working on GTDB Release 05-RS95 (Chaumeil et al., 2019; Parks et al., 2020) to the order Opitutales and to the family UBA953.	
Description of Candidatus Spyradosoma merdigallinarum sp. nov.	
Candidatus Spyradosoma merdigallinarum (mer.di.gal.li.na’rum. L. fem. n. merda faeces; L. fem. n. gallina hen; N.L. gen. n. merdigallinarum of hen faeces)	
A bacterial species identified by metagenomic analyses. This species includes all bacteria with genomes that show ≥95% average nucleotide identity (ANI) to the type genome, which has been assigned the MAG ID 10669 and which is available via NCBI BioSample SAMN15817156. The GC content of the type genome is 62.43% and the genome length is 1.6 Mbp.	
Description of Candidatus Stackebrandtia excrementipullorum sp. nov.	
Candidatus Stackebrandtia excrementipullorum (ex.cre.men.ti.pul.lo’rum. L. neut. n. excrementum excrement; L. masc. n. pullus a young chicken; N.L. gen. n. excrementipullorum of young chicken excrement)	
A bacterial species identified by metagenomic analyses. This species includes all bacteria with genomes that show ≥95% average nucleotide identity (ANI) to the type genome, which has been assigned the MAG ID ChiHjej8B7-33794 and which is available via NCBI BioSample SAMN15816761. The GC content of the type genome is 64.17% and the genome length is 4.1 Mbp.	
Description of Candidatus Stackebrandtia faecavium sp. nov.	
Candidatus Stackebrandtia faecavium (faec.a’vi.um. L. fem. n. faex, faecis excrement; L. fem. n. avis bird; N.L. gen. n. faecavium of bird faeces)	
A bacterial species identified by metagenomic analyses. This species includes all bacteria with genomes that show ≥95% average nucleotide identity (ANI) to the type genome, which has been assigned the MAG ID ChiGjej4B4-770 and which is available via NCBI BioSample SAMN15816782. The GC content of the type genome is 62.84% and the genome length is 3.4 Mbp.	
Description of Candidatus Stercoripulliclostridium gen. nov.	
Candidatus Stercoripulliclostridium (Ster.co.ri.pul.li.clos.tri’di.um. L. neut. n. stercus dung; N.L. masc. n. pullus a young chicken; N.L. neut. n. Clostridium a genus name; N.L. neut. n. Stercoripulliclostridium a genus related to the genus Clostridium but distinct from it and found in poultry faeces)	
A bacterial genus identified by metagenomic analyses. The genus includes all bacteria with genomes that show ≥60% average amino acid identity (AAI) to the type genome from the type species Candidatus Stercoripulliclostridium merdipullorum. This genus has been assigned by GTDB-Tk v1.3.0 working on GTDB Release 05-RS95 (Chaumeil et al., 2019; Parks et al., 2020) to the order Christensenellales and to the family DTU072.	
Description of Candidatus Stercoripulliclostridium merdigallinarum sp. nov.	
Candidatus Stercoripulliclostridium merdigallinarum (mer.di.gal.li.na’rum. L. fem. n. merda faeces; L. fem. n. gallina hen; N.L. gen. n. merdigallinarum of hen faeces)	
A bacterial species identified by metagenomic analyses. This species includes all bacteria with genomes that show ≥95% average nucleotide identity (ANI) to the type genome, which has been assigned the MAG ID 18911 and which is available via NCBI BioSample SAMN15816986. The GC content of the type genome is 48.67% and the genome length is 1.5 Mbp.	
Description of Candidatus Stercoripulliclostridium merdipullorum sp. nov.	
Candidatus Stercoripulliclostridium merdipullorum (mer.di.pul.lo’rum. L. fem. n. merda faeces; L. masc. n. pullus a young chicken; N.L. gen. n. merdipullorum of the faeces of young chickens)	
A bacterial species identified by metagenomic analyses. This species includes all bacteria with genomes that show ≥95% average nucleotide identity (ANI) to the type genome, which has been assigned the MAG ID 23406 and which is available via NCBI BioSample SAMN15816995. The GC content of the type genome is 53.33% and the genome length is 1.7 Mbp.	
Description of Candidatus Stercoripulliclostridium pullicola sp. nov.	
Candidatus Stercoripulliclostridium pullicola (pul.li’co.la. L. masc. n. pullus a young chicken; L. suff. -cola inhabitant of; N.L. n. pullicola an inhabitant of young chickens)	
A bacterial species identified by metagenomic analyses. This species includes all bacteria with genomes that show ≥95% average nucleotide identity (ANI) to the type genome, which has been assigned the MAG ID 517 and which is available via NCBI BioSample SAMN15816997. The GC content of the type genome is 52.41% and the genome length is 1.6 Mbp.	

Description of Candidatus Stercorousia gen. nov.	
Candidatus Stercorousia (Ster.cor.ou’si.a. L. neut. n. stercus dung; Gr. fem. n. ousia an essence; N.L. fem. n. Stercorousia a microbe associated with the intestines)	
A bacterial genus identified by metagenomic analyses. The genus includes all bacteria with genomes that show ≥60% average amino acid identity (AAI) to the type genome from the type species Candidatus Stercorousia faecigallinarum. This is a name for the alphanumeric GTDB genus Zag1. This genus has been assigned by GTDB-Tk v1.3.0 working on GTDB Release 05-RS95 (Chaumeil et al., 2019; Parks et al., 2020) to the order Gastranaerophilales and to the family Gastranaerophilaceae.	
Description of Candidatus Stercorousia faecigallinarum sp. nov.	
Candidatus Stercorousia faecigallinarum (fae.ci.gal.li.na’rum. L. fem. n. faex, faecis excrement; L. fem. n. gallina hen; N.L. gen. n. faecigallinarum of hen faeces)	
A bacterial species identified by metagenomic analyses. This species includes all bacteria with genomes that show ≥95% average nucleotide identity (ANI) to the type genome, which has been assigned the MAG ID CHK154-323 and which is available via NCBI BioSample SAMN15817192. This is a new name for the alphanumeric GTDB species sp000438175. The GC content of the type genome is 34.85% and the genome length is 2.1 Mbp.	
Description of Candidatus Streptococcus faecavium sp. nov.	
Candidatus Streptococcus faecavium (faec.a’vi.um. L. fem. n. faex, faecis excrement; L. fem. n. avis bird; N.L. gen. n. faecavium of bird faeces)	
A bacterial species identified by metagenomic analyses. This species includes all bacteria with genomes that show ≥95% average nucleotide identity (ANI) to the type genome, which has been assigned the MAG ID ChiBcolR9-63 and which is available via NCBI BioSample SAMN15816845. This is a new name for the alphanumeric GTDB species sp002300045. The GC content of the type genome is 40.90% and the genome length is 1.4 Mbp.	
Description of Candidatus Sutterella merdavium sp. nov.	
Candidatus Sutterella merdavium (merd.a’vi.um. L. fem. n. merda faeces; L. fem. n. avis bird; N.L. gen. n. merdavium of bird faeces)	
A bacterial species identified by metagenomic analyses. This species includes all bacteria with genomes that show ≥95% average nucleotide identity (ANI) to the type genome, which has been assigned the MAG ID ChiGjej6B6-11950 and which is available via NCBI BioSample SAMN15816882. This is a new name for the alphanumeric GTDB species sp900543805. The GC content of the type genome is 62.35% and the genome length is 2.1 Mbp.	
Description of Candidatus Tetragenococcus pullicola sp. nov.	
Candidatus Tetragenococcus pullicola (pul.li’co.la. L. masc. n. pullus a young chicken; L. suff. -cola inhabitant of; N.L. n. pullicola an inhabitant of young chickens)	
A bacterial species identified by metagenomic analyses. This species includes all bacteria with genomes that show ≥95% average nucleotide identity (ANI) to the type genome, which has been assigned the MAG ID CHK175-10598 and which is available via NCBI BioSample SAMN15816709. The GC content of the type genome is 36.35% and the genome length is 2.6 Mbp.	
Description of Candidatus Tidjanibacter faecipullorum sp. nov.	
Candidatus Tidjanibacter faecipullorum (fae.ci.pul.lo’rum. L. fem. n. faex, faecis excrement; L. masc. n. pullus a young chicken; N.L. gen. n. faecipullorum of young chicken faeces)	
A bacterial species identified by metagenomic analyses. This species includes all bacteria with genomes that show ≥95% average nucleotide identity (ANI) to the type genome, which has been assigned the MAG ID ChiHjej11B10-19426 and which is available via NCBI BioSample SAMN15816696. The GC content of the type genome is 60.46% and the genome length is 1.8 Mbp.	
Description of Candidatus Tidjanibacter gallistercoris sp. nov.	
Candidatus Tidjanibacter gallistercoris (gal.li.ster’co.ris. L. masc. n. gallus chicken; L. neut. n. stercus dung; N.L. gen. n. gallistercoris of chicken faeces)	
A bacterial species identified by metagenomic analyses. This species includes all bacteria with genomes that show ≥95% average nucleotide identity (ANI) to the type genome, which has been assigned the MAG ID ChiSxjej4B16-7142 and which is available via NCBI BioSample SAMN15816746. The GC content of the type genome is 58.50% and the genome length is 2.0 Mbp.	
Description of Candidatus Treponema excrementipullorum sp. nov.	
Candidatus Treponema excrementipullorum (ex.cre.men.ti.pul.lo’rum. L. neut. n. excrementum excrement; L. masc. n. pullus a young chicken; N.L. gen. n. excrementipullorum of young chicken excrement)	
A bacterial species identified by metagenomic analyses. This species includes all bacteria with genomes that show ≥95% average nucleotide identity (ANI) to the type genome, which has been assigned the MAG ID Gambia15-2214 and which is available via NCBI BioSample SAMN15816896. Although GTDB has assigned this species to the genus it calls Treponema_F, this genus designation cannot be incorporated into a well-formed binomial, so in naming this species, we have used the current validly published name for the genus. The GC content of the type genome is 39.91% and the genome length is 2.3 Mbp.	

Description of Candidatus Treponema faecavium sp. nov.	
Candidatus Treponema faecavium (faec.a’vi.um. L. fem. n. faex, faecis excrement; L. fem. n. avis bird; N.L. gen. n. faecavium of bird faeces)	
A bacterial species identified by metagenomic analyses. This species includes all bacteria with genomes that show ≥95% average nucleotide identity (ANI) to the type genome, which has been assigned the MAG ID USASDec8-330 and which is available via NCBI BioSample SAMN15816910. Although GTDB has assigned this species to the genus it calls Treponema_F, this genus designation cannot be incorporated into a well-formed binomial, so in naming this species, we have used the current validly published name for the genus. The GC content of the type genome is 53.41% and the genome length is 2.2 Mbp.	
Description of Candidatus Ureaplasma intestinipullorum sp. nov.	
Candidatus Ureaplasma intestinipullorum (in.tes.ti.ni.pul.lo’rum. L. neut. n. intestinum gut; L. masc. n. pullus a young chicken; N.L. gen. n. intestinipullorum of the gut of young chickens)	
A bacterial species identified by metagenomic analyses. This species includes all bacteria with genomes that show ≥95% average nucleotide identity (ANI) to the type genome, which has been assigned the MAG ID A5-1222 and which is available via NCBI BioSample SAMN15816777. The GC content of the type genome is 24.43% and the genome length is 0.6 Mbp.	
Description of Candidatus Ventrenecus gen. nov.	
Candidatus Ventrenecus (Ventr.en.e’cus. L. masc. n. venter the belly; Gr. masc. enoikos inhabitant; N.L. masc. n. Ventrenecus a microbe associated with faeces)	
A bacterial genus identified by metagenomic analyses. The genus includes all bacteria with genomes that show ≥60% average amino acid identity (AAI) to the type genome from the type species Candidatus Ventrenecus avicola. This is a name for the alphanumeric GTDB genus UMGS1217. This genus has been assigned by GTDB-Tk v1.3.0 working on GTDB Release 05-RS95 (Chaumeil et al., 2019; Parks et al., 2020) to the order RF39 and to the family CAG-1000.	
Description of Candidatus Ventrenecus avicola sp. nov.	
Candidatus Ventrenecus avicola (a.vi’co.la. L. fem. n. avis bird; L. suff. -cola inhabitant of; N.L. n. avicola inhabitant of birds)	
A bacterial species identified by metagenomic analyses. This species includes all bacteria with genomes that show ≥95% average nucleotide identity (ANI) to the type genome, which has been assigned the MAG ID ChiW22-487 and which is available via NCBI BioSample SAMN15817076. The GC content of the type genome is 31.09% and the genome length is 1.3 Mbp.	
Description of Candidatus Ventrenecus stercoripullorum sp. nov.	
Candidatus Ventrenecus stercoripullorum (ster.co.ri.pul.lo’rum. L. neut. n. stercus dung; L. masc. n. pullus a young chicken; N.L. gen. n. stercoripullorum of the faceces of young chickens)	
A bacterial species identified by metagenomic analyses. This species includes all bacteria with genomes that show ≥95% average nucleotide identity (ANI) to the type genome, which has been assigned the MAG ID CHK197-17881 and which is available via NCBI BioSample SAMN15817010. The GC content of the type genome is 30.90% and the genome length is 1.4 Mbp.	
Description of Candidatus Ventricola gen. nov.	
Candidatus Ventricola (Ven.tri’co.la. L. masc. n. venter the belly; L. suff. -cola inhabitant of; N.L. fem. n. Ventricola a microbe associated with faeces)	
A bacterial genus identified by metagenomic analyses. The genus includes all bacteria with genomes that show ≥60% average amino acid identity (AAI) to the type genome from the type species Candidatus Ventricola intestinavium. This is a name for the alphanumeric GTDB genus SFFH01. This genus has been assigned by GTDB-Tk v1.3.0 working on GTDB Release 05-RS95 (Chaumeil et al., 2019; Parks et al., 2020) to the order Christensenellales and to the family CAG-74.	
Description of Candidatus Ventricola gallistercoris sp. nov.	
Candidatus Ventricola gallistercoris (gal.li.ster’co.ris. L. masc. n gallus chicken; L. neut. n. stercus dung; N.L. gen. n. gallistercoris of chicken faeces)	
A bacterial species identified by metagenomic analyses. This species includes all bacteria with genomes that show ≥95% average nucleotide identity (ANI) to the type genome, which has been assigned the MAG ID ChiHcec16-310 and which is available via NCBI BioSample SAMN15817100. The GC content of the type genome is 60.63% and the genome length is 2.4 Mbp.	
Description of Candidatus Ventricola intestinavium sp. nov.	
Candidatus Ventricola intestinavium (in.tes.tin.a’vi.um. L. neut. n. intestinum gut; L. fem. n. avis bird; N.L. gen. n. intestinavium of the gut of birds)	
A bacterial species identified by metagenomic analyses. This species includes all bacteria with genomes that show ≥95% average nucleotide identity (ANI) to the type genome, which has been assigned the MAG ID 8987 and which is available via NCBI BioSample SAMN15817130. The GC content of the type genome is 59.95% and the genome length is 2.2 Mbp.	
Description of Candidatus Ventrimonas gen. nov.	
Candidatus Ventrimonas (Ven.tri.mo’nas. L. masc. n. venter the belly; L. fem. n. monas a monad; N.L. fem. n. Ventrimonas a microbe associated with faeces)	
A bacterial genus identified by metagenomic analyses. The genus includes all bacteria with genomes that show ≥60% average amino acid identity (AAI) to the type genome from the type species Candidatus Ventrimonas merdavium. This is a name for the alphanumeric GTDB genus UBA9502. This genus has been assigned by GTDB-Tk v1.3.0 working on GTDB Release 05-RS95 (Chaumeil et al., 2019; Parks et al., 2020) to the order Lachnospirales and to the family Lachnospiraceae.	
Description of Candidatus Ventrimonas merdavium sp. nov.	
Candidatus Ventrimonas merdavium (merd.a’vi.um. L. fem. n. merda faeces; L. fem. n. avis bird; N.L. gen. n. merdavium of bird faeces)	
A bacterial species identified by metagenomic analyses. This species includes all bacteria with genomes that show ≥95% average nucleotide identity (ANI) to the type genome, which has been assigned the MAG ID USAMLcec2-739 and which is available via NCBI BioSample SAMN15817118. The GC content of the type genome is 57.43% and the genome length is 3.0 Mbp.	
Description of Candidatus Ventrisoma gen. nov.	
Candidatus Ventrisoma (Ven.tri.so’ma. L. masc. n. venter the belly; Gr. neut. n. soma a body; N.L. neut. n. Ventrisoma a microbe associated with faeces)	
A bacterial genus identified by metagenomic analyses. The genus includes all bacteria with genomes that show ≥60% average amino acid identity (AAI) to the type genome from the type species Candidatus Ventrisoma faecale. This is a name for the alphanumeric GTDB genus UC5-1-2E3. This genus has been assigned by GTDB-Tk v1.3.0 working on GTDB Release 05-RS95 (Chaumeil et al., 2019; Parks et al., 2020) to the order Lachnospirales and to the family Lachnospiraceae.	
Description of Candidatus Ventrisoma faecale sp. nov.	
Candidatus Ventrisoma faecale (fae.ca’le. L. neut. adj. faecale of faeces)	
A bacterial species identified by metagenomic analyses. This species includes all bacteria with genomes that show ≥95% average nucleotide identity (ANI) to the type genome, which has been assigned the MAG ID ChiHcolR19-5415 and which is available via NCBI BioSample SAMN15817173. This is a new name for the alphanumeric GTDB species sp001304875. The GC content of the type genome is 55.91% and the genome length is 2.8 Mbp.	
Description of Candidatus Ventrousia gen. nov.	
Candidatus Ventrousia (Ventr.ou’si.a. L. masc. n. venter the belly; Gr. fem. n. ousia an essence; N.L. fem. n. Ventrousia a microbe associated with faeces)	
A bacterial genus identified by metagenomic analyses. The genus includes all bacteria with genomes that show ≥60% average amino acid identity (AAI) to the type genome from the type species Candidatus Ventrousia excrementavium. This is a name for the alphanumeric GTDB genus SCN-57-10. This genus has been assigned by GTDB-Tk v1.3.0 working on GTDB Release 05-RS95 (Chaumeil et al., 2019; Parks et al., 2020) to the order Oscillospirales and to the family Butyricicoccaceae.	
Description of Candidatus Ventrousia excrementavium sp. nov.	
Candidatus Ventrousia excrementavium (ex.cre.ment.a’vi.um. L. neut. n. excrementum excrement; L. fem. n. avis bird; N.L. gen. n. excrementavium of bird excrement)	
A bacterial species identified by metagenomic analyses. This species includes all bacteria with genomes that show ≥95% average nucleotide identity (ANI) to the type genome, which has been assigned the MAG ID CHK191-8634 and which is available via NCBI BioSample SAMN15817048. The GC content of the type genome is 57.61% and the genome length is 2.1 Mbp.	
Description of Candidatus Yaniella excrementavium sp. nov.	
Candidatus Yaniella excrementavium (ex.cre.ment.a’vi.um. L. neut. n. excrementum excrement; L. fem. n. avis bird; N.L. gen. n. excrementavium of bird excrement)	
A bacterial species identified by metagenomic analyses. This species includes all bacteria with genomes that show ≥95% average nucleotide identity (ANI) to the type genome, which has been assigned the MAG ID ChiHjej13B12-778 and which is available via NCBI BioSample SAMN15816702. The GC content of the type genome is 55.30% and the genome length is 2.5 Mbp.	
Description of Candidatus Yaniella excrementigallinarum sp. nov.	
Candidatus Yaniella excrementigallinarum (ex.cre.men.ti.gal.li.na’rum. L. neut. n. excrementum excrement; L. fem. n. gallina hen; N.L. gen. n. excrementigallinarum of hen excrement)	
A bacterial species identified by metagenomic analyses. This species includes all bacteria with genomes that show ≥95% average nucleotide identity (ANI) to the type genome, which has been assigned the MAG ID 4905 and which is available via NCBI BioSample SAMN15816764. The GC content of the type genome is 53.17% and the genome length is 2.6 Mbp.	

Taxonomic diversity of cultured bacterial isolates

To extend our metagenomics analyses, we applied culture-based methods to six faecal samples that appeared species-rich in Kraken 2 analyses and in so doing obtained 282 isolates from aerobic culture (~80% of isolates) and anaerobic culture (~20% of isolates) (Table S16). All isolates underwent genome sequencing on the Illumina platform and phylogenetic analysis to enable taxonomic assignment. The resulting chicken gut culture collection was found to contain 56 genera, 93 species and 162 strains drawn from five phyla. These included thirty novel species, with all novel species confirmed to originate from a monophyletic group through phylogenetic analysis against all available reference genomes of their respective genus (Fig. S2). Curiously, there was no overlap between the species that we obtained and those reported by Medvecky et al. (2018), suggesting that we are far from exhausting the set of species that can be cultured from this habitat. As with the metagenomic species, all novel or previously unnamed genera and species from cultured isolates were assigned Linnaean binomials (Table 2; Table S17). Species-level ANI clustering of all MAGs and all cultured isolates according to phylum is provided in Fig. S3.

Table 2 Protologues for new taxa cultured from chicken faeces.

Description of Acinetobacter pecorum sp. nov.	
(pe.co’rum. L. gen. pl. n. pecorum of flocks of sheep, birds etc., as this species has been isolated from chickens and sheep)	
The type strain for this bacterial species has been cultured from chicken faeces after overnight incubation on brain-heart infusion agar at 37 °C under aerobic conditions. The species has been assigned to this genus by the software tool GTDB-Tk v1.3.0 applied to the Genome Taxonomy Database (GTDB) Release 05-RS95 (Chaumeil et al., 2019; Parks et al., 2020). Its status as a species within this genus has been confirmed by phylogenetic analysis of all available reference genomes from the genus (Fig. S3).	
The species includes all bacteria with genomes that show ≥95% average nucleotide identity (ANI) to the genome of the type strain, which is available via GenBank accession GCA_014837015.1. The GC content of the type strain is 42.9% and the genome size is 3.2 Mbp. GTDB has given this species the alphanumerical designation sp001647535 and the two other genomes assigned to this species are derived from sheep isolates (RefSeq assembly accessions GCF_001647535.1, GCF_001647575.1) (Gupta et al., 2016). The type strain has been deposited in NCTC and DSMZ, where it can be identified via the original strain reference Sa1BUA6.	
Description of Arthrobacter gallicola sp. nov.	
(gal.li’co.la. L. masc. n. gallus a cock; N.L. suff. -cola an inhabitant of; N.L. masc. or fem. n. gallicola an inhabitant of the chicken)	
The type strain for this bacterial species has been cultured from chicken faeces after overnight incubation on Columbia blood agar at 37 °C under aerobic conditions. The species has been assigned to this genus by the software tool GTDB-Tk v1.3.0 applied to the Genome Taxonomy Database (GTDB) Release 05-RS95 (Chaumeil et al., 2019; Parks et al., 2020). Its status as a species within this genus has been confirmed by phylogenetic analysis of all available reference genomes from the genus (Fig. S3).	
The species includes all bacteria with genomes that show ≥95% average nucleotide identity (ANI) to the genome of the type strain, which is available via GenBank accession GCA_014836775.1. The GC content of the type strain is 65.5% and the genome size is 3.7 Mbp. Although GTDB has assigned this species to the genus it calls Arthrobacter_B, this genus designation cannot be incorporated into a well-formed binomial, so in naming this species, we have used the basonym for the genus. The type strain has been deposited in NCTC and DSMZ, where it can be identified via the original strain reference Sa2CUA1.	
Description of Arthrobacter pullicola sp. nov.	
(pul.li’co.la. L. masc. n. pullus a young chicken; N.L. suff. -cola an inhabitant of; N.L. masc. or fem. n. pullicola an inhabitant of young chickens)	
The type strain for this bacterial species has been cultured from chicken faeces after overnight incubation on brain-heart infusion agar at 37 °C under aerobic conditions. The species has been assigned to this genus by the software tool GTDB-Tk v1.3.0 applied to the Genome Taxonomy Database (GTDB) Release 05-RS95 (Chaumeil et al., 2019; Parks et al., 2020). Its status as a species within this genus has been confirmed by phylogenetic analysis of all available reference genomes from the genus (Fig. S3).	
The species includes all bacteria with genomes that show ≥95% average nucleotide identity (ANI) to the genome of the type strain, which is available via GenBank accession GCA_014836875.1. The GC content of the type strain is 65.7% and the genome size is 3.7 Mbp. Although GTDB has assigned this species to the genus it calls Arthrobacter_B, this genus designation cannot be incorporated into a well-formed binomial, so in naming this species, we have used the basonym for the genus. The type strain has been deposited in NCTC and DSMZ, where it can be identified via the original strain reference Sa2BUA2.	
Description of Bacillus norwichensis sp. nov.	
(nor.wich.en’sis. N.L. masc. adj. norwichensis pertaining to English city of Norwich, where the organism was isolated)	
The type strain for this bacterial species has been cultured from chicken faeces after overnight incubation on Columbia blood agar at 37 °C under aerobic conditions. The species has been assigned to this genus by the software tool GTDB-Tk v1.3.0 applied to the Genome Taxonomy Database (GTDB) Release 05-RS95 (Chaumeil et al., 2019; Parks et al., 2020). Its status as a species within this genus has been confirmed by phylogenetic analysis of all available reference genomes from the genus (Fig. S3).	
The species includes all bacteria with genomes that show ≥95% average nucleotide identity (ANI) to the genome of the type strain, which is available via GenBank accession GCA_014836955.1. The GC content of the type strain is 40.2% and the genome size is 4.7 Mbp. Although GTDB has assigned this species to the genus it calls Bacillus_AM, this genus designation cannot be incorporated into a well-formed binomial, so in naming this species, we have used the basonym for the genus. The type strain has been deposited in NCTC and DSMZ, where it can be identified via the original strain reference Sa1BUA2	
Description of Brevibacterium gallinarum sp. nov.	
(gal.li.na’rum. L. pl. gen. n. gallinarum of hens)	
The type strain for this bacterial species has been cultured from chicken faeces after overnight incubation on brain-heart infusion agar at 37 °C under aerobic conditions. The species has been assigned to this genus by the software tool GTDB-Tk v1.3.0 applied to the Genome Taxonomy Database (GTDB) Release 05-RS95 (Chaumeil et al., 2019; Parks et al., 2020). Its status as a species within this genus has been confirmed by phylogenetic analysis of all available reference genomes from the genus (Fig. S3).	
The species includes all bacteria with genomes that show ≥95% average nucleotide identity (ANI) to the genome of the type strain, which is available via GenBank accession GCA_014836885.1. The GC content of the type strain is 67.0% and the genome size is 3.2 Mbp. The type strain has been deposited in NCTC and DSMZ, where it can be identified via the original strain reference Re57	
Description of Brevundimonas guildfordensis sp. nov.	
(guild.ford.en’sis. N.L. fem. adj. guildfordensis pertaining to English town Guildford, home to the University of Surrey, where the samples were processed)	
The type strain for this bacterial species has been cultured from chicken faeces after overnight incubation on Columbia blood agar at 37 °C under aerobic conditions. The species has been assigned to this genus by the software tool GTDB-Tk v1.3.0 applied to the Genome Taxonomy Database (GTDB) Release 05-RS95 (Chaumeil et al., 2019; Parks et al., 2020). Its status as a species within this genus has been confirmed by phylogenetic analysis of all available reference genomes from the genus (Fig. S3).	
The species includes all bacteria with genomes that show ≥95% average nucleotide identity (ANI) to the genome of the type strain, which is available via GenBank accession GCA_014836405.1. The GC content of the type strain is 67.3% and the genome size is 2.9 Mbp. The type strain has been deposited in NCTC and DSMZ, where it can be identified via the original strain reference Sa3CVA3	
Description of Cellulomonas avistercoris sp. nov.	
(a.vi.ster’co.ris. L. fem. n. avis bird; L. neut. n. stercus dung; N.L. gen. n. avistercoris of bird faeces)	
The type strain for this bacterial species has been cultured from chicken faeces after overnight incubation on Columbia blood agar at 37 °C under aerobic conditions. The species has been assigned to this genus by the software tool GTDB-Tk v1.3.0 applied to the Genome Taxonomy Database (GTDB) Release 05-RS95 (Chaumeil et al., 2019; Parks et al., 2020). Its status as a species within this genus has been confirmed by phylogenetic analysis of all available reference genomes from the genus (Fig. S3).	
The species includes all bacteria with genomes that show ≥95% average nucleotide identity (ANI) to the genome of the type strain, which is available via GenBank accession GCA_014836445.1. The GC content of the type strain is 74.5% and the genome size is 4.2 Mbp. The type strain has been deposited in NCTC and DSMZ, where it can be identified via the original strain reference Sa3CUA2	
Description of Clostridium cibarium sp. nov.	
(ci.ba’ri.um. L. neut. adj. cibarium pertaining to food, as this species has been isolated from chickens and zha-chili)	
The type strain for this bacterial species has been cultured from chicken faeces after overnight incubation on Columbia blood agar at 37 °C under anaerobic conditions. The species has been assigned to this genus by the software tool GTDB-Tk v1.3.0 applied to the Genome Taxonomy Database (GTDB) Release 05-RS95 (Chaumeil et al., 2019; Parks et al., 2020). Its status as a species within this genus has been confirmed by phylogenetic analysis of all available reference genomes from the genus (Fig. S3).	
The species includes all bacteria with genomes that show ≥95% average nucleotide identity (ANI) to the genome of the type strain, which is available via GenBank accession GCA_014836335.1. The GC content of the type strain is 29.8% and the genome size is 4.3 Mbp. GTDB has given this species the alphanumerical designation sp007115085. One other isolate from this species (RefSeq accession GCA_007115085.1) has been cultured from zha-chili, a Chinese fermented food. The type strain has been deposited in NCTC and DSMZ, where it can be identified via the original strain reference Sa3CVN1.	
Description of Clostridium gallinarum sp. nov.	
(gal.li.na’rum. L. pl. gen. n. gallinarum of hens)	
The type strain for this bacterial species has been cultured from chicken faeces after overnight incubation on Columbia blood agar at 37 °C under anaerobic conditions. The species has been assigned to this genus by the software tool GTDB-Tk v1.3.0 applied to the Genome Taxonomy Database (GTDB) Release 05-RS95 (Chaumeil et al., 2019; Parks et al., 2020). Its status as a species within this genus has been confirmed by phylogenetic analysis of all available reference genomes from the genus (Fig. S3).	
The species includes all bacteria with genomes that show ≥95% average nucleotide identity (ANI) to the genome of the type strain, which is available via GenBank accession GCA_014836325.1. The GC content of the type strain is 27.2% and the genome size is 3.4 Mbp. The type strain has been deposited in NCTC and DSMZ, where it can be identified via the original strain reference Sa3CUN1.	
Description of Clostridium faecium sp. nov.	
(fae’ci.um. N.L. gen. pl. n. faecium, of faeces, as this species has been isolated from chicken and human faeces)	
The type strain for this bacterial species has been cultured from chicken faeces after overnight incubation on brain-heart infusion agar at 37 °C under anaerobic conditions. The species has been assigned to this genus by the software tool GTDB-Tk v1.3.0 applied to the Genome Taxonomy Database (GTDB) Release 05-RS95 (Chaumeil et al., 2019; Parks et al., 2020). Its status as a species within this genus has been confirmed by phylogenetic analysis of all available reference genomes from the genus (Fig. S3).	
The species includes all bacteria with genomes that show ≥95% average nucleotide identity (ANI) to the genome of the type strain, which is available via GenBank accession GCA_014836835.1. The GC content of the type strain is 28.7% and the genome size is 3.9 Mbp. Although GTDB has assigned this species to the genus it calls Clostridium_J, this genus designation cannot be incorporated into a well-formed binomial, so in naming this species, we have used the basonym for the genus. GTDB has given this species the alphanumerical designation sp900547625, which includes a gut isolate from a preterm human infant (GenBank accession GCA_900547625.1). The type strain has been deposited in NCTC and DSMZ, where it can be identified via the original strain reference N37	
Description of Comamonas avium sp. nov.	
(a’vi.um. L. gen. pl. n. avium of birds)	
The type strain for this bacterial species has been cultured from chicken faeces after overnight incubation on brain-heart infusion agar at 37 °C under aerobic conditions. The species has been assigned to this genus by the software tool GTDB-Tk v1.3.0 applied to the Genome Taxonomy Database (GTDB) Release 05-RS95 (Chaumeil et al., 2019; Parks et al., 2020). Its status as a species within this genus has been confirmed by phylogenetic analysis of all available reference genomes from the genus (Fig. S3).	
The species includes all bacteria with genomes that show ≥95% average nucleotide identity (ANI) to the genome of the type strain, which is available via GenBank accession GCA_014836675.1. The GC content of the type strain is 57.5% and the genome size is 3.9 Mbp. The type strain has been deposited in NCTC and DSMZ, where it can be identified via the original strain reference Sa2CVA6	
Description of Corynebacterium gallinarum sp. nov.	
(gal.li.na’rum. L. pl. gen. n. gallinarum of hens)	
The type strain for this bacterial species has been cultured from chicken faeces after overnight incubation on YCFA (yeast extract, casitone and fatty acid) agar at 37 °C under aerobic conditions. The species has been assigned to this genus by the software tool GTDB-Tk v1.3.0 applied to the Genome Taxonomy Database (GTDB) Release 05-RS95 (Chaumeil et al., 2019; Parks et al., 2020). Its status as a species within this genus has been confirmed by phylogenetic analysis of all available reference genomes from the genus (Fig. S3).	
The species includes all bacteria with genomes that show ≥95% average nucleotide identity (ANI) to the genome of the type strain, which is available via GenBank accession GCA_014837045.1. The GC content of the type strain is 63.1% and the genome size is 3.1 Mbp. The type strain has been deposited in NCTC and DSMZ, where it can be identified via the original strain reference Sa1YVA5	
Description of Cytobacillus stercorigallinarum sp. nov.	
(ster.co.ri.gal.li.na’rum. L. neut. n. stercus faeces; L. fem. n. gallina a hen; N.L. gen. n. stercorigallinarum of hen faeces)	
The type strain for this bacterial species has been cultured from chicken faeces after overnight incubation on brain-heart infusion agar at 37 °C under aerobic conditions. The species has been assigned to this genus by the software tool GTDB-Tk v1.3.0 applied to the Genome Taxonomy Database (GTDB) Release 05-RS95 (Chaumeil et al., 2019; Parks et al., 2020). Its status as a species within this genus has been confirmed by phylogenetic analysis of all available reference genomes from the genus (Fig. S3).	
The species includes all bacteria with genomes that show ≥95% average nucleotide identity (ANI) to the genome of the type strain, which is available via GenBank accession GCA_014836495.1. The GC content of the type strain is 36.8% and the genome size is 4.4 Mbp. GTDB has assigned this species to the genus it calls Bacillus_AA, which cannot be incorporated into a well-formed binomial. However, according to (Patel & Gupta, 2020), the newly named genus Cytobacillus encompasses other species classified by GTDB within the genus designation Bacillus_AA and therefore Cytobacillus is treated as a synonym of GTDB Bacillus_AA. The type strain has been deposited in NCTC and DSMZ, where it can be identified via the original strain reference Sa5YUA1.	
Description of Escherichia whittamii sp. nov.	
(whit.tam’i.i. N.L. gen. n. whittamii, named in honour of American microbiologist Thomas S. Whittam)	
The type strain for this bacterial species has been cultured from chicken faeces after overnight incubation on Columbia blood agar at 37 °C under anaerobic conditions. The species has been assigned to this genus by the software tool GTDB-Tk v1.3.0 applied to the Genome Taxonomy Database (GTDB) Release 05-RS95 (Chaumeil et al., 2019; Parks et al., 2020). Its status as a species within this genus has been confirmed by phylogenetic analysis of reference genomes from the genus (Fig. 4).	
The species includes all bacteria with genomes that show ≥95% average nucleotide identity (ANI) to the genome of the type strain, which is available via GenBank accession GCA_014836715.1. The GC content of the type strain is 50.6% and the genome size is 4.6 Mbp. GTDB has given this species the alphanumerical designation sp001660175 and has assigned two other cultured isolates to this species, both of which come from birds (RefSeq assembly accessions GCF_001660175.1, GCF_002965485.1) (Clermont et al., 2011; Gangiredla et al., 2018). The type strain has been deposited in NCTC and DSMZ, where it can be identified via the original strain reference Sa2BVA5.	
Description of Fictibacillus norfolkensis sp. nov.	
(nor.folk.en’sis. N.L. masc. adj. norfolkensis pertaining to the English county of Norfolk)	
The type strain for this bacterial species has been cultured from chicken faeces after overnight incubation on YCFA (yeast extract, casitone and fatty acid) agar at 37 °C under aerobic conditions. The species has been assigned to this genus by the software tool GTDB-Tk v1.3.0 applied to the Genome Taxonomy Database (GTDB) Release 05-RS95 (Chaumeil et al., 2019; Parks et al., 2020). Its status as a species within this genus has been confirmed by phylogenetic analysis of all available reference genomes from the genus (Fig. S3).	
The species includes all bacteria with genomes that show ≥95% average nucleotide identity (ANI) to the genome of the type strain, which is available via GenBank accession GCA_014836645.1. The GC content of the type strain is 39.5% and the genome size is 4.0 Mbp. Although GTDB has assigned this species to the genus it calls Fictibacillus_B, this genus designation cannot be incorporated into a well-formed binomial, so in naming this species, we have used the basonym for the genus. The type strain has been deposited in NCTC and DSMZ, where it can be identified via the original strain reference Sa2CUA10	
Description of Kaistella pullorum sp. nov.	
(pul.lo’rum. L. gen. pl. n. pullorum of chickens)	
The type strain for this bacterial species has been cultured from chicken faeces after overnight incubation on Columbia blood agar at 37 °C under aerobic conditions. The species has been assigned to this genus by the software tool GTDB-Tk v1.3.0 applied to the Genome Taxonomy Database (GTDB) Release 05-RS95 (Chaumeil et al., 2019; Parks et al., 2020). Its status as a species within this genus has been confirmed by phylogenetic analysis of all available reference genomes from the genus (Fig. S3).	
The species includes all bacteria with genomes that show ≥95% average nucleotide identity (ANI) to the genome of the type strain, which is available via GenBank accession GCA_014837035.1. The GC content of the type strain is 42.9% and the genome size is 2.6 Mbp. The type strain has been deposited in NCTC and DSMZ, where it can be identified via the original strain reference Sa1CVA4	
Description of Limosilactobacillus avistercoris sp. nov.	
(a.vi.ster’co.ris. L. fem. n. avis bird; L. neut. n. stercus dung; N.L. gen. n. avistercoris of bird faeces)	
The type strain for this bacterial species has been cultured from chicken faeces after overnight incubation on brain-heart infusion agar at 37 °C under anaerobic conditions. The species has been assigned to this genus by the software tool GTDB-Tk v1.3.0 applied to the Genome Taxonomy Database (GTDB) Release 05-RS95 (Chaumeil et al., 2019; Parks et al., 2020). Its status as a species within this genus has been confirmed by phylogenetic analysis of all available reference genomes from the genus (Fig. S3).	
The species includes all bacteria with genomes that show ≥95% average nucleotide identity (ANI) to the genome of the type strain, which is available via GenBank accession GCA_014836425.1. The GC content of the type strain is 39.9% and the genome size is 1.8 Mbp. GTDB has assigned this species to the genus it calls Lactobacillus_H, which cannot be incorporated into a well-formed binomial. However, according to (Zheng et al., 2020) Limosilactobacillus encompasses other species classified by GTDB within the genus designation Lactobacillus_H and therefore Limosilactobacillus is treated as a synonym for GDTB designation Lactobacillus_H. The type strain has been deposited in NCTC and DSMZ, where it can be identified via the original strain reference Sa3CUN2	
Description of Luteimonas colneyensis sp. nov.	
(col.ney.en’sis. N.L. fem. adj. colneyensis pertaining to the English village of Colney, home to the Quadram Institute, where the species was first described)	
The type strain for this bacterial species has been cultured from chicken faeces after overnight incubation on Columbia blood agar at 37 °C under aerobic conditions. The species has been assigned to this genus by the software tool GTDB-Tk v1.3.0 applied to the Genome Taxonomy Database (GTDB) Release 05-RS95 (Chaumeil et al., 2019; Parks et al., 2020). Its status as a species within this genus has been confirmed by phylogenetic analysis of all available reference genomes from the genus Fig. S3).	
The species includes all bacteria with genomes that show ≥95% average nucleotide identity (ANI) to the genome of the type strain, which is available via GenBank accession GCA_014836665.1. The GC content of the type strain is 71.0% and the genome size is 3.0 Mbp. The type strain has been deposited in NCTC and DSMZ, where it can be identified via the original strain reference Sa2BVA3	
Description of Microbacterium commune sp. nov.	
(com.mu’ne. L. neut. adj. commune common, referring to diverse habitats, as this species has been isolated from mosquitos and chicken)	
The type strain for this bacterial species has been cultured from chicken faeces after overnight incubation on Columbia blood agar at 37 °C under aerobic conditions. The species has been assigned to this genus by the software tool GTDB-Tk v1.3.0 applied to the Genome Taxonomy Database (GTDB) Release 05-RS95 (Chaumeil et al., 2019; Parks et al., 2020). Its status as a species within this genus has been confirmed by phylogenetic analysis of all available reference genomes from the genus (Fig. S3).	
The species includes all bacteria with genomes that show ≥95% average nucleotide identity (ANI) to the genome of the type strain, which is available via GenBank accession GCA_014836945.1. The GC content of the type strain is 70.3% and the genome size is 3.3 Mbp. GTDB has given this species the alphanumerical designation sp001878835, which also contains a mosquito isolate (RefSeq accession GCF_001878835.1). The type strain has been deposited in NCTC and DSMZ, where it can be identified via the original strain reference Re1.	
Description of Microbacterium gallinarum sp. nov.	
(gal.li.na’rum. L. pl. gen. n. gallinarum of hens)	
The type strain for this bacterial species has been cultured from chicken faeces after overnight incubation on brain-heart infusion agar at 37 °C under aerobic conditions. The species has been assigned to this genus by the software tool GTDB-Tk v1.3.0 applied to the Genome Taxonomy Database (GTDB) Release 05-RS95 (Chaumeil et al., 2019; Parks et al., 2020). Its status as a species within this genus has been confirmed by phylogenetic analysis of all available reference genomes from the genus (Fig. S3).	
The species includes all bacteria with genomes that show ≥95% average nucleotide identity (ANI) to the genome of the type strain, which is available via GenBank accession GCA_014837165.1. The GC content of the type strain is 69.4% and the genome size is 2.8 Mbp. The type strain has been deposited in NCTC and DSMZ, where it can be identified via the original strain reference Sa1CUA4	
Description of Microbacterium pullorum sp. nov.	
(pul.lo’rum. L. gen. pl. n. pullorum of chickens)	
The type strain for this bacterial species has been cultured from chicken faeces after overnight incubation on Columbia blood agar at 37 °C under aerobic conditions. The species has been assigned to this genus by the software tool GTDB-Tk v1.3.0 applied to the Genome Taxonomy Database (GTDB) Release 05-RS95 (Chaumeil et al., 2019; Parks et al., 2020). Its status as a species within this genus has been confirmed by phylogenetic analysis of all available reference genomes from the genus (Fig. S3).	
The species includes all bacteria with genomes that show ≥95% average nucleotide identity (ANI) to the genome of the type strain, which is available via GenBank accession GCA_014836535.1. The GC content of the type strain is 70.1% and the genome size is 3.1 Mbp. The type strain has been deposited in NCTC and DSMZ, where it can be identified via the original strain reference Sa4CUA7	
Description of Oceanitalea stevensii sp. nov.	
(ste.ven’si.i. N.L. gen. n. stevensii, named in honour of British microbiologist Mark Stevens)	
The type strain for this bacterial species has been cultured from chicken faeces after overnight incubation on brain-heart infusion agar at 37 °C under aerobic conditions. The species has been assigned to this genus by the software tool GTDB-Tk v1.3.0 applied to the Genome Taxonomy Database (GTDB) Release 05-RS95 (Chaumeil et al., 2019; Parks et al., 2020). Its status as a species within this genus has been confirmed by phylogenetic analysis of all available reference genomes from the genus (Fig. S3).	
The species includes all bacteria with genomes that show ≥95% average nucleotide identity (ANI) to the genome of the type strain, which is available via GenBank accession GCA_014837105.1. The GC content of the type strain is 73.4% and the genome size is 3.5 Mbp. The type strain has been deposited in NCTC and DSMZ, where it can be identified via the original strain reference Sa1BUA1.	
Description of Ochrobactrum gallinarum sp. nov.	
(gal.li.na’rum. L. pl. gen. n. gallinarum of hens)	
The type strain for this bacterial species has been cultured from chicken faeces after overnight incubation on brain-heart infusion at 37 °C under aerobic conditions. The species has been assigned to this genus by the software tool GTDB-Tk v1.3.0 applied to the Genome Taxonomy Database (GTDB) Release 05-RS95 (Chaumeil et al., 2019; Parks et al., 2020). Its status as a species within this genus has been confirmed by phylogenetic analysis of all available reference genomes from the genus (Fig. S3).	
The species includes all bacteria with genomes that show ≥95% average nucleotide identity (ANI) to the genome of the type strain, which is available via GenBank accession GCA_014836735.1. The GC content of the type strain is 53.5% and the genome size is 5.0 Mbp. The type strain has been deposited in NCTC and DSMZ, where it can be identified via the original strain reference Sa2BUA5	
Description of Oerskovia douganii sp. nov.	
(dou.ga’ni.i. N.L. gen. n. douganii named in honour of British microbiologist Gordon Dougan)	
The type strain for this bacterial species has been cultured from chicken faeces after overnight incubation on brain-heart infusion at 37 °C under aerobic conditions. The species has been assigned to this genus by the software tool GTDB-Tk v1.3.0 applied to the Genome Taxonomy Database (GTDB) Release 05-RS95 (Chaumeil et al., 2019; Parks et al., 2020). Its status as a species within this genus has been confirmed by phylogenetic analysis of all available reference genomes from the genus (Fig. S3).	
The species includes all bacteria with genomes that show ≥95% average nucleotide identity (ANI) to the genome of the type strain, which is available via GenBank accession GCA_015142735.1. The GC content of the type strain is 72.5% and the genome size is 4.3 Mbp. The type strain has been deposited in NCTC and DSMZ, where it can be identified via the original strain reference Sa1BUA8	
Description of Oerskovia gallyi sp. nov.	
(gal’ly.i. N.L. gen. n. gallyi named in honour of British microbiologist David Gally)	
The type strain for this bacterial species has been cultured from chicken faeces after overnight incubation on brain-heart infusion agar at 37 °C under aerobic conditions. The species has been assigned to this genus by the software tool GTDB-Tk v1.3.0 applied to the Genome Taxonomy Database (GTDB) Release 05-RS95 (Chaumeil et al., 2019; Parks et al., 2020). Its status as a species within this genus has been confirmed by phylogenetic analysis of all available reference genomes from the genus (Fig. S3).	
The species includes all bacteria with genomes that show ≥95% average nucleotide identity (ANI) to the genome of the type strain, which is available via GenBank accession GCA_014836745.1. The GC content of the type strain is 72.5% and the genome size is 4.3 Mbp. The type strain has been deposited in NCTC and DSMZ, where it can be identified via the original strain reference Sa2CUA8	
Description of Oerskovia merdavium sp. nov.	
(merd.a’vi.um. L. fem. n. merda faeces; L. fem. n. avis bird; N.L. gen. n. merdavium of bird faeces)	
The type strain for this bacterial species has been cultured from chicken faeces after overnight incubation on Columbia blood agar at 37 °C under aerobic conditions. The species has been assigned to this genus by the software tool GTDB-Tk v1.3.0 applied to the Genome Taxonomy Database (GTDB) Release 05-RS95 (Chaumeil et al., 2019; Parks et al., 2020). Its status as a species within this genus has been confirmed by phylogenetic analysis of all available reference genomes from the genus (Fig. S3).	
The species includes all bacteria with genomes that show ≥95% average nucleotide identity (ANI) to the genome of the type strain, which is available via GenBank accession GCA_014836755.1. The GC content of the type strain is 72.1% and the genome size is 4.5 Mbp. The type strain has been deposited in NCTC and DSMZ, where it can be identified via the original strain reference Sa2CUA9.	
Description of Oerskovia rustica sp. nov.	
(rus’ti.ca. L fem adj. rustica of the countryside, as isolates have been obtained from soil and chickens)	
The type strain for this bacterial species has been cultured from chicken faeces after overnight incubation on Columbia blood agar at 37 °C under aerobic conditions. The species has been assigned to this genus by the software tool GTDB-Tk v1.3.0 applied to the Genome Taxonomy Database (GTDB) Release 05-RS95 (Chaumeil et al., 2019; Parks et al., 2020). Its status as a species within this genus has been confirmed by phylogenetic analysis of all available reference genomes from the genus (Fig. S3).	
The species includes all bacteria with genomes that show ≥95% average nucleotide identity (ANI) to the genome of the type strain, which is available via GenBank accession GCA_014836555.1. The GC content of the type strain is 72.5% and the genome size is 4.4 Mbp. GTDB has given this species the alphanumerical designation sp005937995, which includes a soil isolate (RefSeq accession GCF_005937995.2). The type strain has been deposited in NCTC and DSMZ, where it can be identified via the original strain reference Sa4CUA1.	
Description of Paenibacillus gallinarum sp. nov.	
(gal.li.na’rum. L. pl. gen. n. gallinarum of hens)	
The type strain for this bacterial species has been cultured from chicken faeces after overnight incubation on Columbia blood agar at 37 °C under aerobic conditions. The species has been assigned to this genus by the software tool GTDB-Tk v1.3.0 applied to the Genome Taxonomy Database (GTDB) Release 05-RS95 (Chaumeil et al., 2019; Parks et al., 2020). Its status as a species within this genus has been confirmed by phylogenetic analysis of all available reference genomes from the genus (Fig. S3).	
The species includes all bacteria with genomes that show ≥95% average nucleotide identity (ANI) to the genome of the type strain, which is available via GenBank accession GCA_014836635.1. The GC content of the type strain is 41.2% and the genome size is 5.4 Mbp. The type strain has been deposited in NCTC and DSMZ, where it can be identified via the original strain reference Sa2BVA9	
Description of Phocaeicola faecium sp. nov.	
(fae’ci.um L. gen. pl. n. faecium, of faeces)	
The type strain for this bacterial species has been cultured from chicken faeces after overnight incubation on YCFA (yeast extract, casitone and fatty acid) agar at 37 °C under anaerobic conditions. The species has been assigned to this genus by the software tool GTDB-Tk v1.3.0 applied to the Genome Taxonomy Database (GTDB) Release 05-RS95 (Chaumeil et al., 2019; Parks et al., 2020). Its status as a species within this genus has been confirmed by phylogenetic analysis of all available reference genomes from the genus (Fig. S3).	
The species includes all bacteria with genomes that show ≥95% average nucleotide identity (ANI) to the genome of the type strain, which is available via GenBank accession GCA_014837055.1. The GC content of the type strain is 45.6% and the genome size is 3.5 Mbp. GTDB has given this species the alphanumerical designation sp900540105, which includes a gut isolate from an infant human (GenBank accession GCA_900540105.1). The type strain has been deposited in NCTC and DSMZ, where it can be identified via the original strain reference Sa1YUN3	
Description of Phocaeicola intestinalis sp. nov.	
(in.tes.ti.na’lis. N.L. masc./fem. adj. intestinalis, pertaining to the intestines)	
The type strain for this bacterial species has been cultured from chicken faeces after overnight incubation on Columbia blood agar at 37 °C under anaerobic conditions. The species has been assigned to this genus by the software tool GTDB-Tk v1.3.0 applied to the Genome Taxonomy Database (GTDB) Release 05-RS95 (Chaumeil et al., 2019; Parks et al., 2020). Its status as a species within this genus has been confirmed by phylogenetic analysis of all available reference genomes from the genus (Fig. S3).	
The species includes all bacteria with genomes that show ≥95% average nucleotide identity (ANI) to the genome of the type strain, which is available via GenBank accession GCA_014837065.1. The GC content of the type strain is 45.7% and the genome size is 4.1 Mbp. GTDB has given this species the alphanumerical designation sp002161565, which includes isolates from the human and chicken guts (GenBank accession numbers GCA_000432695.1, GCA_900540165.1, GCF_002159615.1, GCF_002159755.1, GCF_002160215.1, GCF_002161565.1). The type strain has been deposited in NCTC and DSMZ, where it can be identified via the original strain reference Sa1CVN1.	
Description of Planococcus wigleyi sp. nov.	
(wig’ley.i. N.L. masc. gen. n. wigleyi named in honour of British microbiologist Paul Wigley)	
The type strain for this bacterial species has been cultured from chicken faeces after overnight incubation on brain-heart infusion agar at 37 °C under aerobic conditions. The species has been assigned to this genus by the software tool GTDB-Tk v1.3.0 applied to the Genome Taxonomy Database (GTDB) Release 05-RS95 (Chaumeil et al., 2019; Parks et al., 2020). Its status as a species within this genus has been confirmed by phylogenetic analysis of all available reference genomes from the genus (Fig. S3).	
The species includes all bacteria with genomes that show ≥95% average nucleotide identity (ANI) to the genome of the type strain, which is available via GenBank accession GCA_014836985.1. The GC content of the type strain is 45.0% and the genome size is 3.8 Mbp. Although GTDB has assigned a genus name with an alphabetic suffix Planococcus_A, this genus designation cannot be incorporated into a well-formed binomial, so in naming this species, we have used the basonym for the genus. The type strain has been deposited in NCTC and DSMZ, where it can be identified via the original strain reference Sa1BUA13	
Description of Psychrobacillus faecigallinarum sp. nov.	
(fae.ci.gal.li.na’rum. L. fem. n. faex, faecis faeces; L. fem. n. gallina a hen; N.L. gen. n. faecigallinarum of hen faeces)	
The type strain for this bacterial species has been cultured from chicken faeces after overnight incubation on brain-heart infusion agar at 37 °C under aerobic conditions. The species has been assigned to this genus by the software tool GTDB-Tk v1.3.0 applied to the Genome Taxonomy Database (GTDB) Release 05-RS95 (Chaumeil et al., 2019; Parks et al., 2020). Its status as a species within this genus has been confirmed by phylogenetic analysis of all available reference genomes from the genus (Fig. S3).	
The species includes all bacteria with genomes that show ≥95% average nucleotide identity (ANI) to the genome of the type strain, which is available via GenBank accession GCA_014836595.1. The GC content of the type strain is 36.5% and the genome size is 4.0 Mbp. The type strain has been deposited in NCTC and DSMZ, where it can be identified via the original strain reference Sa2BUA9	
Description of Psychrobacter communis sp. nov.	
(com.mu’nis. L. masc. adj. communis common, referring to diverse habitats from which this species has been isolated, including chickens and soil)	
The type strain for this bacterial species has been cultured from chicken faeces after overnight incubation on xyz medium at 37 °C under aerobic conditions. The species has been assigned to this genus by the software tool GTDB-Tk v1.3.0 applied to the Genome Taxonomy Database (GTDB) Release 05-RS95 (Chaumeil et al., 2019; Parks et al., 2020). Its status as a species within this genus has been confirmed by phylogenetic analysis of all available reference genomes from the genus (Fig. S3).	
The species includes all bacteria with genomes that show ≥95% average nucleotide identity (ANI) to the genome of the type strain, which is available via GenBank accession GCA_014836505.1. The GC content of the type strain is 43.7% and the genome size is 3.0 Mbp. GTDB has given this species the alphanumerical designation sp001652315, which contains six environmental isolates from a variety of sources including soil (GCA_002332465.1, GCA_002439405.1, GCA_003524605.1, GCA_007280595.1, GCF_001652315.1, GCF_002836335.1) The type strain has been deposited in NCTC and DSMZ, where it can be identified via the original strain reference Sa4CVA2	
Description of Serpens gallinarum sp. nov.	
(gal.li.na’rum. L. pl. gen. n. gallinarum of hens)	
The type strain for this bacterial species has been cultured from chicken faeces after overnight incubation on Columbia blood agar at 37 °C under aerobic conditions. The species has been assigned to this genus by the software tool GTDB-Tk v1.3.0 applied to the Genome Taxonomy Database (GTDB) Release 05-RS95 (Chaumeil et al., 2019; Parks et al., 2020). Its status as a species within this genus has been confirmed by phylogenetic analysis of all available reference genomes from the genus (Fig. S3).	
The species includes all bacteria with genomes that show ≥95% average nucleotide identity (ANI) to the genome of the type strain, which is available via GenBank accession GCA_014836765.1. The GC content of the type strain is 61.0% and the genome size is 3.9 Mbp. Although GTDB has assigned a genus name with an alphabetic suffix Pseudomonas_H, which cannot be incorporated into a well-formed binomial. However, GDTB genus Pseudomonas_H includes Pseudomonas flexibilis, where the basonym is Serpens (Hespell, 1977), so we have used this genus name. The type strain has been deposited in NCTC and DSMZ, where it can be identified via the original strain reference Sa2CUA2	
Description of Solibacillus faecavium sp. nov.	
(faec.a’vi.um. L. fem. n. faex, faecis faeces; L. fem. n. avis bird; N.L. gen. n. faecavium of bird faeces)	
The type strain for this bacterial species has been cultured from chicken faeces after overnight incubation on brain-heart infusion agar at 37 °C under aerobic conditions. The species has been assigned to this genus by the software tool GTDB-Tk v1.3.0 applied to the Genome Taxonomy Database (GTDB) Release 05-RS95 (Chaumeil et al., 2019; Parks et al., 2020). Its status as a species within this genus has been confirmed by phylogenetic analysis of all available reference genomes from the genus (Fig. S3).	
The species includes all bacteria with genomes that show ≥95% average nucleotide identity (ANI) to the genome of the type strain, which is available via GenBank accession GCA_014836905.1. The GC content of the type strain is 37.1% and the genome size is 3.8 Mbp. The type strain has been deposited in NCTC and DSMZ, where it can be identified via the original strain reference A46	
Description of Solibacillus merdavium sp. nov.	
(merd.a’vi.um. L. fem. n. merda faeces; L. fem. n. avis bird; N.L. gen. n. merdavium of bird faeces)	
The type strain for this bacterial species has been cultured from chicken faeces after overnight incubation on YCFA (yeast extract, casitone and fatty acid) agar at 37 °C under aerobic conditions. The species has been assigned to this genus by the software tool GTDB-Tk v1.3.0 applied to the Genome Taxonomy Database (GTDB) Release 05-RS95 (Chaumeil et al., 2019; Parks et al., 2020). Its status as a species within this genus has been confirmed by phylogenetic analysis of all available reference genomes from the genus (Fig. S3).	
The species includes all bacteria with genomes that show ≥95% average nucleotide identity (ANI) to the genome of the type strain, which is available via GenBank accession GCA_014836935.1. The GC content of the type strain is 37.0% and the genome size is 3.8 Mbp. The type strain has been deposited in NCTC and DSMZ, where it can be identified via the original strain reference Sa1YVA6	
Description of Sporosarcina gallistercoris sp. nov.	
(gal.li.ster’co.ris. L. masc. n. gallus a cock; L. neut. n. stercus dung; N.L. gen. n. gallistercoris of faeces of a cock)	
The type strain for this bacterial species has been cultured from chicken faeces after overnight incubation on YCFA (yeast extract, casitone and fatty acid) agar at 37 °C under aerobic conditions. The species has been assigned to this genus by the software tool GTDB-Tk v1.3.0 applied to the Genome Taxonomy Database (GTDB) Release 05-RS95 (Chaumeil et al., 2019; Parks et al., 2020). Its status as a species within this genus has been confirmed by phylogenetic analysis of all available reference genomes from the genus (Fig. S3).	
The species includes all bacteria with genomes that show ≥95% average nucleotide identity (ANI) to the genome of the type strain, which is available via GenBank accession GCA_014836415.1. The GC content of the type strain is 44.1% and the genome size is 3.1 Mbp. Although GTDB has assigned a genus name with an alphabetic suffix Sporosarcina_A, this genus designation cannot be incorporated into a well-formed binomial, so in naming this species, we have used the basonym for the genus. The type strain has been deposited in NCTC and DSMZ, where it can be identified via the original strain reference Sa3CUA8	
Description of Sporosarcina quadrami sp. nov.	
(qua.dra’mi. N.L. gen. n. quadrami, of the Quadram Institute, where the species was first cultured.)	
The type strain for this bacterial species has been cultured from chicken faeces after overnight incubation on brain-heart infusion agar at 37 °C under aerobic conditions. The species has been assigned to this genus by the software tool GTDB-Tk v1.3.0 applied to the Genome Taxonomy Database (GTDB) Release 05-RS95 (Chaumeil et al., 2019; Parks et al., 2020). Its status as a species within this genus has been confirmed by phylogenetic analysis of all available reference genomes from the genus (Fig. S3).	
The species includes all bacteria with genomes that show ≥95% average nucleotide identity (ANI) to the genome of the type strain, which is available via GenBank accession GCA_014836615.1. The GC content of the type strain is 41.4% and the genome size is 3.6 Mbp. Although GTDB has assigned a genus name with an alphabetic suffix Sporosarcina_B, this genus designation cannot be incorporated into a well-formed binomial, so in naming this species, we have used the basonym for the genus. The type strain has been deposited in NCTC and DSMZ, where it can be identified via the original strain reference Sa2YVA2	
Description of Stenotrophomonas pennii sp. nov.	
(pen’ni.i. N.L. gen. n. pennii, named in honour of British microbiologist Charles W. Penn)	
The type strain for this bacterial species has been cultured from chicken faeces after overnight incubation on brain-heart infusion agar at 37 °C under aerobic conditions. The species has been assigned to this genus by the software tool GTDB-Tk v1.3.0 applied to the Genome Taxonomy Database (GTDB) Release 05-RS95 (Chaumeil et al., 2019; Parks et al., 2020). Its status as a species within this genus has been confirmed by phylogenetic analysis of all available reference genomes from the genus (Fig. S3).	
The species includes all bacteria with genomes that show ≥95% average nucleotide identity (ANI) to the genome of the type strain, which is available via GenBank accession GCA_014836545.1. The GC content of the type strain is 66.4% and the genome size is 3.9 Mbp. GTDB has given this species the alphanumerical designation sp002836635, which includes four environmental isolates (RefSeq accessions GCF_000834105.1, GCF_002836635.1, GCF_002836645.1, GCF_002836675.1). The type strain has been deposited in NCTC and DSMZ, where it can be identified via the original strain reference Sa5BUN4	
Description of Ureibacillus galli sp. nov.	
(gal’li. L. masc. gen. n. galli of a chicken)	
The type strain for this bacterial species has been cultured from chicken faeces after overnight incubation on brain-heart infusion agar at 37 °C under aerobic conditions. The species has been assigned to this genus by the software tool GTDB-Tk v1.3.0 applied to the Genome Taxonomy Database (GTDB) Release 05-RS95 (Chaumeil et al., 2019; Parks et al., 2020). Its status as a species within this genus has been confirmed by phylogenetic analysis of all available reference genomes from the genus (Fig. S3).	
The species includes all bacteria with genomes that show ≥95% average nucleotide identity (ANI) to the genome of the type strain, which is available via GenBank accession GCA_014836845.1. The GC content of the type strain is 35.2% and the genome size is 3.7 Mbp. The type strain has been deposited in NCTC and DSMZ, where it can be identified via the original strain reference Re31.	
Description of Xanthomonas surreyensis sp. nov.	
(sur.rey.en’sis. N.L. fem. adj. surreyensis pertaining to the English county of Surrey, where the samples were obtained)	
The type strain for this bacterial species has been cultured from chicken faeces after overnight incubation on brain-heart infusion agar at 37 °C under aerobic conditions. The species has been assigned to this genus by the software tool GTDB-Tk v1.3.0 applied to the Genome Taxonomy Database (GTDB) Release 05-RS95 (Chaumeil et al., 2019; Parks et al., 2020). Its status as a species within this genus has been confirmed by phylogenetic analysis of all available reference genomes from the genus (Fig. S3).	
The species includes all bacteria with genomes that show ≥95% average nucleotide identity (ANI) to the genome of the type strain, which is available via GenBank accession GCA_014836395.1. The GC content of the type strain is 68.8% and the genome size is 5.4 Mbp. Although GTDB has assigned a genus name with an alphabetic suffix Xanthomonas_A, this genus designation cannot be incorporated into a well-formed binomial, so in naming this species, we have used the basonym for the genus. The type strain has been deposited in NCTC and DSMZ, where it can be identified via the original strain reference Sa3BUA13.	

Interestingly, alongside ten cultured isolates of the well-characterised species Escherichia coli, we recovered three isolates from Escherichia marmotae (a species recently described in Himalayan marmots (Liu et al., 2015)). As previously reported, the E. marmotae strains cluster closely with the Escherichia Clade V (Liu et al., 2019; Walk, 2015), so all members of this clade should be considered members of this species (Fig. 5; Table S18). Further analysis of the GTDB species designated Escherichia sp001660175 (https://gtdb.ecogenomic.org/searches?s=al&q=sp001660175) confirmed that this species forms a monophyletic lineage that corresponds to Clade II, among the cryptic environmental clades described by Walk et al. (2009), which has subsequently been documented in birds (Clermont et al., 2011). As Clade II is comparable in divergence to the other Escherichia spp. and cryptic clades, we have therefore assigned the Linnaean binomial Escherichia whittamii to designate a new species (Table 2), honouring the outstanding contribution of Thomas S. Whittam to the study of Escherichia spp. (Walk & Feng, 2011).

Figure 5 Phylogenetic tree showing the relationships between Escherichia marmotae, Escherichia whittamii and the other Escherichia species and cryptic clades.

The tree was constructed by RAxML maximum likelihood analysis of a core genome alignment generated using Mugsy. The scale bar indicates the number of substitutions per site represented by the branch length shown. Numbers on branches indicate the percentage bootstrap support out of 100 replicates. Strains sequenced as part of this study are highlighted in red.

We found that only 16 species were common to our cultured isolates and our MGS. Subsequent sequence mapping allowed us to detect a further two cultured species at ≥1× coverage in at least one metagenomic sample (Fig. 6A; Table S19). The genomes from cultured isolates were on average 20% larger than the corresponding MAG sequences retrieved from the same source sample (Table S20), which is in line with the completeness threshold of 80% we adopted in quality assurance of the MAGs. However, when we performed detailed gene content analyses on three abundant species in both cultured and metagenomic datasets—Lactobacillus reuteri (with the synonym Limosilactobacillus reuteri), Escherichia coli (including the synonym Escherichia flexneri) and Enterococcus faecium—we found that >99% of the genes from the core genomes and nearly half of the genes in the accessory genomes of cultured species were represented in at least one MAG. These observations suggest that our high-quality MAGs are sufficiently complete to warrant Candidatus names.

Figure 6 Sequence novelty.

(A) Venn diagram showing shared and unique taxonomic species among three data sources; cultured isolates derived from six chicken faecal samples (Cultured species), metagenomic species identified from a combined dataset of >630 chicken gastrointestinal metagenome samples (Metagenomic species); MAGs also found by Glendinning et al. (2020). (B) Percentage of classified metagenomic reads derived from 50 chicken faecal samples according to a standard Kraken 2 database (Previously) and to a standard Kraken 2 database with the addition of the 2,344 genomic and metagenomic sequences derived from this study (Now).

We analysed our chicken faecal metagenomes with a Kraken 2 database derived from genomes representing our candidate metagenomic and cultured species; this yielded a considerable improvement in the number of reads that can be classified through rapid phylogenetic profiling (Fig. 6B).

Distribution of microbial species

An analysis of the distribution of 820 MGSs across the entire metagenomic dataset revealed marked variation between samples, with not a single species present at ≥1× coverage in all samples and only 39 species present in >90% of samples—although 441 species were present in >50% of samples at ≥1× coverage (Table S21). At ≥1× coverage, co-occurrence of nearly 300 species (n = 295) was identified across all 10 BioProjects (Fig. 7A), with no species identified in all BioProjects at ≥10× coverage (Fig. 7B). Focusing on samples from distinct anatomical or physiological sites (faeces, caecum etc.), we found no species present in all faecal samples at >1× coverage and only two species were found in all caecal samples at >1× coverage: both of them newly named in this study: Candidatus Paraprevotella stercoravium and Candidatus Blautia pullistercoris (the latter identified but not named by Glendinning et al. (2020)). These findings rule out the concept of a core chicken gut microbiome. Studies on the human gut microbiome provide a useful comparison in that, in a recent study, only 14 genera were found to be shared across 95% of samples from the human gut (Falony et al., 2016).

Figure 7 UpSet plots depicting presence of 820 metagenomic species across all BioProjects included within this study.

(A) 1× coverage b. (B) 10× coverage. Bars are stacked according to taxonomic species novelty, with black-stacked bars depicting novel species and grey depicting species previously described in public databases or published studies. Only intersections with five or more species are shown.

Among the species with high coverage, frequency is clearly linked to Bioproject. Although species quantification curves showed that the number of species identified increased rapidly with the number of samples, species discovery appeared to plateau at approximately 230 species after including only 50 metagenomes (Fig. S2A). Only two species appeared to be restricted (at ≥1× coverage) to just a single sample: Aliarcobacter thereius and Candidatus Avibacteroides faecavium. Correlation clustering confirmed structure in the data linked to BioProject (Fig. S2B)—for example, the BioProject from the study by Glendinning et al. (2020) clearly shows enhancement of clostridial species compared to other BioProjects, which reflects the fact that samples in that study were sourced from chicks with no post-hatching contact with an adult bird. However, the BioSamples do not appear to cluster by country (Fig. S2C) and show only limited clustering by anatomical/physiological sample site (Fig. S2D). Unfortunately, there is insufficient metadata for other potentially important factors, such as breed, age or diet to draw conclusions on how these might influence clustering.

Discussion

Given the dominance of chickens in the planetary biomass, the chicken gut microbiome ranks as one of the most abundant microbial communities on the planet. Here, we have exploited two complementary approaches—metagenomics and culture—to create an extensive catalogue of genes, genomes and isolates from this important ecosystem. Our work illustrates the value of combining culture-dependent and culture-independent approaches in analysing microbiomes.

We have clearly demonstrated the advantages of shotgun metagenomic sequencing when applied to the chicken gut microbiome, providing catalogues of genes and genome sequences that takes us well beyond what can be achieved using 16S ribosomal RNA gene sequences. Similarly, the current study is much wider in scope than the previous study by Glendinning et al. (2020), not just including analyses of viral genomes and cultured isolates, while also incorporating MAGs built from data not just from that study but from all publicly available metagenomic datasets. Furthermore, the limited overlap between bacterial species represented among our cultured isolates and in our MGS reinforces the utility of the combined approach. Nonetheless, the substantial co-linearity between genomes obtained by the two approaches—and with those from another similar metagenomic study (Glendinning et al., 2020)—confirms the reliability of our binning approaches.

We were surprised to find such a remarkable phylogenetic diversity within this commonplace livestock ecosystem—diversity that rivals that associated with the human gut. Our work has more than doubled the number of bacterial species known to reside in the chicken gut and has resulted in the creation of an unprecedented number of new Candidatus species. By including well-formed Latin binomials with the genomes we have uploaded into public repositories, we have ensured that the new proposed names and associated sequences will be integrated into commonly used online taxonomies and databases and will provide a stable taxonomic nomenclature for future studies. In addition, we have provided proof-of-principle for a scalable approach to Linnaean nomenclature that could be applied to species recovered from other metagenomic assembly projects.

Given that we did not recover by culture some of the organisms that appear most abundant by metagenomics, there is clearly scope for additional culture-based investigations, using a wider range of cultural conditions—perhaps drawing on the precedent of the Human Microbiome Project to create and target a list of the ‘most-wanted-for-culture’ organisms documented by metagenomics (Fodor et al., 2012). The fact that novel metagenomic species are still being recovered from human gut datasets that include tens of thousands of metagenomes (Almeida et al., 2019)—twinned with the promise of novel long-read and proximity-capture approaches to metagenome analyses (Stewart et al., 2018)—make it clear that our attempts here to analyse all currently available chicken gut metagenomes provide far from the last word on microbial diversity in this abundant and important ecosystem. Nonetheless, the availability of so many novel genes, genome and species represents a substantial step forward.

Conclusions

The extensive catalogue of genes, genomes and isolates we have created here substantially improves the coverage of the chicken gut microbiome in the public databases and will make it possible to profile sequences from the chicken gut much more rapidly, easily and comprehensively, providing a valuable resource that lays the ground-work for future comparative and intervention studies. This study also sets a provocative precedent—relevant not just to animal microbiomes, but to studies on all microbiomes—assigning well-formed Latin binomials to hundreds of metagenomic species in a scalable alternative to the automated use of bland, unstable, user-unfriendly alphanumerical designations. Drawing on the precedent set by the current study, we have recently extended this approach to encompass creation of more than a million new names for Bacteria and Archaea (Pallen, Telatin & Oren, 2020). Thus, the time is now ripe to bring Linnaeus right into the heart of microbiome studies.

Supplemental Information

Supplemental Information 1 Supplemental Data.

Table S1. Culture media used for culture of isolates from six chicken faecal samples.

Table S2. Summary of sequencing data from 50 chicken faecal samples (BioProject ID PRJNA543206) using Illumina NextSeq sequencing platform

Table S3. Bracken read based and relative abundance values for 50 chicken faecal samples from BioProject ID PRJNA543206.

Table S4. Summary of reads classified by Kraken 2 for 50 chicken faecal samples from BioProject ID PRJNA543206.

Table S5. 1,455 dereplicated Scaffold sequences ≥10kb identified as Category 1 or Category 2 by VirSorter found in 50 metagenomic samples form chicken faeces.

Table S6. BLASTN analysis for 10 bacteriophage genomes showing similarity to known reference bacteriophage sequences at nucleotide level (percentage identity >70%; query covering >50%). For each query sequence, the three best HITs (when ordered according to E-value) have been described.

Table S7. Coverage values for four identified coliphages showing similarity to known bacteriophage genomes of Escherichia coli across the 50 metagenomic samples from which they were derived.

Table S8. Family-level taxonomic classification for nearly 600 de-replicated, bacteriophage genomes of near-full completion.

Table S9. Metadata and sequencing statistics associated with 582 publicly available metagenomic samples derived from the domesticated chicken.

Table S10. Genome statistics for MAGs of >80% completion and <5% contamination constructed from 632 metagenomic samples and binned using MetaBAT2.

Table S11. Genome statistics for Metagenomic Species of >80% completion and <5% contamination derived from gene-based clustering from 632 metagenomic samples.

Table S12. Taxonomic analysis for Metagenomic Species according to GTDB (release 95), Kraken 2 and SpecI classification.

Table S13. Final genus and species level taxonomic classification for all de-replicated metagenomically-derived genome sequences. Newly assigned Latin binomials, alongside their respective etymologies have been provided where appropriate.

Table S14. PubMed hits associated with identified species and the chicken.

Table S15. Latin and Greek roots used combinatorically to create new genus and species names.

Table S16. Genome statistics, clustering statistics at 95% and 99% ANI and final taxonomic classification for all cultured isolates.

Table S17. Final genus and species level taxonomic classification for all de-replicated genome sequences from cultured isolates. Newly assigned Latin binomials, alongside their respective etymologies have been provided where appropriate.

Table S18. Delineation of E. whittamii by ANI using genomes representing the full diversity of the genus Escherichia

Table S19. Redundancy of MGS, cultured species and a comparable dataset of published MAGs (Glendinning et al., 2020).

Table S20. Comparison of genome statistics associated with species found in abundance in both metagenomic and cultured datasets

Table S21. Coverage statistics for 820 MGS sequences across 632 metagenomic samples

Click here for additional data file.

Supplemental Information 2 Supplemental Figures.

Figure S1. Comparisons between metagenomic samples according to BioProject from which they were sourced with regard to a. Number of paired reads input and b. Assembled scaffold N50. c. Completeness and contamination of MAGs and MGS derived from single assembly or co-assembly of all metagenomic samples. Quality estimates were estimated by CheckM and are reported for all Low Quality (designated Reconstructed) (≥50% completion, <5% contamination) Medium Quality (≥80% - ≤90% completion, <5% contamination) and High Quality (>90% completion, <5% contamination) MAGs, with MAGs clustered to form MGS being shown in blue.

Figure S2a. Species accumulation curve for 632 samples over 10 publicly available BioProjects accounting for 820 MGS. Only studies with > 1 sample are included in this plot. A further curve has been plotted in black to depict species accumulation for all metagenomic samples from all BioProjects. Inset plot shows magnification of original plot for species 0-250 (.pdf format). b. A heatmap depicting the abundance of the top 300 MGS, ordered by maximum sample abundance, across 632 samples derived from 10 BioProjects. Rows (MGS) were ordered by correlation clustering while columns (metagenomic sample) were ordered by source BioProject. Metagenomic species have been annotated with their taxonomic class. All data were Log10 transformed with Blue colour depicting species of low abundance and Red showing high abundance. c. Principle component analysis (PCA) based on Bray Curtis dissimilarity matrices between country of origin for MGS relative abundance. d. PCA based on Bray Curtis dissimilarity matrices based on GIT sample site for MGS relative abundance.

Figure S3. Phylogenetic trees reconstructed for genomes of each novel cultured isolate against all available reference genomes of the respective genus obtained from NCBI. Trees were reconstructed using PhyloPhlAn 3.0.58 (Asnicar et al., 2020) against 400 marker genes before reconstruction using FastTree and RAxML of a MAFFT sequence alignment. Genomes of isolates cultured in this study have been highlighted in red and detailed with their respective strain identifier.

Figure S4. Primary ANI clustering of all MAGs and cultured isolates separated according to phylum as determined by dREP. Each plot clusters at species-level according to 95% ANI, as shown by a continual dotted line. Sequence clusters are assigned colours according to designated species-level cluster.

Click here for additional data file.

The authors thank the farmers for collecting the chicken faecal samples for the study.

Additional Information and Declarations

Competing Interests

Author Contributions

Data Availability

New Species Registration

Arss Secka is employed by the West Africa Livestock Innovation Centre.

Rachel Gilroy analysed the data, prepared figures and/or tables, authored or reviewed drafts of the paper, and approved the final draft.

Anuradha Ravi analysed the data, authored or reviewed drafts of the paper, and approved the final draft.

Maria Getino performed the experiments, authored or reviewed drafts of the paper, and approved the final draft.

Isabella Pursley performed the experiments, authored or reviewed drafts of the paper, and approved the final draft.

Daniel L. Horton performed the experiments, authored or reviewed drafts of the paper, and approved the final draft.

Nabil-Fareed Alikhan analysed the data, authored or reviewed drafts of the paper, and approved the final draft.

Dave Baker performed the experiments, authored or reviewed drafts of the paper, and approved the final draft.

Karim Gharbi conceived and designed the experiments, performed the experiments, authored or reviewed drafts of the paper, and approved the final draft.

Neil Hall conceived and designed the experiments, authored or reviewed drafts of the paper, and approved the final draft.

Mick Watson conceived and designed the experiments, authored or reviewed drafts of the paper, and approved the final draft.

Evelien M. Adriaenssens conceived and designed the experiments, analysed the data, authored or reviewed drafts of the paper, and approved the final draft.

Ebenezer Foster-Nyarko performed the experiments, analysed the data, authored or reviewed drafts of the paper, and approved the final draft.

Sheikh Jarju performed the experiments, authored or reviewed drafts of the paper, and approved the final draft.

Arss Secka performed the experiments, authored or reviewed drafts of the paper, and approved the final draft.

Martin Antonio conceived and designed the experiments, authored or reviewed drafts of the paper, and approved the final draft.

Aharon Oren analysed the data, prepared figures and/or tables, authored or reviewed drafts of the paper, and approved the final draft.

Roy R. Chaudhuri analysed the data, authored or reviewed drafts of the paper, and approved the final draft.

Roberto La Ragione conceived and designed the experiments, authored or reviewed drafts of the paper, and approved the final draft.

Falk Hildebrand conceived and designed the experiments, analysed the data, prepared figures and/or tables, authored or reviewed drafts of the paper, and approved the final draft.

Mark J. Pallen conceived and designed the experiments, analysed the data, prepared figures and/or tables, authored or reviewed drafts of the paper, and approved the final draft.

The following information was supplied regarding data availability:

Data are available at BioProject ID PRJNA543206 and Figshare:

Gilroy, Rachel (2020): Gene catalogue and MAGs. figshare. Dataset. DOI 10.6084/m9.figshare.13116809.v4.

Gilroy, Rachel (2020): Cultured isolates. figshare. Dataset. DOI 10.6084/m9.figshare.13234556.v3.

Gilroy, Rachel (2020): Gene catalogue and MAGs. figshare. Dataset. DOI 10.6084/m9.figshare.13116809.v4.

Gilroy, Rachel (2020): Cultured isolates. figshare. Dataset. DOI 10.6084/m9.figshare.13234556.v3.

Gilroy, Rachel (2020): MAGs. figshare. Dataset. DOI 10.6084/m9.figshare.13187396.v1.

Gilroy, Rachel (2020): MAGs10. figshare. Dataset. DOI 10.6084/m9.figshare.13234478.v1.

The following information was supplied regarding the registration of a newly described species:

Hundreads of new candidatus taxa and dozens of new cultured bacterial taxa are detailed in the article and Supplemental Material.

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
