# Peer review of "Extensive microbial diversity within the chicken gut microbiome revealed by metagenomics and culture"

_PeerJ, doi:10.7717/peerj.10941_

## Round 0.1 · original submission · Minor Revisions

It is my opinion that this manuscript represents a significant body of work and an important contribution, and requires only minor revisions for me to recommend for publication. Please see reviewer comments for specific issues with wording and/or grammar.

There are two minor areas of concern that I believe you should address. First, the title suggests a functional characterization of the chicken fecal microbiome, and this is somewhat misleading (as noted by one reviewer) as you don't provide gene annotations. I don't consider annotations necessary for the current work; however, a rewrite of the title to focus on taxonomic diversity characterization would make it more palatable. Second, please ensure that your figures are high-quality (high dpi or vector graphics) so they can be understood at high magnification; as one reviewer noted, Figure 3 in particular was not legible.

Reviewer 1 ·

Basic reporting

This is a worthwhile contribution that reports new metagenomes and representative genomes that, very likely, will be used by others in the future. However, I felt that in most lines of research that the paper presented, the reported results fell a bit short of what could be accomplished. Please see below for more details.

Experimental design

This is good overall but I felt the paper did not reach the expectation set by its title. Please see below for more details.

Validity of the findings

The methods are well executed; the results are valid overall.

Additional comments

This is a worthwhile contribution that reports new metagenomes and representative genomes that, very likely, will be used by others in the future. However, I felt that in most lines of research that the paper presented, the reported results fell a bit short of what could be accomplished. For instance, the paper claims to report a gene catalogue of the chicken microbiome but the gene sequences released lack functional annotations and metadata, and there was no discussion of what are the key, core functions that characterize this microbiome, and distinguish it from that of other animals. The paper attempts to name several new species recovered by isolates and metagenomes but the taxa descriptions are really minimum (too “shallow” in my view) as they do not include any ecological or functional characteristics (or diagnostic phenotypes) of the newly proposed taxa. So, I am not sure this is a good practice (seems more like “stamp collection” that could possibly prevent future, more detailed taxonomic descriptions). See also specific examples below. Further, I have an issue with the title because the results reported do not really represent much of a global analysis (this was very limited). That said, the methods are well described and robust. The data reported will be valuable to the scientific community, which is quite large for this specific area.

Abstract: the authors seem to focus on the taxonomic novelty of the members of the chicken microbiome but, given its title, I was also expecting to read about variation among chicken individuals and functions characterizing the chicken vs. other animal microbiomes.

Line 179. It is good that the authors released the gene catalogue and the sequences but the information is not useful without functional predictions and other metadata. Seems like only the fasta file with unique identifiers were included in Figshare.

Lines 218-222. The second approach for defining genomospecies (or de-replicating MAGs) is more straightforward than the first one reported just above. How consistent its results were compared to approach 1? Later in Results, it is reported that the approach 1 found seven new genomospecies, but how abundant and prevalent these were, how many genomospecies the second method found that the approach #1 did not, and what is the evidence that the uniquely found 7 are not chimeric, were not provided.

Lines 340-342. Can you define presence? If it was a couple reads in one sample, I would not call it a core member of the chicken microbiome and hence, these findings carry less weight. This applies to other parts of the study. In general, presence and prevalence should be clearly defined and what is core, or almost core, members should also be clearly outlined in order for the whole paper to read more clear.

Lines 348-350. Is it possible that these reads represent close relatives of E. coli as opposed to E. coli, e.g. what is the threshold for Bracken to assign a read to a genome (if the relatives are missing from the database -as it is likely the case here- Bracken/Kracken will assign reads to the closest available relative, if close enough)? Also was the underlying analysis done in a read competitive manner with all reference genomes?

Lines 418-420. I have an issue with the title because I would argue that some of the previous studies represent the global chicken microbiome much more than the present study which is focused on ~50 fecal samples form a single site in Great Brittan and two chicken lines.

Line 437. I think it is important to mention the prevalence of these species in the chicken samples and probably why the other approach (95% ANI de-replication of MAGs), which is much more popular, missed them. See also Rodriguez-R et al, Env. Microbiol. 2020, for some useful definitions of presence, prevalence etc., based on metagenomic data.

Line 445-450. I think this could be reported better (more quantitively) in terms of gains in phylogenetic diversity by the new MAGs. And to report these gains in terms of (or against) i) named species (isolates and Candidatus) and ii) all available MAG in GTDB.

Line 454-457. I think it is important to mention the prevalence of these species in the chicken samples, e.g., do they represent “spurious matches” in a couple samples or a truly prevalent and core members of the chicken microbiome?

Line 489-490. I am not sure this effort to name all novel taxa in the chicken microbiome does a good service to science. I like the approach to create names and I believe it solves a significant problem we have in the field, but I would prefer we name the taxa we care (not necessarily all extant taxa), and study the taxa in more depth their function or ecology and not just trying to put our stamp on each genomospecies recovered. For instance, what if it is proven that a species named here is much more abundant in other birds than chicken and by chance only was found in the chicken samples? The proposed names would not be successful in such case, and the existence of the names will deter other people for proposing new, most successful names, I believe.

Line 542-545. I think this is an important topic to expand upon.

Line 556. Several punctuation marks are missing like here; or a comma is in place of a period. But text is overall very good.

Reviewer 2 ·

Basic reporting

A few figures require minor attention:

FIG 3 Unfortunately, this figure was not legible in the reviewer PDF. The symbols called out (triangles, hallow symbols, or stars) can’t be seen. Not sure if this is something the authors could have fixed or a problem with the PeerJ submission system, just be sure in the final print the image can be increased for the reader.


FIG S2a Please include more details regarding this panel legend. It is not abundantly clear what the inset graph is representing. I believe the information is already in the figure legend, it should just be labeled more distinctly.

Experimental design

No comment

Validity of the findings

No comment

Additional comments

This paper represents an exhaustive look into the chicken gut microbiome. The manuscript was highly detailed in its description and justification for the experimental design, as well as included extraordinarily detailed and thorough descriptions of the methods employed. This manuscript is a great example of the utility of combining metagenomics and culture techniques to gain deep insight into microbial communities.

---

## Round 0.2 · accepted · Accept

Thank you for contributing this important work!